# Scalable, Explainable and Provably Robust Anomaly Detection with One-Step Flow Matching

**Zhong Li**♣,♡    **Qi Huang**♣    **Yuxuan Zhu**▲    **Lincen Yang**♣(✉)
**Mohammad Mohammadi Amiri**▲    **Niki van Stein**♣    **Matthijs van Leeuwen**♣

♣The Leiden Institute of Advanced Computer Science (LIACS), Leiden University
▲Department of Computer Science, Rensselaer Polytechnic Institute
♡The Intelligent Computing Research Center, Great Bay University
Corresponding Author (✉): `l.yang@liacs.leidenuniv.nl` (Lincen Yang)

## Abstract

We introduce Time-Conditioned Contraction Matching (TCCM), a novel method for semi-supervised anomaly detection in tabular data. TCCM is inspired by flow matching, a recent generative modeling framework that learns velocity fields between probability distributions and has shown strong performance compared to diffusion models and generative adversarial networks. Instead of directly applying flow matching as originally formulated, TCCM builds on its core idea—learning velocity fields between distributions—but simplifies the framework by predicting a time-conditioned contraction vector toward a fixed target (the origin) at each sampled time step. This design offers three key advantages: (1) a lightweight and scalable training objective that removes the need for solving ordinary differential equations during training and inference; (2) an efficient scoring strategy called one time-step deviation, which quantifies deviation from expected contraction behavior in a single forward pass, addressing the inference bottleneck of existing continuous-time models such as DTE (a diffusion-based model with leading anomaly detection accuracy but heavy inference cost); and (3) explainability and provable robustness, as the learned velocity field operates directly in input space, making the anomaly score inherently feature-wise attributable; moreover, the score function is Lipschitz-continuous with respect to the input, providing theoretical guarantees under small perturbations. Extensive experiments on the ADBench benchmark show that TCCM strikes a favorable balance between detection accuracy and inference cost, outperforming state-of-the-art methods—especially on high-dimensional and large-scale datasets. The source code is provided at `https://github.com/ZhongLIFR/TCCM-NIPS`.

## 1 Introduction

**Background.** Anomaly detection in tabular data is the task of identifying data instances (or patterns) that deviate significantly from expected behavior (Chandola et al., 2009; Aggarwal and Aggarwal, 2017a; Pang et al., 2021). It has found widespread applications in various domains, such as fraud detection in finance (Hilal et al., 2022), fault detection in manufacturing (Yu and Zhang, 2023), intrusion detection in cybersecurity (Chou and Jiang, 2021), and medical diagnosis in healthcare (Fernando et al., 2021). In these high-stakes domains, data is growing rapidly in both size and dimensionality, calling for approaches that are not only effective but also scalable. Equally important, decisions made in these settings often have critical consequences, making interpretability an ethical and regulatory necessity (Li et al., 2023). Therefore, anomaly detection methods should also be able to provide meaningful explanations alongside accurate predictions.

39th Conference on Neural Information Processing Systems (NeurIPS 2025).

**Positioning our work.** Existing anomaly detection methods can be broadly categorized into classical machine learning approaches and deep learning-based techniques. Classical methods—such as OCSVM (Schölkopf et al., 1999), LOF (Breunig et al., 2000), PCA (Shyu et al., 2003), and KDE (Latecki et al., 2007)—often struggle with high-dimensional data due to the curse of dimensionality, and with large-scale datasets due to limited computational scalability. To address these limitations, deep learning-based anomaly detection methods have gained research attention and achieved strong performance across various domains (Pang et al., 2021). Deep methods can be grouped into two categories: (1) *two-stage approaches*, which first learn a low-dimensional representation (e.g., via an autoencoder) and then apply off-the-shelf anomaly detectors. However, such decoupled training strategies often struggle to learn task-effective features due to the lack of joint optimization (Nguyen and Vien, 2019); and (2) *end-to-end trained approaches*, which integrate representation learning and anomaly detection into a unified training objective, achieving better performance. Meanwhile, given that labeled anomalies are both scarce and expensive to obtain in many real-world applications—such as system failures, fraud, or clinical anomalies—many existing methods, both classical and deep, adopt a *semi-supervised* setting. In this paradigm, models are trained solely on normal data and tasked with identifying deviations at test time. Although claimed as "unsupervised" in some studies (Goodge et al., 2022), we rigorously refer to this setup as *semi-supervised anomaly detection* following An and Cho (2015); Ruff et al. (2018); Akcay et al. (2018); Bergman and Hoshen (2020). Our work is situated within this setting, adopting a deep[1], end-to-end, and semi-supervised approach that learns the structure of normality during training and identifies deviations at inference.

**Limitations of Existing Studies.** Despite recent advances in end-to-end deep anomaly detection, many existing approaches face fundamental limitations. Adversarial models such as AnoGAN (Schlegl et al., 2017) and GANomaly (Akcay et al., 2018) often suffer from training instability due to their reliance on min-max optimization. Density-based methods like DAGMM (Zong et al., 2018) introduce complex architectures to approximate latent distributions. Diffusion-based approaches such as Anomaly-DDPM and DTE (Livernoche et al., 2023) may depend heavily on carefully tuned noise schedules and sampling hyperparameters; in practice, they also suffer from extremely slow inference on large-scale datasets due to their iterative nature. Normalizing flows (e.g., OneFlow (Maziarka et al., 2021)) require invertibility and Jacobian computations, leading to trade-offs between model expressivity and computational efficiency. LUNAR (Goodge et al., 2022), which employs graph neural networks to capture relational structures, incurs high training costs and scales poorly with data size. Methods such as DeepSVDD (Ruff et al., 2018), which focus on compact representation learning, rely on restrictive architectural constraints (e.g., no biases, bounded activations) to avoid representation collapse, and often depend on numerous training heuristics to yield satisfactory results. Another major limitation lies in the lack of interpretability—most deep models offer little insight into why a sample is considered anomalous. While a few methods have made progress in this direction—e.g., AE-1SVM (Nguyen and Vien, 2019) using gradient-based attribution, ICL (Shenkar and Wolf, 2022) identifying key contributing features, MCM (Yin et al., 2024) modeling both feature-level abnormality and inter-feature correlations, and DTE (Livernoche et al., 2023) providing denoised reconstructions as explanations—such interpretable designs remain the exception rather than the norm. The vast majority of deep anomaly detection models continue to operate as black boxes, limiting their utility in high-stakes domains where interpretability is critical.

**Flow Matching.** Flow matching has emerged as a promising generative modeling framework that retains the training stability and expressivity of diffusion models, while offering improved computational efficiency (Liu et al., 2022; Lee et al., 2023). Instead of relying on stochastic differential equations (SDEs), it learns an ordinary differential equation (ODE) that deterministically maps samples from a source to a target distribution, enabling faster sampling and easier optimization. Flow matching also bypasses the need for a forward noising process and explicit density functions, making it well-suited for settings with implicit or intractable data distributions (Albergo and Vanden-Eijnden, 2023; CSAIL, 2024). Its interpolant formulation further allows empirical analysis of learned velocity fields over time (Albergo and Vanden-Eijnden, 2023), enhancing interpretability for downstream tasks. Despite its recent success in generative modeling, flow matching has not been explored for anomaly detection as of this writing, to the best of our knowledge.

---

[1]Following common practice in the machine learning community, we use the term "deep" to indicate deep learning-based, end-to-end models, even when the employed neural architecture is relatively shallow (e.g., a multi-layer perceptron with two hidden layers).

**Contributions.** Motivated by the limitations of existing deep anomaly detection methods and benefiting from recent advances in generative modeling—particularly flow matching—we introduce *Time-Conditioned Contraction Matching* (TCCM), a novel flow matching-inspired approach for semi-supervised anomaly detection. Specifically, TCCM learns a time-conditioned velocity field that contracts normal data, drawn from a source distribution $\rho_{\text{source}}$, towards a degenerate target distribution $\rho_{\text{target}}$, defined as a Dirac delta at the origin. Unlike previous approaches that simulate full continuous trajectories via ODE or SDE integration, TCCM avoids trajectory simulation entirely by directly learning a velocity field that approximates contraction dynamics from any input point at any time (details are described in Section 3). At test time, samples are scored based on how much their predicted velocity field deviates from the expected contraction pattern—an idea illustrated in Figure 1. This mismatch in velocity magnitudes and directions forms the basis of our anomaly score. TCCM inherits the scalability and simplicity of flow matching (Liu et al., 2022), and is trained using an unconstrained least-squares objective. It avoids adversarial instability (as in AnoGAN (Schlegl et al., 2017), GANomaly (Akcay et al., 2018)), complex density modeling (as in DAGMM (Zong et al., 2018) or KDE (Latecki et al., 2007)), and slow sampling-based inference (as in DTE (Livernoche et al., 2023)). Unlike normalizing flows (Maziarka et al., 2021), it requires neither invertibility nor Jacobian computation, and unlike DeepSVDD (Ruff et al., 2018), it does not rely on restrictive architectural constraints to avoid collapse. Furthermore, compared to graph-based methods like LUNAR (Goodge et al., 2022), TCCM achieves significantly faster training on large-scale datasets. Crucially, TCCM is inherently interpretable—its velocity field lives in the input space, supporting feature-wise attribution—and provably robust, with a Lipschitz-continuous anomaly score under small input perturbations.

**Findings.** We evaluate TCCM on 47 benchmark datasets from the ADBench suite (Han et al., 2022), comparing it against 44 baseline methods (23 deep learning-based and 21 classical), for a total of **10,575 runs** across five seeds. Our results demonstrate five key strengths: (1) *Accuracy*: TCCM achieves **top-1 performance** in both AUPRC and AUROC scores (see Appendix B.4 for definitions) across all evaluated methods (see Figures 2a and 2b for aggregated results). (2) *Scalability*: The model is highly efficient in both training and inference on high-dimensional and large-scale datasets—achieving, on average, **1573× faster inference** than DTE-NonParametric (top-2 in AUROC and AUPRC), and **85× faster inference** than LUNAR (top-3 in both metrics), while maintaining superior detection performance (see Figure 3a). (3) *Explainability*: TCCM enables feature-level attribution for anomaly scores, supporting interpretable diagnosis—an aspect largely absent in existing deep anomaly detection models (see Figure 4). (4) *Robustness*: We theoretically prove that the anomaly score satisfies a Lipschitz continuity condition, offering provable robustness guarantees under input perturbations (see Proposition 1). (5) *Simplicity of training*: TCCM requires no adversarial losses, density estimation, or noise schedules—making it simple to train, stable to optimize, and easy to reproduce (see Eq. 4). Together, these findings establish TCCM as a principled, highly effective, scalable, explainable, and provably robust solution for semi-supervised anomaly detection in tabular data.

## 2 Preliminaries

Due to space constraints, we defer a detailed discussion of related work—including anomaly detection methods and flow matching—to Appendix A, and begin with a general problem statement.

### 2.1 Problem Statement

**Notations.** Bold lowercase letters (e.g., $\boldsymbol{x}$) denote vectors; bold uppercase letters (e.g., $\boldsymbol{X}$) represent matrices. Calligraphic symbols (e.g., $\mathcal{X}$) denote sets, and standard italic letters (e.g., $x$) are used for scalars, unless otherwise specified. Besides, these symbols may be used to denote both random variables and their realizations; this dual use is common in the machine learning literature and will be made explicit whenever necessary.

**Problem Setting.** We consider a semi-supervised anomaly detection scenario, where only normal samples are available during training. Let $\mathcal{X} = \{\boldsymbol{x}_i\}_{i=1}^N \subset \mathbb{R}^d$ be a dataset of $d$-dimensional observations, partitioned into a training set $\mathcal{X}_{\text{train}}$ containing only normal instances sampled from an unknown distribution $p_{\text{data}}(\boldsymbol{x})$, and a test set $\mathcal{X}_{\text{test}}$ that may include both normal and anomalous samples.

**Problem 1** (Semi-Supervised Anomaly Detection). *Given access to normal training data $\mathcal{X}_{train} \subset \mathbb{R}^d$, the goal is to learn an anomaly scoring function $S : \mathbb{R}^d \to \mathbb{R}$ that quantifies the deviation of any test input $\boldsymbol{x} \in \mathcal{X}_{test}$ from the learned notion of normality.*

To solve this problem, we aim to learn the structure of normal data using only unlabeled normal instances. At test time, deviations from this learned structure are quantified and assigned anomaly scores, allowing the detection of abnormal inputs without access to anomalous data during training.

## 2.2 Recap of Flow Matching

Flow matching (or stochastic interpolant) (Lipman et al., 2022; Albergo and Vanden-Eijnden, 2023; Liu et al., 2022) provides a principled and flexible framework for learning neural ODE-based transport maps between two empirical distributions. Given samples from a source distribution $\boldsymbol{x}_0 \sim p_0$ and a target distribution $\boldsymbol{x}_1 \sim p_1$, the goal is to learn a time-dependent velocity field $v(\boldsymbol{x}_t, t)$ such that the following ODE governs the evolution between the two: $d\boldsymbol{x}_t = v(\boldsymbol{x}_t, t)dt$ for $t \in [0, 1]$. This Lagrangian formulation describes the motion of particles from $\boldsymbol{x}_0$ to $\boldsymbol{x}_1$, implicitly defining the coupling $\pi(p_0, p_1)$ between the distributions. The velocity field is parameterized as $v_\theta(\boldsymbol{x}_t, t)$ via a neural network and trained to match a reference velocity field using a simple least-squares objective:

$$\min_\theta \mathbb{E}_{t,\boldsymbol{x}_t} \left[ \|v(\boldsymbol{x}_t, t) - v_\theta(\boldsymbol{x}_t, t)\|_2^2 \right]. \tag{1}$$

Different choices of interpolation path $\boldsymbol{x}_t$ and reference velocity $v(\boldsymbol{x}_t, t)$ give rise to different flow matching models. A widely used class of methods adopts the *probability flow ODE* formulation (Song et al., 2020), where the velocity incorporates the score function $\nabla \log p_t$ and corresponds to a deterministic trajectory derived from an underlying SDE. In this case, the path is often defined via a variance-preserving schedule:

$$\boldsymbol{x}_t = \alpha_t \boldsymbol{x}_0 + \sqrt{1 - \alpha_t^2}\, \boldsymbol{x}_1, \quad \text{with} \quad \alpha_t = \exp\left(-\frac{1}{2}\int_0^t \beta(s)\, ds\right),$$

where $\beta(s)$ is a pre-defined noise schedule that controls the rate of variance increase over time. This form allows equivalence to score matching under certain conditions (Lee et al., 2023; Zheng et al., 2023), and is popular in diffusion-based generative models. However, the resulting curved trajectory can complicate optimization and slow down sampling (Liu et al., 2022).

To address these issues, Liu et al. (2022) have proposed a *constant velocity ODE* approach, where the interpolation path is simply linear: $\boldsymbol{x}_t = (1 - t)\boldsymbol{x}_0 + t\boldsymbol{x}_1$. In this case, the reference velocity becomes a constant vector $\boldsymbol{x}_1 - \boldsymbol{x}_0$, and the flow matching objective reduces to:

$$\min_\theta \mathbb{E}_{t,\boldsymbol{x}_t} \left[ \|\boldsymbol{x}_1 - \boldsymbol{x}_0 - v_\theta(\boldsymbol{x}_t, t)\|_2^2 \right]. \tag{2}$$

This variant is known as the *rectified flow* model, and has been shown to improve training efficiency and reduce curvature in the learned trajectories, facilitating both forward simulation and backward sampling. Our proposed method builds on this formulation, leveraging its simplicity and scalability while adapting it for the anomaly detection setting.

## 3 Methodology: Time-Conditioned Contraction Matching (TCCM)

**Core idea.** Conventional flow matching models (Lipman et al., 2022; Liu et al., 2022) construct continuous-time trajectories that gradually transport samples from a source distribution (at $t = 0$) to a target distribution (at $t = 1$) by integrating a learned velocity field over the entire time interval. In contrast, our method departs from this paradigm both conceptually and technically. Rather than relying on the full trajectory across time to learn the transformation, we directly learn a *contraction vector field* at each time step—one that immediately points from the current position to the fixed target (the origin). This allows the model to predict the contraction behavior independently at each time point, avoiding the need for simulating or reconstructing the full flow path. This provides a powerful yet simple framework for anomaly detection: every point learns how to contract back to the origin over time, which motivates the name *Time-Conditioned Contraction Matching* (TCCM).

Formally, we treat the data distribution as the source, $\boldsymbol{z}_0 := \boldsymbol{z} \sim p_{\text{data}}$, and consider the target as a degenerate Dirac distribution at the origin, $\boldsymbol{z}_1 := \boldsymbol{0}$. While this setup may suggest a flow-like

interpretation, we emphasize that our model is not tasked with approximating the full solution of a dynamical system such as:

$$dz(t) = -z(t)dt, \quad \text{with} \quad z(0) = z, \tag{3}$$

whose analytical solution would be $z(t) = z \cdot e^{-t}$. However, our model does not supervise or simulate $z(t)$ across time. Instead, we adopt a simplified training strategy that uses a constant target direction $-z$ for supervision at all time steps. To achieve this, we learn a neural velocity field $f_{\theta}(\cdot)$ on an augmented space $\tilde{z} = [z; \text{Embed}(t)]$. Specifically, $f_{\theta}(\cdot)$ is conditioned on both the input $z$ and a time variable $t \in [0, 1]$. The time is encoded using sinusoidal embeddings (Vaswani et al., 2017), which are concatenated with the input: $\tilde{z} = [z; \text{Embed}(t)]$, and passed through the model $f_{\theta}$ to predict a contraction vector.

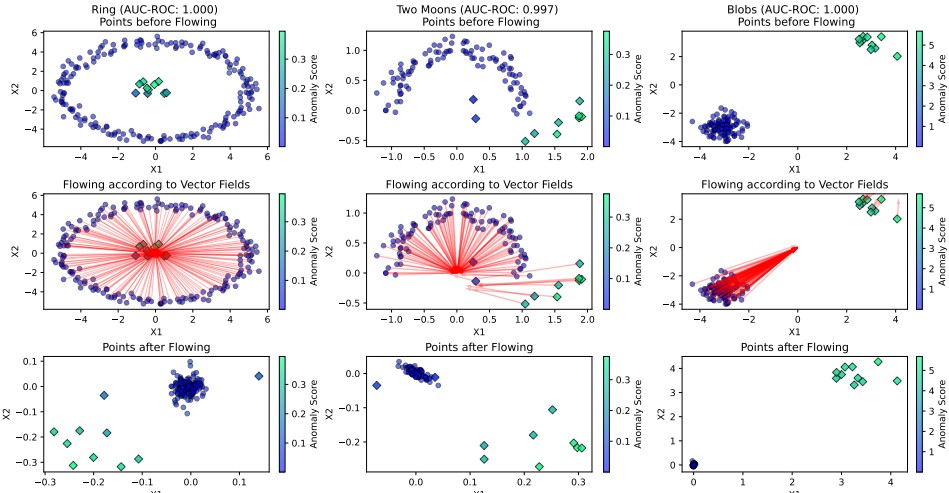

Figure 1: **Core idea of TCCM:** TCCM learns a time-conditioned velocity (vector) field that contracts normal data points, sampled from a source distribution $\rho_{\text{source}}$, towards a degenerate target distribution $\rho_{\text{target}}$, defined as a Dirac delta at the origin. At test time, anomalies are detected by measuring inconsistency with this learned contraction field. **Illustrative examples:** We visualize TCCM behavior on synthetic 2D datasets with varying normal (circles) and anomalous (squares) distributions. *Left*: Normal data form a ring; anomalies are sampled from a central Gaussian. *Middle*: Normals follow an upper moon; anomalies form a sparse lower moon. *Right*: Normals are clustered bottom-left; anomalies are drawn from a distinct Gaussian in the top-right. In all cases, TCCM successfully distinguishes anomalies based on their deviation from the expected contraction vector.

**Training Objective.** The training minimizes the following loss:

$$\min_{\theta} \mathbb{E}_{z \sim p_{\text{data}}, \, t \sim \mathcal{U}(0,1)} \left[ \| f_{\theta}([z; \text{Embed}(t)]) + z \|_2 \right]. \tag{4}$$

Optimizing objective (4) encourages the model to predict a velocity vector that approximates the negation of the current state, i.e., $f_{\theta}([z; \text{Embed}(t)]) \approx -z$. This guides the system to evolve toward the origin by learning both the direction and magnitude of motion in a time-dependent manner. To achieve this, the neural velocity field is required to extract the common factors of variation present in normal data. As a result, normal samples following their predicted velocity fields can approach the origin at any given time, while anomalous instances, due to their deviation from the learned structure, fail to do so. The pseudocode for training is given in Algorithm 1 in Appendix B.5.

**Interpretation and Motivation.** Although not derived from an explicit ODE, our method can be viewed as learning a time-aware vector field that approximates the contraction dynamics toward a shared target. This formulation provides several advantages: *(1) Time-Conditioned Consistency:* The model learns to predict contraction vectors across time steps that consistently guide inputs toward the origin, promoting geometric alignment and stability; *(2) Simplified Supervision:* Using a fixed supervision target $-z$ removes the need for trajectory supervision, leading to a simpler and smoother optimization process; *(3) No ODE Solvers Required:* Unlike conventional flow-based models, TCCM avoids numerical integration during both training and inference, resulting in substantial computational

efficiency; *(4) Learnable Temporal Dynamics:* Sinusoidal time embeddings allow the model to modulate both the magnitude and direction of contraction vectors over time, enabling rich, non-linear temporal behavior.

**Anomaly Scoring at Inference Time.** Given a test input $z \in \mathcal{X}_{\text{test}}$ and a fixed evaluation time $t_{\text{fixed}} \in (0, 1]$, we define the anomaly score as:

$$S_{\text{fixed}}(z; t_{\text{fixed}}) = \| f_{\boldsymbol{\theta}}([z; \text{Embed}(t_{\text{fixed}})]) + z \|_2, \tag{5}$$

where $f_{\boldsymbol{\theta}}([z; \text{Embed}(t)])$ denotes the learned velocity field conditioned on both the input feature $z$ and the time variable $t$, encoded via sinusoidal embeddings and concatenated with $z$ before being passed into a multilayer perceptron (MLP). This scoring strategy is grounded in the following expectations: (1) *Normal instances* are trained to follow a contraction path toward the origin. Since supervision is based on a constant target $-z$ across all time steps, a well-aligned normal sample satisfies $f_{\boldsymbol{\theta}}([z; \text{Embed}(t)]) \approx -z$, leading to a small residual norm. (2) *Anomalous instances*, which deviate from the learned contraction pattern, yield misaligned velocities and hence higher residuals. (3) This approach is computationally efficient, as it avoids solving ODEs and requires only a single forward pass through the network at a chosen time step $t_{\text{fixed}}$, which overcomes the primary bottleneck of high evaluation cost found in existing continuous-time ODE/SDE models such as Anomaly-DDPM (Livernoche et al., 2023) and DTE-NonParametric (Livernoche et al., 2023). (4) Importantly, because the residual vector $f_{\boldsymbol{\theta}}([z; \text{Embed}(t_{\text{fixed}})]) + z$ lies in the original feature space, the absolute values of its entries directly quantify how much each feature contributes to the anomaly score. This provides intrinsic feature-level interpretability, in contrast to post-hoc explanation methods such as SHAP (Lundberg and Lee, 2017) and LIME (Ribeiro et al., 2016).

We refer to this anomaly scoring procedure as *one-step flow matching* because, unlike classical flow matching models that integrate velocity fields across time to compute transformation paths, our method makes a single-time-point evaluation to determine alignment with the learned contraction dynamics. While the underlying model is termed *TCCM*, this scoring mechanism captures the spirit of flow matching—comparing learned dynamics to an ideal contraction vector—yet does so in a highly scalable one-step formulation. Although the evaluation time $t_{\text{fixed}}$ in Eq 5 can be any value in $(0, 1]$, we set $t_{\text{fixed}} = 1$ by default throughout our experiments for simplicity. Particularly, we provide a sensitivity analysis (see Figure 13) showing that the anomaly detection performance is largely stable across different values of $t$, validating the temporal consistency of the learned flow field and its ability to produce meaningful predictions at any time step. The pseudocode for inference is given in Algorithm 2 in Appendix B.5.

## 4 Theoretical Properties of TCCM

In this section, we establish two key theoretical properties of our method: *(i)* Lipschitz continuity of the anomaly score, which leads to provable robustness guarantees under input perturbations; and *(ii)* discriminative behavior of the score function under distributional shift, explained via a stylized Gaussian mixture setting. These results offer both certifiability of robustness and theoretical insight into the score function's discriminative behavior, complementing our empirical findings.

**Proposition 1** (Lipschitz Continuity and Robustness). *Let $f_{\boldsymbol{\theta}}(\cdot, t_{\text{fixed}})$ be L-Lipschitz continuous in its first argument (for a fixed time $t_{\text{fixed}} \in (0, 1]$). Then the anomaly score*

$$S_{\text{fixed}}(\boldsymbol{x}) := \| f_{\boldsymbol{\theta}}([\boldsymbol{x}; \text{Embed}(t_{\text{fixed}})]) + \boldsymbol{x} \|_2$$

*is $(L + 1)$-Lipschitz continuous with respect to $\boldsymbol{x}$, i.e.,*

$$|S_{\text{fixed}}(\boldsymbol{x}_1) - S_{\text{fixed}}(\boldsymbol{x}_2)| \leq (L + 1)\|\boldsymbol{x}_1 - \boldsymbol{x}_2\|_2.$$

*Proof.* Define $g(\boldsymbol{x}) := f_{\boldsymbol{\theta}}([\boldsymbol{x}; \text{Embed}(t_{\text{fixed}})]) + \boldsymbol{x}$. Then:

$$\|g(\boldsymbol{x}_1) - g(\boldsymbol{x}_2)\|_2 \leq \|f_{\boldsymbol{\theta}}([\boldsymbol{x}_1; \text{Embed}(t)]) - f_{\boldsymbol{\theta}}([\boldsymbol{x}_2; \text{Embed}(t)])\|_2 + \|\boldsymbol{x}_1 - \boldsymbol{x}_2\|_2 \leq (L+1)\|\boldsymbol{x}_1 - \boldsymbol{x}_2\|_2.$$

Finally, since the $\ell_2$ norm is 1-Lipschitz, we have $|S_{\text{fixed}}(\boldsymbol{x}_1) - S_{\text{fixed}}(\boldsymbol{x}_2)| \leq \|g(\boldsymbol{x}_1) - g(\boldsymbol{x}_2)\|_2$. $\square$

**Remark and Implications.** The assumption that $f_{\boldsymbol{\theta}}(\cdot, t_{\text{fixed}})$ is Lipschitz is both theoretically and practically reasonable. In continuous normalizing flows (CNFs) and flow-matching models, such smoothness is often required to ensure existence and uniqueness of solutions (via Picard–Lindelöf

theorem (Murray and Miller, 2013)) or to stabilize ODE solvers. More specifically, the function $f_{\boldsymbol{\theta}}$ is implemented as a multilayer perceptron with ReLU activations and a fixed architecture, making it piecewise linear and hence Lipschitz continuous. The Lipschitz constant $L$ can be further controlled through spectral normalization, gradient penalties, or other regularization techniques. Moreover, this proposition has the following two **implications**: (1) *robustness:* the score is stable under small perturbations, enhancing reliability in noisy or adversarial environments; and (2) *certifiability:* the bound implies that $|S(\boldsymbol{x} + \delta) - S(\boldsymbol{x})| \leq (L+1)\varepsilon$ if $\|\delta\| \leq \varepsilon$, providing a certifiable safety margin.

To theoretically support the discriminative power of our anomaly score, we analyze an idealized setting where normal and anomalous instances are drawn from two disjoint Gaussian mixture models (GMMs) with shared isotropic covariance. Although simplified, this setup enables a clean analysis of how the learned score function behaves on out-of-distribution samples. Under the assumption that the model has learned a noisy contraction field of the form $f_{\boldsymbol{\theta}}([\boldsymbol{x}; \text{Embed}(1)]) = -\boldsymbol{x} + \boldsymbol{\epsilon}$ for normal training data, we establish the following result:

**Proposition 2** (Discriminative Power under GMM-to-GMM Shift). *Let normal samples be drawn from a Gaussian mixture $p_{normal}(\boldsymbol{x}) = \sum_{r=1}^{R} \pi_r \cdot \mathcal{N}(\boldsymbol{\mu}_r, \sigma^2 I_d)$, and anomalous samples from a disjoint mixture $p_{anom}(\boldsymbol{z}) = \sum_{s=1}^{S} \eta_s \cdot \mathcal{N}(\boldsymbol{\nu}_s, \sigma^2 I_d)$, with $\boldsymbol{\nu}_s \notin \{\boldsymbol{\mu}_r\}_{r=1}^{R}$. Assume the learned contraction field satisfies $f_{\boldsymbol{\theta}}([\boldsymbol{x}; Embed(1)]) = -\boldsymbol{x} + \boldsymbol{\epsilon}$, where $\boldsymbol{\epsilon} \sim \mathcal{N}(\boldsymbol{0}, \sigma_f^2 I_d)$; and the learned velocity field is mismatched for anomalies. Define the anomaly score: $S(\boldsymbol{x}) = \|f_{\boldsymbol{\theta}}([\boldsymbol{x}; Embed(1)]) + \boldsymbol{x}\|_2 = \|\boldsymbol{\epsilon}\|_2$. Then, it holds that: (1) for normal samples, $S(\boldsymbol{x}) \sim \chi_d \cdot \sigma_f$, and $\mathbb{E}[S(\boldsymbol{x})] = \sigma_f \cdot \sqrt{2} \cdot \frac{\Gamma(\frac{d+1}{2})}{\Gamma(\frac{d}{2})}$; (2) for anomalies, let $\lambda_s = \frac{\|\boldsymbol{\nu}_s - \boldsymbol{\mu}_{r^*(s)}\|_2^2}{\sigma_f^2}$, where $r^*(s) := \arg\min_r \|\boldsymbol{\nu}_s - \boldsymbol{\mu}_r\|_2$, we have $S(\boldsymbol{z}) \sim \sum_{s=1}^{S} \eta_s \cdot \chi_d(\lambda_s)$; and (3) the expected anomaly scores of normal and anomalous instances satisfy: $\mathbb{E}[S(\boldsymbol{z})] > \mathbb{E}[S(\boldsymbol{x})]$. This implies that our score function assigns, in expectation, higher values to anomalies than to normal points—providing a theoretical foundation for its discriminative capability.*

The proof, provided in Appendix C.2, shows that the anomaly score corresponds to the norm of a central chi-distributed variable for normal samples and a non-central chi-distributed one for anomalies. The non-centrality parameter captures the squared distance between each anomaly and the closest normal-mode center, leading to systematically larger scores.

**Implications.** This result provides a theoretical lens into why our anomaly score increases for distributional outliers. Even though real-world data may not exactly follow Gaussian mixtures, the underlying intuition persists: samples that deviate from the structure captured by the contraction field are naturally assigned larger residuals.

In addition, Appendix C.3 further analyzes the model's representation dynamics and verifies that TCCM avoids degenerate or collapsed mappings in practice, complementing the above theoretical guarantees with empirical evidence of stable and discriminative behavior.

## 5 Experiments

We conduct comprehensive experiments to address the following research questions: (1) **Effectiveness**—Can TCCM outperform existing baselines in anomaly detection? (2) **Scalability**—How does TCCM compare to the strongest baselines in detection accuracy in terms of training and inference efficiency? (3) **Explainability**—Are the explanations generated by TCCM intuitive and meaningful to human users? (4) **Ablation Studies and Sensitivity Analysis**—How do various design choices impact the performance of TCCM?

### 5.1 Experiment Setup

**Datasets Description and Processing.** (1) **Dataset Description**: A summary of the datasets used in our study is provided in Table 1. We adopt 47 benchmark datasets from the well-established ADBENCH benchmark (Han et al., 2022), spanning diverse domains including sociology, finance, linguistics, physics, and healthcare. To enable a comprehensive evaluation of different anomaly detectors, including our proposed method, we categorize the datasets into four groups based on their scale and dimensionality: (a) *High-dimensional* datasets, with more than 50 features; (b)

*Large-scale* datasets (but not high-dimensional), containing more than 10,000 instances and fewer than 50 features; (c) *Medium-scale* datasets (not high-dimensional), with 1,000 to 10,000 instances; and (d) *Small-scale* datasets (not high-dimensional), containing fewer than 1,000 instances. This categorization facilitates a nuanced analysis of model performance across varying data regimes. (2) **Data Processing**: We adopt a semi-supervised anomaly detection setting, where models are trained solely on normal instances. Specifically, we apply a stratified split to the normal data, using 50% for training and holding out the rest for testing. The test set includes both normal and anomalous samples. All features are standardized using a `StandardScaler` (Pedregosa et al., 2011) fitted on the training data (see Figure 16 in Appendix D.3 for an ablation study on the effect of feature normalization). This protocol is consistent with common practices in anomaly detection (e.g., (Zong et al., 2018; Bergman and Hoshen, 2020; Shenkar and Wolf, 2022; Yin et al., 2024)) and ensures a fair evaluation.

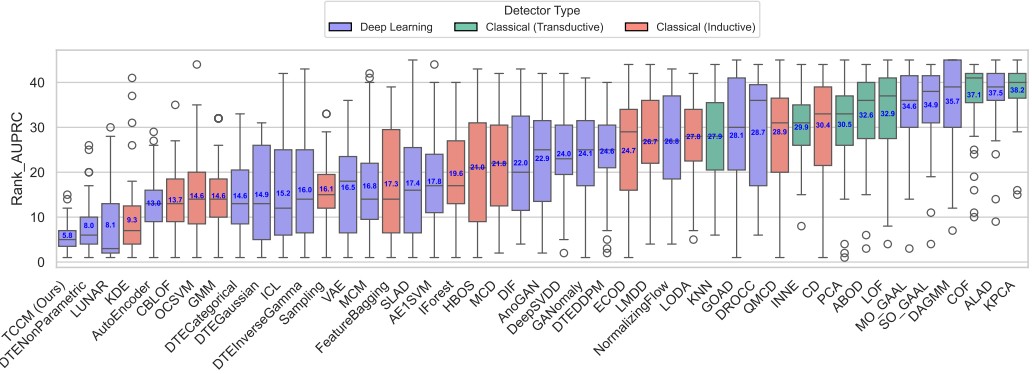

(a) AUPRC ranking distribution across 47 datasets for 45 anomaly detectors.

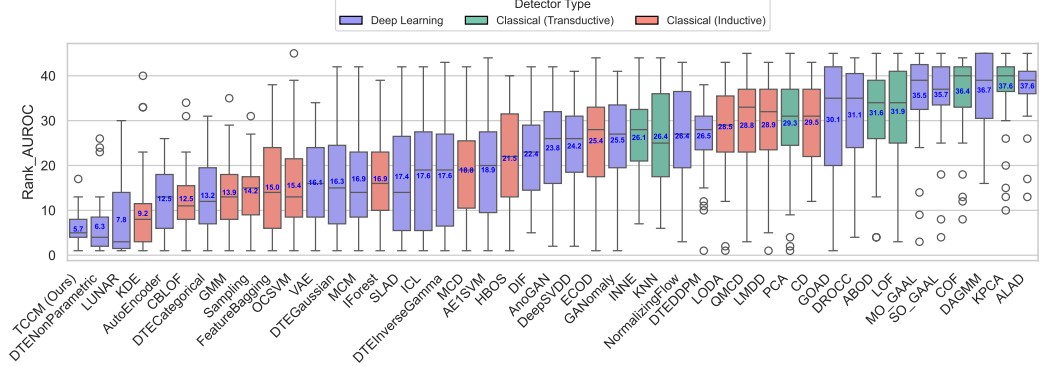

(b) AUROC ranking distribution across 47 datasets for 45 anomaly detectors..

Figure 2: Box plots of detector rankings based on AUPRC and AUROC scores across 47 datasets. Medians are marked by horizontal lines; means are shown as numbers.

**Baselines and Evaluation Metrics.** (1) **Baselines**: We evaluate our method against 44 baselines, including 21 classical (shallow) and 23 deep anomaly detection algorithms. Detailed descriptions of these baselines are provided in Appendix B.2. In particular, we offer a critical review of each deep method, highlighting their limitations in comparison to our approach in Appendix A.1. (2) **Evaluation metrics**: We adopt two standard metrics—Area Under the Receiver Operating Characteristic curve (AUROC) and Area Under the Precision-Recall Curve (AUPRC)—with higher values indicating better performance (see Appendix B.4 for more information).

**Configurations.** The details of architectures and hyperparameters will be postponed to Appendix B.3, while we highlight some of the main characteristics of our model here: the vector field $f_\theta(\boldsymbol{x}, t)$ is parameterized by a 3-layer multilayer perceptron (MLP), where each hidden layer contains 256 units followed by ReLU activations. To incorporate time information, we use a fixed sinusoidal embedding of the scalar time input $t \in [0, 1]$, following the positional encoding scheme used in transformer models (Vaswani et al., 2017). The time embedding is concatenated with the input vector $\boldsymbol{x}$, and the combined representation is passed through the MLP to produce the predicted velocity field.

## 5.2 Results Analysis

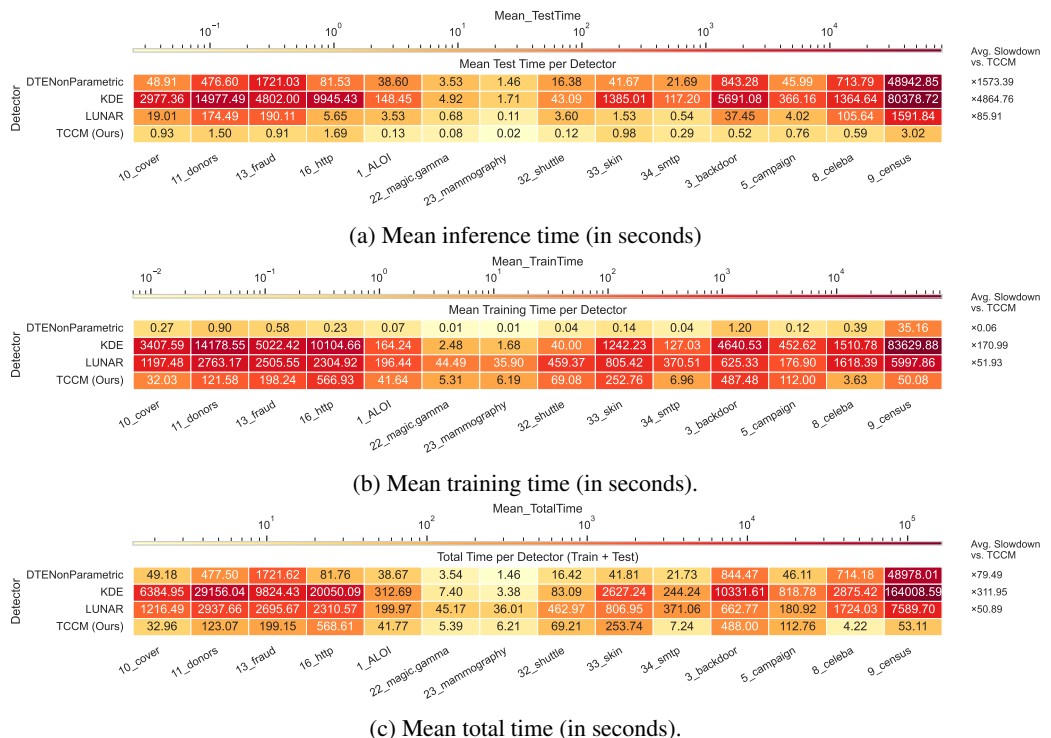

(a) Mean inference time (in seconds)

(b) Mean training time (in seconds).

(c) Mean total time (in seconds).

Figure 3: Mean run time (in seconds) across large-scale datasets for TCCM and other top-performing baselines in detection accuracy.

**(1) Effectiveness.** Figures 2a and 2b present the aggregated results based on AUPRC and AUROC scores, respectively. Due to the large scale of our experiments—covering 45 anomaly detectors across 47 datasets with 5 different random seeds, resulting in a total of 10,575 runs—it is impractical to include all individual results in the main paper. We thus report the complete results in Tables 6–13 in Appendix D. Particularly, we evaluate each method by reporting the distribution of its rankings across the 47 datasets. Rankings are computed based on the average AUPRC (respectively, AUROC) across the 5 seeds. As shown in Figures 2a and 2b, our method, TCCM, achieves the best overall performance in terms of both AUPRC (with an average rank of 5.8) and AUROC (with an average rank of 5.7). While DTE-NonParametric (second in both AUPRC and AUROC), LUNAR (third in both), and KDE (fourth in both) also demonstrate strong detection accuracy, we will show later that these methods suffer from poor scalability in training and/or inference, making them less favorable for large-scale deployment compared to TCCM. A more detailed analysis is deferred to Appendix D.1 due to space constraint. We further perform statistical significance testing using the Friedman (Friedman, 1937) and Nemenyi tests (Nemenyi, 1963) to assess whether the observed ranking differences are statistically meaningful; detailed results are provided in Appendix D.5.

**(2) Scalability.** TCCM achieves considerably faster inference than most deep learning baselines, particularly on large-scale and high-dimensional datasets. As shown in Figure 3a, it significantly outpaces other *high-accuracy* methods in inference speed—being 1,573.39× faster than DTE-NonParametric, 4,864.76× faster than KDE, and 85.91× faster than LUNAR on average. On the largest dataset, *census* (299,285 samples × 500 dimensions), TCCM takes just 1.50 seconds, while DTE-NonParametric requires 48,942 seconds. These results highlight TCCM's scalability and suitability for real-time anomaly detection in big-data environments. Beyond inference efficiency, we further provide an analysis on training time and total runtime to evaluate the end-to-end deployability of TCCM. As shown in Figure 3b, TCCM maintains competitive training efficiency—while DTE-NonParametric trains faster (requiring only 0.06× the training time of TCCM), KDE and LUNAR are **170.99×** and **51.93×** slower, respectively. When considering the overall cost, TCCM exhibits the lowest total runtime among all top-performing baselines (Figure 3c), outperforming DTE-NonParametric, KDE,

and LUNAR by **79.49**×, **311.95**×, and **50.89**×, respectively. This balanced efficiency across both training and inference phases underscores TCCM's suitability for real-world, large-scale anomaly detection deployments, where both accuracy and runtime constraints are critical. To further contextualize the trade-off between speed and performance, we include scatter plots comparing average inference time versus average AUROC (or AUPRC) across all 44 baselines (see Figures 7 and 8 in Appendix D.2.1). The results demonstrate that TCCM achieves one of the best balances between detection accuracy and inference efficiency among all evaluated methods. A detailed breakdown and additional comparisons across all 45 anomaly detection methods are provided in Appendix D.2.

**(3) Explainability.** TCCM is designed for tabular data and inherently supports self-explanation by producing feature-wise importance scores derived from its learned residual velocity field, which characterizes deviation from expected normal contraction behavior. To provide a more intuitive illustration of this property, we apply TCCM to image data (MNIST (Deng, 2012)), treating each pixel as a feature in a flattened tabular vector. We use digit 1 as the normal class and digit 7 as the anomaly (achieving an AUROC of 0.76). As shown in Figure 4, the model highlights the additional horizontal stroke that distinguishes 7 from 1, demonstrating that the learned importance scores align well with human-interpretable cues. Importantly, these explanations are *intrinsic* to TCCM rather than post hoc approximations such as SHAP (Lundberg and Lee, 2017) or LIME (Ribeiro et al., 2016): the residual vector itself encodes per-feature contributions to the anomaly score, faithfully reflecting the model's internal reasoning. For this, we provide a controlled synthetic study in Appendix D.4.2, which quantitatively validates the accuracy of these feature-level attributions and further substantiates TCCM's intrinsic interpretability. This makes the explanations directly actionable in practice, enabling domain experts in areas such as fraud detection, healthcare, or industrial monitoring to identify not only *which* instances are anomalous but also *why*.

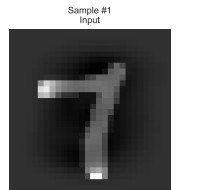 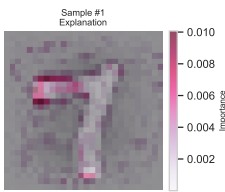 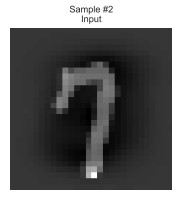 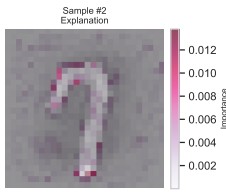

Figure 4: Illustrative examples of anomalous images and their explanations, where digit '1' is treated as the normal class and digit '7' as the anomaly. The highlighted regions correspond to structural differences between '7' and '1', which the model identifies as key contributors to the anomaly score.

**(4) Ablation Studies and Sensitivity Analysis.** We study how major design and data factors influence TCCM (see Appendix D.3): (1) Time embedding and inference time (Figures 12–13): results are nearly unchanged across choices, showing strong robustness; (2) Noise injection (Figure 14): deterministic training consistently performs better; (3) Training contamination (Figure 15): higher anomaly ratios reduce accuracy, underscoring the need for clean supervision; (4) Feature normalization (Figure 16): z-score normalization is generally beneficial and improves robustness; (5) Time-interpolated inputs (Figure 17): interpolation offers no gain and may add noise; (6) Comparison with Autoencoder +Time Embedding: confirms that TCCM learns a time-conditioned velocity field rather than a reconstruction mapping. Overall, TCCM remains stable and efficient across all variations.

# 6 Conclusion

We presented *Time-Conditioned Contraction Matching* (TCCM), a novel method for semi-supervised anomaly detection in tabular data. By learning a time-conditioned contraction field grounded in flow matching, TCCM avoids adversarial training, trajectory simulation, and density modeling—offering a lightweight yet expressive alternative to existing generative-based anomaly detection methods. On the ADBench benchmark, TCCM outperforms 44 classical and deep baselines in both AUROC and AUPRC, while achieving orders-of-magnitude faster inference than the strongest diffusion-based competitor (namely DTE-NonParametric). It also provides feature-level interpretability via its learned vector field. Theoretical analysis confirms the Lipschitz continuity and discriminative power of its scoring function. Together, these results position TCCM as an highly effective, scalable, interpretable, and robust solution for large-scale anomaly detection. Future directions include extending to other data modalities. The limitations and broader impacts of our work are further discussed in Appendix D.6.

## Acknowledgment

We thank all anonymous reviewers for their time and efforts in reviewing this paper and their constructive comments to improve this paper. **Qi Huang, Niki van Stein**: This publication is partly sponsored by the XAIPre project (with project number 19455) of the research program Smart Industry 2020 which is (partly) financed by the Dutch Research Council (NWO).

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

## Appendix

## Table of Contents

## A  Related Work

### A.1  Anomaly Detection Methods

**Unsupervised VS Semi-Supervised.** Labeled anomalies are often scarce in practice, as they typically correspond to rare and costly events—such as aerospace system crashes (Marzat et al., 2012; Nanduri and Sherry, 2016), faults in industrial systems (Li and van Leeuwen, 2022; Li et al., 2024b), financial fraud (Hilal et al., 2022; Khodabandehlou and Golpayegani, 2024), or critical health incidents (Šabić et al., 2021; Fernando et al., 2021). As a result, anomaly detection is commonly framed as an unsupervised or semi-supervised task. Compared to supervised settings, most unsupervised approaches face a fundamental challenge: the absence of labeled input-output pairs renders standard regression or classification techniques inapplicable (Liu et al., 2022). Anomaly detection inherits this difficulty, requiring alternative methods to detect abnormality without explicit supervision. To address this, many recent deep learning-based anomaly detection approaches adopt a *semi-supervised* setting (or one-class classification), where the training data consists exclusively of normal instances, which are relatively easier to collect. The underlying principle is that a model trained solely on normal data should learn only normal patterns. When evaluated on test data containing both normal and anomalous instances, those deviating from the learned normal patterns are expected to exhibit larger fitting errors (e.g., reconstruction errors (An and Cho, 2015; Zong et al., 2018), prediction residuals (Lai et al., 2021; Hundman et al., 2018), or likelihood-based scores (Zenati et al., 2018; Ren et al., 2019)), leading to higher anomaly scores. Although many works refer to this setup as *unsupervised anomaly detection*, we use the term *semi-supervised anomaly detection* for conceptual clarity and rigor. This setting is widely adopted in recent deep learning-based anomaly detection

studies such as DeepSVDD (Ruff et al., 2018), VAE (An and Cho, 2015), GANomaly (Akcay et al., 2018), ALAD (Zenati et al., 2018).

**Shallow vs. Deep.** *Shallow methods* refer to classical anomaly detection approaches that do not rely on neural networks. Representative examples include One-Class SVM (OCSVM) (Schölkopf et al., 1999), Support Vector Data Description (SVDD) (Tax and Duin, 2004), Kernel Density Estimation (KDE) (Latecki et al., 2007), Isolation Forest (IForest) (Liu et al., 2008), and distance-based methods such as k-Nearest Neighbors (KNN) (Angiulli and Pizzuti, 2002), Local Outlier Factor (LOF) (Breunig et al., 2000), and Connectivity-based Outlier Factor (COF) (Tang et al., 2002). Kernel-based methods often suffer from poor scalability, as they require constructing large kernel matrices and storing support vectors during inference (Ruff et al., 2018). IForest, while efficient in low dimensions, tends to degrade in high-dimensional settings due to its reliance on random projections and axis-aligned splits, which may fail to capture meaningful structure in complex data. Similarly, KNN-based methods are sensitive to the curse of dimensionality, where distance metrics lose discriminative power, and their computational complexity scales poorly with dataset size. Overall, these limitations have motivated the development of deep learning-based anomaly detection methods, which can better handle large-scale and high-dimensional data by learning expressive representations.

*Deep learning-based anomaly detection methods* can be broadly categorized into two groups Ruff et al. (2018): (1) *two-stage approaches*, which utilize neural networks to learn feature representations, followed by classical anomaly detection algorithms applied to these representations; and (2) *end-to-end trained approaches*, which integrate representation learning and anomaly detection objectives within a unified deep learning framework. This paper focuses on the latter category, which has received increasing attention due to its potential for end-to-end optimization and adaptability to complex data modalities. We structure our review around four major lines of *end-to-end trained* approaches: (i) one-class classification paradigms (e.g., Deep SVDD), (ii) generative based models (e.g., GANs-based methods, VAEs-based methods, diffusion models-based methods),(iii) reconstruction-based methods (e.g., autoencoders), and (iv) emerging variants including self-supervised based methods, graph-based methods, and hybrid models. For each group, we review some representative methods and highlight their respective limitations in the following.

### A.1.1   One-Class Classification Methods

This line of work draws inspiration from classical one-class classification, such as Support Vector Data Description (SVDD) (Tax and Duin, 2004). For instance, Deep SVDD (Ruff et al., 2018) and its variants train a neural network to map normal data into a compact region in latent space, typically minimizing the distance to a center point or hypersphere. Anomalies are then identified based on their deviation from this learned region. Other examples include AE-1SVM (Nguyen and Vien, 2019), Deep Robust One-Class Classification (DROCC) (Goyal et al., 2020), OneFlow (Maziarka et al., 2021), and they will be reviewed in more details as follows.

**DeepSVDD** (Ruff et al., 2018). They train a neural network to learn a hypersphere of minimum volume to enclose the embeddings of normal instances, while the embeddings of abnormal instances tend to lie outside the hypersphere. However, it may suffer from the problem of hypersphere collapse, where the hypersphere collapses to a single point. To alleviate this collapse problem, they authors propose that: 1) all-zero-weights solution cannot be used for the center of hypersphere, 2) only unbounded activations should be used, and 3) bias terms need to be omitted, which may lead to sub-optimal feature representation as bias terms are mandatory to shift activation values in neural networks. In contrast, our method TCCM deliberately learns to flow to such a "collapsed" single point, without suffering from any model collapse problem. Moreover, we operate on the input space without learning an explicit latent space, maintaining the explainability of anomaly scores.

**AE-1SVM** (Nguyen and Vien, 2019). They combine autoencoder (for representation learning) with OCSVM (for anomaly detection) in an end-to-end training manner. Moreover, they extend gradient-based attribution methods to analyze the contribution of input features on anomaly scores. Particularly, to solve the scalability issues with kernel machines (which has a complexity of $O(N^2)$ with $N$ the number of samples) in the original SVM, they employ random Fourier features (Rahimi and Recht, 2007) to approximate the kernel function. They also point out that "the biggest issue of OCSVM is their (poor) capability to handle large and high-dimensional datatsets due to optimization complexity."

**DROCC** (Goyal et al., 2020). They assume that instances from the normal class lie on a locally linear low dimensional manifold, which is well-sampled in the training data. A test instance is considered as anomalous if it is outside the union of small $l_2$ balls around the typical normal instances. Particularly, they convert the anomaly detection problem from an unsupervised learning setting to supervised setting as follows: they first generate synthetic anomalous instances (based on the above assumption) to add into the training set, and then train a supervised classifier to distinguish the embeddings of synthetical anomalous instances and those of typical normal instances. This method is robust to representation collapse as mapping all instances into a single point will lead to poor classification results. This method is applicable to tabular data, image, time series, audio, etc. DROCC's optimization is formulated as a saddle point problem, which is solved using standard gradient descent-ascent algorithm (which may be unstable like in adversarial training).

**OneFlow** (Maziarka et al., 2021). They introduce a one-class anomaly detection method based on NICE flows (Dinh et al., 2014), a type of normalizing flow with a volume-preserving transformation (i.e., constant Jacobian determinant). The core idea is to learn an invertible transformation that maps nominal data to a latent space, and then fit a minimal-volume hypersphere that encloses a fixed proportion of the mapped points. Anomalies are detected based on their distance from the hypersphere center. This approach avoids full density estimation and focuses on directly modeling the support of normal data. A Bernstein polynomial estimator is used to ensure smooth quantile estimation, and the training loss only depends on boundary-adjacent points—resembling support vector behavior. While effective in modeling low-density support, OneFlow has several limitations compared to flow matching methods: the use of volume-preserving flows like NICE limits the expressivity of the learned transformation and the capacity to model complex data distributions.

### A.1.2 Generative based Approaches

Generative models, particularly those based on Generative Adversarial Networks (GANs) (Goodfellow et al., 2014), have been widely adapted for anomaly detection by learning to approximate the distribution of normal data (Schlegl et al., 2017; Akcay et al., 2018). These methods typically detect anomalies by measuring reconstruction error, discriminator scores, or deviations in latent space, leveraging GANs' capacity to generate realistic high-dimensional samples. Variational Autoencoders (VAEs) (Kingma et al., 2013) offer an alternative probabilistic formulation, modeling normality via reconstruction likelihood and latent priors. More recently, denoising diffusion probabilistic models (DDPMs) (Ho et al., 2020) have been applied to anomaly detection, often by reconstructing inputs through reverse diffusion trajectories and using the reconstruction residual as an anomaly score (Livernoche et al., 2023). However, such models often suffer from high inference latency due to the iterative nature of reverse sampling. Most notably, a newer class of generative models known as *flow matching* (Lipman et al., 2022; Albergo and Vanden-Eijnden, 2023; Liu et al., 2022) has recently emerged as a powerful and stable alternative to diffusion models. Despite its strong theoretical foundation and demonstrated success in generative modeling, flow matching has not yet been systematically explored for anomaly detection, especially in the tabular setting. This leaves a promising research gap, motivating our development of a flow matching-inspired framework tailored specifically to the needs of semi-supervised anomaly detection.

**AnoGAN** (Schlegl et al., 2017). This is the first work to employ GANs for anomaly detection. Specifically, they first utilize only normal instances to train a GAN model (including a generator and a discriminator). At test time, given a query image $x$, they iteratively search for a latent embedding $z$ such that the generated image $G(z)$ closely resembles $x$ and lies on the learned data manifold. This is done by minimizing a joint loss consisting of a residual loss and a discrimination loss through $\Gamma$ steps of backpropagation. However, such per-instance optimization leads to a high inference cost, making the method less practical for real-time applications. To quantify the abnormality of a test sample, AnoGAN defines an *anomaly score* as a weighted sum of the residual loss and the discrimination loss obtained after $\Gamma$ optimization steps in the latent space. Specifically, the residual loss captures the pixel-wise difference between the input image $x$ and the generated image $G(z_\Gamma)$, while the discrimination loss measures how well the generated image fits on the learned data manifold. A high anomaly score indicates poor reconstruction and/or low likelihood under the discriminator, both of which suggest the input is dissimilar to the normal training distribution. In addition to scalar scoring, AnoGAN also provides visual explanation by computing a residual image $x_R = |x - G(z_\Gamma)|$, highlighting regions that contribute most to the anomaly.

**ALAD** (Zenati et al., 2018). They propose Adversarially Learned Anomaly Detection (ALAD), a reconstruction-based GAN framework tailored for efficient unsupervised anomaly detection. Unlike earlier methods such as AnoGAN that require costly per-instance optimization during inference, ALAD incorporates an encoder network that directly maps inputs to the latent space, enabling fast, one-pass anomaly scoring. The model builds upon the BiGAN framework (Donahue et al., 2016) by jointly training a generator, discriminator, and encoder, with additional cycle-consistency regularizations to enforce accurate reconstruction. Specifically, it introduces two auxiliary discriminators to enforce consistency in both data and latent spaces, improving the quality of reconstructions. To detect anomalies, ALAD defines an anomaly score based on the feature difference between the input and its reconstruction, extracted from an intermediate layer of the discriminator operating on sample pairs. This feature-level distance serves as a more robust indicator than raw pixel differences, especially for high-dimensional data. While ALAD achieves fast inference and strong performance, it relies on adversarially balancing multiple networks and loss components during training, which may introduce stability challenges and increased complexity compared to simpler reconstruction-based approaches.

**GANomaly** (Akcay et al., 2018). It is a representative GAN-based anomaly detection framework that enhances vanilla GANs by incorporating an explicit encoder–decoder structure. Instead of relying on latent space search at test time, GANomaly introduces a dual-encoder architecture to enable efficient, feedforward inference. During training, only normal data are used to train a generator network composed of an encoder and decoder ($G = G_D \circ G_E$), which learns to reconstruct the input images. In parallel, a second encoder $E$ is trained to map the reconstructed images back into the latent space. The key assumption is that, for normal samples, the original and reconstructed latent vectors ($z = G_E(x)$ and $\hat{z} = E(G(x))$) should be close, whereas for anomalous inputs, their discrepancy will be larger. The training objective combines three losses: (i) a contextual loss that encourages pixel-wise similarity between input and reconstruction, (ii) a feature matching loss computed from intermediate discriminator activations to stabilize adversarial learning, and (iii) a latent encoder loss that penalizes the difference between $z$ and $\hat{z}$. At test time, the anomaly score is defined based solely on the latent distance $\mathcal{A}(x) = |G_E(x) - E(G(x))|_1$, allowing for fast, one-pass anomaly detection without the iterative inference used in methods like AnoGAN. While GANomaly achieves a good trade-off between accuracy and efficiency, it has two notable limitations. First, its anomaly score depends entirely on the latent discrepancy, which may be vulnerable to representation collapse or ambiguous reconstructions. Second, the three-part loss requires balancing multiple objectives, and tuning the corresponding weights without labeled data can be challenging in unsupervised settings.

**SO-GAAL** and **MO-GAAL** (Liu et al., 2019) . They propose two GAN-based outlier detection methods that approach anomaly detection as an adversarial learning process. In *Single-Objective Generative Adversarial Active Learning* (SO-GAAL), a generator is trained to synthesize informative potential outliers, while a discriminator attempts to distinguish these from the real data. This adversarial interplay gradually improves the quality of the generated outliers, enabling the discriminator to carve a tighter decision boundary around normal data. However, SO-GAAL may suffer from mode collapse and performance degradation once the generator overfits the data manifold. To address this, the authors further propose *Multiple-Objective GAAL* (MO-GAAL), which introduces multiple sub-generators, each responsible for generating outliers relative to a specific subset of the data. By doing so, MO-GAAL builds a more diverse and comprehensive reference distribution, allowing the discriminator to maintain stable and accurate detection even when dealing with multi-modal or high-dimensional data. Both approaches ultimately compute an outlier score based on the discriminator's output, with higher scores indicating greater deviation from the learned normal distribution.

**VAE** (Kingma et al., 2013). They introduce a principled probabilistic framework for learning latent representations via a variational autoencoder. For anomaly detection, the VAE is trained solely on normal instances, learning to encode data into a low-dimensional latent space and reconstruct inputs through a decoder (An and Cho, 2015; Zhou et al., 2020). During training, the model jointly minimizes a reconstruction loss and a KL divergence regularizer, which encourages the latent codes to follow a standard Gaussian prior. At test time, given a new input $x$, the model produces a reconstructed sample $\hat{x}$ by encoding and decoding it through the learned latent space. The anomaly score is then computed based on the reconstruction error, typically using an $\ell_2$ distance between $x$ and $\hat{x}$. Since the model is trained to reconstruct normal patterns well, higher reconstruction errors suggest that the input deviates from the normal data distribution. Compared to GAN-based approaches like AnoGAN, VAE offers faster inference as no iterative optimization is required at test time, making it more practical for real-time applications. However, the generative quality of

VAEs is often inferior to GANs, especially when dealing with complex or high-dimensional data. In some cases, even anomalous inputs can be reconstructed with low error, leading to false negatives. Furthermore, the balance between reconstruction fidelity and latent regularization (e.g., via the KL term or the $\beta$ coefficient in $\beta$-VAE) is sensitive, and improper tuning may lead to posterior collapse or over-regularization, which degrades anomaly detection performance.

**DTE (Livernoche et al., 2023).** They propose a novel use of diffusion processes for anomaly detection by predicting the noise level—or diffusion timestep—associated with an input sample. The core intuition is that normal instances lie close to the data manifold and hence resemble samples with low diffusion noise, while anomalies lie further away and mimic samples diffused with stronger noise. During training, DTE simulates noisy samples via a predefined forward diffusion process (e.g., variance-preserving), and learns a neural network to regress or classify the corresponding diffusion time, using only normal data. At test time, inputs that are harder to explain as low-noise samples receive higher predicted diffusion times and are flagged as anomalies. Notably, this approach avoids learning the full reverse process as in DDPMs and instead frames anomaly detection as a diffusion time estimation task. However, DTE—particularly its non-parametric variant—faces major scalability bottlenecks. DTE-NonParametric estimates the diffusion time of a test sample by computing a posterior over all training points, using distance-based kernel density approximations. This requires comparing each test input against a large training set, making inference prohibitively slow for high-volume or high-dimensional data. Moreover, the diffusion process used to synthesize training data adds to the overall preprocessing overhead, limiting DTE's suitability for real-time or resource-constrained settings. Despite its strong detection performance, these computational costs hinder its broad applicability in large-scale anomaly detection pipelines.

### A.1.3 Reconstruction-based Methods

Reconstruction-based methods constitute a major paradigm in anomaly detection. The central idea is that models trained to accurately reconstruct normal data will fail to do so for anomalous inputs, which typically deviate from the learned data manifold. The reconstruction error—measured in input space or latent space—is then used as an anomaly score. Among the most widely used reconstruction-based models are Autoencoders (AEs) and their variants (Sakurada and Yairi, 2014; Zhou and Paffenroth, 2017). These models learn compact representations through a bottleneck architecture and are trained to minimize the reconstruction loss on normal data. Anomalies are expected to produce larger reconstruction errors due to their poor alignment with the learned representation space. Variational Autoencoders (VAEs) (An and Cho, 2015) further extend this idea by introducing a probabilistic latent space, allowing uncertainty-aware reconstructions.

Beyond classical AEs, many generative models also incorporate reconstruction-based objectives. For example, GAN-based methods such as GANomaly (Akcay et al., 2018) and ALAD (Zenati et al., 2018) utilize an encoder–decoder–discriminator pipeline, where anomaly scores are derived from reconstruction fidelity or latent-space consistency. Similarly, recent diffusion-based methods (Livernoche et al., 2023) detect anomalies by reconstructing inputs through reverse diffusion trajectories. As other methods have been (or will be) reviewed in other parts, we will review a representative reconstruction-based approach, DAGMM (Zong et al., 2018), in the following.

**DAGMM (Zong et al., 2018).** They propose the Deep Autoencoding Gaussian Mixture Model (DAGMM), a unified deep framework for unsupervised anomaly detection that jointly learns low-dimensional representations and density estimation. Specifically, DAGMM integrates two key components: a compression network, which is a deep autoencoder producing both latent features and reconstruction error metrics; and an estimation network, which models a Gaussian Mixture Model (GMM) over the concatenated features to estimate sample energy (i.e., negative log-likelihood). To avoid traditional two-step training, DAGMM jointly optimizes the autoencoder and the GMM via a shared objective that includes reconstruction loss, sample energy, and a regularization term to prevent degenerate covariance matrices. During inference, an anomaly score is computed as the energy of a test sample under the learned GMM, with higher energy indicating greater anomaly likelihood. Unlike conventional approaches that rely only on reconstruction error or pre-trained representations, DAGMM is trained end-to-end, enabling the autoencoder to adapt its compression strategy in favor of improved density estimation. This results in enhanced ability to detect subtle or "lurking" anomalies that might not have high reconstruction errors but reside in low-density regions. One limitation of DAGMM is that its network configuration—such as the number of mixture components in the GMM, and the architectures of the compression and estimation networks—needs to be selected in a

data-dependent manner. These choices often require dataset-specific tuning, which may affect the model's ease of deployment and generalizability across tasks.

**Limitations of Autoencoders.** Autoencoders are typically trained to reconstruct the input data while enforcing an intermediate low-dimensional representation, which acts as an information bottleneck to encourage the neural network to extract salient features from the training data. In the context of semi-supervised anomaly detection, this promotes learning the underlying factors of variation shared among normal instances. However, since autoencoders do not directly optimize for anomaly detection, their effectiveness heavily depends on how well the latent space captures relevant structure in the data. In particular, the choice of latent dimensionality becomes critical: if it is too high, the model may simply memorize the input; if it is too low, essential information may be lost. This hyperparameter is often data-dependent and difficult to tune due to the unsupervised nature of the task and the challenges in estimating the intrinsic dimensionality of the data (Bengio et al., 2013).

### A.1.4 Self-Supervised based Methods and Other Miscellaneous Methods

Recent studies explore alternative deep paradigms for anomaly detection, including self-supervised learning-based methods such as GOAD (Bergman and Hoshen, 2020), ICL (Shenkar and Wolf, 2022), SLAD (Xu et al., 2023b), and MCM (Yin et al., 2024), graph neural network-based method such as LUNAR (Goodge et al., 2022), and hybrid models such as DIF (Xu et al., 2023a) that combine deep feature learning with traditional detectors. They will be reviewed in more detail as follows.

**GOAD** (Bergman and Hoshen, 2020). They introduce a classification-based approach for anomaly detection that unifies one-class and transformation-based paradigms. It first applies a set of $M$ geometric or affine transformations to each normal training instance and learns a shared feature extractor $f$ that maps transformed inputs to a representation space. Each transformed variant is encouraged to cluster around a distinct center using a triplet-style loss, promoting intra-class compactness and inter-class separation. At inference, GOAD computes the transformation prediction likelihood for each transformed test instance and aggregates these into a final anomaly score: samples that are poorly aligned with any learned transformation subspace are considered anomalous. Unlike earlier transformation-based methods (e.g., GEOM (Golan and El-Yaniv, 2018)) that suffer from unreliable extrapolation to unseen anomalies, GOAD regularizes prediction confidence on out-of-distribution regions and generalizes to non-image data via learnable affine transformations. However, GOAD's effectiveness heavily depends on the quality and diversity of the transformations used, and its performance is sensitive to the choice of the number of transformations $M$, which must be manually specified and tuned per dataset—potentially limiting its practicability across domains.

**ICL** (Shenkar and Wolf, 2022). They introduce a novel approach to anomaly detection in tabular data by leveraging contrastive learning on internal feature partitions. Unlike methods that depend on external transformations or assume data structure (e.g., spatial correlations in images), ICL operates under the premise that dependencies among feature subsets are class-specific. Specifically, given a single-class training set, the method slides a window of size $k$ over each input vector $\boldsymbol{x} \in \mathbb{R}^d$ to generate a set of $m = d - k + 1$ paired sub-vectors $(\boldsymbol{a}_j, \boldsymbol{b}_j)$, where $\boldsymbol{a}_j$ is a segment of $k$ consecutive features and $\boldsymbol{b}_j$ is its complement. Two neural networks $G$ and $F$ embed $\boldsymbol{a}_j$ and $\boldsymbol{b}_j$, respectively, into a shared latent space $\mathbb{R}^u$, trained to maximize mutual information between matching pairs via a noise contrastive loss. At test time, the anomaly score for a sample $\boldsymbol{x}$ is defined as the sum of contrastive losses across all $j$, directly measuring how class-consistent its internal structure is under the learned embeddings. Importantly, the method is interpretable by design: the local loss for each feature subset allows pinpointing which attributes contribute most to an anomaly. While hyperparameters such as the window size $k$ and latent dimension $u$ must be set (in a data-dependent way), empirical evidence shows that performance is robust across a wide range of values, and no dataset-specific tuning is required.

**SLAD** (Xu et al., 2023b). They propose Scale Learning-based Anomaly Detection (SLAD), a self-supervised framework for tabular anomaly detection that avoids reliance on reconstruction losses. SLAD introduces a novel supervision signal—scale—which quantifies the relationship between subspace dimensionality and representation complexity. Specifically, it samples subspaces of each input, maps them to fixed-length representations, and defines scale labels to supervise the learning of a ranking function via distribution alignment. During inference, anomaly scores are computed by measuring the divergence between predicted and target scale distributions, based on the assumption that normal samples produce more consistent and predictable scales. Despite its conceptual novelty

and strong empirical performance, SLAD presents several limitations. First, the generation of scale labels and the assumption that anomalies inherently exhibit scale inconsistencies may not hold uniformly across datasets or domains, particularly when anomalies are subtle or lie near the decision boundary. Second, SLAD's reliance on distribution alignment adds algorithmic complexity and hyperparameter sensitivity, which may impact robustness in practical deployment. Finally, while the model is self-supervised, its interpretability remains limited, as the learned scale concept is abstract and may not directly correspond to human-understandable explanations.

**MCM** (Yin et al., 2024). This is a novel self-supervised framework for anomaly detection in tabular data. Inspired by masked modeling techniques in NLP and vision (e.g., BERT and MAE), MCM learns to reconstruct randomly masked subsets of input features using only the unmasked portions. The core hypothesis is that normal instances exhibit strong internal feature correlations, which the model can learn to reconstruct effectively—while anomalies violate such correlations, leading to higher reconstruction error. To enhance robustness, MCM introduces a *learnable masking strategy* that dynamically generates multiple soft masks per instance. These masks are trained end-to-end via a mask generator network. A *diversity loss* is used to ensure that different masks capture complementary correlations among features, thereby improving the model's discriminative power. The final anomaly score is computed as the average reconstruction error across all masked versions. Compared to prior methods such as contrastive learning (e.g., ICL), which often rely on engineered transformations, MCM provides a more data-driven and flexible mechanism to capture normality. Moreover, the ensemble of masks makes the method more expressive while maintaining a lightweight architecture based on an encoder-decoder MLP. MCM also offers interpretability through both per-mask and per-feature contributions, facilitating insight into which correlations are violated by a given anomaly.

**LUNAR** (Goodge et al., 2022). They propose a local outlier detection framework based on graph neural networks (GNNs) with learnable message aggregation. Instead of relying on raw feature vectors, LUNAR constructs a $k$-nearest neighbor graph from the training data and uses pairwise distances as edge features. A single-layer GNN is trained to distinguish normal points from synthetic negative samples by aggregating the distance vector from each node's neighborhood. This design enables the model to learn a parametric anomaly scoring function that adapts better than classical non-trainable local methods such as LOF or KNN. Negative samples are generated using a mix of uniform sampling and feature-space perturbation, which prevents the model from collapsing to trivial solutions. While LUNAR demonstrates strong empirical performance, its reliance on $k$-NN graph construction introduces scalability challenges in high-dimensional settings, where distance metrics become less meaningful. Moreover, because it avoids using raw features and instead encodes distance information through a learnable aggregation function, it can incur additional computational cost in preprocessing and training—especially on large-scale datasets with many neighbors per node. As the nearest neighbor graph must be recomputed for each new input distribution and all neighborhood distances are fed into the model, the method may be less suitable for real-time or streaming applications compared to embedding-based approaches.

**DIF** (Xu et al., 2023a). They propose the *Deep Isolation Forest* (DIF), a scalable anomaly detection method that extends isolation-based techniques by incorporating random deep representations. Instead of applying axis-aligned splits on raw features (as in iForest (Liu et al., 2008)), DIF first transforms input data into multiple representation spaces using randomly initialized, optimization-free neural networks. These transformations enable nonlinear partitioning in the original space via simple axis-parallel cuts in the projected space. To ensure efficiency, DIF introduces a computation-efficient ensemble mechanism (CERE) that allows all ensemble members to be computed simultaneously in a mini-batch. For scoring, DIF further proposes a deviation-enhanced anomaly scoring function (DEAS) that combines traditional path length with deviation from split thresholds to reflect local density and isolation difficulty. The authors show that DIF generalizes both iForest and Extended Isolation Forest (EIF) (Hariri et al., 2019), while preserving linear scalability and offering stronger expressive power. However, DIF relies heavily on the randomness and diversity of representations for performance, which may lead to instability without sufficient ensemble size or structure-aware initialization.

## A.2    Generative Models

Generative models span a broad spectrum of paradigms, including energy-based models (LeCun et al., 2006), variational autoencoders (VAEs) (An and Cho, 2015), generative adversarial networks

(GANs) (Goodfellow et al., 2014), normalizing flows (Papamakarios et al., 2021), autoregressive models (e.g., Transformers (Vaswani et al., 2017)), diffusion models (Ho et al., 2020), and the more recent flow matching framework (Lipman et al., 2022; Albergo and Vanden-Eijnden, 2023; Liu et al., 2022). While each of these approaches offers unique modeling capabilities, we focus our discussion on flow matching, as it is most directly relevant to the methodology developed in this paper.

**GANs and Diffusion Models.** Generative adversarial networks (GANs) (Goodfellow et al., 2014) have long been the de facto choice for high-fidelity image generation, but diffusion models have recently surpassed them in terms of mode coverage and conditional flexibility. Despite their effectiveness, diffusion models (Yang et al., 2023) are notoriously computationally intensive, often requiring hundreds of iterative denoising steps to generate a single sample. This has spurred efforts to accelerate training and inference (Dockhorn et al., 2022; Jolicoeur-Martineau et al., 2021); however, many of these approaches still suffer from slow convergence and rely on carefully constructed probability paths, which limit scalability on high-dimensional or large-scale datasets (Dao et al., 2023). Particularly, there is a recent survey paper on diffusion models for tabular data (Li et al., 2025a), which systematically reviewed existing diffusion models for tabular data modeling (including anomaly detection).

**Normalizing Flows and Continuous Normalizing Flows.** Continuous normalizing flows (CNFs) (Chen et al., 2018; Grathwohl et al., 2018) model invertible transformations between distributions using neural ordinary differential equations (ODEs). While theoretically elegant, training such models is computationally intensive, as it involves solving ODEs during each forward and backward pass, making scalability a major bottleneck. To overcome this, recent works on *flow matching* (Albergo and Vanden-Eijnden, 2023; Lipman et al., 2022; Liu et al., 2022) propose simulation-free alternatives that avoid explicit trajectory integration. Inspired by ideas from score-based diffusion models, these methods enable more efficient training of CNFs by directly learning velocity fields without solving full ODE systems.

**Flow Matching.** Flow matching has emerged as a promising alternative that inherits many of the desirable properties of diffusion models—such as robustness and expressivity—while avoiding their main drawbacks. Rather than relying on stochastic differential equations (SDEs), flow matching learns an ordinary differential equation (ODE) that deterministically maps samples from a source distribution $\rho_0$ ($\rho_{\text{source}}$) to a target distribution $\rho_1$ ($\rho_{\text{target}}$). This shift from stochastic to deterministic dynamics leads to lower curvature in generative trajectories, which translates into improved stability, faster sampling, and easier optimization (Liu et al., 2022; Lee et al., 2023). The simplicity of its training framework also facilitates broader adoption in settings where computational efficiency is critical. Beyond efficiency, flow matching offers notable modeling flexibility and interpretability. It eliminates the need for a forward diffusion process, and training can be performed by directly matching vector fields between arbitrary distributions (CSAIL, 2024; Albergo and Vanden-Eijnden, 2023). This generality stands in contrast to denoising diffusion models, which often assume Gaussian base distributions and Gaussian interpolants. In addition, the interpolant formulation of flow matching allows evaluation of intermediate densities $\rho_t$ at arbitrary time points $t \in [0, 1]$, enabling empirical inspection of the learned velocity field throughout the trajectory (Albergo and Vanden-Eijnden, 2023). Although this capability may not always be required—e.g., in anomaly detection we often focus only on the terminal velocity at $t = 1$—it enhances interpretability. Lastly, because flow matching only requires samples from the source and target distributions (not their explicit densities), it is particularly well-suited to scenarios with implicit or intractable data distributions.

**Difference from Generative Models.** While the main objective of normalizing flows (Rezende and Mohamed, 2015) or flow matching (Lipman et al., 2022) is to transform a simple initial distribution into a complex, often multimodal, target distribution—either through a sequence of invertible mappings (in the case of normalizing flows) or via velocity fields (in flow matching)—our focus, by contrast, is on measuring the distance to the target distribution after applying the learned one step velocity field (at any given time). Importantly, this target distribution is a degenerate distribution, which stands in stark contrast to the typical emphasis in generative modeling on the validity and richness of the target distribution.

# B  Experiment setups

## B.1  Datasets

A summary of the datasets used in our study is provided in Table 1. We adopt 47 benchmark datasets from the well-established ADBENCH benchmark (Han et al., 2022), spanning diverse domains including sociology, finance, linguistics, physics, and healthcare. To enable a comprehensive evaluation of different anomaly detectors, including our proposed method, we categorize the datasets into four groups based on their scale and dimensionality: (a) *High-dimensional* datasets, with more than 50 features; (b) *Large-scale* datasets (but not high-dimensional), containing more than 10,000 instances and fewer than 50 features; (c) *Medium-scale* datasets (not high-dimensional), with 1,000 to 10,000 instances; and (d) *Small-scale* datasets (not high-dimensional), containing fewer than 1,000 instances. This categorization facilitates a nuanced analysis of model performance across varying data regimes.

## B.2  Baselines

Before introducing the baseline methods, we clarify an important distinction in anomaly detection paradigms: **inductive** vs. **transductive** approaches. Inductive methods learn a generalizable decision function from the training set and apply it directly to unseen test data. In contrast, transductive methods rely on the distribution of the test set during inference, often scoring anomalies relative to the entire evaluation batch. While inductive approaches are generally preferred in deployment scenarios where test data is unavailable during training, transductive methods are still commonly included in benchmarking for historical and comparative purposes.

Our proposed method TCCM is an inductive method, as it learns a model using training data and then computes anomaly scores with the unseen test data. Although comparing inductive and transductive methods directly may not always be ideal due to differing assumptions, we include both for completeness. We categorize the anomaly detection baselines into two main groups:

- 21 Classical Machine Learning-based Methods (Transductive and Inductive). (1) Transductive Methods: **ABOD** (Kriegel et al., 2008), **COF** Tang et al. (2002), **LOF** (Breunig et al., 2000), **PCA** (Shyu et al., 2003), **KPCA** (Hoffmann, 2007), **KNN** (Ramaswamy et al., 2000), **INNE** (Bandaragoda et al., 2018); and (2) Inductive Methods: **CBLOF**(He et al., 2003), **CD** (Cook, 1977), **ECOD** (Li et al., 2022) **FeatureBagging** (Lazarevic and Kumar, 2005), **GMM** (Agarwal, 2007), **HBOS** (Goldstein and Dengel, 2012), **IForest** (Liu et al., 2008), **KDE** (Latecki et al., 2007), **LMDD** (Arning et al., 1996), **LODA** (Pevný, 2016), **MCD** (Fauconnier and Haesbroeck, 2009), **OCSVM** (Schölkopf et al., 1999), **QMCD** (Fang and Ma, 2001), and **Sampling** (Sugiyama and Borgwardt, 2013).

- 23 Deep Learning-based Methods (All are inductive). **AutoEncoder** (Sakurada and Yairi, 2014; Aggarwal and Aggarwal, 2017b), **ALAD** (Zenati et al., 2018), **DIF** (Xu et al., 2023a), **DeepSVDD** (Ruff et al., 2018), **LUNAR** (Goodge et al., 2022), **MOGAAL** (Liu et al., 2019), **SOGAAL** (Liu et al., 2019), **VAE** (An and Cho, 2015), **AE-1SVM** (Nguyen and Vien, 2019), **AnoGAN** (Schlegl et al., 2017), **DAGMM** (Zong et al., 2018), **PlanarFlow** (Normalizing Flows) (Rezende and Mohamed, 2015), **SLAD** (Xu et al., 2023b), **MCM** (Yin et al., 2024), **ICL**(Shenkar and Wolf, 2022), **GOAD**(Bergman and Hoshen, 2020), **GANomaly**(Akcay et al., 2018), **DTE-Categorical** (Livernoche et al., 2023), **DTE-Gaussian** (Livernoche et al., 2023), **DTE-InverseGamma** (Livernoche et al., 2023), **DTE-NonParametric** (Livernoche et al., 2023), **DROCC** (Goyal et al., 2020), **Anomaly-DDPM** (Livernoche et al., 2023).

## B.3  Configurations

All experiments are independently performed five times with different random seeds (0, 1, 2, 3, and 4) on each dataset for all 44 baselines and our proposed TCCM with high reproducibility to ensure high robustness, and account for variability due to random initialization.

**Implementation Details of TCCM**[2]**.** The time-conditioned velocity field $f_\theta(\boldsymbol{x}, t)$ is parameterized by a 3-layer multilayer perceptron (MLP) with 256 hidden units per layer and ReLU activations. To

---

[2]Code available at: `https://github.com/ZhongLIFR/TCCM-NIPS`

Table 1: Summary of Datasets. To systematically evaluate the performance of various anomaly detectors, we categorize the datasets into four groups based on their data scale and dimensionality: (a) *high-dimensional* datasets, which contain more than 50 features; (b) *large-scale* datasets (but not high-dimensional), with more than 10,000 instances and fewer than 50 features; (c) *medium-scale* datasets (not high-dimensional), with between 1,000 and 10,000 samples; and (d) *small-scale* (not high-dimensional) datasets, consisting of fewer than 1,000 instances. This categorization allows for a nuanced analysis of model behavior under different data regimes.

| Dataset | # Samples | # Features | # Anomaly | % Anomaly | Domain | Category |
|---|---|---|---|---|---|---|
| census | 299285 | 500 | 18568 | 6.2 | Sociology | High-dimensional |
| backdoor | 95329 | 196 | 2329 | 2.44 | Network | High-dimensional |
| campaign | 41188 | 62 | 4640 | 11.27 | Finance | High-dimensional |
| mnist | 7603 | 100 | 700 | 9.21 | Image | High-dimensional |
| speech | 3686 | 400 | 61 | 1.65 | Linguistics | High-dimensional |
| optdigits | 5216 | 64 | 150 | 2.88 | Image | High-dimensional |
| SpamBase | 4207 | 57 | 1679 | 39.91 | Document | High-dimensional |
| musk | 3062 | 166 | 97 | 3.17 | Chemistry | High-dimensional |
| InternetAds | 1966 | 1555 | 368 | 18.72 | Image | High-dimensional |
| donors | 619326 | 10 | 36710 | 5.93 | Sociology | Large |
| http | 567498 | 3 | 2211 | 0.39 | Web | Large |
| cover | 286048 | 10 | 2747 | 0.96 | Botany | Large |
| fraud | 284807 | 29 | 492 | 0.17 | Finance | Large |
| skin | 245057 | 3 | 50859 | 20.75 | Image | Large |
| celeba | 202599 | 39 | 4547 | 2.24 | Image | Large |
| smtp | 95156 | 3 | 30 | 0.03 | Web | Large |
| ALOI | 49534 | 27 | 1508 | 3.04 | Image | Large |
| shuttle | 49097 | 9 | 3511 | 7.15 | Astronautics | Large |
| magic.gamma | 19020 | 10 | 6688 | 35.16 | Physical | Large |
| mammography | 11183 | 6 | 260 | 2.32 | Healthcare | Large |
| annthyroid | 7200 | 6 | 534 | 7.42 | Healthcare | Medium |
| pendigits | 6870 | 16 | 156 | 2.27 | Image | Medium |
| satellite | 6435 | 36 | 2036 | 31.64 | Astronautics | Medium |
| landsat | 6435 | 36 | 1333 | 20.71 | Astronautics | Medium |
| satimage-2 | 5803 | 36 | 71 | 1.22 | Astronautics | Medium |
| PageBlocks | 5393 | 10 | 510 | 9.46 | Document | Medium |
| Wilt | 4819 | 5 | 257 | 5.33 | Botany | Medium |
| thyroid | 3772 | 6 | 93 | 2.47 | Healthcare | Medium |
| Waveform | 3443 | 21 | 100 | 2.9 | Physics | Medium |
| Cardiotocography | 2114 | 21 | 466 | 22.04 | Healthcare | Medium |
| fault | 1941 | 27 | 673 | 34.67 | Physical | Medium |
| cardio | 1831 | 21 | 176 | 9.61 | Healthcare | Medium |
| letter | 1600 | 32 | 100 | 6.25 | Image | Medium |
| yeast | 1484 | 8 | 507 | 34.16 | Biology | Medium |
| vowels | 1456 | 12 | 50 | 3.43 | Linguistics | Medium |
| Pima | 768 | 8 | 268 | 34.9 | Healthcare | Small |
| breastw | 683 | 9 | 239 | 34.99 | Healthcare | Small |
| WDBC | 367 | 30 | 10 | 2.72 | Healthcare | Small |
| Ionosphere | 351 | 32 | 126 | 35.9 | Oryctognosy | Small |
| Stamps | 340 | 9 | 31 | 9.12 | Document | Small |
| vertebral | 240 | 6 | 30 | 12.5 | Biology | Small |
| WBC | 223 | 9 | 10 | 4.48 | Healthcare | Small |
| glass | 214 | 7 | 9 | 4.21 | Forensic | Small |
| WPBC | 198 | 33 | 47 | 23.74 | Healthcare | Small |
| Lymphography | 148 | 18 | 6 | 4.05 | Healthcare | Small |
| wine | 129 | 13 | 10 | 7.75 | Chemistry | Small |
| Hepatitis | 80 | 19 | 13 | 16.25 | Healthcare | Small |

incorporate time information, we apply fixed sinusoidal embeddings (Vaswani et al., 2017) to the scalar input $t \in [0, 1]$, following the positional encoding strategy used in transformer architectures. The time embedding (default dimension: 128) is concatenated with the input vector $\boldsymbol{x}$, and the combined representation is passed through the MLP to produce the predicted flow vector. We use the Adam optimizer with a learning rate of 0.005. The batch size is set to 1024 for datasets with more than 10,000 samples and to $\min(512, \#\text{training instances})$ for smaller datasets. The number of training epochs is determined empirically using the unsupervised hyperparameter selection method proposed by Li et al. (2025b), which requires no access to anomaly labels. While their method supports per-seed tuning, for consistency and fair evaluation, we fix the number of epochs across different random seeds. Notably, thanks to the efficiency of TCCM, tuning this single hyperparameter incurs minimal computational overhead. This is also the only data-dependent hyperparameter in our setup. The choices of key hyperparameters for our TCCM are presented in Table 5.

**Implementations of Other Baselines.** We utilise the well-established PyOD package (Zhao et al., 2019) for implementing (1) all classical anomaly detectors such as IForest (Liu et al., 2008), KDE (Latecki et al., 2007), etc., and (2) some deep anomaly detectors, including AutoEncoder (Aggarwal and Aggarwal, 2017b), ALAD (Zenati et al., 2018), DIF (Xu et al., 2023a), DeepSVDD (Ruff et al., 2018), LUNAR (Goodge et al., 2022), MOGAAL (Liu et al., 2019), SOGAAL (Liu et al., 2019), VAE (An and Cho, 2015), AE-1SVM (Nguyen and Vien, 2019), and AnoGAN (Schlegl et al., 2017); Additionally, we adapt the implementations from ADBench [3] for DAGMM (Zong et al., 2018) and GANAnomaly (Akcay et al., 2018); Besides, we include various advanced deep detectors, DROCC (Goyal et al., 2020)[4], GOAD (Bergman and Hoshen, 2020)[5], ICL (Shenkar and Wolf, 2022)[6], SLAD (Xu et al., 2023b)[7], MCM (Yin et al., 2024)[8], DTE (with four variants DTE-Categorical, DTE-Gaussian, DTE-InverseGamma, and DTE-NonParametric and a modified DDPM) (Livernoche et al., 2023)[9]; The implementation of planar flows (Rezende and Mohamed, 2015), a normalizing-flows-based detector, is also taken from (Livernoche et al., 2023). For all baseline detectors, we use their default configurations and hyperparameters as provided by their source implementations.

**Hardware and Software.** All experiments are conducted on machines equipped with Intel Xeon Gold 6430 CPUs (3.4 GHz, same model across runs, though not necessarily the same physical unit) and 256 GB RAM. No GPU acceleration is used. To ensure a fair comparison, each model is restricted to run on a *single* CPU core, allocated up to 10 GB of RAM, and a maximum runtime of 3 days per dataset. Our implementation is based on Python 3.9.21 with PyTorch 2.0, and experiments are executed within a conda-managed environment running Ubuntu 22.04.

## B.4  Evaluation metrics

We evaluate our proposed method and baselines using two standard metrics: Area Under the Receiver Operating Characteristic curve (AUROC) and Area Under the Precision-Recall Curve (AUPRC) (McDermott et al., 2024). Both metrics range from 0 to 1, with higher values indicating better performance. AUROC reflects a method's ability to distinguish between normal and anomalous instances: a score near 1 indicates near-perfect performance, 0.5 corresponds to random guessing, and values below 0.5 imply worse-than-random behavior. For AUPRC, which is more informative in imbalanced settings, higher values reflect better precision-recall trade-offs. All experiments are repeated over 5 independent runs with different random seeds, and we report the mean and standard deviation of each metric for every (dataset, anomaly detector) pair. For each dataset, we also compute detector rankings based on their mean AUROC and AUPRC. Due to the scale of the experiments, we present the complete tables of AUROC and AUPRC scores in the appendix. In the main paper, we visualize the distribution of ranks—computed from the mean AUROC and AUPRC scores across 5 runs—over all datasets using box plots. These plots are ordered by overall performance, defined as

---

[3]ADBench: `https://github.com/Minqi824/ADBench/tree/main/adbench/baseline`

[4]DROCC: `https://github.com/microsoft/EdgeML/blob/master/pytorch/edgeml_pytorch`

[5]GOAD: `https://github.com/lironber/GOAD`

[6]ICL: in the supplementary material of `https://openreview.net/forum?id=_hszZbt46bT`

[7]SLAD: `https://github.com/xuhongzuo/scale-learning`

[8]MCM: `https://github.com/JXYin24/MCM`

[9]DTE & DDPM: `https://github.com/vicliv/DTE`

the average rank across datasets, where each rank is based on the per-dataset mean score aggregated over the 5 runs.

## B.5    Pseudo-code of TCCM

---

**Algorithm 1** TCCM Training

---

1: **Input:** Training data samples $z \sim p_{\text{data}}$, neural network $f_{\boldsymbol{\theta}}$ with parameters $\boldsymbol{\theta}$, number of training epochs $N_{\text{epochs}}$, batch size $B$, learning rate $\eta$.
2: Initialize model parameters $\boldsymbol{\theta}$.
3: **for** epoch = 1 to $N_{\text{epochs}}$ **do**
4:       Shuffle training data.
5:       **for** each batch $\{z^{(i)}\}_{i=1}^{B}$ from $p_{\text{data}}$ **do**
6:             Sample time steps $\{t^{(i)}\}_{i=1}^{B}$ where each $t^{(i)} \sim \mathcal{U}(0,1)$.
7:             Generate time embeddings: $e_t^{(i)} \leftarrow \text{SinusoidalEmbedding}(t^{(i)})$ for $i = 1, \ldots, B$.
8:             Form augmented inputs: $\tilde{z}^{(i)} \leftarrow [z^{(i)}; e_t^{(i)}]$ for $i = 1, \ldots, B$.
9:             Predict contraction vectors: $\hat{v}^{(i)} \leftarrow f_{\boldsymbol{\theta}}(\tilde{z}^{(i)})$ for $i = 1, \ldots, B$.
10:            Compute batch loss: $\mathcal{L}(\boldsymbol{\theta}) \leftarrow \frac{1}{B} \sum_{i=1}^{B} \left\| \hat{v}^{(i)} + z^{(i)} \right\|_2$.       ▷ Corresponds to Eq. 4
11:            Update parameters: $\boldsymbol{\theta} \leftarrow \boldsymbol{\theta} - \eta \nabla_{\boldsymbol{\theta}} \mathcal{L}(\boldsymbol{\theta})$.
12: **Output:** Trained model $f_{\boldsymbol{\theta}}$.

---

---

**Algorithm 2** TCCM Inference (Anomaly Scoring)

---

1: **Input:** Test sample $z_{\text{test}}$, trained model $f_{\boldsymbol{\theta}}$, fixed evaluation time $t_{\text{fixed}} \in (0,1]$ (default $t_{\text{fixed}} = 1$).
2: Generate time embedding: $e_{t_{\text{fixed}}} \leftarrow \text{SinusoidalEmbedding}(t_{\text{fixed}})$.
3: Form augmented input: $\tilde{z}_{\text{test}} \leftarrow [z_{\text{test}}; e_{t_{\text{fixed}}}]$.
4: Predict contraction vector: $\hat{v}_{\text{test}} \leftarrow f_{\boldsymbol{\theta}}(\tilde{z}_{\text{test}})$.
5: Compute anomaly score: $S(z_{\text{test}}; t_{\text{fixed}}) \leftarrow \|\hat{v}_{\text{test}} + z_{\text{test}}\|_2$.       ▷ Corresponds to Eq. 5
6: **Output:** Anomaly score $S(z_{\text{test}}; t_{\text{fixed}})$.

---

## B.6    Unsupervised Epoch Selection Strategy

In the main paper, the architecture of TCCM is fixed as a lightweight MLP ($2 \times 256$ ReLU) across all datasets. While this choice ensures efficiency and comparability, the number of training epochs is not arbitrarily hardcoded. Instead, we adopt a principled and largely automated protocol for unsupervised hyperparameter selection. For completeness, we provide additional details below.

**Epoch Selection Protocol.**    For each dataset, we first examine the empirical training loss curve to identify a rough convergence threshold. Using this as a lower bound, we define a bounded search space of candidate epochs. Within this space, we apply the unsupervised hyperparameter tuning method introduced by Li et al. (2025b), based on the *Improved Contrast Score Margin (CSM)* criterion. This criterion evaluates the margin between top-$k$ predicted anomalous and normal samples solely from the distribution of model outputs, without requiring any ground-truth labels:

$$T(f) = \frac{\hat{\mu}_O - \hat{\mu}_I}{\sqrt{\hat{\sigma}_O^2 + \hat{\sigma}_I^2}},$$

where $\hat{\mu}_O, \hat{\sigma}_O^2$ denote the mean and variance of anomaly scores for the top-$k$ predicted anomalies, and $\hat{\mu}_I, \hat{\sigma}_I^2$ correspond to the remaining $n - k$ presumed inliers. For each candidate epoch, we compute $T(f)$ and select the configuration maximizing this criterion.

**Hardcoding for Simplicity.**    Although this procedure yields dataset-specific and random-seed-dependent epoch values, we ultimately fix the selected epoch across all seeds of a given dataset for simplicity and reproducibility. We note that using per-seed dynamically tuned epochs can sometimes further improve performance, but we chose not to report this to avoid inflating results and to ensure fair comparison across baselines.

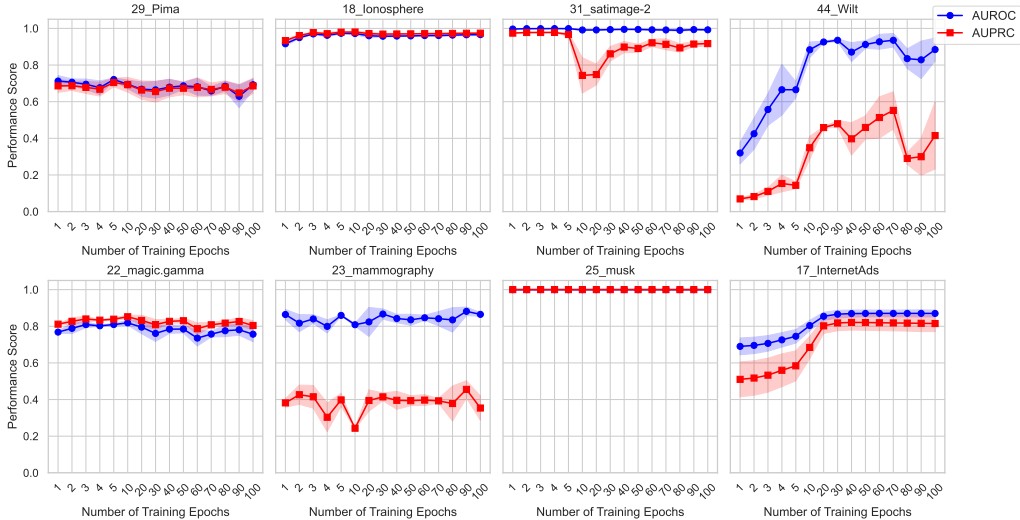

Figure 5: Sensitivity of TCCM to the number of training epochs. For each dataset, we evaluate AUROC and AUPRC across a wide range of epoch values. Results show a **stable plateau** on most datasets (e.g., `Pima`, `Ionosphere`, `Musk`, `InternetAds`), where performance converges early and further training offers minimal gain but adds runtime cost. A similar plateau also appears on `magic.gamma` and `mammography`, albeit with minor fluctuations. On some datasets (e.g., `satimage`), excessive training leads to **overfitting**, while others (e.g., `Wilt`) exhibit stronger fluctuations, reflecting less stable convergence. These findings highlight that: for most datasets, TCCM does not rely on finely tuned epoch numbers and remains robust once a reasonable training horizon is reached.

**Sensitivity Analysis.** To further address this point, we include a sensitivity analysis over representative datasets. Two consistent patterns emerge:

- **Stable Plateau:** For most datasets (e.g., `Pima`, `Ionosphere`, `Musk`, `InternetAds`), model performance stabilizes after a certain number of epochs, where further training brings little to no improvement but increases runtime. A similar plateau is also observed on `magic.gamma` and `mammography`, though with minor fluctuations.

- **Overfitting Risk:** On some datasets (notably `satimage`), training beyond the plateau results in performance degradation, suggesting overfitting. In contrast, `Wilt` shows larger fluctuations, indicating less stable convergence rather than clear overfitting.

These findings justify our principled choice of using CSM-based epoch selection combined with early convergence boundaries.

**Discussion.** We emphasize that unsupervised hyperparameter tuning remains an under-explored but important challenge in anomaly detection. Our approach leverages a recently proposed and validated criterion, but we believe future work should explore more adaptive and automated tuning protocols.

# C   Property Analysis

**Philosophical Analogy.** Our method encodes a natural inductive bias: no matter where a sample lies along the temporal axis, it is always guided by the same high-level goal—movement toward the origin. This echoes the classical proverb *"Only by staying true to our original aspiration can we reach our final destination"*, embodying a consistency that is both geometrically meaningful and empirically effective. In this section, we provide some theoretical analyses of our anomaly detection framework in addition to the analyses given in the main paper as well as their proof (when applicable).

## C.1 Relation to Flow Matching and Diffusion Modeling

Our proposed TCCM can be viewed as a task-specific simplification of flow-based learning frameworks, adapted to the semi-supervised anomaly detection setting.

Conventional flow matching methods (Lipman et al., 2022; Liu et al., 2022) aim to learn a continuous-time vector field $v(x, t)$ that transforms samples from a source to a target distribution, often supervised using interpolated trajectories. Extensions to stochastic generative modeling, such as diffusion models, further describe the evolution of data via a stochastic differential equation (SDE):

$$\frac{dx(t)}{dt} = -\alpha(t)x(t) + \sigma(t)\epsilon, \quad \epsilon \sim \mathcal{N}(0, I), \tag{6}$$

where $\alpha(t)$ defines the contraction rate toward the origin, and $\sigma(t)$ controls the injection of Gaussian noise over time. While such stochasticity improves sample diversity in generative tasks, it may hinder anomaly detection—especially in the semi-supervised setting where only normal data is observed. The added noise may obscure the underlying structure of normal instances, reducing their separability from anomalies.

TCCM can be interpreted as a deterministic limit of this framework, where we fix $\alpha(t) := 1$ and set $\sigma(t) := 0$. Moreover, instead of simulating time-evolving trajectories, we directly supervise the velocity field at the initial state $x$, using the fixed target vector $-x$ at each sampled time step. This design retains the inductive bias of contraction toward normality while significantly simplifying training and inference, avoiding both trajectory supervision and numerical integration.

**Ablation on Noise Injection during Training.** To evaluate the effect of noise, we compare TCCM with a noisy variant that injects Gaussian perturbations during training (emulating the SDE in Eq. 6). As shown in Appendix D.3, noise consistently harms anomaly detection performance across AUPRC and AUROC. This supports our hypothesis that deterministic dynamics better preserve structural regularities in normal data.

## C.2 Anomaly Score Expectation under Distributional Shift

To theoretically justify the discriminative behavior of our anomaly score, we analyze its expected value under a Gaussian distributional shift. For notation simplicity, we utilise $f_\theta(x, 1)$ for $f_\theta([x; \text{Embed}(1)])$ and $f_\theta(z, 1)$ for $f_\theta([z; \text{Embed}(1)])$ in the following. We consider the case where the learned contraction field satisfies, for all $x \in \mathbb{R}^d$ in the training set,

$$f_\theta(x, 1) = -x + \epsilon, \quad \epsilon \sim \mathcal{N}(0, \sigma_f^2 I_d). \tag{7}$$

The anomaly score then becomes

$$S(x) = \|f_\theta(x, 1) + x\|_2 = \|\epsilon\|_2. \tag{8}$$

**Proposition 3** (Discriminative Power under Gaussian-to-Gaussian Shift). *Let normal samples be drawn from $x \sim \mathcal{N}(0, \sigma^2 I_d)$, and anomalous samples from $z \sim \mathcal{N}(\mu, \sigma^2 I_d)$ with $\mu \neq 0$. Assume the learned contraction field satisfies $f_\theta(x, 1) = -x + \epsilon$, where $\epsilon \sim \mathcal{N}(0, \sigma_f^2 I_d)$. Assume that the learned velocity field is mismatched for anomalies. Then the corresponding anomaly scores satisfy:*

$$S(x) \sim \chi_d \cdot \sigma_f, \tag{9}$$

$$S(z) \sim \chi_d(\lambda), \quad with \quad \lambda = \frac{\|\mu\|_2^2}{\sigma_f^2}, \tag{10}$$

*where $\chi_d$ and $\chi_d(\lambda)$ denote the central and non-central chi distributions with $d$ degrees of freedom and non-centrality parameter $\lambda$, respectively. Moreover, the expected values satisfy:*

$$\mathbb{E}[S(x)] = \sigma_f \cdot \sqrt{2} \cdot \frac{\Gamma\left(\frac{d+1}{2}\right)}{\Gamma\left(\frac{d}{2}\right)}, \quad \mathbb{E}[S(z)] > \mathbb{E}[S(x)]. \tag{11}$$

*Proof.* For normal samples $x \sim \mathcal{N}(0, \sigma^2 I_d)$, the anomaly score reduces to

$$S(x) = \|f_\theta(x, 1) + x\|_2 = \|\epsilon\|_2,$$

where $\epsilon \sim \mathcal{N}(\mathbf{0}, \sigma_f^2 I_d)$. Thus,

$$S(\boldsymbol{x}) \sim \sigma_f \cdot \chi_d,$$

and the expected value is

$$\mathbb{E}[S(\boldsymbol{x})] = \sigma_f \cdot \mathbb{E}[\chi_d] = \sigma_f \cdot \sqrt{2} \cdot \frac{\Gamma\left(\frac{d+1}{2}\right)}{\Gamma\left(\frac{d}{2}\right)}.$$

For anomalous samples $\boldsymbol{z} \sim \mathcal{N}(\boldsymbol{\mu}, \sigma^2 I_d)$, we again have

$$f_\theta(\boldsymbol{z}, 1) = -\boldsymbol{z} + \boldsymbol{\epsilon}, \quad \Rightarrow \quad S(\boldsymbol{z}) = \|f_\theta(\boldsymbol{z}, 1) + \boldsymbol{z}\|_2 = \|\boldsymbol{\epsilon}\|_2,$$

but now $\epsilon \sim \mathcal{N}(\mathbf{0}, \sigma_f^2 I_d)$ is independent of $\boldsymbol{z} \sim \mathcal{N}(\boldsymbol{\mu}, \sigma^2 I_d)$, and $\epsilon$ is added to the fixed vector $\boldsymbol{z}$.

Thus, conditional on a sample $\boldsymbol{z}$, the score becomes

$$S(\boldsymbol{z}) = \|\boldsymbol{z} - \boldsymbol{z} + \boldsymbol{\epsilon}\|_2 = \|\boldsymbol{\epsilon}\|_2,$$

but this is misleading. In practice, the model is trained to approximate contraction on normal data. For anomalous inputs, the field is mismatched, and we express this by assuming:

$$f_\theta(\boldsymbol{z}, 1) = -\boldsymbol{z}_{\text{proj}} + \boldsymbol{\epsilon},$$

with $\boldsymbol{z}_{\text{proj}}$ being the projection of $\boldsymbol{z}$ onto the normal data manifold. Then:

$$S(\boldsymbol{z}) = \|f_\theta(\boldsymbol{z}, 1) + \boldsymbol{z}\|_2 = \|\boldsymbol{z} - \boldsymbol{z}_{\text{proj}} + \boldsymbol{\epsilon}\|_2.$$

Let $\boldsymbol{\delta} := \boldsymbol{z} - \boldsymbol{z}_{\text{proj}}$, then:

$$S(\boldsymbol{z}) = \|\boldsymbol{\delta} + \boldsymbol{\epsilon}\|_2.$$

Since $\epsilon \sim \mathcal{N}(\mathbf{0}, \sigma_f^2 I_d)$, and $\boldsymbol{\delta} \in \mathbb{R}^d$ is fixed conditional on $\boldsymbol{z}$, it follows that:

$$S(\boldsymbol{z}) \sim \chi_d(\lambda), \quad \text{with } \lambda = \frac{\|\boldsymbol{\delta}\|_2^2}{\sigma_f^2}.$$

From standard properties of the non-central chi distribution, we know:

$$\mathbb{E}[\chi_d(\lambda)] > \mathbb{E}[\chi_d] \quad \text{for all } \lambda > 0.$$

Thus:

$$\mathbb{E}[S(\boldsymbol{z})] > \mathbb{E}[S(\boldsymbol{x})],$$

completing the proof. $\qquad \square$

**Proposition 4** (Discriminative Power under GMM-to-Gaussian Shift)**.** *Let normal data be sampled from a Gaussian mixture model (GMM)*

$$\boldsymbol{x} \sim \sum_{r=1}^R \pi_r \cdot \mathcal{N}(\boldsymbol{\mu}_r, \sigma^2 I_d), \quad \sum_{r=1}^R \pi_r = 1,$$

*and assume the learned contraction field satisfies*

$$f_\theta(\boldsymbol{x}, 1) = -\boldsymbol{x} + \boldsymbol{\epsilon}, \quad \epsilon \sim \mathcal{N}(\mathbf{0}, \sigma_f^2 I_d).$$

*Assume that the learned velocity field is mismatched for anomalies. Define the anomaly score as*

$$S(\boldsymbol{x}) := \|f_\theta(\boldsymbol{x}, 1) + \boldsymbol{x}\|_2 = \|\boldsymbol{\epsilon}\|_2.$$

*Then the anomaly score for normal data follows a central chi distribution:*

$$S(\boldsymbol{x}) \sim \chi_d \cdot \sigma_f,$$

*and its expected value is*

$$\mathbb{E}[S(\boldsymbol{x})] = \sigma_f \cdot \sqrt{2} \cdot \frac{\Gamma\left(\frac{d+1}{2}\right)}{\Gamma\left(\frac{d}{2}\right)}.$$

Let anomalous samples be drawn from $z \sim \mathcal{N}(\mu_z, \sigma^2 I_d)$, with $\mu_z \notin \{\mu_1, \dots, \mu_R\}$. Then the anomaly score for $z$ satisfies

$$S(z) := \|f_\theta(z, 1) + z\|_2 = \|\delta + \epsilon\|_2,$$

where $\delta := z - z_{proj}$ is the mismatch between $z$ and the normal manifold. Then

$$S(z) \sim \chi_d(\lambda), \quad \lambda = \frac{\|\delta\|_2^2}{\sigma_f^2},$$

and the expected anomaly score satisfies

$$\mathbb{E}[S(z)] > \mathbb{E}[S(x)].$$

*Proof.* For normal samples $x \sim \mathcal{N}(\mu_r, \sigma^2 I_d)$, the anomaly score is

$$S(x) = \|f_\theta(x, 1) + x\|_2 = \| - x + \epsilon + x\|_2 = \|\epsilon\|_2.$$

Since $\epsilon \sim \mathcal{N}(0, \sigma_f^2 I_d)$, it follows that

$$S(x) \sim \sigma_f \cdot \chi_d.$$

Hence,

$$\mathbb{E}[S(x)] = \sigma_f \cdot \mathbb{E}[\chi_d] = \sigma_f \cdot \sqrt{2} \cdot \frac{\Gamma\left(\frac{d+1}{2}\right)}{\Gamma\left(\frac{d}{2}\right)}.$$

Now consider an anomalous input $z \sim \mathcal{N}(\mu_z, \sigma^2 I_d)$, not seen during training. The contraction field, trained only on normal GMM components, is not well aligned with $z$. Let $z_{proj} \in \text{supp}(p_{data})$ be the closest point on the normal manifold, then we model the output as:

$$f_\theta(z, 1) \approx -z_{proj} + \epsilon \quad \Rightarrow \quad S(z) = \|z + f_\theta(z, 1)\|_2 = \|z - z_{proj} + \epsilon\|_2.$$

Let $\delta := z - z_{proj}$, which is fixed conditioned on $z$, then

$$S(z) \sim \chi_d(\lambda), \quad \text{with } \lambda = \frac{\|\delta\|_2^2}{\sigma_f^2}.$$

It is a known result that for all $\lambda > 0$, the non-central chi distribution satisfies:

$$\mathbb{E}[\chi_d(\lambda)] > \mathbb{E}[\chi_d],$$

implying that

$$\mathbb{E}[S(z)] > \mathbb{E}[S(x)].$$

$\square$

**Proposition 5** (Namely Proposition 2, Discriminative Power under GMM-to-GMM Shift). *Let normal samples be drawn from a Gaussian mixture model:*

$$x \sim \sum_{r=1}^{R} \pi_r \cdot \mathcal{N}(\mu_r, \sigma^2 I_d), \quad \sum_{r=1}^{R} \pi_r = 1.$$

*Let anomalous samples be drawn from another Gaussian mixture model with distinct component means:*

$$z \sim \sum_{s=1}^{S} \eta_s \cdot \mathcal{N}(\nu_s, \sigma^2 I_d), \quad \sum_{s=1}^{S} \eta_s = 1,$$

*with $\nu_s \notin \{\mu_1, \dots, \mu_R\}$ for all $s$. Assume the learned contraction field satisfies:*

$$f_\theta(x, 1) = -x + \epsilon, \quad \epsilon \sim \mathcal{N}(0, \sigma_f^2 I_d).$$

*Assume that the learned velocity field is mismatched for anomalies.* [10] *Define the anomaly score as:*

$$S(\boldsymbol{x}) := \|f_\theta(\boldsymbol{x}, 1) + \boldsymbol{x}\|_2.$$

*Then:*

1. *For normal samples:*

$$S(\boldsymbol{x}) \sim \chi_d \cdot \sigma_f, \quad \mathbb{E}[S(\boldsymbol{x})] = \sigma_f \cdot \sqrt{2} \cdot \frac{\Gamma\left(\frac{d+1}{2}\right)}{\Gamma\left(\frac{d}{2}\right)}.$$

2. *For anomalous samples, each component satisfies:*

$$S(\boldsymbol{z}) \sim \chi_d(\lambda_s), \quad \lambda_s = \frac{\|\boldsymbol{\nu}_s - \boldsymbol{\mu}_{r^*(s)}\|_2^2}{\sigma_f^2},$$

*where $\boldsymbol{\mu}_{r^*(s)} := \arg\min_{\boldsymbol{\mu}_r} \|\boldsymbol{\nu}_s - \boldsymbol{\mu}_r\|_2$. Then:*

$$\mathbb{E}[S(\boldsymbol{z})] = \sum_{s=1}^{S} \eta_s \cdot \mathbb{E}[\chi_d(\lambda_s)] > \mathbb{E}[S(\boldsymbol{x})].$$

*Proof.* **Step 1: Normal samples.**

For any $\boldsymbol{x} \sim \mathcal{N}(\boldsymbol{\mu}_r, \sigma^2 I_d)$, since the contraction field is learned from normal data, we assume it satisfies:

$$f_\theta(\boldsymbol{x}, 1) = -\boldsymbol{x} + \boldsymbol{\epsilon}, \quad \boldsymbol{\epsilon} \sim \mathcal{N}(\boldsymbol{0}, \sigma_f^2 I_d).$$

Therefore,

$$S(\boldsymbol{x}) = \|f_\theta(\boldsymbol{x}, 1) + \boldsymbol{x}\|_2 = \|\boldsymbol{\epsilon}\|_2 \sim \chi_d \cdot \sigma_f.$$

Thus, for normal samples, the anomaly score distribution is a central chi distribution with scale $\sigma_f$. Its expectation is given by:

$$\mathbb{E}[S(\boldsymbol{x})] = \sigma_f \cdot \sqrt{2} \cdot \frac{\Gamma\left(\frac{d+1}{2}\right)}{\Gamma\left(\frac{d}{2}\right)}.$$

**Step 2: Anomalous samples.**

Each anomalous component is $\boldsymbol{z} \sim \mathcal{N}(\boldsymbol{\nu}_s, \sigma^2 I_d)$. Since the model is trained only on normal components $\boldsymbol{\mu}_r$, it cannot learn a correct contraction vector for $\boldsymbol{z}$. As an approximation, we model the field as:

$$f_\theta(\boldsymbol{z}, 1) \approx -\boldsymbol{z}_{\text{proj}} + \boldsymbol{\epsilon},$$

where $\boldsymbol{z}_{\text{proj}}$ is the projection of $\boldsymbol{z}$ onto the nearest normal cluster center:

$$\boldsymbol{z}_{\text{proj}} := \boldsymbol{\mu}_{r^*(s)} = \arg\min_{\boldsymbol{\mu}_r} \|\boldsymbol{z} - \boldsymbol{\mu}_r\|_2.$$

Then the anomaly score becomes:

$$S(\boldsymbol{z}) = \|f_\theta(\boldsymbol{z}, 1) + \boldsymbol{z}\|_2 = \|\boldsymbol{z} - \boldsymbol{z}_{\text{proj}} + \boldsymbol{\epsilon}\|_2.$$

---

[10]Regarding empirical support for this assumption, we offer three points of clarification: (1) Direct visual validation: Figure 1 provides visualizations of the learned contraction vectors on synthetic 2D datasets. These examples clearly demonstrate that anomalous points consistently deviate from the expected contraction field, validating the mismatch assumption in a controlled and interpretable setting. (2) Indirect support through benchmark results: Across 47 real-world tabular datasets, TCCM consistently achieves strong AUROC and AUPRC scores. This level of performance would be difficult to attain if the model failed to differentiate between normal and anomalous points during inference—thus indirectly supporting the presence and utility of the mismatch behavior assumed in our analysis. (3) Controlled synthetic validation: We aslo provide a dedicated empirical study based on the Gaussian mixture setup. By comparing anomaly score distributions for normal and anomalous points across multiple dimensions ($d = 2, 5, 10, 15, 20$), we show that anomalies consistently yield higher scores. This directly validates the mismatch assumption in a controlled setting aligned with our theoretical analysis.

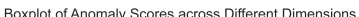

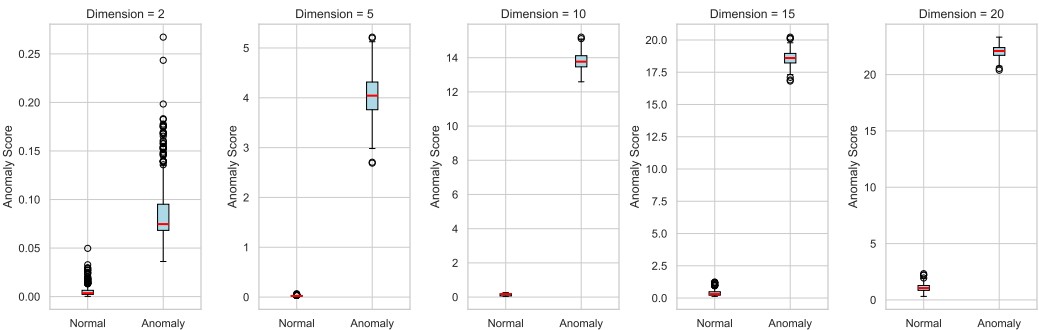

Figure 6: Empirical validation of the mismatch assumption in Propositions 3, 4, and 5. Boxplots show anomaly score distributions for normal and anomalous samples under different data dimensions ($d = 2, 5, 10, 15, 20$). Across all cases, anomalous points consistently yield higher scores than normal points, supporting the assumption that anomalies incur a systematic mismatch under the learned contraction field.

Let $\boldsymbol{\delta}_s := \boldsymbol{z} - \boldsymbol{\mu}_{r^*(s)}$, which satisfies $\boldsymbol{\delta}_s \sim \mathcal{N}(\boldsymbol{\nu}_s - \boldsymbol{\mu}_{r^*(s)}, \sigma^2 I_d)$. Since $\boldsymbol{\epsilon} \sim \mathcal{N}(\boldsymbol{0}, \sigma_f^2 I_d)$, the sum $\boldsymbol{\delta}_s + \boldsymbol{\epsilon} \sim \mathcal{N}(\boldsymbol{\nu}_s - \boldsymbol{\mu}_{r^*(s)}, (\sigma^2 + \sigma_f^2)I_d)$. Hence,

$$S(\boldsymbol{z}) \sim \chi_d(\lambda_s), \quad \lambda_s = \frac{\|\boldsymbol{\nu}_s - \boldsymbol{\mu}_{r^*(s)}\|_2^2}{\sigma_f^2}.$$

Then the overall anomaly score distribution for anomalous samples (from the mixture) is:

$$S(\boldsymbol{z}) \sim \sum_{s=1}^{S} \eta_s \cdot \chi_d(\lambda_s), \quad \text{with } \lambda_s > 0.$$

It is a standard result that:
$$\mathbb{E}[\chi_d(\lambda_s)] > \mathbb{E}[\chi_d] \quad \forall \lambda_s > 0.$$

Therefore,

$$\mathbb{E}[S(\boldsymbol{z})] = \sum_{s=1}^{S} \eta_s \cdot \mathbb{E}[\chi_d(\lambda_s)] > \mathbb{E}[\chi_d] = \frac{1}{\sigma_f} \cdot \mathbb{E}[S(\boldsymbol{x})],$$

which implies:
$$\mathbb{E}[S(\boldsymbol{z})] > \mathbb{E}[S(\boldsymbol{x})],$$

completing the proof. $\square$

**Empirical Study: Validating the Mismatch Assumption.** To empirically validate the key assumption in Propositions 3, 4, and 5—that the learned contraction field is mismatched for anomalies—we conduct an experiment on synthetic Gaussian mixture data. Normal samples are drawn from a GMM with $R = 3$ components (each with isotropic covariance $\sigma^2 I_d$), while anomalous samples are generated from a single Gaussian $\mathcal{N}(\mu_z, \sigma^2 I_d)$ whose mean is located outside the mixture centers. The contraction field $f_\theta$ is trained exclusively on normal data using our objective function (see Eq. 4), and anomaly scores $S(x) = \|f_\theta(x, 1) + x\|_2$ are computed for both groups. Figure 6 shows boxplots of the score distributions for normals and anomalies across $d \in \{2, 5, 10, 15, 20\}$. In all cases, anomalous points exhibit consistently higher scores than normal points, with AUROC values exceeding 0.9 regardless of dimension. These results provide direct empirical evidence that anomalies indeed incur a systematic mismatch under the learned contraction field, thereby justifying the modeling assumption made in Propositions 3, 4, and 5.

**Limitation of Theoretical Analysis and Future Work.** We acknowledge that Proposition 5 is derived under a simplified setting involving Gaussian mixture models (GMMs). We would like to clarify that the use of GMMs in this theoretical result is a deliberate and well-motivated modeling choice:

(1) GMMs have been widely adopted as analytical tools in the machine learning literature—not only in anomaly detection but also in clustering and density modeling. For example, the work of (Zong et al., 2018) introduces the DAGMM model for unsupervised anomaly detection based on similar distributional assumptions. Likewise, GMMs serve as the theoretical backbone in clustering studies such as (Chen and Zhang, 2024), which performs theoretical analysis under anisotropic GMMs. These works demonstrate that GMM-based settings are not only standard but also provide valuable theoretical insight despite being idealized; (2) Our aim is not to suggest that real-world data exactly follow GMMs, but rather to use this setup as a clean, analyzable lens to understand the discriminative behavior of the TCCM scoring function. The result shows that under mild assumptions, the anomaly score is provably larger in expectation for out-of-distribution samples drawn from disjoint mixtures—thus justifying the use of the residual norm as a discriminative signal; (3) Deriving general results under arbitrary data distributions is typically intractable, especially for deep models. Our theoretical analysis strikes a practical balance by providing provable insight under realistic yet analyzable settings. In future work, we aim to explore theoretical extensions to broader classes of distributions, but we believe the current result already provides valuable intuition and justification for the observed empirical behavior.

### C.3 Analysis of Representation Collapse

One potential concern for our model is the possibility of *representation collapse*, where the learned mapping trivially reproduces the input or converges to a constant output, thereby failing to distinguish between normal and anomalous data. To verify that TCCM does not suffer from this issue, we provide both theoretical and empirical evidence.

**Architectural Considerations.** The trivial mapping case, e.g., $f_\theta([\mathbf{z}, \mathrm{Embed}(t)]) = [\mathbf{I}, \mathbf{0}][\mathbf{z}; \mathrm{Embed}(t)]^T = \mathbf{z}$, corresponds to a highly restricted setting where the model degenerates to a single-layer linear transformation without bias or activation, and where the time embedding has no influence. However, this configuration does not reflect the architecture used in TCCM. In practice, TCCM employs a multi-layer MLP with RELU activations and high-dimensional sinusoidal time embeddings that are explicitly concatenated to the input. These design choices allow the model to learn complex, time-varying contraction dynamics, making identity or partial-identity mappings highly unlikely.

**Implicit Regularization.** Unlike previous methods such as DeepSVDD (Ruff et al., 2018) that prevent collapse by imposing explicit architectural constraints (e.g., bounded activations or bias removal), TCCM avoids such restrictions and instead discourages collapse through *time-conditioned supervision*, multi-time-step optimization, and implicit regularization induced by nonlinear transformations. The temporal embedding ensures that each training instance is contextually distinct across time, which prevents the network from converging to a single trivial representation.

**Empirical Verification.** To further examine this, we track training dynamics and representation diversity throughout training. Empirically, we observe no evidence of collapse: training loss decreases smoothly without flattening, and anomaly scores exhibit non-degenerate distributions across both normal and anomalous samples. Additionally, the learned feature representations maintain high variance across dimensions, and anomaly detection performance remains stable across datasets (see Figure 2). These observations collectively confirm that TCCM learns meaningful, discriminative representations rather than degenerate identity mappings.

**Summary.** Overall, TCCM's design—combining multi-layer nonlinear mappings, explicit temporal conditioning, and implicit regularization—effectively mitigates representation collapse without relying on handcrafted architectural constraints. This ensures that the learned contraction field remains expressive and discriminative, supporting robust anomaly detection across diverse data regimes.

# D Full Results and Analysis

## D.1 Full Analysis of Effectiveness

**(1) Effectiveness on Small-scale Dataset.** As shown in Table 6, TCCM achieves strong performance in terms of AUROC on small-scale datasets, ranking in the top 10 on 10 out of 12 datasets, with an average rank of 4.42—the best among all evaluated methods. This result highlights the effectiveness of TCCM in low-data regimes. Despite being a deep learning method—which are typically considered data-hungry and prone to underperformance on small datasets—TCCM consistently outperforms both classical and deep baselines. Similar conclusions hold for AUPRC, as shown in Table 7.

**(2) Effectiveness on Medium-scale Datasets.** As shown in Table 8, TCCM demonstrates strong performance on medium-scale datasets in terms of AUROC, ranking in the top 10 on 13 out of 15 datasets, with an average rank of 6.80—the second best among all anomaly detectors (slightly outperformed by DTE-NonParametric with an average of 6.20). Similarly, Table 9 shows that TCCM ranks in the top 10 on 13 out of 15 datasets in terms of AUPRC, with an average rank of 6.60, achieving the best position overall (followed by DTE-NonParametric with an average rank of 7.13). In both cases, DTE-NonParametric achieves strong performance, but suffers from poor explainability and lacks provable robustness, limiting its practical deployment in sensitive or high-stakes applications.

**(3) Effectiveness on Large-scale Datasets.** From Table 10, we can see that TCCM gives good performance in terms of AUROC score: it gives top-10 results 9 out of 11 datasets, with an average ranking of 7.36, the second highest ranking among all anomaly detectors (slightly outperformed by DTE-NonParametric with a rank of 6.36). Meanwhile, Table 11 shows that: concerning AUPRC score, it gives top-10 results 9 out of 11 datasets, with an average ranking of 7.18 (followed by DTE-NonParametric with a rank of 8.45), the highest ranking among all anomaly detectors. Note that DTE-NonParametric suffers from low scalability and lack of provable robustness and explainability.

**(4) Effectiveness on High-dimensional Datasets.** From Table 12, we can see that TCCM gives good performance in terms of AUROC score: it gives top-10 results 9 out of 9 datasets, with an average ranking of 4.89, the second highest ranking among all anomaly detectors (slightly outperformed by DTE-NonParametric with an average rank of 4.44). Meanwhile, Table 13 shows that: concerning AUPRC score, it gives top-10 results on 8 out of 9 datasets, with an average ranking of 5.56 (followed by DTE-NonParametric with a rank of 6.56), the highest ranking among all anomaly detectors. Note that DTE-NonParametric suffers from low scalability at inference and lack of provable robustness and explainability.

## D.2 Full Results on Scalability Analysis

### D.2.1 Analysis of the Trade-off Between Inference Speed and Accuracy

To further contextualize the efficiency of TCCM, we follow the evaluation practice introduced by DTE (Livernoche et al., 2023) and analyze the relationship between average inference time and detection performance (measured by AUROC and AUPRC) across all 46 anomaly detection methods. As shown in Figures 7 and 8 , TCCM occupies the lower-left region of the plot, indicating simultaneously high accuracy and low inference latency.

Most competing methods fall into one of two regimes: (1) *slow but accurate* models such as DTE-NonParametric, LUNAR, and KDE, which achieve comparable AUROC and AUPRC but require several orders of magnitude longer inference; and (2) *fast but less accurate* methods such as GMM, CBLOF, and Sampling, which exhibit shorter inference but substantially reduced detection accuracy. In contrast, TCCM provides a favorable middle ground—delivering high detection accuracy without compromising inference efficiency.

These results reinforce our claim that TCCM achieves one of the best overall balances between accuracy and computational cost among deep learning-based approaches, demonstrating its strong potential for deployment in real-time and resource-constrained anomaly detection scenarios.

### D.2.2 Scalability Analysis on All Algorithms

To assess the practical deployability of TCCM, we perform a comprehensive scalability analysis across three runtime dimensions: **training time**, **inference time**, and **total execution time** (training

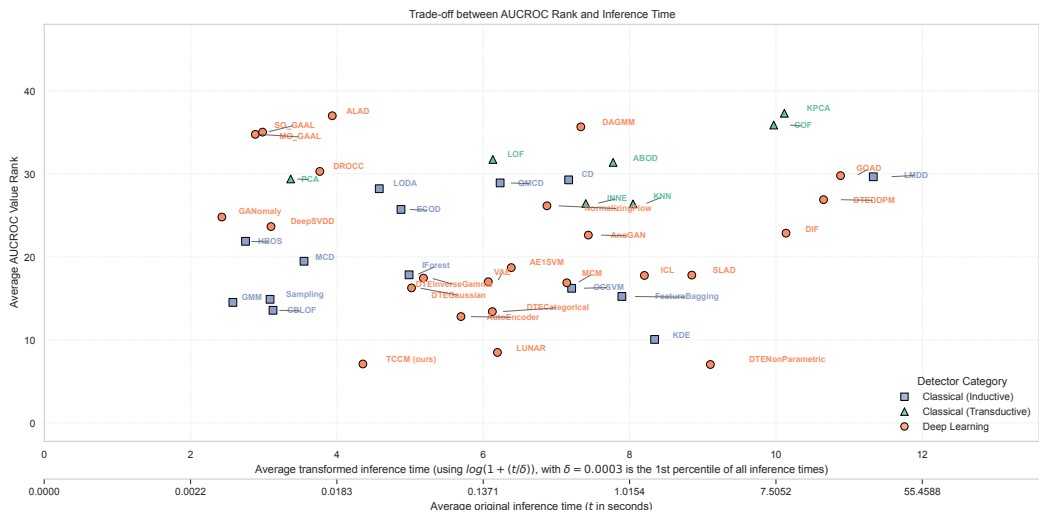

Figure 7: Distribution of *average inference time* (transformed with $\log\left(1 + \frac{t}{\delta}\right)$ to achieve better visualization, with $\delta$ is the 1st percentile of all inference times $t$) vs. *average AUROC rank* across all 45 anomaly detection methods. The ticks corresponding to the original average inference time are also displayed underneath. TCCM achieves the best balance between inference speed and detection accuracy, outperforming both slow but accurate (e.g., DTE-Nonparametric, KDE) and fast but less accurate (e.g., GMM, CBLOF, Sampling) methods.

+ testing). Unlike the main paper, which focuses on comparisons with the strongest deep baselines (DTE-NonParametric, LUNAR, and KDE), here we extend the evaluation to **all 44 baselines**—including classical (transductive and inductive) and deep learning-based methods—to provide a complete view of computational efficiency.

Our analysis centers on **large and high-dimensional datasets**, where runtime differences become most pronounced. Smaller datasets tend to produce negligible timing gaps, as even slower models finish within seconds. In contrast, the large-scale datasets (with hundreds of thousands of samples or high feature dimensionality) amplify differences in efficiency, offering a realistic measure of scalability in deployment settings.

**(1) Training Time.** As shown in Figure 9, TCCM achieves one of the lowest training times within the deep learning group. Its distribution centers near fast, lightweight models such as AutoEncoder and DeepSVDD, while being faster than most other deep learning methods (e.g., ANOGAN, DTE-Categorical, GOAD). Some classical algorithms (e.g., KDE, OCSVM, LMDD) display higher variability and much longer training durations due to nonparametric or pairwise computations. Overall, TCCM provides a strong balance between model capacity and training efficiency, confirming its practicality for large-scale learning.

**(2) Inference Time.** Figure 10 presents the distribution of inference times across detectors. Within the deep learning group, TCCM ranks among the fastest methods, close to simple methods such as ALAD and DROCC, but with markedly higher detection accuracy. By contrast, diffusion-based approaches such as DTE-NonParametric, and DTE-DDPM lie at the upper end of the runtime spectrum, often exhibiting multi-order magnitude slower inference across large datasets. Compared to classical baselines (e.g., CBLOF, IFOREST, ECOD), TCCM maintains similar or better inference efficiency while achieving superior detection performance.

**(3) Total Runtime.** The total runtime (training + inference), summarized in Figure 11, shows that TCCM achieves one of the best overall efficiency–accuracy trade-offs among all evaluated methods. Within the deep learning family, TCCM clusters in the lower range of the runtime distribution, far outperforming heavier diffusion (namely DTE-Gaussian, DTE-InverseGamma) and kernel-based methods (namely KDE). Its compact and stable distribution across large datasets highlights its consistent computational advantage. These results demonstrate that TCCM maintains both scalability and practicality for real-world, high-volume anomaly detection deployments.

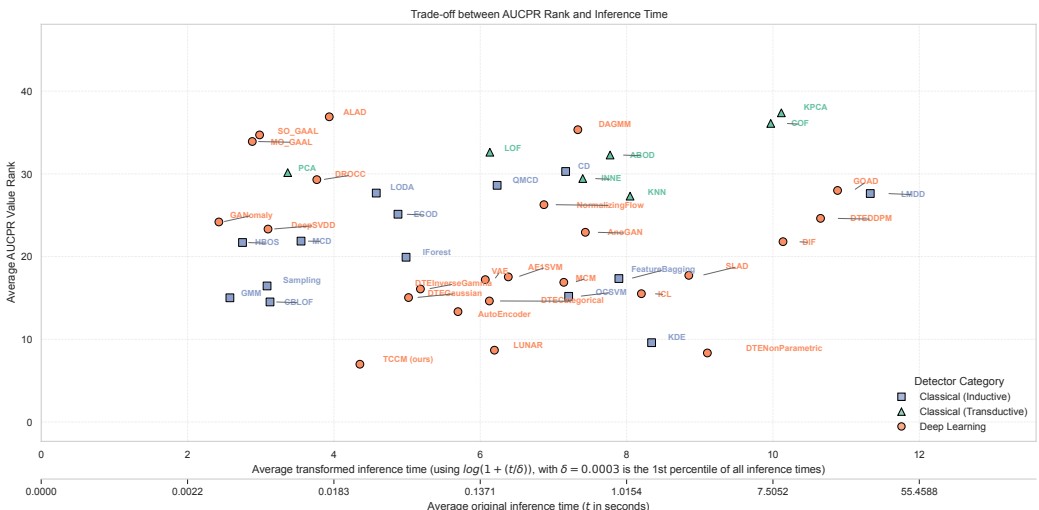

Figure 8: Distribution of *average inference time* (transformed with $\log\left(1 + \frac{t}{\delta}\right)$ to achieve better visualization, with $\delta$ is the 1st percentile of all all inference times $t$) vs. *average AUPRC rank* across all 45 anomaly detection methods. The ticks corresponding to the original average inference time are also displayed underneath. TCCM achieves the best balance between inference speed and detection accuracy, outperforming both slow but accurate (e.g., DTE-Nonparametric, KDE) and fast but less accurate (e.g., GMM, CBLOF, Sampling) methods.

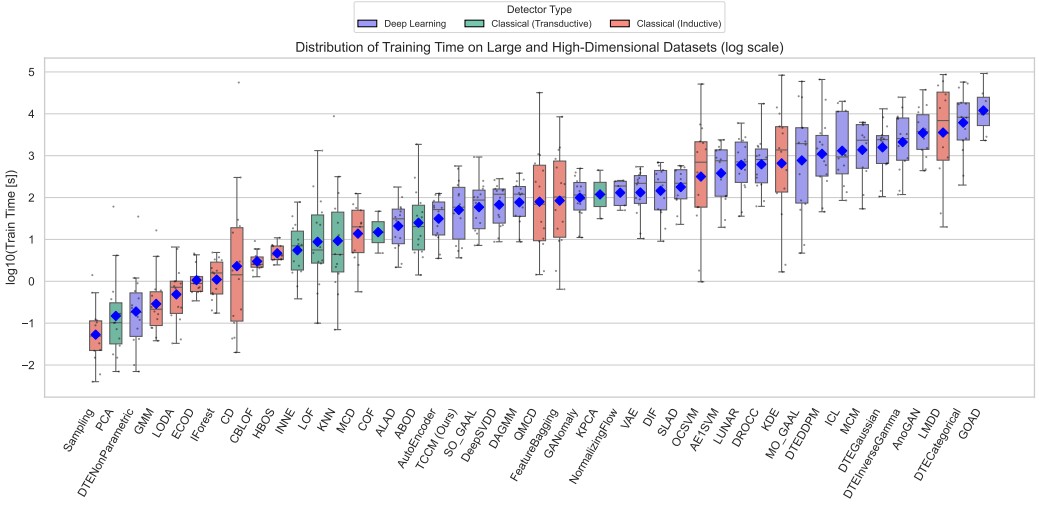

Figure 9: Distribution of training times (log-scale) on 14 large and high-dimensional datasets across all 45 anomaly detectors. TCCM achieves one of the lowest training times among deep learning models, comparable to simple architectures such as AutoEncoder and DeepSVDD, while significantly faster than most other deep learning methods. Some classical models (e.g., KDE, LMDD) show much higher variability and longer training durations.

**Discussion on Ultra-Large-Scale Scenarios.** While our experiments already include a wide spectrum of realistic datasets—with 14 datasets exceeding 10,000 samples and 6 high-dimensional datasets—two cases are particularly noteworthy: *donors* (619K samples, 10 dimensions) and *census* (299K samples, 500 dimensions). On these datasets, TCCM completes inference in merely **1.05s** and **3.02s**, respectively, whereas the most accurate competitor (DTE-NonParametric) requires **476.60s** and **48,942.85s**. This striking difference (3–4 orders of magnitude) highlights the practical scalability of TCCM in both sample size and feature dimensionality. Furthermore, several conventional and

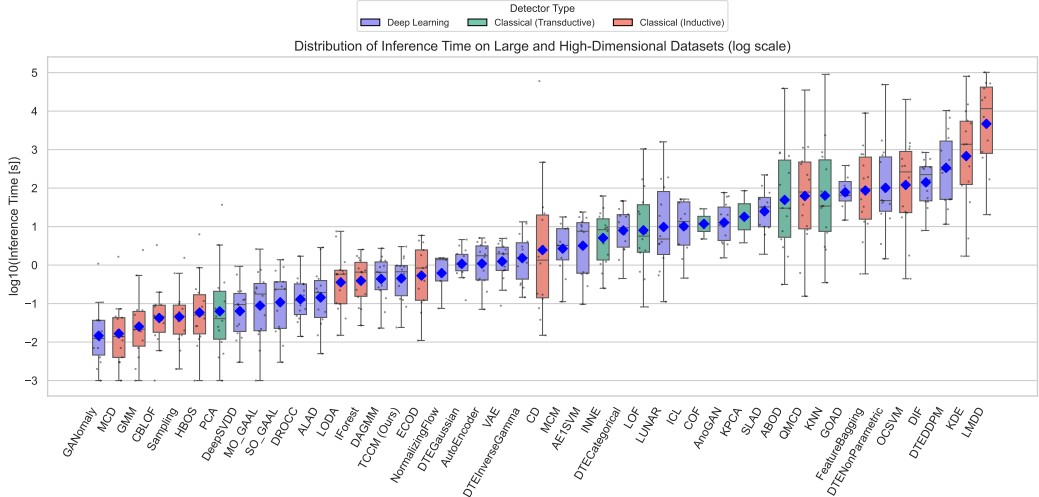

Figure 10: Distribution of inference times (log-scale) on 14 large and high-dimensional datasets. TCCM ranks among the fastest deep learning methods, close to methods such as ALAD and DROCC, but far more accurate. In contrast, diffusion-based baselines (i.e., DTE-NonParametric, DTE-DDPM) occupy the slowest end of the spectrum, illustrating TCCM's superior efficiency for large-scale inference.

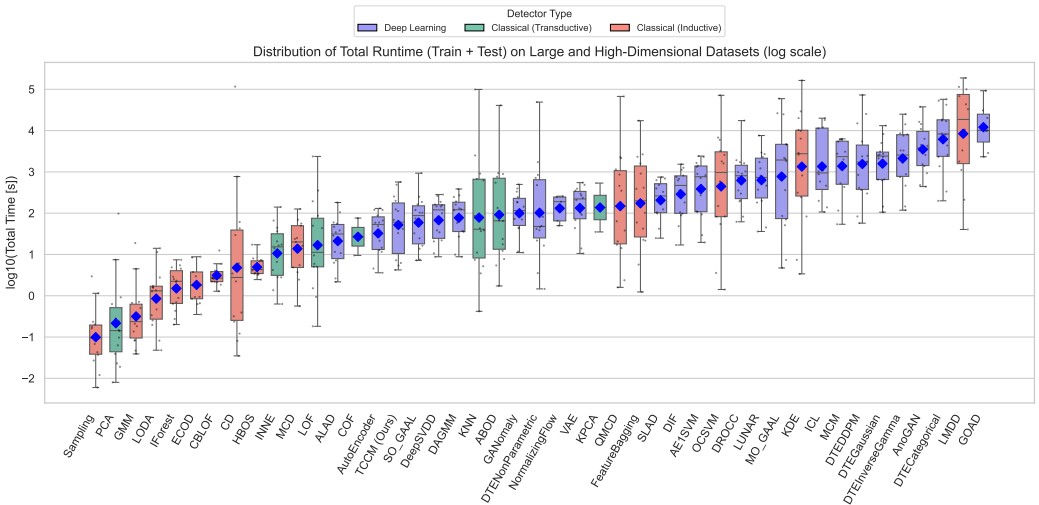

Figure 11: Distribution of total runtimes (training + inference, log-scale) across the 14 large and high-dimensional datasets. TCCM demonstrates one of the best efficiency–accuracy trade-offs within the deep learning category, remaining among the overall fastest detectors. Its compact runtime distribution contrasts sharply with heavier diffusion (namely DTE-Gaussian, DTE-InverseGamma) and kernel-based methods (namely KDE), confirming its scalability and deployability in high-volume environments.

deep baselines fail to process such large-scale inputs within reasonable resource constraints (e.g., memory overflow or exceeding 72h runtime; see Tables 10 and 12). Although evaluating on tens of millions of samples remains an exciting direction for future work, our current results already provide compelling evidence that TCCM is well-suited for real-world, large-scale anomaly detection deployments.

## D.3 Ablation Studies and Sensitivity Analysis: Full Results and Analysis

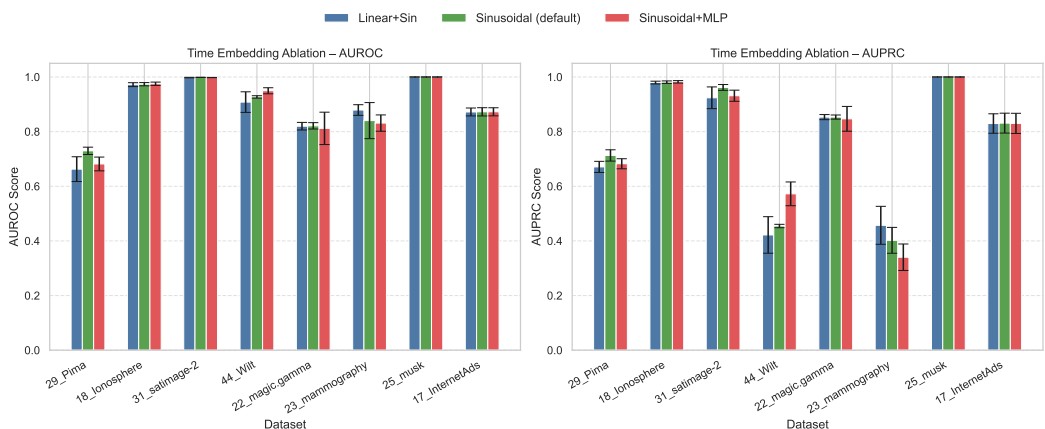

Figure 12: Ablation Study on Time Embedding Methods. We compare three different time embedding strategies used in our flow-based anomaly detection model: *Linear+Sin*, *Sinusoidal (default)*, and *Sinusoidal+MLP*, across eight representative datasets spanning four categories: *Small* (29_Pima, 18_Ionosphere), *Medium* (31_satimage-2, 44_Wilt), *Large* (22_magic.gamma , 23_mammography), and *High-dimensional* (25_musk, 17_InternetAds). The figure shows AUROC (left) and AUPRC (right) scores on the y-axis versus dataset names on the x-axis. Bars are grouped by embedding method and include standard deviation as error bars. Results show that our model is robust across all embedding types, with the default *Sinusoidal* embedding generally offering strong and stable performance.

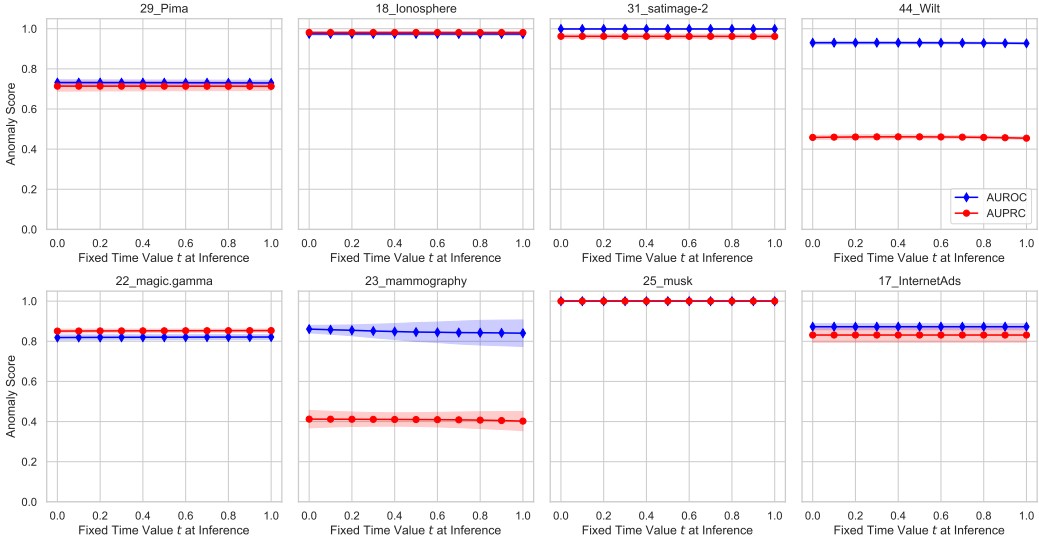

Figure 13: Sensitivity Analysis on Time Value at Inference. We evaluate the sensitivity of our model to different fixed time inputs $t \in [0.0, 1.0]$ at inference across four categories of datasets: *Small* (29_Pima, 18_Ionosphere), *Medium* (31_satimage-2, 44_Wilt), *Large* (22_magic.gamma , 23_mammography), and *High-dimensional* (25_musk, 17_InternetAds). Each plot shows the average AUROC (blue) and AUPRC (red) across 5 random seeds, with individual points marked on each curve. Shaded regions indicate one standard deviation. The **x-axis** represents the fixed value of time $t$, while the **y-axis** reports the detection performance (AUROC or AUPRC). The results demonstrate that our method is insensitive to the specific choice of $t$. Other datasets show similar behavior and are omitted for brevity.

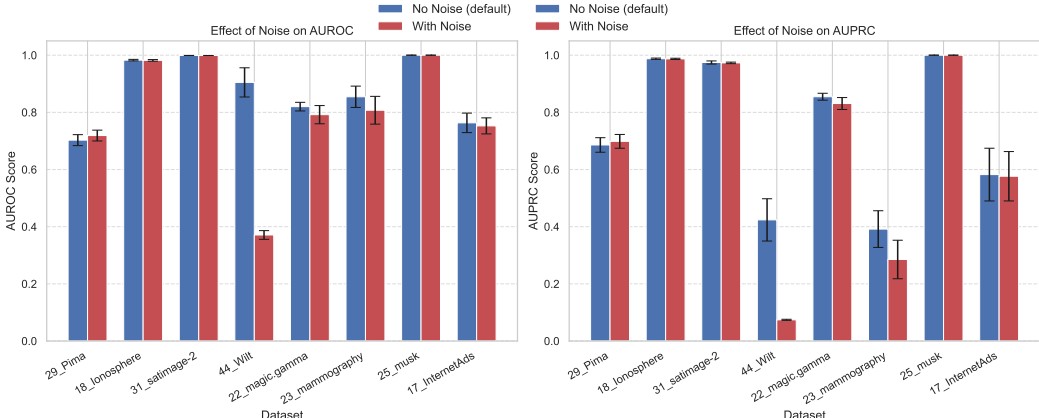

Figure 14: Ablation Study on the Effect of Injecting Noise during Training. We compare the anomaly detection performance (AUROC and AUPRC) of TCCM trained **with** and **without noise** perturbation. We report results across 8 representative datasets spanning four categories (small, medium, large, and high-dimensional). Each bar shows the average score over 5 random seeds, with error bars indicating standard deviation. **Key findings:** (1) On most datasets, adding noise during training does not significantly impact performance; (2) However, in some cases (e.g., `Wilt`, `mammography`), injecting noise leads to a substantial drop in AUROC and/or AUPRC. This indicates that noise injection, while helpful in diffusion based generative modeling, may hinder learning in deterministic anomaly detection tasks.

**Study 1: Time Embedding Variants Used in TCCM.** We consider three different time embedding strategies within the TCCM architecture, each representing a trade-off between simplicity and expressiveness:

- **Linear + Sin:** This basic approach applies a single linear layer followed by a sine transformation to the scalar time input $t$. It is defined as $\phi(t) = \sin(Wt + b)$, where $W$ and $b$ are learnable parameters. This encoding is computationally efficient and empirically fast to converge, making it suitable for lightweight applications.

- **Sinusoidal (default):** Inspired by the positional encoding in Transformers, this method maps time to a fixed set of sinusoidal functions at different frequencies. It is defined as

$$\phi(t) = [\sin(\omega_1 t), \cos(\omega_1 t), \dots, \sin(\omega_d t), \cos(\omega_d t)],$$

where frequencies $\omega_i$ are logarithmically spaced. This embedding captures richer periodic structure without additional learnable parameters.

- **Sinusoidal + MLP:** To enhance the expressiveness of the sinusoidal embedding, we append a two-layer feedforward MLP to it. This allows the model to learn nonlinear combinations of the sinusoidal basis, which is often beneficial when modeling more complex dynamics. However, it introduces more parameters and increases training time.

All three variants are seamlessly plugged into the same backbone network, differing only in the time embedding module. In our experiments, we observe consistent performance across them, while the sinusoidal embedding offers a good balance between performance and simplicity.

**Study 2: Sensitivity to Fixed Time $t$ during Inference.** In the TCCM framework, the final anomaly score is computed based on the model output at a specific time $t$, typically fixed to $t = 1.0$ during inference. To assess the robustness of our method to the choice of $t$, we conduct a sensitivity analysis by varying $t$ uniformly in the range $[0.0, 1.0]$ and measuring the performance in terms of AUROC and AUPRC on eight representative datasets.

For each dataset, we fix $t$ to different values and compute the anomaly score as $S(\boldsymbol{x}) = \| f_{\boldsymbol{\theta}}(\boldsymbol{x}, t) + \boldsymbol{x} \|_2$, where $f_{\boldsymbol{\theta}}(\boldsymbol{x}, t)$ is the predicted vector field. This formulation relies on the observation that, for normal samples, the model learns to approximate $f_{\boldsymbol{\theta}}(\boldsymbol{x}, t) \approx -\boldsymbol{x}$, such that the residual becomes small. Anomalous samples, being out-of-distribution, typically incur larger residuals.

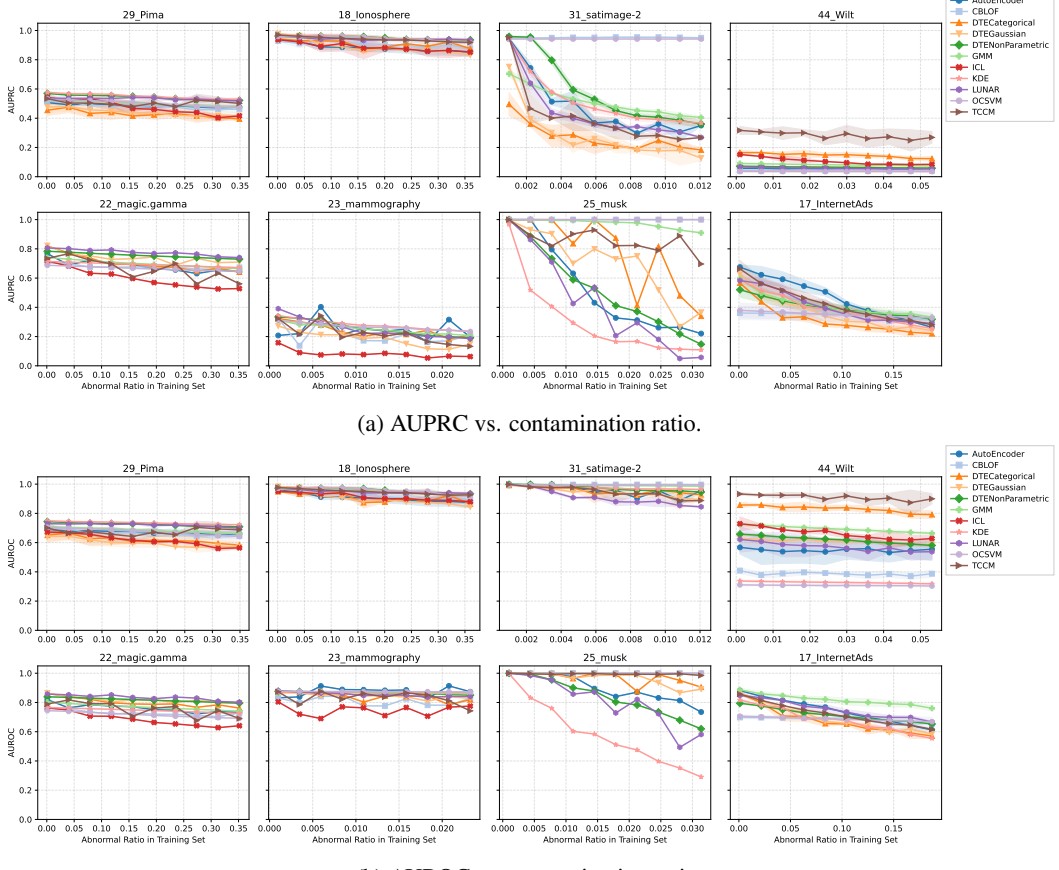

(a) AUPRC vs. contamination ratio.

(b) AUROC vs. contamination ratio.

Figure 15: Sensitivity to training-set contamination for TCCM and 10 top-performing baselines. For each dataset, we fix the train/test split by using 50% of normal samples for training and progressively inject anomalies into the training set (up to the dataset's natural anomaly ratio). The x-axis denotes the abnormal ratio in training; curves summarize mean and variance across multiple seeds (see legends for methods). Overall, increasing contamination tends to reduce performance for most methods; TCCM remains among the most robust, though degradation can still occur on some datasets. We present AUPRC (top) and AUROC (bottom) separately to improve readability.

The results (see Figure 13) demonstrate that the detection performance is largely invariant to the specific value of $t$, indicating that our method is not sensitive to this hyperparameter. This makes the approach more robust and practical, as it avoids the need for tuning $t$ at inference. The shaded areas in the figure denote standard deviation over five random seeds, further confirming the stability of the results.

**Study 3: Effect of Noise Injection during Training.** To assess the role of stochasticity in our framework, we compare two variants of TCCM: one trained with Gaussian noise injection—motivated by the SDE formulation in Appendix C.1—and another trained deterministically without noise. In the noisy case, input samples are perturbed as $\tilde{\boldsymbol{x}}(t) = \boldsymbol{x} + t\boldsymbol{\epsilon}$ with $\boldsymbol{\epsilon} \sim \mathcal{N}(\boldsymbol{0}, \boldsymbol{I})$, but the model is still supervised to predict the residual vector $-\boldsymbol{x}$. This setup preserves the inductive bias toward contraction while introducing input-level stochasticity during training. In contrast, the deterministic variant trains on unperturbed inputs using the same supervision.

Empirical results, summarized in Figure 14, show that noise injection does not consistently improve performance and in many cases leads to a significant drop in AUROC and AUPRC—particularly for datasets such as `Wilt` and `mammography`. These findings validate our theoretical motivation: while noise injection can enhance sample diversity in generative modeling, anomaly detection—especially

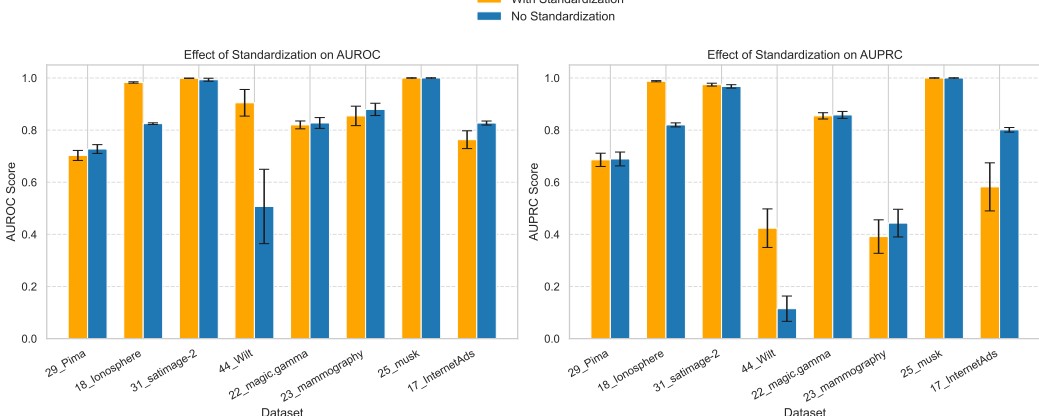

Figure 16: Ablation Study on the Effect of Performing z-score Normalization before Model Training. We compare the anomaly detection performance (AUROC and AUPRC) of TCCM trained **with** and **without noise** z-score normalization. We report results across 8 representative datasets spanning four categories (small, medium, large, and high-dimensional). Each bar shows the average score over 5 random seeds, with error bars indicating standard deviation. **Key findings:** (1) On most datasets, performing z-score normalization does not significantly impact performance; (2) In some cases (e.g., `Ionosphere`, `Wilt`), performing z-score normalization leads to a substantial increase in AUROC and/or AUPRC. This indicates that without normalization, the anomaly scores may be dominated by features with larger scales, leading to suboptimal ranking behavior. However, we note that performing z-score normalization on a few datasets (e.g., `InternetAds`) leads to a drop in AUPRC.

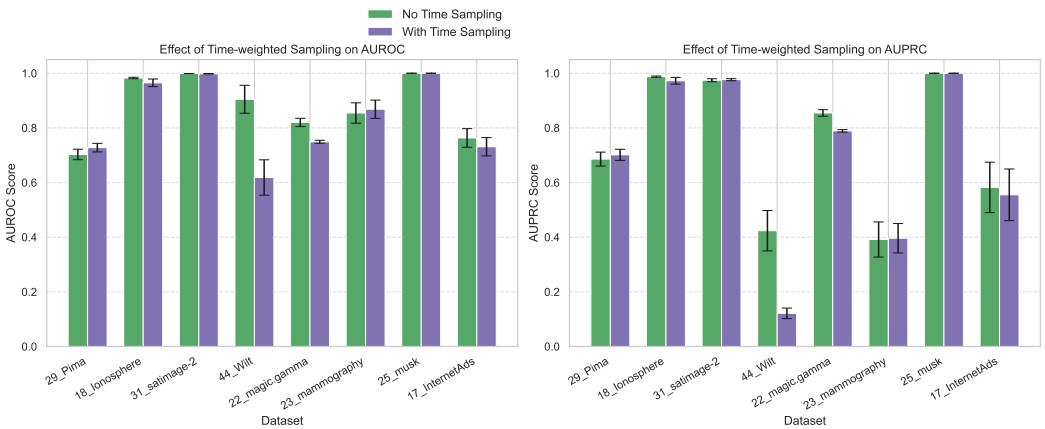

Figure 17: Ablation Study on the Effect of Training with Interpolated Time-Dependent Inputs. We compare the anomaly detection performance (AUROC and AUPRC) of TCCM trained **with** and **without** interpolated inputs $\mathbf{z}_t = t\mathbf{z}$, while keeping all other configurations identical. Results are reported across 8 representative datasets spanning four categories (small, medium, large, and high-dimensional). Each bar shows the average performance over 5 random seeds, with error bars indicating standard deviation. **Key findings:** (1) Introducing time-interpolated samples generally does *not improve* anomaly detection performance, with results remaining approximately unchanged or moderately degraded on most datasets; (2) The degradation is more evident on certain datasets (e.g., `Wilt`, `magic_gamma`), suggesting that interpolated trajectories may introduce undesirable temporal supervision signals; (3) These results empirically support our design choice of directly supervising time-conditioned vector fields at fixed input locations, as discussed in Section 3.

under a semi-supervised regime—relies on preserving the precise structure of normal data. Even mild stochastic perturbations may obscure this structure, weakening the learned vector field. Consequently, our deterministic training procedure yields more stable and effective results for anomaly detection.

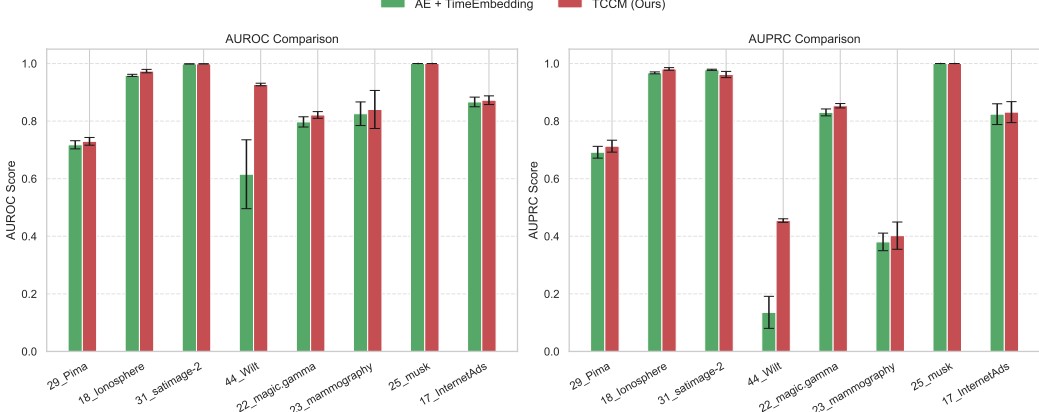

Figure 18: Comparison between TCCM and Autoencoder with Time Embedding (AE+TE) across eight datasets. Bars represent mean AUROC and AUPRC values averaged over 5 random seeds; error bars indicate standard deviation. TCCM consistently performs on par or better, especially on `Wilt` and `magic_gamma`, demonstrating that direct residual learning without reconstruction bottlenecks better captures anomaly-relevant dynamics.

**Study 4: Effect of Contamination in Training Data.** This study examines how varying degrees of contamination—i.e., abnormal samples present in the training set—affect model performance. Unlike the main experimental setup, where **50% of normal data** is used for training and the test set contains the remaining normal and all abnormal samples, here we **fix the train/test split** by randomly dividing both normal and abnormal data in half: 50% of the normals are used for training, and evaluation is conducted on the remaining normals plus half of the anomalies. We then progressively inject additional anomalies into the training set, increasing the abnormal ratio from near-zero up to each dataset's intrinsic anomaly rate. This protocol normalizes the contamination range across datasets and isolates its effect under the semi-supervised assumption.

Empirical results (Figures 15a–15b) compare TCCM with ten top-performing baselines. While TCCM maintains stable AUROC and AUPRC on several datasets, its performance—like that of most methods—deteriorates as contamination increases, sometimes substantially. This degradation is particularly evident when the abnormal ratio approaches the dataset's natural contamination level, suggesting that even small amounts of anomaly leakage can distort learned decision boundaries. Overall, the results indicate that no method is entirely immune to contaminated supervision and reinforce the practical importance of maintaining a clean training set in semi-supervised anomaly detection.

**Study 5: Effectiveness of Conventional Flow Matching on Anomaly Detection.** While in principle conventional Flow Matching can be adapted for anomaly detection, our preliminary experiments indicate two natural strategies yield suboptimal results: (i) using a standard Flow Matching model to reconstruct the input $x$ from a learned trajectory and computing the final-step reconstruction error as the anomaly score; (ii) computing a cumulative reconstruction error across multiple time steps along the trajectory. We implemented both approaches and found them to be worse than our proposed TCCM in terms of both detection accuracy and inference efficiency. In particular, trajectory simulation requires numerical integration and multiple model evaluations, incurring high computational cost at inference time. In contrast, TCCM performs anomaly scoring with a single forward pass at a fixed time step, offering both speed and accuracy advantages.

**Study 6: Effect of Feature Normalization.** In practice, we apply z-score normalization (zero mean, unit variance) to all features before training and inference, which aligns the origin with the center of the normal data distribution. To further evaluate the impact of normalization, we conduct an ablation study comparing TCCM trained **with** and **without** z-score normalization. Results across eight representative datasets (see Figure 16) show that normalization does not significantly affect performance on most datasets but leads to substantial gains in some cases (e.g., `Ionosphere`, `Wilt`). This suggests that without normalization, features with larger scales may dominate the anomaly score,

degrading ranking quality. On a few datasets (e.g., `InternetAds`), normalization slightly reduces AUPRC, indicating dataset-specific effects. Overall, these findings demonstrate that normalization generally improves robustness and provides a principled justification for contracting toward the origin in our framework.

**Study 7: Effect of Time-Interpolated Inputs.**  A key design choice in TCCM is to supervise the time-conditioned vector field directly at fixed input locations, without using time-interpolated samples. To examine whether incorporating interpolated inputs (i.e., $\mathbf{z}_t = t\mathbf{z}$) influences performance, we conduct an ablation study comparing models trained **with** and **without** time interpolation. Results across eight representative datasets (see Figure 17) show that introducing interpolated samples generally does *not improve* anomaly detection performance, with results remaining approximately unchanged or moderately degraded on most datasets. The degradation is more evident on certain datasets (e.g., `Wilt`, `magic_gamma`), suggesting that interpolated trajectories may introduce undesirable or redundant temporal supervision signals, which can interfere with the learning of stable contraction dynamics. Overall, these findings empirically support our design choice of training with fixed inputs and time-conditioned supervision, confirming that TCCM effectively captures temporal dependencies without requiring explicit trajectory interpolation.

**Study 8: Distinction Between TCCM and Autoencoder with Time Embedding (AE+TE).**  A remaining concern pertains to the conceptual distinction between TCCM and an autoencoder (AE) architecture applied to time-augmented data. While both methods may take the same input form $[\mathbf{x}, \text{Embed}(t)]$, their *training objectives*, *architectural principles*, and *learned representations* differ fundamentally.

**Conceptual Comparison.** Autoencoders aim to minimize a reconstruction loss (e.g., $\|\hat{\mathbf{z}} - \mathbf{z}\|_2^2$), learning to reproduce the input itself. In contrast, TCCM learns a *time-conditioned velocity field* $f_\theta([\mathbf{z}, \text{Embed}(t)])$ that is explicitly supervised toward a fixed contraction direction ($-\mathbf{z}$). This distinction fundamentally alters both the optimization target and the semantics of the learned mapping: TCCM predicts the *instantaneous contraction dynamics* of the data manifold, rather than reconstructing input values. Consequently, the model operates under the framework of *vector field learning*, akin to score-based diffusion or flow-matching methods, not under the reconstruction paradigm of autoencoders.

**Architectural Comparison.** TCCM predicts the residual vector field directly in the input space using a 3-layer MLP *without* bottleneck compression, maintaining full dimensionality throughout. In contrast, the AE+TE baseline employs a symmetric encoder–decoder architecture with a latent bottleneck layer, formulated as:

- **Encoder:** Linear(input_dim + time_embed_dim $\rightarrow$ 256) $\rightarrow$ ReLU $\rightarrow$ Linear(256 $\rightarrow$ bottleneck_dim) $\rightarrow$ ReLU

- **Decoder:** Linear(bottleneck_dim $\rightarrow$ 256) $\rightarrow$ ReLU $\rightarrow$ Linear(256 $\rightarrow$ input_dim + time_embed_dim)

This bottleneck compression can discard anomaly-related signals, particularly in early training, whereas TCCM preserves feature-level information and learns residual dynamics directly.

**Experimental Results.** We compare both models on eight representative datasets (see Figure 18), spanning small, medium, large, and high-dimensional settings. TCCM consistently matches or outperforms AE+TimeEmbedding in both AUROC and AUPRC metrics. Notably, the AE+TE baseline exhibits pronounced performance degradation on datasets such as `Wilt` and `magic_gamma`, confirming that its reconstruction-oriented learning objective is less suited for capturing contraction-based anomalies. These results demonstrate that the advantages of TCCM arise not from architectural complexity but from its fundamentally different learning principle.

**Conclusion.** Both empirically and conceptually, TCCM is *not* an autoencoder. Its flow-inspired supervision, residual prediction mechanism, and non-bottleneck design collectively enable it to model anomaly-relevant dynamics more effectively than reconstruction-based alternatives.

### D.4 Empirical Studies on Robustness and Interpretability

#### D.4.1 Empirical Studies on Robustness

Although has been shown theoretically, we provide an empirical study on robustness here. By following (Bergman and Hoshen, 2020), we utilize PGD (Madry et al., 2017) to create adversarial examples, aiming to make anomalies appear like normal instances (or make normal instances look like anomalies). We measure the increase of false negative rate (i.e., the decrease of anomaly score) or false positive rate (i.e., the increase of anomaly score) on the adversarial examples. To make sure that the attacks are non-trivial, we must limit the allowed budget to use.

**Experiment Setup.** The experiment is conducted on a suite of synthetic datasets generated from two disjoint Gaussian mixture models. Details of the dataset construction are provided in Table 2. In line with the 65–95–99.7 rule for standard normal distributions, this setup largely satisfies the assumptions of Proposition 5 (namely Proposition 2 in the main paper), while enabling us to evaluate the robustness of the proposed TCCM method. The training set consists of 5,000 samples randomly drawn from the mixture distribution $\mathbb{P}$. The test set comprises 4,000 samples from $\mathbb{P}$ and 1,000 samples from $\mathbb{Q}$, resulting in an anomaly ratio of 0.2. Following our experimental setup used in ADBench, both training and test sets are standardized prior to attack. We consider two types of attacks: 1) False positive attack, where normal samples are perturbed to appear anomalous; 2) False negative attack, where anomalous samples are perturbed to resemble normal data. For the PGD-based attack setup, we evaluate 30 levels of perturbation budgets, $\epsilon \in \{0.1, 0.2, 0.3 \ldots, 2.8, 2.9, 3.0\}$, under the $L_\infty$ norm. Each attack is performed with a step size of 0.01 and a maximum of $\lceil 200 \cdot \epsilon \rceil$ iterations. It is worth noting that, given the data is standardized, a perturbation budget of $\epsilon = 3$ corresponds to a substantial shift in feature space. For both attack types, we track how AUROC and AUPRC evolve with increasing perturbation strength. All experiments are independently repeated five times using different random seeds to ensure statistical robustness.

Table 2: Specifications of synthetic data for robustness verification. $I_d$ is an identity matrix with size $d$. The normal data is sampled from a GMM with three modes, where $\mu_1 = -3 \times \mathbf{1}_d$, $\mu_2 = \mathbf{0}_d$, and $\mu_3 = 3 \times \mathbf{1}_d$. The anomaly data is sampled from a two-mode GMM, where $\nu_1 = -9 \times \mathbf{1}_d$ and $\nu_1 = 9 \times \mathbf{1}_d$. The experiments are performed across 5 different dimensionalities, i.e., $d \in \{2, 10, 20, 50, 100\}$.

| **Datasets** | Normal $\mathbb{P}$ | Anomaly $\mathbb{Q}$ |
|---|---|---|
| *Robustness* | $\sum_{r=1}^{3} \frac{1}{3}\mathcal{N}(\mu_r,\, I_d)$ | $\sum_{s=1}^{2} \frac{1}{2}\mathcal{N}(\nu_s,\, I_d)$ |

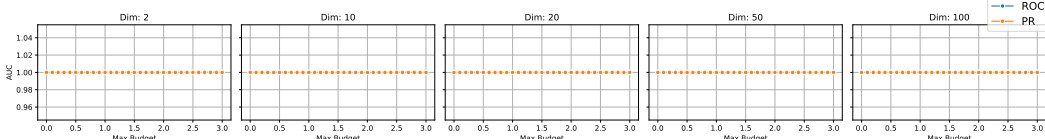

Figure 19: Results of false **negative** attacks (attack on *anomaly* samples) on synthetic GMM-to-GMM shift datasets across five different dimensionalities. The horizontal axis represents the maximum perturbation budget (measured in the $L_\infty$ norm), while the vertical axis indicates the area under the curve (AUC) value.

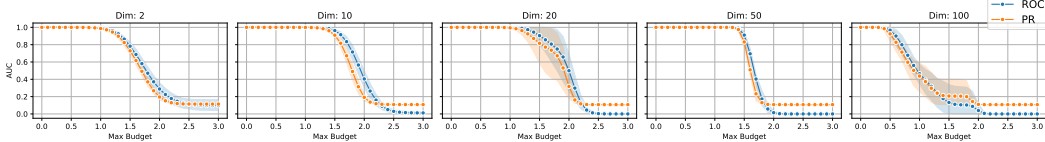

Figure 20: Results of false **positive** attacks (attack on *normal* samples) on synthetic GMM-to-GMM shift datasets across five different dimensionalities. The horizontal axis represents the maximum perturbation budget (measured in the $L_\infty$ norm), while the vertical axis indicates the area under the curve (AUC) value.

The results are presented in Figure 19 and Figure 20, with baseline AUROC and AUPRC scores (i.e., before attack, when the maximum perturbation budget is zero) also indicated for reference. As shown, TCCM is both theoretically justified and empirically validated to be robust under the GMM-to-GMM shift setting. It consistently achieves the highest AUROC and AUPRC across all random seeds and dimensionalities, effectively detecting anomalous samples from $\mathbb{Q}$ in all cases. TCCM demonstrates strong robustness against false negative attacks: both AUROC and AUPRC remain at 1.0, regardless of the perturbation strength. This indicates that adversarial perturbations fail to disguise anomalous inputs as normal. In the case of false positive attacks, TCCM also exhibits a notable degree of robustness. As normal samples are gradually perturbed away from their original distribution, TCCM maintains high AUROC and AUPRC values—particularly up to perturbation levels equivalent to one standard deviation of the standardized data. The only exception occurs in high-dimensional settings (e.g., $d = 100$), where performance slightly degrades. These results suggest that TCCM learns a compact and stable representation of normality, enabling it to ignore semantically meaningless variations within a reasonable margin.

### D.4.2 Empirical Studies on Interpretability

To further validate the reliability of TCCM's feature-level importance scores, we conduct a controlled synthetic experiment designed to quantitatively assess whether the learned residual vector field can accurately identify the features responsible for anomalies. This study complements the qualitative analyses in the main paper by providing direct empirical evidence of the model's intrinsic interpretability.

**Experimental Design.** We construct a well-controlled anomaly detection task based on a Gaussian Mixture Model (GMM) with known anomalous dimensions, enabling precise evaluation of whether TCCM attributes anomalies to the truly perturbed features.

- Normal samples: Drawn from a standard multivariate Gaussian $\mathcal{N}(0, \mathbf{I})$.
- Anomalous samples: Generated from a 3-component GMM:
  - Component 1: 1 dimension shifted,
  - Component 2: 2 dimensions shifted,
  - Component 3: 3 dimensions shifted.
- Shift magnitude: Each shifted feature is perturbed by a random offset uniformly sampled from the range [15, 20].
- Input dimensions: We vary $d \in \{5, 10, 15, 20, 25\}$.
- Training: The model is trained exclusively on normal samples.
- Evaluation: Both anomaly detection and feature-level explanation are assessed on the combined test set.

**Evaluation Metrics.** We employ two complementary metrics that do not rely on any external explainer:

- **Exact Match:** The proportion of anomalies for which the predicted top-$k$ features *exactly* coincide with the ground-truth anomalous dimensions, where $k$ equals the number of shifted dimensions per sample ($k \in \{1, 2, 3\}$).
- **Jaccard Index:** The average intersection-over-union (IoU) between predicted and true anomalous dimensions across all anomalous samples.

Both metrics are derived directly from the model's built-in residual vector field, computed as $\|[f_\phi([\mathbf{x}, \mathrm{Embed}(t)]) + \mathbf{x}]\|_2$, as defined in Eq. 5 of the main paper. This ensures that interpretability is evaluated based on the model's internal reasoning rather than post hoc approximations.

**Results and Discussion.** The outcomes in Table 3 show that TCCM consistently and accurately identifies the ground-truth anomalous features across all tested dimensionalities.

The near-perfect ExactMatch and Jaccard scores confirm that TCCM's residual velocity field yields faithful, fine-grained feature-level attributions. Crucially, this interpretability arises *intrinsically* from

Table 3: Quantitative evaluation of explanation accuracy on synthetic GMM anomalies.

| Setting | ExactMatch | Jaccard | AUROC | AUPRC |
|---------|-----------|---------|-------|-------|
| 5D | 1.000 | 1.000 | 1.000 | 1.000 |
| 10D | 1.000 | 1.000 | 1.000 | 1.000 |
| 15D | 1.000 | 1.000 | 1.000 | 1.000 |
| 20D | 0.996 | 0.998 | 1.000 | 1.000 |
| 25D | 0.996 | 0.997 | 1.000 | 1.000 |

the model's formulation—no auxiliary explanation method (e.g., SHAP or LIME) is required. The residual components directly encode each feature's contribution to the contraction mismatch, offering a transparent and actionable view of the decision process. This property enables practitioners in domains such as fraud analysis, medical diagnostics, and industrial monitoring to understand not only *which* samples are anomalous but also *why*.

## D.5 Statistical Tests

To rigorously assess whether the performance differences between TCCM and competing methods are statistically significant, we conduct non-parametric statistical tests on their rankings across 47 datasets. For each method, we compute its average AUPRC and AUROC rankings over five random seeds on each dataset. Given the multi-method, multi-dataset nature of this evaluation, traditional pairwise tests are inadequate due to increased risk of Type I error. Hence, we follow the protocol proposed by Demšar (2006), which recommends a two-stage procedure:

- First, we apply the Friedman test (Friedman, 1937), a non-parametric alternative to repeated-measures ANOVA, to determine whether there is any statistically significant difference in performance rankings among all methods.

- If the null hypothesis is rejected, we proceed with the Nemenyi post hoc test (Nemenyi, 1963), which compares all classifiers pairwise. Two methods are considered significantly different if their average ranks differ by at least the critical difference (CD).

Compared to alternatives like the Wilcoxon-Holm method (García et al., 2010), which performs pairwise tests between a control method and others with Holm correction, the Nemenyi test is more conservative—it simultaneously controls the family-wise error rate across all pairwise comparisons, not just against a reference. While this often results in fewer significant findings, it provides a stronger guarantee against false positives, especially important in benchmark settings involving many methods.

We report the results using critical difference diagrams (see Figure 21 and 22). For AUPRC, the Nemenyi test indicates that there are no statistically significant differences among the top-performing group, which includes **TCCM** (ranked 5.8), DTE-NonParametric, LUNAR, KDE, AutoEncoder, ICL, CBLOF, DTE-Categorical, GMM, and OCSVM. For AUROC, the top group includes **TCCM** (ranked 5.7), DTE-NonParametric, LUNAR, KDE, AutoEncoder, ICL, CBLOF, DTE-Categorical, GMM, and Sampling. Although TCCM achieves the best average rank in both metrics, the conservative nature of the Nemenyi test explains the lack of statistically significant superiority. Nonetheless, TCCM consistently ranks at the top, reinforcing its robustness and broad effectiveness across diverse datasets.

Table 4: Overall comparison of top-performant anomaly detection algorithms (with top-4 performance in terms of AUROC and AUPRC) across four key dimensions.

| Algorithm | Accuracy | Scalability | Explainability | (Provable) Robustness |
|-----------|----------|-------------|----------------|----------------------|
| **TCCM** | ✓ | ✓ | ✓ (feature contribution) | ✓ |
| DTE-NonParametric (Livernoche et al., 2023) | ✓ | ✗ (slow inference) | ✓ (reconstruction) | ✗ |
| LUNAR (Goodge et al., 2022) | ✓ | ✗ (slow training) | ✗ | ✗ |
| KDE (Latecki et al., 2007) | ✓ | ✗ (slow training) | ✓ (density) | ✗ |

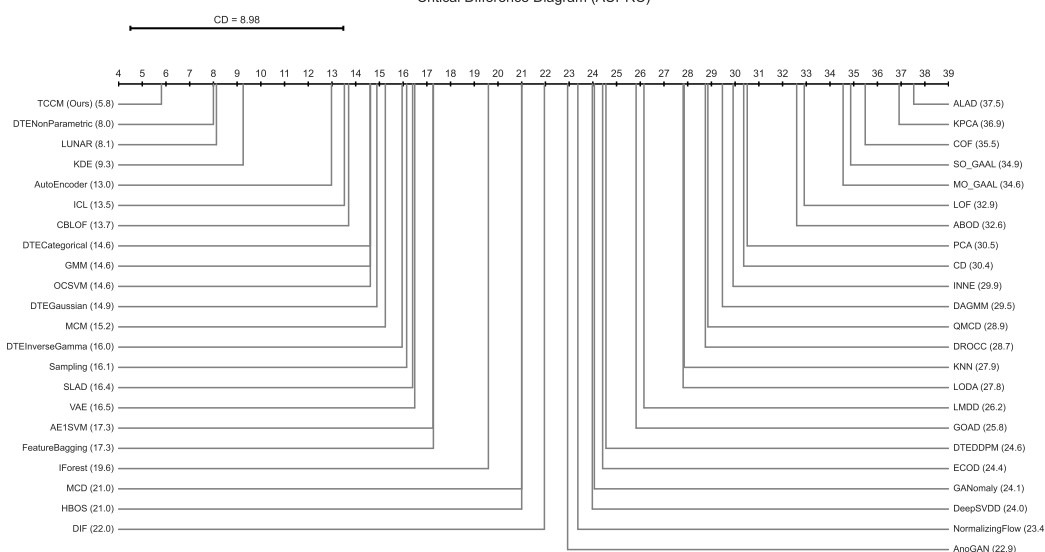

Figure 21: Critical difference (CD) diagram illustrating statistical rank comparisons of the 45 anomaly detection methods based on their AUPRC performance across 47 datasets. Each method is ranked by its mean AUPRC over five random seeds. The CD value, computed via the Nemenyi post-hoc test at significance level $0.05$, indicates the minimum difference in average rank that is statistically significant. Notably, TCCM (ranked 5.8 on average) is part of the top-performing group including DTE-NonParametric, LUNAR, KDE, AutoEncoder, ICL, CBLOF, DTE-Categorical, GMM, and OCSVM.

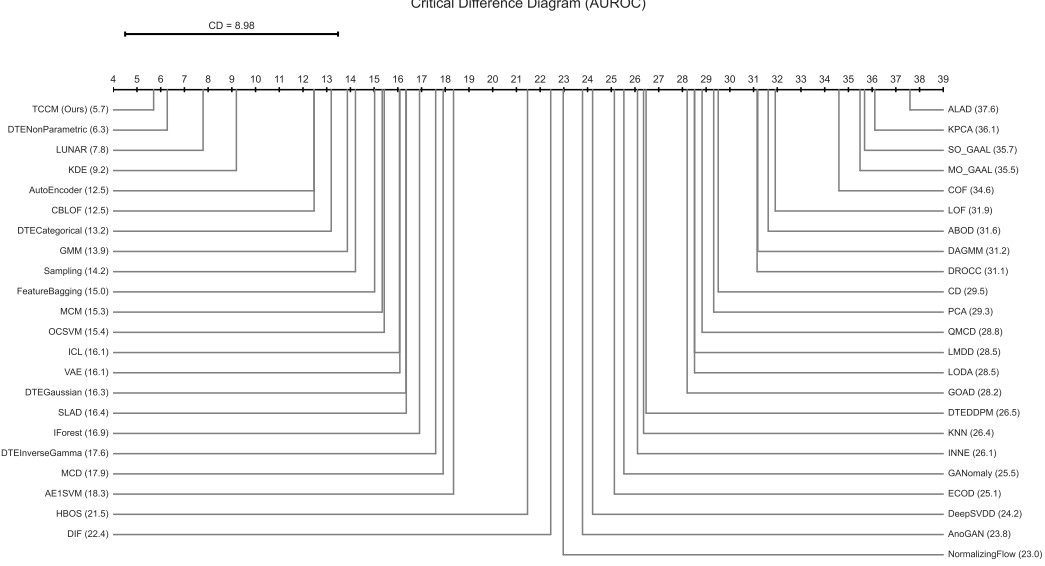

Figure 22: Critical difference (CD) diagram illustrating statistical rank comparisons of the 45 anomaly detection methods based on their AUROC performance across 47 datasets. Each method is ranked by its mean AUROC over five random seeds. The CD value is derived using the Nemenyi post-hoc test with a significance level of $0.05$. TCCM (with an average rank of 5.7) belongs to the top-performing group, which includes DTE-NonParametric, LUNAR, KDE, AutoEncoder, ICL, CBLOF, DTE-Categorical, GMM, and Sampling.

### D.6 Limitations and Broader Impacts

**Limitations**. While TCCM achieves state-of-the-art performance with relatively low computational cost, we outline three limitations that offer promising directions for future research. *(1) Data Modality:* As the first work on adapting flow-matching modeling to anomaly detection, our study focuses exclusively on tabular data. Extending TCCM to other data modalities, such as vision (Liu et al., 2024), time series (Blázquez-García et al., 2021), or graph-structured data (Akoglu et al., 2015; Li et al., 2024a), is an exciting avenue for exploration. *(2) Neural Architecture:* To achieve maximum efficiency, TCCM models the velocity field using a multilayer perceptron. This design choice raises an open question: could more sophisticated neural architectures, such as ResNet (He et al., 2016), further improve performance? *(3) Real-World Usability:* Our evaluation is conducted on ADBench (Han et al., 2022), following the common practice in the anomaly detection research community. Exploring TCCM's effectiveness in real-world high-stakes domains, e.g., finance or healthcare, under more dynamic and complex conditions would be valuable.

**Broader Impacts**. While the TCCM methodology, as presented, is foundational research focused on advancing anomaly detection in tabular data, its limitations inherently shape its potential broader impacts and demarcate avenues for future work that could address these implications. The current focus on tabular data, while demonstrating significant methodological advancements, means that the direct applicability to other prevalent data types like images, time series, or complex graph structures is not yet established. The broader societal impact of anomaly detection often lies in these other domains—such as medical imaging analysis, financial transaction monitoring over time, or social network security. Therefore, until TCCM is extended and validated on these diverse data modalities, its positive impact in such critical areas remains a future prospect, and any potential negative impacts from misuse in these unvalidated contexts are purely speculative but warrant caution.

Furthermore, the utilization of a multilayer perceptron (MLP) for the velocity field, chosen for efficiency, may cap the model's capability to discern highly complex patterns compared to more sophisticated architectures. This architectural limitation could influence its broader impact in scenarios demanding exceptional nuance and accuracy, potentially limiting its deployment in safety-critical applications where the cost of a false negative or positive is extremely high. The ethical implications of deploying a system that might not capture the full complexity of a problem due to architectural constraints should be considered as the research progresses.

Lastly, the evaluation of TCCM primarily on the ADBench benchmark, though a standard practice, means its performance characteristics in messy, dynamic, and potentially adversarial real-world environments are not fully known. The broader impact, particularly concerning fairness, robustness to unforeseen data shifts, privacy implications in data-sensitive fields like finance or healthcare, and security against emergent threats, can only be truly assessed through rigorous testing in such operational settings. Without this, the translation of TCCM into systems with significant societal touchpoints should proceed with a clear understanding of these unevaluated risks. Future work addressing these limitations will be crucial in responsibly broadening the positive societal impact of this line of research.

## E   Results under the Inductive Evaluation Setting

To ensure a protocol that is consistent across *all* methods evaluated together, we additionally report results under a unified inductive (semi-supervised) setting, in which training is performed solely on normal samples without access to anomalous data. Most methods in our benchmark—including the majority of deep learning approaches and many classical baselines—are already inductive by design and thus remain unchanged. For completeness, we *adapt* the following algorithms that were originally formulated as transductive detectors to an inductive training procedure: ABOD, COF, LOF, PCA, KPCA, KNN, and INNE. All other experimental configurations, datasets, and evaluation metrics are identical to those used in the main paper.

### E.1   Effectiveness under the Inductive Setting

Figures 23a and 23b summarize the aggregated rankings of 45 detectors across 47 datasets, based on AUPRC and AUROC, respectively. Each ranking averages over five random seeds.

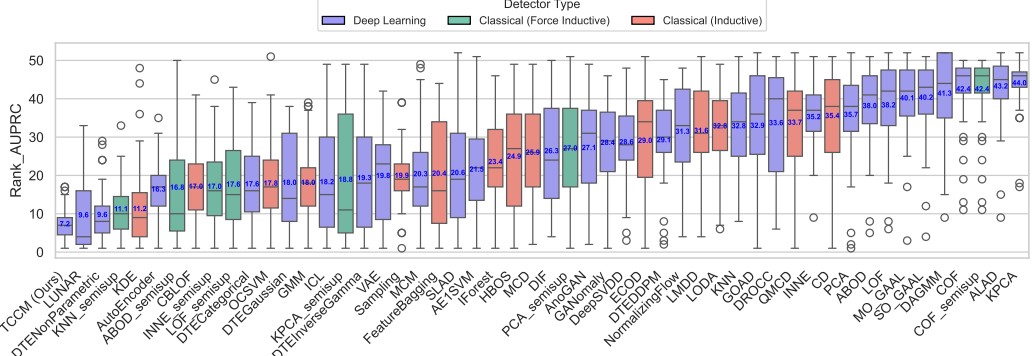

(a) AUPRC ranking distribution across 47 datasets for 45 anomaly detectors under the inductive setting.

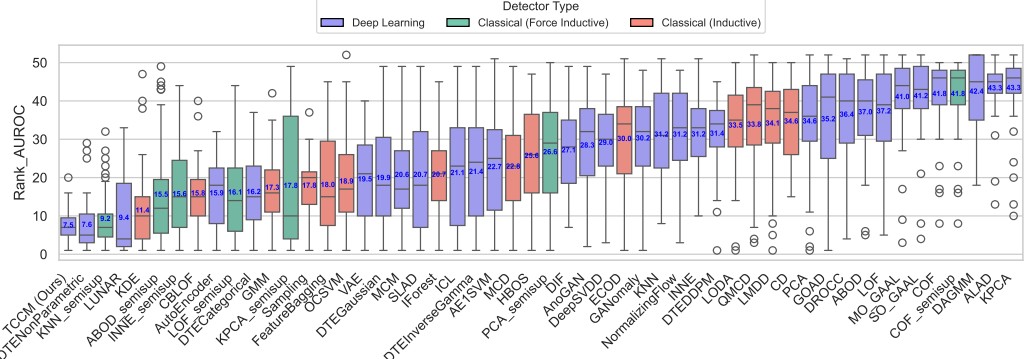

(b) AUROC ranking distribution across 47 datasets for 45 anomaly detectors under the inductive setting.

Figure 23: Updated detector ranking distributions (AUPRC and AUROC) under the inductive (semi-supervised) protocol, where models are trained exclusively on normal data. Medians are indicated by horizontal lines; means are shown as numbers.

**Findings.** The ranking trends for deep learning methods remain highly consistent with the main paper: TCCM continues to rank first overall in both AUPRC and AUROC under the inductive setting, indicating that its performance advantage does not rely on transductive assumptions of other baselines. Other strong baselines (e.g., LUNAR, DTE-NonParametric, KDE) approximately preserve their relative positions.

Notably, several classical methods that we adapted from transductive to inductive exhibit **substantial performance gains**. The improvements are most pronounced for KNN, ABOD, and INNE: for KNN, the average rank in AUPRC improves from 27.9 to 11.1, and in AUROC from 26.4 to 9.2; for ABOD, AUPRC average rank improves from 32.6 to 31.6 and AUROC average rank from 36.1 to 15.5; INNE shows similarly notable gains. These changes indicate that experimental protocol can materially affect certain neighborhood- or structure-based detectors.

## E.2 Scalability, Explainability, and Ablation Analyses

**Scalability.** The scalability conclusions remain aligned with the main paper: TCCM retains its efficiency advantages. Methods that were already inductive (e.g., TCCM, LUNAR, KDE, DTE-NonParametric) are unaffected by the protocol change.

**Interpretability.** The interpretability analysis in Section 5.1 is unchanged, as TCCM's feature-level explanations stem from its model structure.

**Ablation and Sensitivity.** We did not repeat ablation/sensitivity studies under the inductive reformulation, since the algorithms modified here (ABOD, COF, LOF, PCA, KPCA, KNN, INNE) were not part of those studies.

**Why in the Appendix?** We place the inductive-variant results here to keep the main text focused and methodologically consistent: introducing non-canonical inductive variants of originally transductive algorithms into the main tables would complicate the primary comparison without changing our conclusions. The appendix ensures transparency while preserving the clarity of the main results.

In summary, adopting a fully inductive evaluation protocol does not qualitatively change our conclusions. TCCM remains the most effective and scalable detector, with robust performance under semi-supervised training conditions for all baselines.

Table 5: Configuration details for each dataset used in TCCM experiments. To avoid bias from aggressive hyperparameter tuning, we adopt a fixed configuration for all core components (e.g., architecture, time embedding, optimizer) across all datasets. Since our setting is fully unsupervised, we refrain from using label information to optimize hyperparameters. The number of training epochs is adjusted in a dataset-dependent but label-agnostic manner by following the unsupervised internal evaluation strategy in Li et al. (2025b). Note that the backbone architecture is a lightweight 3-layer MLP (two hidden layers), chosen deliberately for efficiency on tabular data; we use the term "deep" in line with common practice to indicate a deep learning-based, end-to-end neural approach rather than architectural depth per se.

| Dataset | Category | Architecture | Time Embedding | Loss Function | Optimizer | #Epochs | Batch Size | Seeds |
|---|---|---|---|---|---|---|---|---|
| census | High-dimensional | MLP (2×256 ReLU) | Sinusoidal (128) | MSE Loss | Adam (lr=0.005) | 5 | 1024 | [0,1,2,3,4] |
| backdoor | High-dimensional | MLP (2×256 ReLU) | Sinusoidal (128) | MSE Loss | Adam (lr=0.005) | 200 | 1024 | [0,1,2,3,4] |
| campaign | High-dimensional | MLP (2×256 ReLU) | Sinusoidal (128) | MSE Loss | Adam (lr=0.005) | 50 | 1024 | [0,1,2,3,4] |
| mnist | High-dimensional | MLP (2×256 ReLU) | Sinusoidal (128) | MSE Loss | Adam (lr=0.005) | 500 | 512 | [0,1,2,3,4] |
| speech | High-dimensional | MLP (2×256 ReLU) | Sinusoidal (128) | MSE Loss | Adam (lr=0.005) | 500 | 512 | [0,1,2,3,4] |
| optdigits | High-dimensional | MLP (2×256 ReLU) | Sinusoidal (128) | MSE Loss | Adam (lr=0.005) | 2000 | 512 | [0,1,2,3,4] |
| SpamBase | High-dimensional | MLP (2×256 ReLU) | Sinusoidal (128) | MSE Loss | Adam (lr=0.005) | 5000 | 512 | [0,1,2,3,4] |
| musk | High-dimensional | MLP (2×256 ReLU) | Sinusoidal (128) | MSE Loss | Adam (lr=0.005) | 5 | 512 | [0,1,2,3,4] |
| InternetAds | High-dimensional | MLP (2×256 ReLU) | Sinusoidal (128) | MSE Loss | Adam (lr=0.005) | 50 | 512 | [0,1,2,3,4] |
| donors | Large | MLP (2×256 ReLU) | Sinusoidal (128) | MSE Loss | Adam (lr=0.005) | 30 | 1024 | [0,1,2,3,4] |
| http | Large | MLP (2×256 ReLU) | Sinusoidal (128) | MSE Loss | Adam (lr=0.005) | 100 | 1024 | [0,1,2,3,4] |
| cover | Large | MLP (2×256 ReLU) | Sinusoidal (128) | MSE Loss | Adam (lr=0.005) | 10 | 1024 | [0,1,2,3,4] |
| fraud | Large | MLP (2×256 ReLU) | Sinusoidal (128) | MSE Loss | Adam (lr=0.005) | 75 | 1024 | [0,1,2,3,4] |
| skin | Large | MLP (2×256 ReLU) | Sinusoidal (128) | MSE Loss | Adam (lr=0.005) | 110 | 1024 | [0,1,2,3,4] |
| celeba | Large | MLP (2×256 ReLU) | Sinusoidal (128) | MSE Loss | Adam (lr=0.005) | 2 | 1024 | [0,1,2,3,4] |
| smtp | Large | MLP (2×256 ReLU) | Sinusoidal (128) | MSE Loss | Adam (lr=0.005) | 2 | 1024 | [0,1,2,3,4] |
| ALOI | Large | MLP (2×256 ReLU) | Sinusoidal (128) | MSE Loss | Adam (lr=0.005) | 100 | 1024 | [0,1,2,3,4] |
| shuttle | Large | MLP (2×256 ReLU) | Sinusoidal (128) | MSE Loss | Adam (lr=0.005) | 200 | 1024 | [0,1,2,3,4] |
| magic.gamma | Large | MLP (2×256 ReLU) | Sinusoidal (128) | MSE Loss | Adam (lr=0.005) | 10 | 1024 | [0,1,2,3,4] |
| mammography | Large | MLP (2×256 ReLU) | Sinusoidal (128) | MSE Loss | Adam (lr=0.005) | 20 | 1024 | [0,1,2,3,4] |
| annthyroid | Medium | MLP (2×256 ReLU) | Sinusoidal (128) | MSE Loss | Adam (lr=0.005) | 2000 | 512 | [0,1,2,3,4] |
| pendigits | Medium | MLP (2×256 ReLU) | Sinusoidal (128) | MSE Loss | Adam (lr=0.005) | 1000 | 512 | [0,1,2,3,4] |
| satellite | Medium | MLP (2×256 ReLU) | Sinusoidal (128) | MSE Loss | Adam (lr=0.005) | 10 | 512 | [0,1,2,3,4] |
| landsat | Medium | MLP (2×256 ReLU) | Sinusoidal (128) | MSE Loss | Adam (lr=0.005) | 6 | 512 | [0,1,2,3,4] |
| satimage-2 | Medium | MLP (2×256 ReLU) | Sinusoidal (128) | MSE Loss | Adam (lr=0.005) | 5 | 512 | [0,1,2,3,4] |
| PageBlocks | Medium | MLP (2×256 ReLU) | Sinusoidal (128) | MSE Loss | Adam (lr=0.005) | 1800 | 512 | [0,1,2,3,4] |
| Wilt | Medium | MLP (2×256 ReLU) | Sinusoidal (128) | MSE Loss | Adam (lr=0.005) | 20 | 512 | [0,1,2,3,4] |
| thyroid | Medium | MLP (2×256 ReLU) | Sinusoidal (128) | MSE Loss | Adam (lr=0.005) | 10 | 512 | [0,1,2,3,4] |
| Waveform | Medium | MLP (2×256 ReLU) | Sinusoidal (128) | MSE Loss | Adam (lr=0.005) | 580 | 512 | [0,1,2,3,4] |
| Cardiotocography | Medium | MLP (2×256 ReLU) | Sinusoidal (128) | MSE Loss | Adam (lr=0.005) | 1 | 512 | [0,1,2,3,4] |
| fault | Medium | MLP (2×256 ReLU) | Sinusoidal (128) | MSE Loss | Adam (lr=0.005) | 5000 | 512 | [0,1,2,3,4] |
| cardio | Medium | MLP (2×256 ReLU) | Sinusoidal (128) | MSE Loss | Adam (lr=0.005) | 2000 | 512 | [0,1,2,3,4] |
| letter | Medium | MLP (2×256 ReLU) | Sinusoidal (128) | MSE Loss | Adam (lr=0.005) | 50 | 512 | [0,1,2,3,4] |
| yeast | Medium | MLP (2×256 ReLU) | Sinusoidal (128) | MSE Loss | Adam (lr=0.005) | 130 | 512 | [0,1,2,3,4] |
| vowels | Medium | MLP (2×256 ReLU) | Sinusoidal (128) | MSE Loss | Adam (lr=0.005) | 20 | 512 | [0,1,2,3,4] |
| Pima | Small | MLP (2×256 ReLU) | Sinusoidal (128) | MSE Loss | Adam (lr=0.005) | 5 | 512 | [0,1,2,3,4] |
| breastw | Small | MLP (2×256 ReLU) | Sinusoidal (128) | MSE Loss | Adam (lr=0.005) | 1 | 512 | [0,1,2,3,4] |
| WDBC | Small | MLP (2×256 ReLU) | Sinusoidal (128) | MSE Loss | Adam (lr=0.005) | 2 | 512 | [0,1,2,3,4] |
| Ionosphere | Small | MLP (2×256 ReLU) | Sinusoidal (128) | MSE Loss | Adam (lr=0.005) | 10 | 512 | [0,1,2,3,4] |
| Stamps | Small | MLP (2×256 ReLU) | Sinusoidal (128) | MSE Loss | Adam (lr=0.005) | 200 | 512 | [0,1,2,3,4] |
| vertebral | Small | MLP (2×256 ReLU) | Sinusoidal (128) | MSE Loss | Adam (lr=0.005) | 25 | 512 | [0,1,2,3,4] |
| WBC | Small | MLP (2×256 ReLU) | Sinusoidal (128) | MSE Loss | Adam (lr=0.005) | 1 | 512 | [0,1,2,3,4] |
| glass | Small | MLP (2×256 ReLU) | Sinusoidal (128) | MSE Loss | Adam (lr=0.005) | 200 | 512 | [0,1,2,3,4] |
| WPBC | Small | MLP (2×256 ReLU) | Sinusoidal (128) | MSE Loss | Adam (lr=0.005) | 6 | 512 | [0,1,2,3,4] |
| Lymphography | Small | MLP (2×256 ReLU) | Sinusoidal (128) | MSE Loss | Adam (lr=0.005) | 3 | 512 | [0,1,2,3,4] |
| wine | Small | MLP (2×256 ReLU) | Sinusoidal (128) | MSE Loss | Adam (lr=0.005) | 20 | 512 | [0,1,2,3,4] |
| Hepatitis | Small | MLP (2×256 ReLU) | Sinusoidal (128) | MSE Loss | Adam (lr=0.005) | 1 | 512 | [0,1,2,3,4] |

Table 6: AUROC results on 12 small datasets, where we compare TCCM to 44 baselines (with 5 independent runs). We report the mean ± std (rank).

| Dataset | TCMM (Ours) | AE | AE-1SVM | ALAD | AnoGAN | DAGMM | DeepSVDD | DIF | DROCC | DTE-Cat | DTE-DDPM | DTE-Gaussian |
|---|---|---|---|---|---|---|---|---|---|---|---|---|
| Pima | 0.735±0.018(4) | 0.698±0.014(17) | 0.668±0.009(23) | 0.528±0.018(41) | 0.695±0.057(18) | NAN | 0.686±0.027(19) | 0.604±0.021(32) | 0.429±0.157(42) | 0.651±0.02(27) | 0.569±0.019(37) | 0.66±0.021(25) |
| breastw | 0.99±0.005(5) | 0.983±0.006(20) | 0.984±0.019(19) | 0.649±0.02(35) | 0.994±0.001(2) | NAN | 0.986±0.004(15) | 0.777±0.065(30) | 0.383±0.315(41) | 0.938±0.01(28) | 0.96±0.01(26) | 0.929±0.023(29) |
| WDBC | 0.993±0.003(3) | 0.993±0.005(3) | 0.98±0.023(24) | 0.615±0.06(41) | 0.99±0.006(12) | NAN | 0.987±0.01(18) | 0.758±0.033(39) | 0.332±0.377(43) | 0.97±0.01(31) | 0.88±0.064(38) | 0.992±0.005(7) |
| Ionosphere | 0.976±0.005(3) | 0.948±0.004(16) | 0.908±0.037(36) | 0.934±0.012(19) | 0.542±0.08(44) | NAN | 0.85±0.042(30) | 0.933±0.007(20) | 0.687±0.167(39) | 0.941±0.011(17) | 0.843±0.029(32) | 0.972±0.004(6) |
| Stamps | 0.935±0.018(8) | 0.926±0.016(14) | 0.944±0.021(4) | 0.597±0.095(41) | 0.741±0.352(35) | NAN | 0.951±0.021(2) | 0.935±0.02(8) | 0.352±0.151(45) | 0.905±0.026(21) | 0.76±0.052(31) | 0.822±0.10(29) |
| vertebral | 0.669±0.072(1) | 0.457±0.026(22) | 0.53±0.041(9) | 0.48±0.071(18) | 0.566±0.104(6) | NAN | 0.418±0.015(36) | 0.541±0.035(8) | 0.435±0.265(29) | 0.527±0.073(10) | 0.418±0.043(31) | 0.518±0.019(11) |
| WBC | 0.986±0.008(8) | 0.979±0.012(16) | 0.975±0.024(21) | 0.585±0.152(40) | 0.991±0.006(4) | NAN | 0.98±0.012(15) | 0.872±0.016(16) | 0.377±0.316(42) | 0.92±0.057(32) | 0.906±0.056(33) | 0.962±0.02(26) |
| glass | 0.903±0.032(2) | 0.748±0.054(27) | 0.763±0.038(22) | 0.501±0.11(44) | 0.694±0.212(32) | NAN | 0.67±0.125(35) | 0.622±0.016(6) | 0.757±0.173(24) | 0.805±0.048(17) | 0.683±0.1(34) | 0.546±0.023(43) |
| WPBC | 0.567±0.013(2) | 0.516±0.024(20) | 0.495±0.033(34) | 0.488±0.067(37) | 0.535±0.063(16) | NAN | 0.499±0.063(32) | 0.471±0.029(43) | 0.474±0.071(42) | 0.548±0.038(10) | 0.537±0.054(15) | 0.487±0.038(38) |
| Lymphography | 0.992±0.008(6) | 0.985±0.019(22) | 0.979±0.028(27) | 0.608±0.166(40) | 0.991±0.008(8) | NAN | 0.97±0.038(32) | 0.899±0.065(38) | 0.434±0.311(42) | 0.986±0.014(21) | 0.971±0.016(18) | 0.981±0.025(25) |
| wine | 0.976±0.01(5) | 0.94±0.025(15) | 0.942±0.033(14) | 0.483±0.156(37) | 0.828±0.125(26) | NAN | 0.825±0.092(27) | 0.665±0.078(33) | 0.427±0.296(40) | 0.957±0.035(13) | 0.629±0.139(35) | 0.896±0.047(24) |
| Hepatitis | 0.831±0.031(6) | 0.831±0.027(6) | NAN | 0.466±0.005(44) | 0.729±0.135(27) | NAN | 0.764±0.069(23) | 0.707±0.056(30) | 0.477±0.206(43) | 0.81±0.036(15) | 0.695±0.073(32) | 0.776±0.035(20) |
| Avg Ranking | 4.42 | 16.5 | 19.82 | 38.5 | 18.33 | NAN | 23.25 | 27.08 | 39.33 | 20.17 | 31.25 | 23.58 |

| Dataset | DTE-IG | DTE-NP | GANomaly | GOAD | ICL | LUNAR | MCM | MO-GAAL | PlanarFlow | SLAD | SO-GAAL | VAE |
|---|---|---|---|---|---|---|---|---|---|---|---|---|
| Pima | 0.62±0.016(30) | 0.742±0.015(2) | 0.594±0.038(36) | 0.385±0.007(44) | 0.651±0.012(27) | 0.727±0.022(6) | 0.721±0.02(9) | 0.363±0.064(45) | 0.71±0.029(13) | 0.558±0.029(39) | 0.425±0.088(43) | 0.681±0.008(20) |
| breastw | 0.689±0.189(33) | 0.987±0.005(14) | 0.951±0.017(27) | 0.721±0.16(32) | 0.965±0.01(25) | 0.989±0.005(8) | 0.986±0.007(15) | 0.04±0.049(45) | 0.971±0.012(24) | 0.99±0.003(5) | 0.23±0.171(44) | 0.99±0.005(5) |
| WDBC | 0.986±0.008(20) | 0.991±0.005(10) | 0.916±0.076(36) | 0.989±0.008(14) | 0.984±0.012(21) | 0.994±0.004(1) | 0.955±0.062(32) | 0.045±0.02(45) | 0.981±0.007(22) | 0.979±0.007(25) | 0.15±0.121(44) | 0.985±0.006(16) |
| Ionosphere | 0.965±0.014(7) | 0.978±0.005(1) | 0.908±0.037(36) | 0.732±0.011(37) | 0.962±0.009(10) | 0.976±0.007(3) | 0.676±0.1(42) | 0.686±0.092(40) | 0.941±0.009(17) | 0.957±0.006(14) | 0.703±0.158(38) | 0.91±0.027(25) |
| Stamps | 0.782±0.114(30) | 0.949±0.014(3) | 0.885±0.07(24) | 0.609±0.252(40) | 0.864±0.025(27) | 0.941±0.023(5) | 0.921±0.06(17) | 0.628±0.23(38) | 0.902±0.03(12) | 0.621±0.054(39) | 0.757±0.135(32) | 0.933±0.02(11) |
| vertebral | 0.616±0.04(2) | 0.45±0.017(25) | 0.457±0.085(22) | 0.449±0.071(27) | 0.505±0.044(12) | 0.45±0.035(25) | 0.462±0.122(21) | 0.612±0.108(3) | 0.581±0.15(5) | 0.479±0.034(19) | 0.594±0.118(4) | 0.487±0.024(16) |
| WBC | 0.806±0.162(37) | 0.981±0.007(14) | 0.924±0.059(30) | 0.893±0.054(35) | 0.935±0.022(29) | 0.978±0.01(11) | 0.978±0.022(11) | 0.024±0.029(45) | 0.959±0.024(27) | 0.987±0.008(7) | 0.064±0.07(44) | 0.986±0.009(8) |
| glass | 0.658±0.085(36) | 0.879±0.013(5) | 0.761±0.095(23) | 0.611±0.187(42) | 0.919±0.09(1) | 0.89±0.02(3) | 0.833±0.037(13) | 0.625±0.234(40) | 0.829±0.055(14) | 0.844±0.032(12) | 0.626±0.2(39) | 0.724±0.03(30) |
| WPBC | 0.506±0.036(28) | 0.561±0.02(5) | 0.545±0.037(11) | 0.564±0.035(3) | 0.511±0.046(24) | 0.564±0.028(3) | 0.495±0.063(34) | 0.482±0.031(40) | 0.502±0.037(29) | 0.542±0.011(14) | 0.499±0.042(32) | 0.509±0.02(25) |
| Lymphography | 0.977±0.013(28) | 0.99±0.006(14) | 0.97±0.016(32) | 0.981±0.023(25) | 0.97±0.025(32) | 0.987±0.001(20) | 0.992±0.006(6) | 0.17±0.263(44) | 0.973±0.008(29) | 0.989±0.011(17) | 0.259±0.324(43) | 0.993±0.006(4) |
| wine | 0.925±0.03(20) | 0.972±0.015(7) | 0.646±0.282(34) | 0.963±0.015(9) | 0.906±0.026(22) | 0.978±0.016(3) | 0.776±0.242(28) | 0.153±0.149(43) | 0.903±0.07(23) | 0.977±0.01(1) | 0.106±0.113(44) | 0.935±0.02(17) |
| Hepatitis | 0.767±0.062(22) | 0.831±0.025(6) | 0.725±0.065(28) | 0.833±0.029(3) | 0.7±0.066(31) | 0.763±0.084(24) | 0.786±0.049(18) | 0.59±0.076(40) | 0.625±0.092(35) | 0.776±0.041(20) | 0.535±0.081(42) | 0.833±0.02(3) |
| Avg Ranking | 24.42 | 8.83 | 27.42 | 24.58 | 21.75 | 9.92 | 21.08 | 39.0 | 21.67 | 17.92 | 37.42 | 15.0 |

| Dataset | CBLOF | CD | ECOD | FB | GMM | HBOS | IForest | KDE | LMDD | LODA | MCD | OCSVM |
|---|---|---|---|---|---|---|---|---|---|---|---|---|
| Pima | 0.71±0.018(13) | 0.676±0.011(22) | 0.595±0.016(35) | 0.706±0.016(15) | 0.752±0.015(11) | 0.727±0.007(6) | 0.731±0.025(5) | 0.754±0.017(1) | 0.599±0.079(33) | 0.654±0.076(26) | 0.719±0.014(12) | 0.701±0.017(16) |
| breastw | 0.989±0.004(8) | 0.976±0.003(22) | 0.991±0.001(3) | 0.581±0.242(38) | 0.985±0.004(18) | 0.991±0.002(3) | 0.995±0.001(1) | 0.989±0.005(8) | 0.628±0.076(36) | 0.981±0.01(21) | 0.989±0.003(8) | 0.989±0.004(8) |
| WDBC | 0.989±0.004(14) | 0.942±0.008(34) | 0.991±0.001(3) | 0.732±0.011(37) | 0.95±0.014(15) | 0.987±0.004(18) | 0.991±0.004(10) | 0.993±0.004(3) | 0.992±0.007(7) | 0.974±0.017(28) | 0.975±0.005(26) | 0.992±0.006(7) |
| Ionosphere | 0.961±0.01(12) | 0.919±0.004(33) | 0.732±0.011(37) | 0.95±0.014(15) | 0.964±0.002(9) | 0.685±0.028(41) | 0.9±0.02(27) | 0.975±0.02(5) | 0.769±0.049(35) | 0.852±0.044(29) | 0.958±0.004(13) | 0.965±0.003(9) |
| Stamps | 0.935±0.026(8) | 0.746±0.02(33) | 0.884±0.011(26) | 0.921±0.04(17) | 0.921±0.017(18) | 0.926±0.014(14) | 0.938±0.016(6) | 0.955±0.01(1) | 0.914±0.04(19) | 0.911±0.032(20) | 0.855±0.026(28) | 0.936±0.02(7) |
| vertebral | 0.487±0.033(16) | 0.455±0.034(24) | 0.416±0.01(33) | 0.611±0.01(35) | 0.489±0.04(14) | 0.362±0.038(40) | 0.426±0.009(30) | 0.412±0.024(34) | 0.387±0.064(38) | 0.386±0.06(39) | 0.469±0.026(20) | 0.502±0.022(13) |
| WBC | 0.979±0.008(16) | 0.971±0.006(24) | 0.993±0.002(2) | 0.709±0.248(39) | 0.978±0.011(18) | 0.988±0.005(6) | 0.994±0.003(1) | 0.982±0.008(13) | 0.99±0.006(5) | 0.97±0.012(25) | 0.986±0.006(8) | 0.985±0.009(11) |
| glass | 0.864±0.016(7) | 0.784±0.043(20) | 0.715±0.025(31) | 0.753±0.067(25) | 0.77±0.027(21) | 0.828±0.027(15) | 0.806±0.014(16) | 0.85±0.021(11) | 0.647±0.052(37) | 0.623±0.1(41) | 0.79±0.006(19) | 0.687±0.02(33) |
| WPBC | 0.527±0.014(18) | 0.465±0.038(44) | 0.5±0.016(31) | 0.543±0.022(12) | 0.508±0.025(26) | 0.578±0.019(1) | 0.549±0.017(9) | 0.558±0.028(6) | 0.495±0.079(34) | 0.543±0.041(12) | 0.515±0.027(21) | 0.528±0.01(17) |
| Lymphography | 0.991±0.008(8) | 0.957±0.013(36) | 0.989±0.009(17) | 0.984±0.008(23) | 0.993±0.001(4) | 0.993±0.006(4) | 0.989±0.011(17) | 0.558±0.028(6) | 0.957±0.047(36) | 0.828±0.162(25) | 0.983±0.006(20) | 0.99±0.006(14) |
| wine | 0.965±0.04(8) | 0.877±0.018(27) | 0.74±0.026(29) | 0.961±0.015(11) | 0.982±0.005(2) | 0.933±0.01(19) | 0.934±0.02(18) | 0.973±0.015(6) | 0.875±0.048(25) | 0.913±0.035(21) | 0.983±0.016(1) | 0.962±0.07(10) |
| Hepatitis | 0.832±0.033(5) | 0.574±0.057(41) | 0.739±0.032(26) | 0.837±0.02(1) | 0.814±0.048(12) | 0.814±0.034(12) | 0.79±0.049(17) | 0.814±0.033(12) | 0.755±0.047(25) | 0.55±0.045(38) | 0.816±0.063(11) | 0.837±0.024(1) |
| Avg Ranking | 11.08 | 30.42 | 23.83 | 18.83 | 15.17 | 14.92 | 11.83 | 9.75 | 32.17 | 27.83 | 15.92 | 12.0 |

| Dataset | QMCD | Sampling | ABOD | COF | INNE | KNN | KPCA | LOF | PCA |
|---|---|---|---|---|---|---|---|---|---|
| Pima | 0.741±0.014(3) | 0.721±0.033(9) | 0.638±0.02(29) | 0.568±0.011(38) | 0.665±0.02(24) | 0.681±0.016(20) | 0.536±0.02(40) | 0.607±0.019(31) |
| breastw | 0.474±0.121(39) | 0.986±0.005(15) | 0.432±0.015(40) | 0.327±0.023(43) | 0.667±0.034(34) | 0.972±0.005(23) | 0.606±0.081(37) | 0.353±0.021(42) | 0.749±0.004(31) |
| WDBC | 0.453±0.14(42) | 0.988±0.01(16) | 0.881±0.017(37) | 0.942±0.011(34) | 0.952±0.002(33) | 0.974±0.009(26) | 0.645±0.333(40) | 0.975±0.009(26) | 0.981±0.005(22) |
| Ionosphere | 0.536±0.02(45) | 0.962±0.018(10) | 0.924±0.004(22) | 0.849±0.008(31) | 0.871±0.011(28) | 0.913±0.005(24) | 0.606±0.048(43) | 0.835±0.011(33) | 0.753±0.009(36) |
| Stamps | 0.887±0.023(23) | 0.931±0.015(12) | 0.653±0.01(37) | 0.434±0.015(44) | 0.742±0.042(34) | 0.706±0.029(36) | 0.516±0.10(42) | 0.447±0.037(43) | 0.885±0.026(24) |
| vertebral | 0.402±0.032(37) | 0.448±0.057(28) | 0.332±0.022(44) | 0.41±0.02(36) | 0.361±0.014(41) | 0.319±0.01(45) | 0.489±0.059(14) | 0.351±0.021(42) | 0.344±0.018(43) |
| WBC | 0.354±0.161(43) | 0.978±0.011(18) | 0.942±0.021(28) | 0.901±0.047(34) | 0.752±0.047(26) | 0.736±0.047(28) | 0.851±0.021(10) | 0.497±0.108(35) | 0.993±0.002(2) |
| glass | 0.857±0.017(8) | 0.855±0.029(9) | 0.796±0.028(18) | 0.752±0.047(26) | 0.736±0.047(28) | 0.851±0.021(10) | 0.412±0.101(45) | 0.725±0.048(29) | 0.638±0.03(38) |
| WPBC | 0.556±0.024(7) | 0.556±0.031(7) | 0.446±0.026(45) | 0.484±0.034(39) | 0.501±0.02(30) | 0.512±0.018(22) | 0.507±0.015(27) | 0.52±0.012(19) | 0.48±0.023(41) |
| Lymphography | 0.086±0.043(45) | 0.991±0.008(8) | 0.969±0.011(35) | 0.991±0.009(7) | 0.973±0.006(29) | 0.991±0.006(8) | 0.604±0.112(41) | 0.99±0.008(14) | 0.996±0.004(1) |
| wine | 0.494±0.136(36) | 0.958±0.035(12) | 0.332±0.065(41) | 0.454±0.14(38) | 0.711±0.091(30) | 0.442±0.067(39) | 0.694±0.277(32) | 0.939±0.025(16) | 0.7±0.011(31) |
| Hepatitis | 0.611±0.125(38) | 0.817±0.039(10) | 0.624±0.063(36) | 0.715±0.025(29) | 0.637±0.029(34) | 0.783±0.028(19) | 0.6±0.107(39) | 0.805±0.015(16) | 0.622±0.043(37) |
| Avg Ranking | 30.5 | 12.83 | 34.33 | 33.33 | 31.75 | 24.33 | 36.75 | 28.75 | 28.08 |

Table 7: AUPRC results on 12 small datasets, where we compare TCCM to 44 baselines (with 5 independent runs). We report the mean ± std (rank).

| Dataset | TCMM (Ours) | AE | AE-1SVM | ALAD | AnoGAN | DAGMM | DeepSVDD | DIF | DROCC | DTE-Cat | DTE-DDPM | DTE-Gaussian |
|---|---|---|---|---|---|---|---|---|---|---|---|---|
| Pima | 0.716±0.029(5) | 0.68±0.023(16) | 0.648±0.009(17) | 0.548±0.007(41) | 0.697±0.046(13) | NAN | 0.665±0.028(21) | 0.578±0.024(38) | 0.502±0.121(43) | 0.636±0.029(28) | 0.594±0.016(35) | 0.657±0.02(20) |
| breastw | 0.987±0.007(8) | 0.976±0.011(20) | 0.98±0.007(17) | 0.692±0.026(35) | 0.993±0.001(2) | NAN | 0.982±0.009(14) | 0.683±0.036(36) | 0.574±0.22(39) | 0.893±0.016(29) | 0.955±0.017(24) | 0.907±0.037(28) |
| WDBC | 0.862±0.074(5) | 0.868±0.091(3) | 0.778±0.217(32) | 0.166±0.105(41) | 0.803±0.122(15) | NAN | 0.782±0.179(21) | 0.12±0.026(43) | 0.126±0.187(42) | 0.565±0.087(32) | 0.432±0.159(33) | 0.865±0.101(4) |
| Ionosphere | 0.983±0.004(2) | 0.959±0.004(15) | 0.946±0.018(18) | 0.595±0.06(44) | 0.806±0.034(34) | NAN | 0.862±0.044(32) | 0.946±0.005(18) | 0.775±0.126(37) | 0.959±0.007(15) | 0.876±0.022(31) | 0.982±0.003(3) |
| Stamps | 0.651±0.096(7) | 0.601±0.075(15) | 0.654±0.112(5) | 0.274±0.092(39) | 0.486±0.231(28) | NAN | 0.68±0.109(2) | 0.626±0.081(8) | 0.186±0.094(41) | 0.568±0.094(21) | 0.28±0.062(32) | 0.491±0.126(27) |
| vertebral | 0.311±0.062(1) | 0.2±0.004(27) | 0.233±0.013(8) | 0.238±0.055(9) | 0.248±0.04(7) | NAN | 0.187±0.017(34) | 0.258±0.01(5) | 0.234±0.117(11) | 0.234±0.027(11) | 0.204±0.02(24) | 0.221±0.007(18) |
| WBC | 0.858±0.092(9) | 0.797±0.104(16) | 0.768±0.118(22) | 0.226±0.177(41) | 0.923±0.086(4) | NAN | 0.832±0.1(14) | 0.173±0.03(42) | 0.234±0.272(40) | 0.462±0.167(33) | 0.615±0.107(29) | 0.66±0.198(27) |
| glass | 0.355±0.054(2) | 0.217±0.046(27) | 0.243±0.047(22) | 0.096±0.032(45) | 0.42±0.065(32) | NAN | 0.185±0.069(38) | 0.327±0.049(5) | 0.263±0.122(13) | 0.258±0.092(15) | 0.242±0.079(20) | 0.208±0.056(29) |
| WPBC | 0.418±0.014(11) | 0.385±0.019(30) | 0.365±0.028(41) | 0.379±0.044(34) | 0.41±0.064(7) | NAN | 0.394±0.053(23) | 0.363±0.014(44) | 0.404±0.073(13) | 0.401±0.019(16) | 0.43±0.051(2) | 0.385±0.02(30) |
| Lymphography | 0.88±0.11(7) | 0.851±0.16(19) | 0.797±0.242(29) | 0.24±0.13(42) | 0.877±0.11(29) | NAN | 0.79±0.161(30) | 0.347±0.234(43) | 0.449±0.182(38) | 0.317±0.033(40) | 0.248±0.06(34) | 0.777±0.18(32) |
| wine | 0.84±0.082(5) | 0.707±0.108(15) | 0.69±0.15(16) | 0.19±0.06(37) | 0.492±0.26(24) | NAN | 0.504±0.17(26) | 0.224±0.007(36) | 0.172±0.123(38) | 0.752±0.15(12) | 0.246±0.06(34) | 0.573±0.127(22) |
| Hepatitis | 0.676±0.062(3) | 0.67±0.058(6) | NAN | 0.278±0.038(44) | 0.518±0.116(27) | NAN | 0.568±0.043(21) | 0.446±0.065(33) | 0.331±0.133(41) | 0.653±0.05(10) | 0.532±0.107(26) | 0.652±0.029(11) |
| Avg Ranking | 4.83 | 17.42 | 20.73 | 37.67 | 17.08 | NAN | 23.0 | 28.83 | 33.5 | 20.58 | 28.83 | 20.92 |

| Dataset | DTE-IG | DTE-NP | GANomaly | GOAD | ICL | LUNAR | MCM | MO-GAAL | PlanarFlow | SLAD | SO-GAAL | VAE |
|---|---|---|---|---|---|---|---|---|---|---|---|---|
| Pima | 0.624±0.009(31) | 0.723±0.027(4) | 0.602±0.039(34) | 0.464±0.044(44) | 0.666±0.019(20) | 0.713±0.028(7) | 0.704±0.029(9) | 0.46±0.033(45) | 0.698±0.022(12) | 0.568±0.018(39) | 0.504±0.08(42) | 0.659±0.012(23) |
| breastw | 0.722±0.137(34) | 0.98±0.01(17) | 0.942±0.017(26) | 0.833±0.087(31) | 0.942±0.025(26) | 0.986±0.009(11) | 0.982±0.011(14) | 0.337±0.024(45) | 0.96±0.024(23) | 0.989±0.004(4) | 0.465±0.139(41) | 0.988±0.007(6) |
| WDBC | 0.82±0.111(11) | 0.839±0.089(9) | 0.42±0.175(34) | 0.808±0.127(12) | 0.789±0.143(19) | 0.892±0.08(1) | 0.653±0.256(26) | 0.03±0.01(45) | 0.7±0.065(23) | 0.698±0.12(24) | 0.036±0.006(44) | 0.787±0.14(20) |
| Ionosphere | 0.974±0.009(8) | 0.982±0.004(3) | 0.918±0.058(26) | 0.943±0.013(23) | 0.968±0.007(12) | 0.981±0.006(5) | 0.638±0.101(42) | 0.728±0.095(41) | 0.943±0.021(23) | 0.967±0.004(13) | 0.755±0.081(39) | 0.924±0.019(25) |
| Stamps | 0.461±0.12(31) | 0.566±0.09(23) | 0.42±0.175(34) | 0.389±0.235(35) | 0.503±0.26(26) | 0.652±0.11(6) | 0.584±0.062(37) | 0.395±0.262(36) | 0.272±0.022(40) | 0.54±0.075(24) | 0.424±0.07(33) | 0.615±0.06(10) |
| vertebral | 0.285±0.042(2) | 0.2±0.005(27) | 0.204±0.02(24) | 0.211±0.027(19) | 0.248±0.04(7) | 0.194±0.01(32) | 0.277±0.06(3) | 0.257±0.05(5) | 0.205±0.042(23) | 0.205±0.009(23) | 0.276±0.07(4) | 0.212±0.01(17) |
| WBC | 0.35±0.2(36) | 0.771±0.105(21) | 0.612±0.192(30) | 0.686±0.095(26) | 0.537±0.153(31) | 0.779±0.101(20) | 0.767±0.148(23) | 0.049±0.09(45) | 0.711±0.151(25) | 0.866±0.099(6) | 0.05±0.005(44) | 0.866±0.09(6) |
| glass | 0.243±0.039(19) | 0.302±0.05(7) | 0.223±0.07(26) | 0.2±0.11(32) | 0.447±0.072(1) | 0.345±0.051(3) | 0.266±0.045(12) | 0.198±0.079(34) | 0.241±0.065(22) | 0.267±0.06(11) | 0.212±0.075(28) | 0.189±0.015(37) |
| WPBC | 0.395±0.029(22) | 0.409±0.021(10) | 0.43±0.041(2) | 0.434±0.044(1) | 0.406±0.033(12) | 0.412±0.019(5) | 0.387±0.052(28) | 0.377±0.034(36) | 0.403±0.042(14) | 0.401±0.007(16) | 0.396±0.024(20) | 0.383±0.018(32) |
| Lymphography | 0.754±0.166(33) | 0.832±0.153(21) | 0.748±0.11(34) | 0.823±0.149(26) | 0.873±0.062(13) | 0.852±0.079(15) | 0.9±0.07(9) | 0.1±0.115(44) | 0.798±0.266(28) | 0.853±0.14(17) | 0.126±0.144(43) | 0.91±0.10(5) |
| wine | 0.648±0.128(19) | 0.809±0.184(7) | 0.31±0.192(31) | 0.768±0.082(10) | 0.506±0.07(25) | 0.543±0.091(4) | 0.459±0.295(28) | 0.094±0.015(43) | 0.572±0.197(23) | 0.88±0.04(1) | 0.089±0.009(44) | 0.651±0.09(18) |
| Hepatitis | 0.591±0.079(17) | 0.66±0.029(7) | 0.546±0.052(34) | 0.4±0.063(23) | 0.458±0.093(31) | 0.547±0.122(24) | 0.576±0.057(20) | 0.388±0.07(37) | 0.385±0.10(38) | 0.594±0.083(16) | 0.321±0.07(41) | 0.675±0.09(5) |
| Avg Ranking | 21.92 | 11.5 | 25.5 | 22.25 | 18.58 | 11.0 | 20.0 | 37.67 | 21.67 | 17.5 | 35.17 | 17.08 |

| Dataset | CBLOF | CD | ECOD | FB | GMM | HBOS | IForest | KDE | LMDD | LODA | MCD | OCSVM |
|---|---|---|---|---|---|---|---|---|---|---|---|---|
| Pima | 0.691±0.024(11) | 0.662±0.02(22) | 0.627±0.022(30) | 0.668±0.019(18) | 0.7±0.025(10) | 0.736±0.024(2) | 0.715±0.018(6) | 0.732±0.027(3) | 0.633±0.054(29) | 0.619±0.068(32) | 0.677±0.025(17) | 0.691±0.021(11) |
| breastw | 0.987±0.007(8) | 0.976±0.002(20) | 0.992±0.001(3) | 0.558±0.216(40) | 0.979±0.006(19) | 0.989±0.002(4) | 0.995±0.001(1) | 0.987±0.009(8) | 0.798±0.044(32) | 0.97±0.01(23) | 0.989±0.003(8) | 0.986±0.01(11) |
| WDBC | 0.807±0.095(13) | 0.414±0.038(35) | 0.648±0.07(27) | 0.883±0.653(2) | 0.797±0.1(17) | 0.801±0.075(16) | 0.792±0.094(18) | 0.861±0.047(7) | 0.822±0.14(10) | 0.58±0.191(31) | 0.598±0.062(29) | 0.843±0.096(8) |
| Ionosphere | 0.966±0.01(14) | 0.944±0.003(22) | 0.766±0.011(38) | 0.958±0.011(17) | 0.974±0.002(8) | 0.61±0.023(43) | 0.907±0.018(27) | 0.979±0.002(6) | 0.801±0.046(36) | 0.837±0.07(33) | 0.969±0.003(10) | 0.975±0.003(7) |
| Stamps | 0.664±0.083(4) | 0.325±0.015(38) | 0.516±0.043(25) | 0.579±0.108(18) | 0.603±0.091(14) | 0.56±0.056(22) | 0.68±0.1(10) | 0.688±0.06(1) | 0.577±0.075(19) | 0.54±0.075(24) | 0.424±0.072(33) | 0.613±0.09(10) |
| vertebral | 0.211±0.012(19) | 0.208±0.019(22) | 0.191±0.004(31) | 0.19±0.009(32) | 0.21±0.005(21) | 0.174±0.011(40) | 0.189±0.002(38) | 0.184±0.007(38) | 0.186±0.027(35) | 0.18±0.016(39) | 0.204±0.01(24) | 0.213±0.007(16) |
| WBC | 0.797±0.067(16) | 0.805±0.02(15) | 0.925±0.02(3) | 0.319±0.32(37) | 0.78±0.11(19) | 0.861±0.06(8) | 0.956±0.02(1) | 0.849±0.08(12) | 0.912±0.05(5) | 0.356±0.07(10) | 0.85±0.115(11) | 0.805±0.07(16) |
| glass | 0.268±0.01(10) | 0.241±0.042(22) | 0.255±0.01(16) | 0.204±0.06(31) | 0.236±0.06(18) | 0.236±0.07(18) | 0.208±0.02(29) | 0.209±0.018(29) | 0.169±0.05(40) | 0.118±0.029(43) | 0.197±0.045(35) | 0.226±0.05(25) |
| WPBC | 0.391±0.01(24) | 0.364±0.013(43) | 0.368±0.012(39) | 0.403±0.02(14) | 0.389±0.024(26) | 0.412±0.017(5) | 0.401±0.07(16) | 0.41±0.03(7) | 0.386±0.042(28) | 0.41±0.02(7) | 0.396±0.02(20) | 0.39±0.013(25) |
| Lymphography | 0.877±0.112(7) | 0.673±0.145(36) | 0.97±0.043(3) | 0.849±0.13(20) | 0.832±0.019(21) | 0.929±0.046(4) | 0.948±0.045(2) | 0.867±0.112(16) | 0.811±0.145(27) | 0.264±0.26(41) | 0.829±0.041(24) | 0.832±0.153(21) |
| wine | 0.778±0.076(8) | 0.111±0.008(42) | 0.327±0.048(30) | 0.748±0.083(13) | 0.853±0.081(3) | 0.674±0.119(17) | 0.715±0.181(14) | 0.826±0.079(6) | 0.557±0.115(24) | 0.631±0.116(20) | 0.875±0.125(2) | 0.773±0.087(9) |
| Hepatitis | 0.68±0.064(1) | 0.323±0.032(42) | 0.461±0.05(30) | 0.679±0.061(2) | 0.607±0.047(15) | 0.59±0.06(19) | 0.56±0.078(22) | 0.614±0.043(14) | 0.591±0.076(17) | 0.469±0.124(29) | 0.619±0.114(13) | 0.676±0.053(3) |
| Avg Ranking | 11.67 | 29.92 | 22.92 | 20.33 | 16.42 | 15.75 | 14.92 | 10.5 | 25.25 | 29.08 | 18.58 | 13.25 |

| Dataset | QMCD | Sampling | ABOD | COF | INNE | KNN | KPCA | LOF | PCA |
|---|---|---|---|---|---|---|---|---|---|
| Pima | 0.749±0.019(1) | 0.707±0.048(8) | 0.639±0.026(26) | 0.588±0.019(36) | 0.638±0.025(27) | 0.668±0.021(18) | 0.556±0.02(40) | 0.58±0.018(37) | 0.612±0.023(33) |
| breastw | 0.639±0.09(37) | 0.981±0.009(16) | 0.44±0.006(42) | 0.399±0.007(44) | 0.596±0.036(38) | 0.945±0.01(25) | 0.744±0.06(33) | 0.41±0.007(43) | 0.845±0.003(30) |
| WDBC | 0.171±0.06(40) | 0.807±0.15(13) | 0.281±0.032(39) | 0.374±0.07(37) | 0.392±0.034(36) | 0.596±0.099(30) | 0.351±0.315(38) | 0.714±0.02(22) | 0.805±0.004(15) |
| Ionosphere | 0.552±0.025(45) | 0.969±0.014(11) | 0.946±0.009(18) | 0.901±0.009(29) | 0.906±0.008(28) | 0.934±0.004(21) | 0.744±0.02(40) | 0.896±0.009(30) | 0.805±0.004(35) |
| Stamps | 0.57±0.037(20) | 0.607±0.043(12) | 0.27±0.018(41) | 0.313±0.07(42) | 0.34±0.029(36) | 0.332±0.035(37) | 0.212±0.021(17) | 0.169±0.065(42) | 0.484±0.06(29) |
| vertebral | 0.185±0.013(37) | 0.198±0.022(30) | 0.166±0.005(44) | 0.186±0.007(35) | 0.172±0.004(41) | 0.161±0.002(45) | 0.212±0.021(17) | 0.169±0.005(43) | 0.167±0.005(43) |
| WBC | 0.153±0.088(43) | 0.787±0.128(18) | 0.524±0.137(32) | 0.382±0.102(35) | 0.309±0.012(38) | 0.751±0.119(24) | 0.26±0.175(39) | 0.95±0.016(2) |
| glass | 0.327±0.084(5) | 0.26±0.044(14) | 0.244±0.052(18) | 0.183±0.041(39) | 0.196±0.056(36) | 0.254±0.019(17) | 0.107±0.044(44) | 0.167±0.037(42) | 0.169±0.018(40) |
| WPBC | 0.401±0.01(16) | 0.409±0.039(11) | 0.351±0.024(45) | 0.365±0.018(41) | 0.368±0.01(39) | 0.375±0.013(37) | 0.381±0.066(33) | 0.378±0.011(35) | 0.369±0.018(38) |
| Lymphography | 0.05±0.004(45) | 0.872±0.106(14) | 0.67±0.05(37) | 0.612±0.05(12) | 0.697±0.103(35) | 0.872±0.106(14) | 0.351±0.184(39) | 0.878±0.094(8) | 0.963±0.041(1) |
| wine | 0.294±0.117(32) | 0.76±0.135(11) | 0.113±0.04(41) | 0.138±0.032(40) | 0.234±0.057(35) | 0.142±0.016(38) | 0.437±0.305(29) | 0.596±0.096(21) | 0.271±0.043(33) |
| Hepatitis | 0.405±0.101(35) | 0.624±0.07(12) | 0.357±0.054(39) | 0.436±0.052(34) | 0.349±0.025(40) | 0.505±0.054(28) | 0.449±0.12(32) | 0.546±0.022(25) | 0.392±0.051(36) |
| Avg Ranking | 29.67 | 14.08 | 35.17 | 35.33 | 35.75 | 27.92 | 35.5 | 32.42 | 28.75 |

Table 8: AUROC results on 15 medium datasets, where we compare TCCM to 44 baselines (with 5 independent runs). We report the mean ± std (rank in terms of mean among all anomaly detectors).

| Dataset | TCMM (Ours) | AE | AE-1SVM | ALAD | AnoGAN | DAGMM | DeepSVDD | DIF | DROCC | DTE-Cat | DTE-DDPM | DTE-Gaussian |
|---|---|---|---|---|---|---|---|---|---|---|---|---|
| annthyroid | $0.918_{\pm0.032}(8)$ | $0.856_{\pm0.042}(23)$ | $0.86_{\pm0.014}(21)$ | $0.575_{\pm0.019}(43)$ | $0.806_{\pm0.055}(27)$ | $0.602_{\pm0.167}(42)$ | $0.857_{\pm0.042}(22)$ | $0.778_{\pm0.045}(30)$ | $0.876_{\pm0.035}(16)$ | $0.982_{\pm0.001}(1)$ | $0.771_{\pm0.012}(31)$ | $0.956_{\pm0.016}(2)$ |
| pendigits | $0.983_{\pm0.002}(7)$ | $0.977_{\pm0.016}(10)$ | $0.955_{\pm0.012}(18)$ | $0.552_{\pm0.051}(41)$ | $0.704_{\pm0.289}(36)$ | $0.688_{\pm0.149}(37)$ | $0.871_{\pm0.092}(27)$ | $0.982_{\pm0.006}(8)$ | $0.774_{\pm0.142}(34)$ | $0.972_{\pm0.011}(12)$ | $0.83_{\pm0.018}(31)$ | $0.993_{\pm0.001}(5)$ |
| satellite | $0.825_{\pm0.008}(10)$ | $0.796_{\pm0.009}(18)$ | $0.769_{\pm0.06}(22)$ | $0.504_{\pm0.025}(45)$ | $0.641_{\pm0.137}(36)$ | $0.689_{\pm0.071}(32)$ | $0.675_{\pm0.043}(33)$ | $0.778_{\pm0.006}(20)$ | $0.728_{\pm0.095}(27)$ | $0.809_{\pm0.007}(14)$ | $0.772_{\pm0.006}(21)$ | $0.795_{\pm0.006}(19)$ |
| landsat | $0.619_{\pm0.011}(11)$ | $0.57_{\pm0.007}(21)$ | $0.586_{\pm0.006}(16)$ | $0.485_{\pm0.019}(32)$ | $0.437_{\pm0.168}(41)$ | $0.508_{\pm0.078}(29)$ | $0.433_{\pm0.066}(42)$ | $0.577_{\pm0.006}(19)$ | $0.588_{\pm0.023}(15)$ | $0.575_{\pm0.027}(20)$ | $0.521_{\pm0.01}(26)$ | $0.40_{\pm0.021}(37)$ |
| satimage-2 | $0.998_{\pm0.001}(6)$ | $0.999_{\pm0.0}(1)$ | $0.987_{\pm0.0}(23)$ | $0.532_{\pm0.044}(46)$ | $0.927_{\pm0.028}(35)$ | $0.787_{\pm0.145}(40)$ | $0.974_{\pm0.023}(30)$ | $0.997_{\pm0.001}(10)$ | $0.818_{\pm0.126}(39)$ | $0.988_{\pm0.001}(21)$ | $0.975_{\pm0.009}(29)$ | $0.995_{\pm0.001}(16)$ |
| PageBlocks | $0.96_{\pm0.005}(6)$ | $0.95_{\pm0.015}(12)$ | $0.964_{\pm0.006}(2)$ | $0.581_{\pm0.048}(44)$ | $0.859_{\pm0.029}(33)$ | $0.765_{\pm0.09}(39)$ | $0.926_{\pm0.021}(22)$ | $0.935_{\pm0.005}(19)$ | $0.947_{\pm0.024}(15)$ | $0.862_{\pm0.06}(3)$ | $0.875_{\pm0.016}(30)$ | $0.959_{\pm0.006}(8)$ |
| Wilt | $0.939_{\pm0.012}(3)$ | $0.55_{\pm0.033}(19)$ | $0.459_{\pm0.019}(29)$ | $0.468_{\pm0.017}(27)$ | $0.377_{\pm0.079}(38)$ | $0.574_{\pm0.091}(17)$ | $0.353_{\pm0.048}(40)$ | $0.361_{\pm0.014}(39)$ | $0.486_{\pm0.107}(25)$ | $0.863_{\pm0.006}(4)$ | $0.506_{\pm0.014}(23)$ | $0.642_{\pm0.022}(12)$ |
| thyroid | $0.982_{\pm0.009}(13)$ | $0.98_{\pm0.005}(16)$ | $0.99_{\pm0.002}(2)$ | $0.594_{\pm0.047}(42)$ | $0.928_{\pm0.126}(33)$ | $0.737_{\pm0.194}(41)$ | $0.985_{\pm0.003}(8)$ | $0.984_{\pm0.000}(10)$ | $0.921_{\pm0.024}(34)$ | $0.993_{\pm0.001}(1)$ | $0.881_{\pm0.038}(38)$ | $0.945_{\pm0.043}(28)$ |
| Waveform | $0.738_{\pm0.064}(8)$ | $0.687_{\pm0.032}(19)$ | $0.686_{\pm0.01}(20)$ | $0.51_{\pm0.036}(41)$ | $0.684_{\pm0.046}(21)$ | $0.492_{\pm0.015}(42)$ | $0.59_{\pm0.133}(34)$ | $0.722_{\pm0.009}(11)$ | $0.68_{\pm0.452}(22)$ | $0.62_{\pm0.026}(29)$ | $0.523_{\pm0.026}(39)$ | $0.588_{\pm0.01}(35)$ |
| Cardiotocography | $0.829_{\pm0.012}(2)$ | $0.74_{\pm0.016}(18)$ | $0.756_{\pm0.052}(15)$ | $0.557_{\pm0.039}(38)$ | $0.742_{\pm0.125}(17)$ | $NAN$ | $0.799_{\pm0.101}(7)$ | $0.633_{\pm0.019}(31)$ | $0.44_{\pm0.172}(43)$ | $0.738_{\pm0.027}(19)$ | $0.574_{\pm0.016}(36)$ | $0.756_{\pm0.013}(15)$ |
| fault | $0.777_{\pm0.026}(8)$ | $0.737_{\pm0.01}(18)$ | $0.633_{\pm0.012}(22)$ | $0.501_{\pm0.019}(41)$ | $0.528_{\pm0.067}(36)$ | $NAN$ | $0.542_{\pm0.048}(35)$ | $0.718_{\pm0.019}(10)$ | $0.621_{\pm0.043}(25)$ | $0.695_{\pm0.017}(13)$ | $0.618_{\pm0.012}(27)$ | $0.701_{\pm0.007}(11)$ |
| cardio | $0.956_{\pm0.008}(5)$ | $0.93_{\pm0.028}(17)$ | $0.958_{\pm0.007}(3)$ | $0.56_{\pm0.025}(40)$ | $0.907_{\pm0.058}(22)$ | $NAN$ | $0.938_{\pm0.029}(14)$ | $0.951_{\pm0.008}(9)$ | $0.516_{\pm0.236}(41)$ | $0.908_{\pm0.019}(21)$ | $0.73_{\pm0.006}(36)$ | $0.934_{\pm0.008}(16)$ |
| letter | $0.891_{\pm0.012}(6)$ | $0.802_{\pm0.008}(19)$ | $0.615_{\pm0.017}(31)$ | $0.507_{\pm0.041}(41)$ | $0.501_{\pm0.035}(43)$ | $NAN$ | $0.507_{\pm0.06}(41)$ | $0.667_{\pm0.02}(27)$ | $0.649_{\pm0.051}(28)$ | $0.88_{\pm0.006}(7)$ | $0.642_{\pm0.045}(29)$ | $0.868_{\pm0.005}(10)$ |
| yeast | $0.503_{\pm0.026}(4)$ | $0.461_{\pm0.021}(22)$ | $0.447_{\pm0.024}(28)$ | $0.481_{\pm0.014}(14)$ | $0.422_{\pm0.061}(36)$ | $NAN$ | $0.43_{\pm0.032}(33)$ | $0.402_{\pm0.021}(41)$ | $0.503_{\pm0.041}(4)$ | $0.468_{\pm0.025}(19)$ | $0.493_{\pm0.012}(11)$ | $0.474_{\pm0.027}(17)$ |
| vowels | $0.97_{\pm0.008}(7)$ | $0.934_{\pm0.009}(15)$ | $0.709_{\pm0.028}(21)$ | $0.502_{\pm0.028}(42)$ | $0.522_{\pm0.031}(40)$ | $NAN$ | $0.576_{\pm0.085}(38)$ | $0.814_{\pm0.018}(26)$ | $0.519_{\pm0.152}(41)$ | $0.978_{\pm0.006}(5)$ | $0.734_{\pm0.043}(29)$ | $0.964_{\pm0.006}(9)$ |
| Avg Ranking | 6.8 | 15.87 | 18.87 | 38.47 | 32.93 | 35.44 | 28.4 | 20.67 | 27.27 | 12.6 | 29.07 | 16.0 |

| Dataset | DTE-IG | DTE-NP | GANomaly | GOAD | ICL | LUNAR | MCM | MO_GAAL | PlanarFlow | SLAD | SO_GAAL | VAE |
|---|---|---|---|---|---|---|---|---|---|---|---|---|
| annthyroid | $0.909_{\pm0.057}(10)$ | $0.939_{\pm0.005}(4)$ | $0.67_{\pm0.088}(38)$ | $0.654_{\pm0.042}(39)$ | $0.794_{\pm0.032}(28)$ | $0.886_{\pm0.027}(13)$ | $0.89_{\pm0.044}(11)$ | $0.681_{\pm0.032}(37)$ | $0.946_{\pm0.012}(3)$ | $0.933_{\pm0.004}(5)$ | $0.723_{\pm0.027}(35)$ | $0.875_{\pm0.022}(18)$ |
| pendigits | $0.979_{\pm0.018}(9)$ | $0.999_{\pm0.0}(1)$ | $0.631_{\pm0.247}(39)$ | $0.262_{\pm0.224}(46)$ | $0.966_{\pm0.022}(14)$ | $0.999_{\pm0.0}(1)$ | $0.976_{\pm0.013}(11)$ | $0.779_{\pm0.068}(33)$ | $0.821_{\pm0.068}(32)$ | $0.923_{\pm0.018}(24)$ | $0.748_{\pm0.104}(35)$ | $0.947_{\pm0.009}(19)$ |
| satellite | $0.755_{\pm0.051}(26)$ | $0.878_{\pm0.002}(3)$ | $0.813_{\pm0.01}(12)$ | $0.725_{\pm0.023}(28)$ | $0.886_{\pm0.008}(1)$ | $0.878_{\pm0.003}(3)$ | $0.767_{\pm0.062}(23)$ | $0.69_{\pm0.011}(30)$ | $0.69_{\pm0.036}(30)$ | $0.881_{\pm0.006}(2)$ | $0.65_{\pm0.036}(35)$ | $0.762_{\pm0.02}(24)$ |
| landsat | $0.492_{\pm0.013}(31)$ | $0.774_{\pm0.005}(2)$ | $0.618_{\pm0.06}(12)$ | $0.539_{\pm0.013}(23)$ | $0.741_{\pm0.006}(4)$ | $0.783_{\pm0.006}(1)$ | $0.554_{\pm0.096}(22)$ | $0.523_{\pm0.025}(25)$ | $0.474_{\pm0.012}(34)$ | $0.71_{\pm0.004}(6)$ | $0.454_{\pm0.022}(40)$ | $0.581_{\pm0.003}(17)$ |
| satimage-2 | $0.974_{\pm0.017}(30)$ | $0.999_{\pm0.0}(1)$ | $0.982_{\pm0.009}(26)$ | $0.992_{\pm0.001}(20)$ | $0.997_{\pm0.002}(10)$ | $0.998_{\pm0.001}(6)$ | $0.988_{\pm0.017}(21)$ | $0.935_{\pm0.007}(34)$ | $0.959_{\pm0.014}(33)$ | $0.998_{\pm0.001}(6)$ | $0.892_{\pm0.018}(38)$ | $0.985_{\pm0.001}(25)$ |
| PageBlocks | $0.924_{\pm0.031}(23)$ | $0.962_{\pm0.001}(3)$ | $0.751_{\pm0.097}(40)$ | $0.784_{\pm0.034}(36)$ | $0.931_{\pm0.01}(20)$ | $0.938_{\pm0.003}(18)$ | $0.96_{\pm0.002}(6)$ | $0.642_{\pm0.056}(43)$ | $0.901_{\pm0.025}(26)$ | $0.873_{\pm0.014}(31)$ | $0.811_{\pm0.06}(35)$ | $0.951_{\pm0.003}(11)$ |
| Wilt | $0.951_{\pm0.012}(2)$ | $0.661_{\pm0.016}(10)$ | $0.446_{\pm0.064}(30)$ | $0.597_{\pm0.038}(15)$ | $0.75_{\pm0.048}(7)$ | $0.512_{\pm0.044}(22)$ | $0.562_{\pm0.216}(18)$ | $0.484_{\pm0.046}(26)$ | $0.76_{\pm0.07}(6)$ | $0.653_{\pm0.014}(11)$ | $0.426_{\pm0.071}(34)$ | $0.446_{\pm0.006}(30)$ |
| thyroid | $0.887_{\pm0.118}(37)$ | $0.987_{\pm0.002}(5)$ | $0.944_{\pm0.044}(29)$ | $0.565_{\pm0.088}(43)$ | $0.94_{\pm0.032}(32)$ | $0.98_{\pm0.001}(16)$ | $0.978_{\pm0.011}(19)$ | $0.816_{\pm0.038}(40)$ | $0.987_{\pm0.007}(5)$ | $0.944_{\pm0.012}(29)$ | $0.93_{\pm0.02}(36)$ | $0.989_{\pm0.00}(4)$ |
| Waveform | $0.676_{\pm0.022}(23)$ | $0.755_{\pm0.005}(5)$ | $0.752_{\pm0.076}(6)$ | $0.456_{\pm0.097}(45)$ | $0.692_{\pm0.036}(18)$ | $0.76_{\pm0.012}(4)$ | $0.793_{\pm0.121}(1)$ | $0.458_{\pm0.041}(44)$ | $0.63_{\pm0.034}(27)$ | $0.477_{\pm0.011}(43)$ | $0.443_{\pm0.015}(46)$ | $0.699_{\pm0.015}(17)$ |
| Cardiotocography | $0.733_{\pm0.065}(20)$ | $0.765_{\pm0.001}(12)$ | $0.679_{\pm0.06}(28)$ | $0.273_{\pm0.056}(45)$ | $0.69_{\pm0.013}(27)$ | $0.808_{\pm0.015}(4)$ | $0.773_{\pm0.035}(10)$ | $0.599_{\pm0.047}(33)$ | $0.73_{\pm0.038}(21)$ | $0.582_{\pm0.026}(34)$ | $0.568_{\pm0.088}(37)$ | $0.832_{\pm0.005}(1)$ |
| fault | $0.671_{\pm0.031}(17)$ | $0.811_{\pm0.008}(2)$ | $0.613_{\pm0.02}(29)$ | $0.665_{\pm0.039}(18)$ | $0.781_{\pm0.007}(5)$ | $0.807_{\pm0.012}(3)$ | $0.7_{\pm0.035}(12)$ | $0.469_{\pm0.06}(43)$ | $0.511_{\pm0.031}(39)$ | $0.799_{\pm0.007}(4)$ | $0.467_{\pm0.036}(44)$ | $0.618_{\pm0.022}(27)$ |
| cardio | $0.848_{\pm0.046}(30)$ | $0.945_{\pm0.005}(12)$ | $0.883_{\pm0.039}(24)$ | $0.166_{\pm0.048}(45)$ | $0.879_{\pm0.026}(26)$ | $0.958_{\pm0.006}(3)$ | $0.947_{\pm0.013}(10)$ | $0.796_{\pm0.078}(34)$ | $0.909_{\pm0.025}(20)$ | $0.839_{\pm0.036}(33)$ | $0.794_{\pm0.053}(35)$ | $0.971_{\pm0.002}(1)$ |
| letter | $0.846_{\pm0.005}(15)$ | $0.903_{\pm0.009}(5)$ | $0.751_{\pm0.025}(22)$ | $0.764_{\pm0.018}(21)$ | $0.925_{\pm0.006}(2)$ | $0.927_{\pm0.006}(1)$ | $0.86_{\pm0.028}(12)$ | $0.385_{\pm0.011}(45)$ | $0.73_{\pm0.044}(24)$ | $0.513_{\pm0.009}(3)$ | $0.497_{\pm0.041}(8)$ | $0.459_{\pm0.025}(31)$ |
| yeast | $0.516_{\pm0.039}(2)$ | $0.46_{\pm0.021}(24)$ | $0.489_{\pm0.044}(12)$ | $0.592_{\pm0.016}(1)$ | $0.501_{\pm0.027}(7)$ | $0.457_{\pm0.019}(26)$ | $0.424_{\pm0.026}(34)$ | $0.495_{\pm0.039}(9)$ | $0.471_{\pm0.02}(18)$ | $0.497_{\pm0.041}(8)$ | $0.459_{\pm0.0}(25)$ | $0.636_{\pm0.037}(35)$ |
| vowels | $0.984_{\pm0.007}(2)$ | $0.981_{\pm0.005}(4)$ | $0.7_{\pm0.072}(33)$ | $0.831_{\pm0.039}(24)$ | $0.985_{\pm0.005}(1)$ | $0.984_{\pm0.005}(2)$ | $0.964_{\pm0.007}(9)$ | $0.125_{\pm0.082}(45)$ | $0.86_{\pm0.02}(23)$ | $0.966_{\pm0.003}(8)$ | $0.151_{\pm0.01}(44)$ | $0.636_{\pm0.037}(35)$ |
| Avg Ranking | 18.47 | 6.2 | 25.4 | 29.93 | 13.47 | 8.2 | 14.6 | 34.73 | 22.73 | 16.2 | 35.73 | 19.0 |

| Dataset | CBLOF | CD | ECOD | FB | GMM | HBOS | IForest | KDE | LMDD | LODA | MCD | OCSVM |
|---|---|---|---|---|---|---|---|---|---|---|---|---|
| annthyroid | $0.888_{\pm0.033}(12)$ | $0.624_{\pm0.003}(41)$ | $0.788_{\pm0.00}(29)$ | $0.923_{\pm0.024}(6)$ | $0.834_{\pm0.02}(25)$ | $0.71_{\pm0.058}(36)$ | $0.912_{\pm0.011}(9)$ | $0.88_{\pm0.026}(15)$ | $0.748_{\pm0.02}(33)$ | $0.736_{\pm0.06}(34)$ | $0.921_{\pm0.005}(7)$ | $0.875_{\pm0.027}(18)$ |
| pendigits | $0.959_{\pm0.014}(18)$ | $0.552_{\pm0.007}(41)$ | $0.929_{\pm0.001}(22)$ | $0.997_{\pm0.002}(4)$ | $0.847_{\pm0.005}(29)$ | $0.937_{\pm0.001}(20)$ | $0.971_{\pm0.006}(13)$ | $0.971_{\pm0.006}(13)$ | $0.521_{\pm0.057}(42)$ | $0.896_{\pm0.034}(26)$ | $0.93_{\pm0.021}(21)$ | $0.966_{\pm0.004}(14)$ |
| satellite | $0.852_{\pm0.025}(8)$ | $0.577_{\pm0.002}(39)$ | $0.584_{\pm0.004}(38)$ | $0.846_{\pm0.005}(9)$ | $0.802_{\pm0.002}(16)$ | $0.867_{\pm0.004}(6)$ | $0.798_{\pm0.02}(17)$ | $0.875_{\pm0.003}(5)$ | $0.521_{\pm0.057}(42)$ | $0.694_{\pm0.014}(29)$ | $0.809_{\pm0.002}(14)$ | $0.756_{\pm0.003}(25)$ |
| landsat | $0.68_{\pm0.024}(7)$ | $0.458_{\pm0.003}(38)$ | $0.367_{\pm0.002}(45)$ | $0.751_{\pm0.007}(3)$ | $0.495_{\pm0.003}(30)$ | $0.697_{\pm0.007}(7)$ | $0.611_{\pm0.007}(14)$ | $0.739_{\pm0.005}(5)$ | $0.397_{\pm0.033}(44)$ | $0.424_{\pm0.034}(43)$ | $0.613_{\pm0.002}(13)$ | $0.461_{\pm0.002}(36)$ |
| satimage-2 | $0.998_{\pm0.0}(6)$ | $0.922_{\pm0.002}(36)$ | $0.965_{\pm0.001}(32)$ | $0.997_{\pm0.0}(10)$ | $0.995_{\pm0.001}(16)$ | $0.994_{\pm0.001}(19)$ | $0.999_{\pm0.0}(1)$ | $0.95_{\pm0.001}(11)$ | $0.553_{\pm0.029}(44)$ | $0.986_{\pm0.006}(24)$ | $0.996_{\pm0.0}(14)$ | $0.997_{\pm0.0}(10)$ |
| PageBlocks | $0.954_{\pm0.008}(9)$ | $0.876_{\pm0.006}(29)$ | $0.914_{\pm0.004}(25)$ | $0.971_{\pm0.01}(1)$ | $0.959_{\pm0.04}(8)$ | $0.772_{\pm0.008}(37)$ | $0.929_{\pm0.006}(21)$ | $0.95_{\pm0.001}(12)$ | $0.346_{\pm0.06}(44)$ | $0.413_{\pm0.085}(36)$ | $0.871_{\pm0.048}(41)$ | $0.922_{\pm0.002}(24)$ |
| Wilt | $0.416_{\pm0.02}(35)$ | $0.619_{\pm0.019}(13)$ | $0.389_{\pm0.006}(37)$ | $0.742_{\pm0.086}(8)$ | $0.722_{\pm0.02}(9)$ | $0.346_{\pm0.014}(41)$ | $0.492_{\pm0.015}(28)$ | $0.346_{\pm0.06}(44)$ | $0.413_{\pm0.085}(36)$ | $0.329_{\pm0.096}(44)$ | $0.859_{\pm0.004}(5)$ | $0.322_{\pm0.006}(46)$ |
| thyroid | $0.984_{\pm0.002}(10)$ | $0.903_{\pm0.018}(36)$ | $0.978_{\pm0.002}(19)$ | $0.958_{\pm0.012}(25)$ | $0.977_{\pm0.003}(21)$ | $0.981_{\pm0.002}(15)$ | $0.99_{\pm0.001}(2)$ | $0.985_{\pm0.001}(8)$ | $0.959_{\pm0.009}(24)$ | $0.942_{\pm0.014}(31)$ | $0.986_{\pm0.001}(7)$ | $0.984_{\pm0.001}(10)$ |
| Waveform | $0.726_{\pm0.017}(9)$ | $0.552_{\pm0.009}(38)$ | $0.607_{\pm0.005}(32)$ | $0.778_{\pm0.005}(2)$ | $0.583_{\pm0.005}(36)$ | $0.705_{\pm0.005}(13)$ | $0.723_{\pm0.023}(10)$ | $0.772_{\pm0.004}(3)$ | $0.592_{\pm0.025}(33)$ | $0.618_{\pm0.055}(30)$ | $0.583_{\pm0.006}(36)$ | $0.702_{\pm0.004}(15)$ |
| Cardiotocography | $0.718_{\pm0.018}(23)$ | $0.623_{\pm0.032}(32)$ | $0.785_{\pm0.005}(9)$ | $0.787_{\pm0.007}(8)$ | $0.759_{\pm0.004}(13)$ | $0.699_{\pm0.015}(25)$ | $0.801_{\pm0.005}(5)$ | $0.771_{\pm0.002}(11)$ | $0.713_{\pm0.044}(24)$ | $0.757_{\pm0.043}(14)$ | $0.676_{\pm0.02}(29)$ | $0.821_{\pm0.006}(3)$ |
| fault | $0.741_{\pm0.016}(13)$ | $0.623_{\pm0.012}(32)$ | $0.785_{\pm0.009}(9)$ | $0.651_{\pm0.027}(20)$ | $0.69_{\pm0.011}(15)$ | $0.569_{\pm0.011}(34)$ | $0.651_{\pm0.016}(20)$ | $0.815_{\pm0.004}(1)$ | $0.815_{\pm0.004}(1)$ | $0.51_{\pm0.059}(40)$ | $0.63_{\pm0.02}(23)$ | $0.604_{\pm0.00}(30)$ |
| cardio | $0.955_{\pm0.016}(7)$ | $0.856_{\pm0.013}(29)$ | $0.937_{\pm0.005}(15)$ | $0.92_{\pm0.017}(19)$ | $0.946_{\pm0.002}(11)$ | $0.844_{\pm0.009}(32)$ | $0.94_{\pm0.011}(13)$ | $0.956_{\pm0.004}(5)$ | $0.865_{\pm0.081}(28)$ | $0.883_{\pm0.036}(24)$ | $0.87_{\pm0.037}(27)$ | $0.965_{\pm0.001}(2)$ |
| letter | $0.779_{\pm0.014}(20)$ | $0.761_{\pm0.011}(22)$ | $0.572_{\pm0.005}(36)$ | $0.87_{\pm0.006}(8)$ | $0.842_{\pm0.06}(16)$ | $0.599_{\pm0.008}(35)$ | $0.621_{\pm0.015}(30)$ | $0.919_{\pm0.007}(3)$ | $0.519_{\pm0.014}(39)$ | $0.541_{\pm0.041}(37)$ | $0.812_{\pm0.017}(18)$ | $0.615_{\pm0.007}(31)$ |
| yeast | $0.503_{\pm0.024}(4)$ | $0.413_{\pm0.016}(39)$ | $0.446_{\pm0.015}(29)$ | $0.475_{\pm0.017}(16)$ | $0.475_{\pm0.007}(16)$ | $0.445_{\pm0.018}(30)$ | $0.417_{\pm0.015}(38)$ | $0.424_{\pm0.017}(34)$ | $0.481_{\pm0.015}(14)$ | $0.467_{\pm0.015}(20)$ | $0.461_{\pm0.016}(22)$ | $0.454_{\pm0.016}(27)$ |
| vowels | $0.901_{\pm0.009}(17)$ | $0.878_{\pm0.009}(20)$ | $0.59_{\pm0.009}(37)$ | $0.953_{\pm0.011}(12)$ | $0.947_{\pm0.006}(14)$ | $0.698_{\pm0.004}(34)$ | $0.772_{\pm0.026}(27)$ | $0.963_{\pm0.005}(11)$ | $0.539_{\pm0.021}(39)$ | $0.727_{\pm0.036}(30)$ | $0.739_{\pm0.043}(28)$ | $0.823_{\pm0.004}(25)$ |
| Avg Ranking | 12.87 | 31.8 | 30.0 | 10.07 | 19.27 | 26.47 | 17.67 | 10.4 | 33.67 | 30.2 | 19.8 | 20.53 |

| Dataset | QMCD | Sampling | ABOD | COF | INNE | KNN | KPCA | LOF | PCA |
|---|---|---|---|---|---|---|---|---|---|
| annthyroid | $0.759_{\pm0.011}(16)$ | $0.876_{\pm0.003}(16)$ | $0.824_{\pm0.00}(26)$ | $0.517_{\pm0.034}(45)$ | $0.855_{\pm0.016}(24)$ | $0.875_{\pm0.005}(18)$ | $0.16_{\pm0.016}(46)$ | $0.539_{\pm0.043}(44)$ | $0.645_{\pm0.002}(40)$ |
| pendigits | $0.909_{\pm0.003}(25)$ | $0.959_{\pm0.012}(16)$ | $0.575_{\pm0.005}(40)$ | $0.52_{\pm0.004}(44)$ | $0.854_{\pm0.01}(28)$ | $0.64_{\pm0.005}(38)$ | $0.54_{\pm0.182}(43)$ | $0.497_{\pm0.004}(45)$ | $0.929_{\pm0.003}(22)$ |
| satellite | $0.859_{\pm0.004}(7)$ | $0.819_{\pm0.013}(11)$ | $0.503_{\pm0.006}(46)$ | $0.516_{\pm0.006}(43)$ | $0.669_{\pm0.009}(34)$ | $0.629_{\pm0.004}(37)$ | $0.512_{\pm0.05}(44)$ | $0.527_{\pm0.005}(41)$ | $0.577_{\pm0.003}(39)$ |
| landsat | $0.683_{\pm0.001}(8)$ | $0.63_{\pm0.04}(10)$ | $0.456_{\pm0.002}(39)$ | $0.52_{\pm0.005}(28)$ | $0.467_{\pm0.022}(35)$ | $0.53_{\pm0.004}(24)$ | $0.478_{\pm0.032}(33)$ | $0.521_{\pm0.004}(26)$ | $0.343_{\pm0.001}(46)$ |
| satimage-2 | $0.995_{\pm0.04}(16)$ | $0.999_{\pm0.0}(1)$ | $0.727_{\pm0.005}(41)$ | $0.56_{\pm0.005}(43)$ | $0.996_{\pm0.001}(14)$ | $0.906_{\pm0.002}(37)$ | $0.601_{\pm0.144}(42)$ | $0.534_{\pm0.005}(45)$ | $0.976_{\pm0.001}(28)$ |
| PageBlocks | $0.77_{\pm0.03}(38)$ | $0.944_{\pm0.006}(16)$ | $0.819_{\pm0.006}(34)$ | $0.579_{\pm0.009}(45)$ | $0.948_{\pm0.007}(14)$ | $0.891_{\pm0.004}(28)$ | $0.187_{\pm0.011}(46)$ | $0.702_{\pm0.009}(42)$ | $0.897_{\pm0.003}(27)$ |
| Wilt | $0.345_{\pm0.005}(43)$ | $0.43_{\pm0.026}(32)$ | $0.545_{\pm0.006}(20)$ | $0.603_{\pm0.008}(14)$ | $0.323_{\pm0.013}(45)$ | $0.503_{\pm0.004}(24)$ | $0.593_{\pm0.006}(16)$ | $0.54_{\pm0.014}(21)$ | $0.429_{\pm0.005}(33)$ |
| thyroid | $0.827_{\pm0.029}(39)$ | $0.98_{\pm0.005}(16)$ | $0.915_{\pm0.005}(35)$ | $0.438_{\pm0.015}(45)$ | $0.97_{\pm0.007}(22)$ | $0.97_{\pm0.003}(22)$ | $0.166_{\pm0.047}(46)$ | $0.565_{\pm0.01}(43)$ | $0.946_{\pm0.002}(27)$ |
| Waveform | $0.747_{\pm0.005}(7)$ | $0.7_{\pm0.058}(16)$ | $0.612_{\pm0.021}(31)$ | $0.65_{\pm0.022}(26)$ | $0.722_{\pm0.012}(11)$ | $0.704_{\pm0.007}(14)$ | $0.514_{\pm0.045}(40)$ | $0.671_{\pm0.012}(24)$ | $0.63_{\pm0.007}(27)$ |
| Cardiotocography | $0.575_{\pm0.016}(35)$ | $0.729_{\pm0.02}(22)$ | $0.408_{\pm0.094}(44)$ | $0.497_{\pm0.006}(40)$ | $0.524_{\pm0.012}(37)$ | $0.641_{\pm0.013}(30)$ | $0.462_{\pm0.094}(42)$ | $0.525_{\pm0.028}(39)$ | $0.695_{\pm0.008}(26)$ |
| fault | $0.62_{\pm0.043}(26)$ | $0.692_{\pm0.014}(14)$ | $0.656_{\pm0.006}(19)$ | $0.524_{\pm0.012}(37)$ | $0.544_{\pm0.014}(34)$ | $0.677_{\pm0.008}(16)$ | $0.561_{\pm0.055}(32)$ | $0.554_{\pm0.013}(33)$ | $0.471_{\pm0.027}(42)$ |
| cardio | $0.722_{\pm0.027}(37)$ | $0.928_{\pm0.026}(18)$ | $0.506_{\pm0.014}(43)$ | $0.507_{\pm0.012}(42)$ | $0.885_{\pm0.011}(23)$ | $0.668_{\pm0.013}(38)$ | $0.603_{\pm0.073}(39)$ | $0.506_{\pm0.011}(43)$ | $0.846_{\pm0.194}(31)$ |
| letter | $0.612_{\pm0.006}(34)$ | $0.729_{\pm0.022}(25)$ | $0.852_{\pm0.006}(14)$ | $0.853_{\pm0.006}(13)$ | $0.696_{\pm0.009}(26)$ | $0.864_{\pm0.006}(11)$ | $0.517_{\pm0.024}(40)$ | $0.841_{\pm0.007}(17)$ | $0.52_{\pm0.008}(38)$ |
| yeast | $0.406_{\pm0.013}(40)$ | $0.467_{\pm0.037}(20)$ | $0.397_{\pm0.014}(42)$ | $0.441_{\pm0.02}(31)$ | $0.382_{\pm0.015}(45)$ | $0.383_{\pm0.016}(44)$ | $0.494_{\pm0.016}(10)$ | $0.434_{\pm0.02}(32)$ | $0.392_{\pm0.016}(43)$ |
| vowels | $0.709_{\pm0.005}(31)$ | $0.867_{\pm0.037}(21)$ | $0.91_{\pm0.008}(16)$ | $0.879_{\pm0.014}(19)$ | $0.864_{\pm0.014}(21)$ | $0.952_{\pm0.007}(13)$ | $0.892_{\pm0.012}(18)$ | | $0.591_{\pm0.016}(36)$ |
| Avg Ranking | 27.87 | 16.93 | 32.67 | 34.33 | 27.13 | 27.07 | 37.27 | 34.33 | 33.67 |

Table 9: AUPRC results on 15 medium datasets, where we compare TCCM to 44 baselines (with 5 independent runs). We report the mean ± std (rank in terms of mean among all anomaly detectors).

| Dataset | TCMM (Ours) | AE | AE-1SVM | ALAD | AnoGAN | DAGMM | DeepSVDD | DIF | DROCC | DTE-Cat | DTE-DDPM | DTE-Gaussian |
|---|---|---|---|---|---|---|---|---|---|---|---|---|
| annthyroid | 0.695±0.065(6) | 0.624±0.045(12) | 0.592±0.012(17) | 0.202±0.028(43) | 0.5±0.069(26) | 0.262±0.112(40) | 0.588±0.065(19) | 0.506±0.058(24) | 0.638±0.043(9) | 0.861±0.008(1) | 0.504±0.035(25) | 0.721±0.06(2) |
| pendigits | 0.701±0.037(8) | 0.565±0.127(12) | 0.464±0.047(16) | 0.06±0.011(44) | 0.172±0.137(31) | 0.136±0.131(36) | 0.302±0.205(24) | 0.726±0.074(7) | 0.142±0.035(34) | 0.445±0.087(18) | 0.209±0.031(29) | 0.808±0.05(6) |
| satellite | 0.861±0.004(10) | 0.851±0.005(13) | 0.819±0.003(24) | 0.493±0.029(46) | 0.685±0.104(36) | 0.698±0.074(34) | 0.764±0.054(29) | 0.839±0.004(19) | 0.76±0.099(30) | 0.853±0.005(12) | 0.821±0.004(22) | 0.845±0.007(17) |
| landsat | 0.416±0.007(12) | 0.411±0.012(13) | 0.413±0.01(13) | 0.327±0.015(34) | 0.319±0.097(38) | 0.365±0.057(27) | 0.305±0.032(43) | 0.437±0.01(11) | 0.411±0.014(14) | 0.394±0.022(20) | 0.339±0.065(32) | 0.342±0.016(31) |
| satimage-2 | 0.946±0.027(10) | 0.96±0.011(9) | 0.899±0.046(18) | 0.035±0.006(46) | 0.533±0.222(33) | 0.124±0.078(42) | 0.801±0.264(25) | 0.937±0.033(14) | 0.241±0.194(38) | 0.518±0.031(34) | 0.603±0.102(32) | 0.815±0.039(23) |
| PageBlocks | 0.871±0.015(2) | 0.828±0.04(12) | 0.839±0.03(9) | 0.27±0.069(45) | 0.583±0.033(35) | 0.505±0.153(37) | 0.728±0.069(23) | 0.788±0.016(20) | 0.83±0.037(10) | 0.848±0.015(7) | 0.679±0.029(27) | 0.855±0.023(4) |
| Wilt | 0.478±0.029(3) | 0.102±0.008(20) | 0.09±0.016(28) | 0.093±0.005(25) | 0.076±0.009(38) | 0.147±0.047(13) | 0.074±0.007(40) | 0.073±0.001(41) | 0.093±0.019(25) | 0.282±0.01(5) | 0.098±0.004(22) | 0.132±0.008(14) |
| thyroid | 0.802±0.07(8) | 0.787±0.067(13) | 0.859±0.017(2) | 0.106±0.027(44) | 0.732±0.21(21) | 0.276±0.179(41) | 0.814±0.037(7) | 0.823±0.022(5) | 0.703±0.065(22) | 0.873±0.016(1) | 0.647±0.061(24) | 0.539±0.107(31) |
| Waveform | 0.143±0.031(12) | 0.118±0.014(17) | 0.103±0.004(22) | 0.063±0.014(42) | 0.089±0.009(27) | 0.057±0.001(45) | 0.085±0.028(29) | 0.125±0.009(16) | 0.191±0.07(8) | 0.089±0.009(27) | 0.071±0.015(38) | 0.08±0.003(32) |
| Cardiotocography | 0.743±0.016(13) | 0.655±0.014(16) | 0.661±0.006(14) | 0.424±0.014(38) | 0.661±0.117(14) | NAN | 0.718±0.10(6) | 0.597±0.013(26) | 0.41±0.126(41) | 0.624±0.018(23) | 0.489±0.018(35) | 0.686±0.015(11) |
| fault | 0.769±0.005(5) | 0.729±0.068(8) | 0.666±0.012(17) | 0.529±0.03(41) | 0.533±0.074(39) | NAN | 0.576±0.047(33) | 0.717±0.021(10) | 0.641±0.029(20) | 0.697±0.015(14) | 0.629±0.015(24) | 0.724±0.01(9) |
| cardio | 0.847±0.022(4) | 0.763±0.043(17) | 0.824±0.035(9) | 0.233±0.03(43) | 0.761±0.07(18) | NAN | 0.778±0.081(13) | 0.836±0.021(6) | 0.369±0.21(40) | 0.657±0.043(27) | 0.473±0.023(37) | 0.77±0.007(15) |
| letter | 0.573±0.03(5) | 0.358±0.031(18) | 0.485±0.045(21) | 0.971±0.005(7) | 0.124±0.008(43) | NAN | 0.138±0.018(38) | 0.234±0.021(28) | 0.247±0.018(26) | 0.557±0.041(7) | 0.229±0.031(29) | 0.553±0.041(8) |
| yeast | 0.518±0.011(3) | 0.485±0.014(28) | 0.485±0.016(28) | 0.498±0.008(15) | 0.472±0.032(35) | NAN | 0.465±0.018(38) | 0.457±0.012(40) | 0.508±0.037(6) | 0.499±0.022(13) | 0.515±0.012(5) | 0.499±0.02(13) |
| vowels | 0.76±0.021(8) | 0.543±0.03(14) | 0.209±0.052(29) | 0.079±0.032(41) | 0.069±0.011(42) | NAN | 0.111±0.044(39) | 0.276±0.04(26) | 0.111±0.063(39) | 0.809±0.032(3) | 0.191±0.045(31) | 0.771±0.028(7) |
| Avg Ranking | 6.6 | 14.87 | 18.47 | 39.2 | 31.73 | 35.0 | 27.07 | 19.53 | 24.13 | 14.13 | 27.47 | 14.87 |

| Dataset | DTE-IG | DTE-NP | GANomaly | GOAD | ICL | LUNAR | MCM | MO_GAAL | PlanarFlow | SLAD | SO_GAAL | VAE |
|---|---|---|---|---|---|---|---|---|---|---|---|---|
| annthyroid | 0.581±0.067(20) | 0.694±0.012(7) | 0.331±0.104(38) | 0.425±0.034(31) | 0.444±0.015(30) | 0.58±0.054(21) | 0.634±0.081(11) | 0.355±0.051(37) | 0.707±0.041(4) | 0.714±0.01(3) | 0.3±0.05(39) | 0.612±0.033(13) |
| pendigits | 0.647±0.174(9) | 0.976±0.016(2) | 0.094±0.057(38) | 0.036±0.024(46) | 0.643±0.138(10) | 0.983±0.01(1) | 0.561±0.154(13) | 0.232±0.077(28) | 0.739±0.052(32) | 0.884±0.006(5) | 0.185±0.076(30) | 0.419±0.052(20) |
| satellite | 0.807±0.03(26) | 0.893±0.003(3) | 0.831±0.011(20) | 0.782±0.018(28) | 0.899±0.006(1) | 0.896±0.002(2) | 0.82±0.04(23) | 0.748±0.01(31) | 0.739±0.052(32) | 0.895±0.006(7) | 0.722±0.025(33) | 0.817±0.01(25) |
| landsat | 0.37±0.015(25) | 0.614±0.008(4) | 0.402±0.051(18) | 0.377±0.005(24) | 0.687±0.016(2) | 0.657±0.013(3) | 0.394±0.04(20) | 0.344±0.035(30) | 0.319±0.007(38) | 0.495±0.006(7) | 0.302±0.113(44) | 0.382±0.004(22) |
| satimage-2 | 0.647±0.148(31) | 0.979±0.005(1) | 0.869±0.045(21) | 0.971±0.005(7) | 0.941±0.042(12) | 0.973±0.005(5) | 0.75±0.299(28) | 0.517±0.223(35) | 0.655±0.152(30) | 0.927±0.035(17) | 0.147±0.138(39) | 0.877±0.006(20) |
| PageBlocks | 0.811±0.064(16) | 0.859±0.009(3) | 0.472±0.167(39) | 0.642±0.034(31) | 0.821±0.019(15) | 0.83±0.015(10) | 0.853±0.018(5) | 0.444±0.017(40) | 0.687±0.071(26) | 0.691±0.028(25) | 0.585±0.09(34) | 0.794±0.016(18) |
| Wilt | 0.745±0.06(1) | 0.132±0.005(14) | 0.089±0.015(29) | 0.17±0.018(9) | 0.299±0.075(4) | 0.095±0.01(24) | 0.16±0.16(11) | 0.097±0.013(23) | 0.185±0.045(8) | 0.158±0.003(12) | 0.088±0.014(30) | 0.66±0.006(2) |
| thyroid | 0.477±0.158(34) | 0.797±0.03(9) | 0.567±0.13(28) | 0.443±0.061(36) | 0.477±0.13(34) | 0.755±0.037(19) | 0.70±0.071(18) | 0.37±0.119(39) | 0.788±0.017(12) | 0.63±0.006(26) | 0.482±0.047(33) | 0.841±0.017(3) |
| Waveform | 0.139±0.02(13) | 0.283±0.017(5) | 0.132±0.04(14) | 0.069±0.038(40) | 0.164±0.012(10) | 0.306±0.016(3) | 0.393±0.229(1) | 0.062±0.007(43) | 0.223±0.045(7) | 0.051±0.001(46) | 0.058±0.007(44) | 0.1±0.008(24) |
| Cardiotocography | 0.647±0.058(19) | 0.687±0.007(10) | 0.567±0.062(28) | 0.258±0.02(45) | 0.632±0.02(22) | 0.744±0.016(2) | 0.703±0.045(8) | 0.521±0.065(32) | 0.601±0.037(25) | 0.555±0.018(30) | 0.478±0.085(36) | 0.755±0.006(1) |
| fault | 0.702±0.017(13) | 0.792±0.012(4) | 0.61±0.064(29) | 0.64±0.057(21) | 0.767±0.009(6) | 0.796±0.016(2) | 0.707±0.016(12) | 0.537±0.061(36) | 0.54±0.024(34) | 0.793±0.009(3) | 0.535±0.066(38) | 0.629±0.022(24) |
| cardio | 0.603±0.066(29) | 0.814±0.02(11) | 0.687±0.054(21) | 0.12±0.014(45) | 0.688±0.051(23) | 0.853±0.029(2) | 0.827±0.03(8) | 0.568±0.14(31) | 0.735±0.022(19) | 0.677±0.034(24) | 0.52±0.094(33) | 0.875±0.016(1) |
| letter | 0.572±0.026(6) | 0.525±0.031(11) | 0.31±0.09(22) | 0.323±0.032(19) | 0.639±0.032(1) | 0.627±0.04(2) | 0.54±0.079(9) | 0.124±0.016(43) | 0.315±0.067(20) | 0.601±0.04(3) | 0.123±0.016(44) | 0.19±0.012(32) |
| yeast | 0.537±0.029(2) | 0.494±0.014(18) | 0.488±0.027(25) | 0.589±0.017(1) | 0.506±0.019(8) | 0.493±0.015(19) | 0.463±0.016(39) | 0.49±0.022(22) | 0.486±0.012(27) | 0.516±0.009(4) | 0.5±0.02(12) | 0.482±0.015(30) |
| vowels | 0.874±0.039(2) | 0.796±0.064(5) | 0.199±0.121(30) | 0.322±0.08(23) | 0.885±0.03(1) | 0.809±0.062(3) | 0.809±0.062(3) | 0.038±0.029(45) | 0.39±0.102(17) | 0.758±0.029(9) | 0.039±0.007(44) | 0.128±0.029(36) |
| Avg Ranking | 16.4 | 7.13 | 26.67 | 27.07 | 11.93 | 7.87 | 14.53 | 34.33 | 22.27 | 16.07 | 35.67 | 20.07 |

| Dataset | CBLOF | CD | ECOD | FB | GMM | HBOS | IForest | KDE | LMDD | LODA | MCD | OCSVM |
|---|---|---|---|---|---|---|---|---|---|---|---|---|
| annthyroid | 0.636±0.057(10) | 0.226±0.011(42) | 0.402±0.006(34) | 0.54±0.122(23) | 0.543±0.039(22) | 0.419±0.047(32) | 0.612±0.044(13) | 0.601±0.04(15) | 0.46±0.05(28) | 0.404±0.045(33) | 0.645±0.05(8) | 0.592±0.042(17) |
| pendigits | 0.454±0.105(17) | 0.051±0.001(45) | 0.417±0.009(21) | 0.93±0.035(4) | 0.156±0.004(33) | 0.426±0.012(19) | 0.57±0.054(11) | 0.969±0.01(3) | 0.276±0.09(35) | 0.394±0.075(22) | 0.132±0.06(37) | 0.523±0.033(14) |
| satellite | 0.87±0.015(9) | 0.582±0.005(40) | 0.66±0.004(38) | 0.879±0.005(7) | 0.848±0.002(15) | 0.883±0.004(6) | 0.843±0.012(18) | 0.891±0.003(4) | 0.555±0.055(41) | 0.791±0.013(27) | 0.849±0.001(14) | 0.823±0.002(21) |
| landsat | 0.467±0.026(9) | 0.314±0.002(41) | 0.281±0.01(45) | 0.693±0.006(1) | 0.337±0.002(33) | 0.526±0.01(6) | 0.445±0.025(10) | 0.556±0.005(5) | 0.379±0.075(23) | 0.326±0.043(35) | 0.401±0.002(19) | 0.325±0.02(36) |
| satimage-2 | 0.976±0.002(4) | 0.268±0.014(37) | 0.746±0.015(29) | 0.94±0.009(13) | 0.798±0.04(26) | 0.833±0.009(22) | 0.932±0.005(15) | 0.979±0.004(1) | 0.036±0.013(45) | 0.931±0.015(16) | 0.815±0.008(23) | 0.973±0.002(5) |
| PageBlocks | 0.825±0.038(13) | 0.545±0.011(36) | 0.659±0.011(30) | 0.888±0.007(1) | 0.822±0.02(14) | 0.353±0.011(41) | 0.699±0.02(24) | 0.844±0.009(8) | 0.502±0.092(38) | 0.664±0.071(29) | 0.74±0.015(22) | 0.796±0.011(17) |
| Wilt | 0.079±0.003(36) | 0.121±0.008(18) | 0.079±0.004(36) | 0.19±0.053(7) | 0.163±0.012(10) | 0.076±0.002(38) | 0.076±0.003(38) | 0.111±0.008(20) | 0.089±0.009(29) | 0.071±0.04(42) | 0.261±0.005(6) | 0.069±0.001(45) |
| thyroid | 0.787±0.027(13) | 0.299±0.042(40) | 0.636±0.023(25) | 0.426±0.16(38) | 0.761±0.031(17) | 0.747±0.023(20) | 0.821±0.03(6) | 0.791±0.031(10) | 0.697±0.047(23) | 0.597±0.09(27) | 0.825±0.016(4) | 0.79±0.03(11) |
| Waveform | 0.246±0.008(6) | 0.071±0.001(38) | 0.078±0.003(33) | 0.318±0.16(2) | 0.078±0.002(33) | 0.094±0.003(26) | 0.111±0.008(20) | 0.289±0.01(5) | 0.072±0.009(37) | 0.078±0.003(33) | 0.078±0.002(33) | 0.112±0.004(19) |
| Cardiotocography | 0.644±0.018(20) | 0.511±0.011(33) | 0.654±0.006(17) | 0.683±0.012(12) | 0.676±0.007(13) | 0.584±0.011(27) | 0.689±0.02(9) | 0.709±0.007(7) | 0.605±0.053(24) | 0.639±0.09(21) | 0.531±0.029(31) | 0.731±0.01(5) |
| fault | 0.83±0.099(7) | 0.609±0.033(30) | 0.485±0.006(34) | 0.717±0.009(20) | 0.68±0.04(16) | 0.68±0.016(12) | 0.612±0.015(28) | 0.769±0.032(16) | 0.846±0.015(5) | 0.687±0.114(21) | 0.525±0.056(42) | 0.612±0.026(27) |
| cardio | 0.83±0.028(7) | 0.52±0.028(33) | 0.717±0.007(20) | 0.676±0.05(25) | 0.68±0.01(12) | 0.811±0.01(12) | 0.612±0.015(28) | 0.769±0.032(16) | 0.846±0.015(5) | 0.687±0.114(26) | 0.549±0.041(32) | 0.85±0.013(3) |
| letter | 0.296±0.024(24) | 0.299±0.01(23) | 0.142±0.003(36) | 0.503±0.038(12) | 0.433±0.02(15) | 0.15±0.005(35) | 0.159±0.007(34) | 0.588±0.01(14) | 0.13±0.06(42) | 0.138±0.015(38) | 0.313±0.032(21) | 0.207±0.01(30) |
| yeast | 0.508±0.02(6) | 0.452±0.01(42) | 0.498±0.013(15) | 0.503±0.019(11) | 0.477±0.04(33) | 0.492±0.012(20) | 0.232±0.053(27) | 0.74±0.066(10) | 0.506±0.009(8) | 0.495±0.031(17) | 0.488±0.013(25) | 0.482±0.015(24) |
| vowels | 0.332±0.032(22) | 0.333±0.013(21) | 0.146±0.011(35) | 0.622±0.067(13) | 0.689±0.068(11) | 0.181±0.022(33) | 0.232±0.053(27) | 0.74±0.066(10) | 0.121±0.027(37) | 0.224±0.058(28) | 0.153±0.097(34) | 0.377±0.028(18) |
| Avg Ranking | 13.73 | 34.6 | 30.6 | 13.0 | 19.13 | 25.67 | 19.13 | 10.07 | 31.0 | 29.07 | 22.4 | 19.07 |

| Dataset | QMCD | Sampling | ABOD | COF | INNE | KNN | KPCA | LOF | PCA |
|---|---|---|---|---|---|---|---|---|---|
| annthyroid | 0.373±0.011(35) | 0.598±0.022(16) | 0.356±0.007(36) | 0.143±0.012(45) | 0.456±0.042(29) | 0.488±0.016(27) | 0.082±0.002(46) | 0.178±0.017(44) | 0.262±0.005(40) |
| pendigits | 0.282±0.014(25) | 0.499±0.08(15) | 0.07±0.005(40) | 0.063±0.004(42) | 0.17±0.022(32) | 0.088±0.003(39) | 0.064±0.034(41) | 0.061±0.003(43) | 0.337±0.006(23) |
| satellite | 0.877±0.005(8) | 0.848±0.024(15) | 0.503±0.006(45) | 0.52±0.004(43) | 0.688±0.015(35) | 0.637±0.002(39) | 0.529±0.041(42) | 0.508±0.002(44) | 0.674±0.003(37) |
| landsat | 0.417±0.007(8) | 0.405±0.058(17) | 0.316±0.002(40) | 0.396±0.005(26) | 0.324±0.017(37) | 0.359±0.002(29) | 0.309±0.011(42) | 0.36±0.005(28) | 0.266±0.0(46) |
| satimage-2 | 0.942±0.004(11) | 0.978±0.007(3) | 0.127±0.007(41) | 0.093±0.029(43) | 0.798±0.066(26) | 0.37±0.025(36) | 0.134±0.056(40) | 0.051±0.01(44) | 0.887±0.004(19) |
| PageBlocks | 0.347±0.023(43) | 0.793±0.018(19) | 0.588±0.03(33) | 0.29±0.006(44) | 0.779±0.019(21) | 0.677±0.008(28) | 0.119±0.002(46) | 0.344±0.008(42) | 0.633±0.01(32) |
| Wilt | 0.071±0.001(42) | 0.081±0.004(34) | 0.102±0.002(20) | 0.124±0.002(17) | 0.069±0.001(45) | 0.092±0.001(27) | 0.127±0.006(16) | 0.103±0.004(19) | 0.081±0.001(34) |
| thyroid | 0.483±0.025(32) | 0.784±0.061(15) | 0.228±0.013(42) | 0.048±0.002(45) | 0.564±0.096(29) | 0.549±0.039(30) | 0.026±0.001(46) | 0.11±0.008(43) | 0.429±0.02(37) |
| Waveform | 0.111±0.004(20) | 0.17±0.047(9) | 0.083±0.009(31) | 0.098±0.011(25) | 0.126±0.008(15) | 0.145±0.013(11) | 0.16±0.03(9) | 0.127±0.003(16) | 0.084±0.003(30) |
| Cardiotocography | 0.429±0.013(37) | 0.654±0.072(17) | 0.35±0.004(44) | 0.387±0.011(42) | 0.53±0.007(40) | 0.501±0.008(34) | 0.422±0.007(39) | 0.412±0.02(43) | 0.566±0.006(29) |
| fault | 0.611±0.035(28) | 0.674±0.011(16) | 0.634±0.006(23) | 0.53±0.007(40) | 0.583±0.016(31) | 0.651±0.016(18) | 0.539±0.059(35) | 0.49±0.023(44) | 0.49±0.023(37) |
| cardio | 0.475±0.043(36) | 0.775±0.084(14) | 0.241±0.015(41) | 0.223±0.021(44) | 0.499±0.032(35) | 0.414±0.02(38) | 0.379±0.05(39) | 0.24±0.009(42) | 0.585±0.229(30) |
| letter | 0.167±0.008(33) | 0.237±0.028(27) | 0.398±0.016(17) | 0.536±0.02(10) | 0.276±0.013(25) | 0.419±0.015(16) | 0.141±0.025(37) | 0.44±0.01(14) | 0.136±0.03(40) |
| yeast | 0.49±0.014(22) | 0.491±0.018(21) | 0.453±0.01(41) | 0.468±0.012(37) | 0.447±0.01(44) | 0.446±0.012(45) | 0.481±0.01(31) | 0.47±0.014(36) | 0.452±0.011(42) |
| vowels | 0.186±0.022(32) | 0.307±0.102(24) | 0.461±0.012(16) | 0.335±0.036(20) | 0.345±0.054(19) | 0.498±0.059(15) | 0.301±0.017(25) | 0.064±0.012(38) | 0.377±0.028(18) |
| Avg Ranking | 27.47 | 17.47 | 34.0 | 34.87 | 30.47 | 29.13 | 39.0 | 35.07 | 34.73 |

Table 10: AUROC results on 11 large datasets, where we compare TCCM to 44 baselines (with 5 independent runs). We report the mean $\pm$ std (rank).

| Dataset | TCCM (Ours) | AE | AE-1SVM | ALAD | AnoGAN | DAGMM | DeepSVDD | DIF | DROCC | DTE-Cat | DTE-DDPM | DTE-Gaussian |
|---|---|---|---|---|---|---|---|---|---|---|---|---|
| ALOI | $0.565_{\pm0.014}(8)$ | $0.559_{\pm0.003}(14)$ | $0.552_{\pm0.005}(19)$ | $0.509_{\pm0.012}(39)$ | $0.535_{\pm0.008}(29)$ | $0.507_{\pm0.015}(40)$ | $0.548_{\pm0.012}(24)$ | $0.55_{\pm0.002}(21)$ | $0.5_{\pm0.0}(43)$ | $0.533_{\pm0.003}(30)$ | $0.533_{\pm0.003}(30)$ | $0.565_{\pm0.01}(8)$ |
| celeba | $0.76_{\pm0.049}(14)$ | $0.73_{\pm0.03}(18)$ | $0.764_{\pm0.001}(12)$ | $0.511_{\pm0.011}(37)$ | $0.697_{\pm0.053}(22)$ | $0.585_{\pm0.097}(34)$ | $0.736_{\pm0.085}(17)$ | $0.653_{\pm0.015}(30)$ | $0.5_{\pm0.0}(38)$ | $0.801_{\pm0.027}(2)$ | $0.664_{\pm0.011}(26)$ | $0.656_{\pm0.054}(29)$ |
| cover | $0.983_{\pm0.002}(5)$ | $0.978_{\pm0.012}(9)$ | $0.984_{\pm0.001}(4)$ | $0.521_{\pm0.021}(40)$ | $0.759_{\pm0.16}(30)$ | $0.813_{\pm0.105}(29)$ | $0.874_{\pm0.063}(24)$ | $0.978_{\pm0.007}(9)$ | $0.5_{\pm0.0}(42)$ | $0.979_{\pm0.006}(8)$ | $0.703_{\pm0.021}(33)$ | $0.98_{\pm0.01}(7)$ |
| donors | $0.999_{\pm0.003}(6)$ | $0.93_{\pm0.021}(11)$ | $0.959_{\pm0.003}(3)$ | $0.955_{\pm0.001}(10)$ | $0.569_{\pm0.008}(40)$ | $0.64_{\pm0.217}(31)$ | $0.708_{\pm0.309}(29)$ | $0.836_{\pm0.072}(20)$ | $0.89_{\pm0.014}(16)$ | $0.973_{\pm0.011}(10)$ | $0.814_{\pm0.014}(22)$ | $0.999_{\pm0.0}(3)$ |
| fraud | $0.957_{\pm0.009}(5)$ | $0.959_{\pm0.001}(3)$ | $0.955_{\pm0.001}(10)$ | $0.569_{\pm0.008}(40)$ | $0.954_{\pm0.001}(13)$ | $0.752_{\pm0.085}(38)$ | $0.939_{\pm0.009}(29)$ | $0.95_{\pm0.003}(27)$ | $0.95_{\pm0.002}(26)$ | $0.949_{\pm0.007}(31)$ | $0.94_{\pm0.002}(28)$ | $0.948_{\pm0.006}(24)$ |
| http | $1.0_{\pm0.0}(1)$ | $0.999_{\pm0.001}(6)$ | $0.862_{\pm0.223}(31)$ | $0.634_{\pm0.059}(35)$ | $0.999_{\pm0.0}(6)$ | $0.988_{\pm0.014}(27)$ | $0.998_{\pm0.002}(16)$ | $0.993_{\pm0.0}(24)$ | $0.5_{\pm0.0}(36)$ | $0.995_{\pm0.001}(21)$ | $0.982_{\pm0.027}(28)$ | $0.761_{\pm0.365}(32)$ |
| magic.gamma | $0.837_{\pm0.009}(9)$ | $0.826_{\pm0.007}(10)$ | $0.673_{\pm0.008}(34)$ | $0.537_{\pm0.016}(43)$ | $0.676_{\pm0.063}(33)$ | $0.606_{\pm0.068}(41)$ | $0.677_{\pm0.032}(32)$ | $0.765_{\pm0.04}(15)$ | $0.784_{\pm0.011}(12)$ | $0.874_{\pm0.002}(1)$ | $0.698_{\pm0.007}(30)$ | $0.862_{\pm0.007}(2)$ |
| mammography | $0.888_{\pm0.013}(8)$ | $0.884_{\pm0.034}(6)$ | $0.69_{\pm0.071}(40)$ | $0.513_{\pm0.051}(45)$ | $0.884_{\pm0.018}(6)$ | $0.857_{\pm0.024}(17)$ | $0.852_{\pm0.04}(19)$ | $0.827_{\pm0.01}(25)$ | $0.814_{\pm0.019}(29)$ | $0.859_{\pm0.021}(16)$ | $0.742_{\pm0.012}(36)$ | $0.861_{\pm0.013}(15)$ |
| shuttle | $0.999_{\pm0.0}(5)$ | $0.998_{\pm0.001}(9)$ | $0.996_{\pm0.001}(17)$ | $0.659_{\pm0.04}(38)$ | $0.992_{\pm0.002}(23)$ | $0.979_{\pm0.009}(30)$ | $0.993_{\pm0.001}(21)$ | $0.991_{\pm0.001}(24)$ | $0.5_{\pm0.0}(40)$ | $0.997_{\pm0.0}(11)$ | $0.997_{\pm0.0}(11)$ | $1.0_{\pm0.0}(1)$ |
| skin | $0.847_{\pm0.094}(17)$ | $0.831_{\pm0.074}(21)$ | $0.603_{\pm0.004}(30)$ | $0.528_{\pm0.03}(34)$ | $0.607_{\pm0.111}(29)$ | $0.836_{\pm0.077}(18)$ | $0.618_{\pm0.118}(28)$ | $0.834_{\pm0.004}(19)$ | $0.92_{\pm0.003}(7)$ | $0.92_{\pm0.002}(7)$ | $0.851_{\pm0.006}(16)$ | $0.991_{\pm0.000}(2)$ |
| smtp | $0.912_{\pm0.008}(8)$ | $0.923_{\pm0.008}(5)$ | $0.796_{\pm0.008}(32)$ | $0.613_{\pm0.08}(39)$ | $0.889_{\pm0.021}(15)$ | $0.854_{\pm0.055}(22)$ | $0.808_{\pm0.038}(30)$ | $0.847_{\pm0.004}(24)$ | $0.5_{\pm0.0}(43)$ | $0.923_{\pm0.004}(5)$ | $0.842_{\pm0.025}(26)$ | $0.899_{\pm0.022}(12)$ |
| Avg Ranking | 7.36 | 10.18 | 24.27 | 38.55 | 21.55 | 29.55 | 23.64 | 20.45 | 33.45 | 11.64 | 26.0 | 12.27 |

| Dataset | DTE-IG | DTE-NP | GANomaly | GOAD | ICL | LUNAR | MCM | MO_GAAL | PlanarFlow | SLAD | SO_GAAL | VAE |
|---|---|---|---|---|---|---|---|---|---|---|---|---|
| ALOI | $0.572_{\pm0.013}(7)$ | $0.7_{\pm0.002}(5)$ | $0.547_{\pm0.008}(25)$ | $0.506_{\pm0.005}(41)$ | $0.529_{\pm0.004}(34)$ | $0.734_{\pm0.007}(2)$ | $0.562_{\pm0.007}(11)$ | $0.542_{\pm0.008}(27)$ | $0.506_{\pm0.015}(41)$ | $0.549_{\pm0.004}(22)$ | $0.544_{\pm0.004}(26)$ | $0.555_{\pm0.001}(17)$ |
| celeba | $0.775_{\pm0.10}(9)$ | $0.663_{\pm0.002}(27)$ | $0.368_{\pm0.052}(42)$ | $NAN$ | $0.713_{\pm0.016}(20)$ | $0.629_{\pm0.04}(31)$ | $0.777_{\pm0.02}(8)$ | $0.665_{\pm0.062}(25)$ | $NAN$ | $0.622_{\pm0.015}(32)$ | $0.544_{\pm0.01}(35)$ | $0.767_{\pm0.001}(11)$ |
| cover | $0.982_{\pm0.01}(6)$ | $0.989_{\pm0.0}(2)$ | $0.659_{\pm0.192}(35)$ | $NAN$ | $0.9_{\pm0.046}(22)$ | $0.993_{\pm0.0}(1)$ | $0.826_{\pm0.099}(27)$ | $0.633_{\pm0.017}(36)$ | $NAN$ | $0.83_{\pm0.01}(26)$ | $0.549_{\pm0.016}(39)$ | $0.975_{\pm0.0}(11)$ |
| donors | $0.875_{\pm0.127}(18)$ | $0.999_{\pm0.0}(3)$ | $0.761_{\pm0.232}(25)$ | $NAN$ | $1.0_{\pm0.0}(1)$ | $1.0_{\pm0.0}(1)$ | $0.998_{\pm0.001}(6)$ | $0.065_{\pm0.027}(41)$ | $NAN$ | $0.306_{\pm0.146}(40)$ | $0.813_{\pm0.047}(23)$ | |
| fraud | $0.941_{\pm0.01}(27)$ | $0.963_{\pm0.001}(2)$ | $0.912_{\pm0.043}(33)$ | $NAN$ | $0.937_{\pm0.01}(30)$ | $0.965_{\pm0.002}(1)$ | $0.957_{\pm0.004}(5)$ | $0.775_{\pm0.011}(37)$ | $NAN$ | $0.947_{\pm0.005}(23)$ | $0.598_{\pm0.019}(39)$ | $0.955_{\pm0.001}(10)$ |
| http | $1.0_{\pm0.0}(1)$ | $1.0_{\pm0.0}(1)$ | $0.493_{\pm0.312}(37)$ | $NAN$ | $0.999_{\pm0.000}(6)$ | $0.997_{\pm0.003}(17)$ | $0.997_{\pm0.002}(17)$ | $0.133_{\pm0.18}(41)$ | $NAN$ | $0.736_{\pm0.036}(33)$ | $0.999_{\pm0.0}(6)$ | |
| magic.gamma | $0.552_{\pm0.022}(4)$ | $0.839_{\pm0.001}(8)$ | $0.578_{\pm0.014}(42)$ | $0.684_{\pm0.008}(38)$ | $0.764_{\pm0.004}(16)$ | $0.853_{\pm0.004}(3)$ | $0.844_{\pm0.038}(5)$ | $0.441_{\pm0.049}(46)$ | $0.749_{\pm0.014}(21)$ | $0.725_{\pm0.004}(26)$ | $0.504_{\pm0.034}(43)$ | $0.708_{\pm0.02}(28)$ |
| mammography | $0.869_{\pm0.028}(13)$ | $0.876_{\pm0.04}(12)$ | $0.853_{\pm0.074}(18)$ | $0.681_{\pm0.024}(42)$ | $0.76_{\pm0.03}(35)$ | $0.881_{\pm0.005}(8)$ | $0.848_{\pm0.045}(21)$ | $0.71_{\pm0.023}(39)$ | $0.812_{\pm0.027}(30)$ | $0.765_{\pm0.036}(34)$ | $0.784_{\pm0.035}(33)$ | $0.797_{\pm0.008}(32)$ |
| shuttle | $1.0_{\pm0.0}(1)$ | $0.999_{\pm0.0}(5)$ | $0.974_{\pm0.009}(31)$ | $0.989_{\pm0.004}(26)$ | $1.0_{\pm0.0}(1)$ | $1.0_{\pm0.0}(1)$ | $0.998_{\pm0.000}(9)$ | $0.691_{\pm0.096}(25)$ | $0.534_{\pm0.114}(33)$ | $0.999_{\pm0.0}(5)$ | $0.072_{\pm0.1}(43)$ | $0.996_{\pm0.0}(17)$ |
| skin | $0.984_{\pm0.000}(4)$ | $0.998_{\pm0.0}(1)$ | $0.436_{\pm0.032}(38)$ | $NAN$ | $0.074_{\pm0.011}(42)$ | $0.991_{\pm0.001}(2)$ | $0.691_{\pm0.096}(25)$ | $0.534_{\pm0.114}(33)$ | $NAN$ | $0.929_{\pm0.015}(5)$ | $0.383_{\pm0.088}(40)$ | $0.628_{\pm0.002}(26)$ |
| smtp | $0.857_{\pm0.037}(20)$ | $0.933_{\pm0.008}(4)$ | $0.504_{\pm0.094}(42)$ | $0.906_{\pm0.015}(11)$ | $0.697_{\pm0.11}(38)$ | $0.935_{\pm0.012}(3)$ | $0.845_{\pm0.044}(25)$ | $0.587_{\pm0.028}(40)$ | $NAN$ | $0.922_{\pm0.005}(6)$ | $0.551_{\pm0.017}(41)$ | $0.837_{\pm0.015}(27)$ |
| Avg Ranking | 10.0 | 6.36 | 33.45 | 31.6 | 14.45 | 6.36 | 14.45 | 37.18 | 31.5 | 20.11 | 37.64 | 18.91 |

| Dataset | CBLOF | CD | ECOD | FB | GMM | HBOS | IForest | KDE | LMDD | LODA | MCD | OCSVM |
|---|---|---|---|---|---|---|---|---|---|---|---|---|
| ALOI | $0.557_{\pm0.001}(16)$ | $0.517_{\pm0.002}(37)$ | $0.531_{\pm0.000}(32)$ | $0.765_{\pm0.000}(1)$ | $0.561_{\pm0.000}(12)$ | $0.53_{\pm0.001}(33)$ | $0.542_{\pm0.000}(27)$ | $0.563_{\pm0.000}(10)$ | $0.513_{\pm0.000}(38)$ | $0.486_{\pm0.02}(44)$ | $0.52_{\pm0.000}(36)$ | $0.551_{\pm0.000}(20)$ |
| celeba | $0.781_{\pm0.022}(6)$ | $0.708_{\pm0.005}(21)$ | $0.757_{\pm0.01}(15)$ | $0.534_{\pm0.016}(36)$ | $0.807_{\pm0.0}(1)$ | $0.761_{\pm0.001}(13)$ | $0.718_{\pm0.012}(19)$ | $0.675_{\pm0.002}(24)$ | $0.688_{\pm NAN}(23)$ | $0.661_{\pm0.121}(28)$ | $0.799_{\pm0.042}(3)$ | $0.79_{\pm0.0}(4)$ |
| cover | $0.943_{\pm0.005}(17)$ | $0.744_{\pm0.025}(31)$ | $0.92_{\pm0.001}(20)$ | $0.989_{\pm0.005}(2)$ | $0.95_{\pm0.001}(16)$ | $0.72_{\pm0.001}(32)$ | $0.843_{\pm0.011}(25)$ | $0.956_{\pm0.001}(14)$ | $0.893_{\pm0.048}(23)$ | $0.94_{\pm0.031}(18)$ | $0.702_{\pm0.001}(34)$ | $0.963_{\pm0.001}(13)$ |
| donors | $0.929_{\pm0.006}(12)$ | $0.504_{\pm0.068}(36)$ | $0.889_{\pm0.0}(17)$ | $0.99_{\pm0.008}(8)$ | $0.925_{\pm0.0}(13)$ | $0.797_{\pm0.14}(24)$ | $0.891_{\pm0.012}(15)$ | $0.975_{\pm0.0}(9)$ | $0.725_{\pm NAN}(27)$ | $0.646_{\pm0.318}(30)$ | $0.822_{\pm0.106}(21)$ | $0.921_{\pm0.0}(14)$ |
| fraud | $0.951_{\pm0.008}(15)$ | $0.949_{\pm0.002}(21)$ | $0.95_{\pm0.0}(18)$ | $0.821_{\pm0.042}(35)$ | $0.957_{\pm0.0}(5)$ | $0.951_{\pm0.001}(15)$ | $0.949_{\pm0.002}(21)$ | $0.959_{\pm0.001}(3)$ | $0.942_{\pm NAN}(26)$ | $0.816_{\pm0.168}(36)$ | $0.922_{\pm0.003}(31)$ | $0.956_{\pm0.0}(8)$ |
| http | $0.999_{\pm0.0}(6)$ | $0.949_{\pm0.002}(21)$ | $0.979_{\pm0.0}(29)$ | $0.897_{\pm0.006}(30)$ | $0.999_{\pm0.0}(6)$ | $0.999_{\pm0.0}(6)$ | $0.993_{\pm0.000}(24)$ | $1.0_{\pm0.0}(1)$ | $0.999_{\pm0.0}(6)$ | $0.179_{\pm0.225}(40)$ | $0.999_{\pm0.0}(6)$ | $1.0_{\pm0.0}(1)$ |
| magic.gamma | $0.756_{\pm0.01}(20)$ | $0.738_{\pm0.001}(24)$ | $0.637_{\pm0.000}(37)$ | $0.84_{\pm0.003}(7)$ | $0.804_{\pm0.001}(11)$ | $0.745_{\pm0.004}(22)$ | $0.77_{\pm0.012}(14)$ | $0.763_{\pm0.001}(17)$ | $0.628_{\pm0.007}(39)$ | $0.705_{\pm0.016}(29)$ | $0.738_{\pm0.004}(24)$ | $0.743_{\pm0.001}(23)$ |
| mammography | $0.844_{\pm0.017}(23)$ | $0.823_{\pm0.007}(26)$ | $0.906_{\pm0.002}(1)$ | $0.845_{\pm0.012}(22)$ | $0.878_{\pm0.002}(9)$ | $0.844_{\pm0.006}(23)$ | $0.877_{\pm0.012}(11)$ | $0.878_{\pm0.003}(9)$ | $0.85_{\pm0.018}(20)$ | $0.9_{\pm0.006}(2)$ | $0.723_{\pm0.026}(37)$ | $0.885_{\pm0.003}(4)$ |
| shuttle | $0.997_{\pm0.001}(11)$ | $0.76_{\pm0.0}(35)$ | $0.993_{\pm0.0}(21)$ | $0.873_{\pm0.085}(33)$ | $0.994_{\pm0.000}(20)$ | $0.988_{\pm0.002}(27)$ | $0.997_{\pm0.001}(11)$ | $0.997_{\pm0.0}(11)$ | $0.984_{\pm0.007}(29)$ | $0.683_{\pm0.365}(37)$ | $0.99_{\pm0.000}(25)$ | $0.996_{\pm0.0}(17)$ |
| skin | $0.924_{\pm0.012}(6)$ | $0.734_{\pm0.000}(24)$ | $0.489_{\pm0.001}(36)$ | $0.852_{\pm0.014}(15)$ | $0.888_{\pm0.001}(13)$ | $0.77_{\pm0.001}(22)$ | $0.89_{\pm0.005}(12)$ | $0.891_{\pm0.001}(11)$ | $0.42_{\pm0.06}(39)$ | $0.742_{\pm0.057}(23)$ | $0.884_{\pm0.001}(14)$ | $0.903_{\pm0.001}(10)$ |
| smtp | $0.892_{\pm0.027}(14)$ | $0.784_{\pm0.001}(35)$ | $0.58_{\pm0.017}(41)$ | $0.829_{\pm0.032}(28)$ | $0.815_{\pm0.000}(29)$ | $0.795_{\pm0.007}(30)$ | $0.907_{\pm0.009}(10)$ | $0.881_{\pm0.001}(16)$ | $0.795_{\pm0.073}(33)$ | $0.735_{\pm0.049}(36)$ | $0.95_{\pm0.0}(1)$ | $0.854_{\pm0.000}(22)$ |
| Avg Ranking | 13.27 | 28.45 | 22.09 | 19.73 | 12.27 | 24.36 | 17.36 | 11.36 | 27.55 | 29.36 | 21.09 | 12.36 |

| Dataset | QMCD | Sampling | ABOD | COF | INNE | KNN | KPCA | LOF | PCA | | |
|---|---|---|---|---|---|---|---|---|---|---|---|
| ALOI | $0.526_{\pm0.001}(35)$ | $0.553_{\pm0.004}(18)$ | $0.728_{\pm0.002}(4)$ | $NAN$ | $0.558_{\pm0.002}(15)$ | $0.671_{\pm0.002}(6)$ | $NAN$ | $0.731_{\pm0.002}(3)$ | $0.549_{\pm0.001}(22)$ | | |
| celeba | $0.5_{\pm0.0}(38)$ | $0.787_{\pm0.0}(5)$ | $0.478_{\pm0.000}(40)$ | $NAN$ | $0.755_{\pm0.000}(16)$ | $0.589_{\pm0.000}(33)$ | $NAN$ | $0.443_{\pm0.000}(41)$ | $0.771_{\pm0.0}(10)$ | | |
| cover | $0.82_{\pm0.001}(28)$ | $0.911_{\pm0.041}(21)$ | $0.61_{\pm0.0}(38)$ | $NAN$ | $0.951_{\pm0.000}(15)$ | $0.627_{\pm0.0}(37)$ | $NAN$ | $0.503_{\pm0.000}(41)$ | $0.923_{\pm0.0}(19)$ | | |
| donors | $0.724_{\pm0.000}(28)$ | $0.759_{\pm0.017}(19)$ | $0.441_{\pm0.000}(39)$ | $NAN$ | $0.594_{\pm0.027}(32)$ | $0.592_{\pm0.0}(33)$ | $NAN$ | $0.546_{\pm0.000}(35)$ | $0.746_{\pm0.0}(26)$ | | |
| fraud | $0.955_{\pm0.001}(10)$ | $0.95_{\pm0.006}(18)$ | $0.849_{\pm0.001}(34)$ | $NAN$ | $0.954_{\pm0.000}(14)$ | $0.921_{\pm0.00}(32)$ | $NAN$ | $0.487_{\pm0.002}(42)$ | $0.951_{\pm0.0}(15)$ | | |
| http | $0.997_{\pm0.0}(17)$ | $0.999_{\pm0.0}(6)$ | $0.731_{\pm0.0}(34)$ | $NAN$ | $0.997_{\pm0.000}(17)$ | $0.193_{\pm0.001}(39)$ | $NAN$ | $0.397_{\pm0.002}(38)$ | $0.995_{\pm0.0}(21)$ | | |
| magic.gamma | $0.71_{\pm0.006}(27)$ | $0.759_{\pm0.017}(18)$ | $0.758_{\pm0.002}(19)$ | $0.619_{\pm0.005}(40)$ | $0.691_{\pm0.001}(31)$ | $0.771_{\pm0.001}(13)$ | $0.53_{\pm0.008}(44)$ | $0.661_{\pm0.004}(35)$ | $0.642_{\pm0.001}(36)$ | | |
| mammography | $0.721_{\pm0.01}(38)$ | $0.864_{\pm0.014}(14)$ | $0.514_{\pm0.007}(44)$ | $0.69_{\pm0.003}(41)$ | $0.822_{\pm0.01}(28)$ | $0.823_{\pm0.006}(27)$ | $0.43_{\pm0.034}(46)$ | $0.664_{\pm0.006}(43)$ | $0.885_{\pm0.005}(4)$ | | |
| shuttle | $0.72_{\pm0.002}(36)$ | $0.997_{\pm0.001}(11)$ | $0.496_{\pm0.006}(42)$ | $NAN$ | $0.942_{\pm0.033}(32)$ | $0.499_{\pm0.012}(41)$ | $NAN$ | $0.551_{\pm0.000}(39)$ | $0.988_{\pm0.000}(27)$ | | |
| skin | $0.624_{\pm0.004}(27)$ | $0.906_{\pm0.01}(9)$ | $0.494_{\pm0.001}(35)$ | $NAN$ | $0.461_{\pm0.086}(37)$ | $0.537_{\pm0.001}(32)$ | $NAN$ | $0.539_{\pm0.001}(31)$ | $0.336_{\pm0.002}(41)$ | | |
| smtp | $0.723_{\pm0.032}(37)$ | $0.912_{\pm0.022}(8)$ | $0.874_{\pm0.003}(18)$ | $NAN$ | $0.918_{\pm0.005}(7)$ | $0.897_{\pm0.013}(13)$ | $NAN$ | $0.87_{\pm0.025}(19)$ | $0.807_{\pm0.002}(31)$ | | |
| Avg Ranking | 29.18 | 13.36 | 31.55 | 40.0 | 22.09 | 27.73 | 45.0 | 33.36 | 22.91 | | |

Table 11: AUPRC results on 11 large datasets, where we compare TCCM to 44 baselines (with 5 independent runs). We report the mean $\pm$ std (rank).

| Dataset | TCCM (Ours) | AE | AE-1SVM | ALAD | AnoGAN | DAGMM | DeepSVDD | DIF | DROCC | DTE-Cat | DTE-DDPM | DTE-Gaussian |
|---|---|---|---|---|---|---|---|---|---|---|---|---|
| ALOI | $0.085_{\pm0.005}(11)$ | $0.077_{\pm0.001}(15)$ | $0.075_{\pm0.002}(16)$ | $0.061_{\pm0.001}(40)$ | $0.068_{\pm0.001}(29)$ | $0.061_{\pm0.000}(40)$ | $0.072_{\pm0.002}(20)$ | $0.079_{\pm0.003}(14)$ | $0.059_{\pm0.0}(44)$ | $0.066_{\pm0.001}(31)$ | $0.069_{\pm0.001}(27)$ | $0.099_{\pm0.003}(8)$ |
| celeba | $0.136_{\pm0.036}(16)$ | $0.106_{\pm0.013}(21)$ | $0.199_{\pm0.02}(7)$ | $0.048_{\pm0.002}(37)$ | $0.144_{\pm0.026}(13)$ | $0.066_{\pm0.008}(33)$ | $0.114_{\pm0.065}(14)$ | $0.088_{\pm0.008}(28)$ | $0.044_{\pm0.0}(38)$ | $0.135_{\pm0.017}(17)$ | $0.095_{\pm0.006}(25)$ | $0.084_{\pm0.012}(29)$ |
| cover | $0.835_{\pm0.029}(3)$ | $0.477_{\pm0.135}(10)$ | $0.393_{\pm0.012}(12)$ | $0.021_{\pm0.000}(40)$ | $0.072_{\pm0.039}(30)$ | $0.124_{\pm0.00}(25)$ | $0.138_{\pm0.15}(24)$ | $0.494_{\pm0.075}(9)$ | $0.019_{\pm0.0}(42)$ | $0.721_{\pm0.028}(8)$ | $0.045_{\pm0.005}(33)$ | $0.824_{\pm0.0}(4)$ |
| donors | $0.96_{\pm0.056}(6)$ | $0.447_{\pm0.135}(10)$ | $0.122_{\pm0.008}(36)$ | $0.141_{\pm0.012}(35)$ | $0.215_{\pm0.162}(30)$ | $0.34_{\pm0.214}(19)$ | $0.74_{\pm0.101}(22)$ | $0.328_{\pm0.026}(20)$ | $0.112_{\pm0.0}(38)$ | $0.666_{\pm0.024}(10)$ | $0.293_{\pm0.026}(26)$ | $0.986_{\pm0.0}(3)$ |
| fraud | $0.66_{\pm0.132}(5)$ | $0.627_{\pm0.029}(8)$ | $0.345_{\pm0.014}(21)$ | $0.006_{\pm0.0}(39)$ | $0.29_{\pm0.016}(25)$ | $0.053_{\pm0.089}(35)$ | $0.25_{\pm0.095}(29)$ | $0.538_{\pm0.026}(13)$ | $0.003_{\pm0.0}(42)$ | $0.742_{\pm0.007}(2)$ | $0.723_{\pm0.009}(3)$ | $0.743_{\pm0.017}(1)$ |
| http | $0.99_{\pm0.015}(5)$ | $0.875_{\pm0.073}(15)$ | $0.355_{\pm0.035}(33)$ | $0.9_{\pm0.017}(13)$ | $0.456_{\pm0.212}(27)$ | $0.817_{\pm0.137}(17)$ | $0.515_{\pm0.003}(25)$ | $0.008_{\pm0.0}(41)$ | $0.579_{\pm0.022}(22)$ | $0.4_{\pm0.255}(29)$ | $0.102_{\pm0.056}(31)$ | |
| magic.gamma | $0.864_{\pm0.008}(8)$ | $0.856_{\pm0.006}(10)$ | $0.745_{\pm0.003}(31)$ | $0.567_{\pm0.014}(43)$ | $0.73_{\pm0.04}(33)$ | $0.669_{\pm0.065}(38)$ | $0.729_{\pm0.034}(34)$ | $0.345_{\pm0.073}(23)$ | $0.361_{\pm0.013}(17)$ | $0.829_{\pm0.008}(12)$ | $0.895_{\pm0.001}(1)$ | $0.737_{\pm0.006}(32)$ | $0.887_{\pm0.005}(2)$ |
| mammography | $0.439_{\pm0.04}(6)$ | $0.418_{\pm0.09}(10)$ | $0.35_{\pm0.108}(19)$ | $0.054_{\pm0.014}(44)$ | $0.477_{\pm0.046}(2)$ | $0.426_{\pm0.073}(7)$ | $0.345_{\pm0.03}(23)$ | $0.426_{\pm0.014}(8)$ | $0.37_{\pm0.034}(16)$ | $0.134_{\pm0.006}(41)$ | $0.36_{\pm0.039}(18)$ | |
| shuttle | $0.99_{\pm0.002}(5)$ | $0.974_{\pm0.005}(11)$ | $0.957_{\pm0.006}(23)$ | $0.311_{\pm0.07}(38)$ | $0.967_{\pm0.000}(16)$ | $0.894_{\pm0.009}(31)$ | $0.959_{\pm0.000}(22)$ | $0.98_{\pm0.000}(10)$ | $0.133_{\pm0.04}(42)$ | $0.938_{\pm0.006}(29)$ | $0.969_{\pm0.006}(15)$ | $0.996_{\pm0.000}(2)$ |
| skin | $0.732_{\pm0.119}(6)$ | $0.635_{\pm0.111}(17)$ | $0.358_{\pm0.002}(32)$ | $0.377_{\pm0.039}(28)$ | $0.43_{\pm0.076}(26)$ | $0.587_{\pm0.083}(19)$ | $0.392_{\pm0.084}(27)$ | $0.549_{\pm0.04}(21)$ | $0.716_{\pm0.008}(8)$ | $0.698_{\pm0.004}(11)$ | $0.719_{\pm0.0}(7)$ | $0.972_{\pm0.0}(3)$ |
| smtp | $0.507_{\pm0.102}(8)$ | $0.462_{\pm0.122}(13)$ | $0.147_{\pm0.045}(12)$ | $0.007_{\pm0.000}(37)$ | $0.33_{\pm0.095}(22)$ | $0.214_{\pm0.214}(23)$ | $0.41_{\pm0.092}(80)$ | $0.616_{\pm0.04}(4)$ | $0.001_{\pm0.0}(40)$ | $0.471_{\pm0.042}(11)$ | $0.143_{\pm0.143}(26)$ | $0.102_{\pm0.092}(28)$ |
| Avg Ranking | 7.18 | 13.18 | 21.09 | 37.64 | 21.73 | 27.0 | 22.91 | 15.91 | 34.36 | 14.36 | 24.0 | 11.73 |

| Dataset | DTE-IG | DTE-NP | GANomaly | GOAD | ICL | LUNAR | MCM | MO_GAAL | PlanarFlow | SLAD | SO_GAAL | VAE |
|---|---|---|---|---|---|---|---|---|---|---|---|---|
| ALOI | $0.087_{\pm0.004}(9)$ | $0.133_{\pm0.002}(5)$ | $0.086_{\pm0.007}(10)$ | $0.064_{\pm0.002}(35)$ | $0.082_{\pm0.002}(12)$ | $0.219_{\pm0.004}(1)$ | $0.075_{\pm0.003}(16)$ | $0.069_{\pm0.002}(27)$ | $0.06_{\pm0.003}(43)$ | $0.072_{\pm0.001}(20)$ | $0.068_{\pm0.001}(29)$ | $0.072_{\pm0.000}(20)$ |
| celeba | $0.14_{\pm0.059}(15)$ | $0.09_{\pm0.001}(26)$ | $0.033_{\pm0.004}(42)$ | $NAN$ | $0.1_{\pm0.008}(23)$ | $0.068_{\pm0.001}(31)$ | $0.132_{\pm0.016}(19)$ | $0.071_{\pm0.011}(30)$ | $NAN$ | $0.067_{\pm0.003}(32)$ | $0.048_{\pm0.002}(36)$ | $0.197_{\pm0.001}(3)$ |
| cover | $0.891_{\pm0.013}(2)$ | $0.814_{\pm0.009}(6)$ | $0.196_{\pm0.37}(18)$ | $NAN$ | $0.414_{\pm0.209}(11)$ | $0.903_{\pm0.000}(1)$ | $0.098_{\pm0.065}(27)$ | $0.026_{\pm0.000}(38)$ | $NAN$ | $0.073_{\pm0.009}(29)$ | $0.021_{\pm0.000}(40)$ | $0.283_{\pm0.003}(14)$ |
| donors | $0.566_{\pm0.301}(11)$ | $0.984_{\pm0.001}(5)$ | $0.362_{\pm0.261}(18)$ | $NAN$ | $0.995_{\pm0.002}(2)$ | $0.995_{\pm0.000}(2)$ | $0.051_{\pm0.002}(38)$ | $0.059_{\pm0.001}(41)$ | $NAN$ | $0.081_{\pm0.019}(40)$ | $0.281_{\pm0.006}(27)$ | |
| fraud | $0.638_{\pm0.075}(7)$ | $0.373_{\pm0.017}(17)$ | $0.521_{\pm0.174}(14)$ | $NAN$ | $0.638_{\pm0.035}(6)$ | $0.573_{\pm0.029}(12)$ | $0.693_{\pm0.044}(4)$ | $0.008_{\pm0.0}(38)$ | $NAN$ | $0.349_{\pm0.233}(19)$ | $0.004_{\pm0.0}(40)$ | $0.273_{\pm0.006}(26)$ |
| http | $0.995_{\pm0.000}(3)$ | $0.995_{\pm0.001}(3)$ | $0.021_{\pm0.014}(34)$ | $NAN$ | $0.961_{\pm0.057}(6)$ | $0.821_{\pm0.248}(16)$ | $0.78_{\pm0.222}(18)$ | $0.009_{\pm0.002}(40)$ | $NAN$ | $0.015_{\pm0.002}(35)$ | $0.911_{\pm0.004}(9)$ | |
| magic.gamma | $0.876_{\pm0.017}(3)$ | $0.863_{\pm0.001}(9)$ | $0.647_{\pm0.016}(41)$ | $0.691_{\pm0.009}(36)$ | $0.819_{\pm0.005}(13)$ | $0.876_{\pm0.003}(3)$ | $0.872_{\pm0.031}(5)$ | $0.522_{\pm0.05}(46)$ | $0.798_{\pm0.009}(19)$ | $0.779_{\pm0.004}(23)$ | $0.539_{\pm0.025}(45)$ | $0.764_{\pm0.012}(26)$ |
| mammography | $0.347_{\pm0.08}(21)$ | $0.419_{\pm0.025}(9)$ | $0.339_{\pm0.219}(24)$ | $0.279_{\pm0.025}(26)$ | $0.182_{\pm0.028}(36)$ | $0.459_{\pm0.019}(3)$ | $0.346_{\pm0.124}(22)$ | $0.26_{\pm0.052}(28)$ | $0.169_{\pm0.056}(38)$ | $0.203_{\pm0.064}(34)$ | $0.182_{\pm0.02}(36)$ | $0.375_{\pm0.025}(14)$ |
| shuttle | $0.99_{\pm0.002}(5)$ | $0.989_{\pm0.002}(6)$ | $0.945_{\pm0.043}(27)$ | $0.952_{\pm0.009}(24)$ | $0.996_{\pm0.003}(2)$ | $0.997_{\pm0.002}(1)$ | $0.966_{\pm0.004}(18)$ | $0.074_{\pm0.005}(44)$ | $0.499_{\pm0.128}(35)$ | $0.98_{\pm0.004}(10)$ | $0.077_{\pm0.015}(43)$ | $0.963_{\pm0.00}(21)$ |
| skin | $0.95_{\pm0.023}(4)$ | $0.990_{\pm0.001}(1)$ | $0.295_{\pm0.04}(40)$ | $NAN$ | $0.314_{\pm0.016}(36)$ | $0.975_{\pm0.01}(2)$ | $0.469_{\pm0.12}(24)$ | $0.345_{\pm0.072}(33)$ | $NAN$ | $0.854_{\pm0.0}(5)$ | $0.293_{\pm0.036}(41)$ | $0.962_{\pm0.00}(29)$ |
| smtp | $0.07_{\pm0.084}(30)$ | $0.553_{\pm0.14}(9)$ | $0.001_{\pm0.0}(40)$ | $0.454_{\pm0.053}(15)$ | $0.042_{\pm0.04}(31)$ | $0.662_{\pm0.000}(1)$ | $0.454_{\pm0.05}(15)$ | $0.001_{\pm0.0}(40)$ | $NAN$ | $0.473_{\pm0.049}(10)$ | $0.001_{\pm0.0}(40)$ | $0.46_{\pm0.014}(14)$ |
| Avg Ranking | 9.73 | 8.45 | 28.0 | 27.2 | 16.18 | 6.55 | 15.91 | 36.82 | 33.75 | 20.22 | 38.64 | 18.45 |

| Dataset | CBLOF | CD | ECOD | FB | GMM | HBOS | IForest | KDE | LMDD | LODA | MCD | OCSVM |
|---|---|---|---|---|---|---|---|---|---|---|---|---|
| ALOI | $0.072_{\pm0.001}(20)$ | $0.065_{\pm0.0}(32)$ | $0.064_{\pm0.0}(35)$ | $0.17_{\pm0.006}(2)$ | $0.074_{\pm0.001}(19)$ | $0.065_{\pm0.0}(32)$ | $0.065_{\pm0.0}(32)$ | $0.102_{\pm0.002}(7)$ | $0.064_{\pm0.001}(35)$ | $0.061_{\pm0.000}(40)$ | $0.063_{\pm0.000}(38)$ | $0.075_{\pm0.001}(16)$ |
| celeba | $0.17_{\pm0.0}(7)$ | $0.097_{\pm0.00}(24)$ | $0.171_{\pm0.000}(6)$ | $0.049_{\pm0.000}(35)$ | $0.166_{\pm0.01}(9)$ | $0.17_{\pm0.001}(7)$ | $0.134_{\pm0.008}(18)$ | $0.089_{\pm0.001}(27)$ | $0.15_{\pm NAN}(11)$ | $0.113_{\pm0.006}(20)$ | $0.149_{\pm0.046}(12)$ | $0.203_{\pm0.00}(1)$ |
| cover | $0.168_{\pm0.022}(21)$ | $0.114_{\pm0.016}(37)$ | $0.187_{\pm0.002}(20)$ | $0.818_{\pm0.09}(5)$ | $0.197_{\pm0.000}(17)$ | $0.051_{\pm0.002}(33)$ | $0.081_{\pm0.012}(28)$ | $0.359_{\pm0.000}(13)$ | $0.155_{\pm0.042}(22)$ | $0.208_{\pm0.114}(16)$ | $0.031_{\pm0.04}(36)$ | $0.223_{\pm0.004}(15)$ |
| donors | $0.447_{\pm0.02}(13)$ | $0.114_{\pm0.016}(37)$ | $0.414_{\pm0.001}(16)$ | $0.919_{\pm0.041}(8)$ | $0.432_{\pm0.001}(14)$ | $0.3_{\pm0.038}(25)$ | $0.397_{\pm0.027}(17)$ | $0.713_{\pm0.000}(9)$ | $0.28_{\pm NAN}(28)$ | $0.309_{\pm0.28}(24)$ | $0.31_{\pm0.122}(23)$ | $0.428_{\pm0.001}(15)$ |
| fraud | $0.257_{\pm0.008}(28)$ | $0.242_{\pm0.042}(30)$ | $0.331_{\pm0.006}(22)$ | $0.016_{\pm0.004}(37)$ | $0.587_{\pm0.015}(11)$ | $0.349_{\pm0.018}(19)$ | $0.208_{\pm0.055}(33)$ | $0.389_{\pm0.012}(16)$ | $0.321_{\pm NAN}(23)$ | $0.311_{\pm0.126}(24)$ | $0.601_{\pm0.022}(10)$ | $0.352_{\pm0.01}(18)$ |
| http | $0.902_{\pm0.002}(12)$ | $0.551_{\pm0.003}(24)$ | $0.252_{\pm0.000}(30)$ | $0.069_{\pm0.004}(32)$ | $0.911_{\pm0.003}(9)$ | $0.562_{\pm0.159}(23)$ | $0.493_{\pm0.12}(26)$ | $0.997_{\pm0.003}(2)$ | $0.908_{\pm0.006}(11)$ | $0.01_{\pm0.003}(39)$ | $0.918_{\pm0.005}(7)$ | $0.998_{\pm0.002}(1)$ |
| magic.gamma | $0.798_{\pm0.007}(19)$ | $0.755_{\pm0.002}(28)$ | $0.68_{\pm0.002}(37)$ | $0.868_{\pm0.002}(7)$ | $0.834_{\pm0.001}(11)$ | $0.773_{\pm0.002}(24)$ | $0.802_{\pm0.008}(17)$ | $0.809_{\pm0.001}(15)$ | $0.668_{\pm0.016}(39)$ | $0.758_{\pm0.012}(27)$ | $0.772_{\pm0.000}(25)$ | $0.791_{\pm0.000}(21)$ |
| mammography | $0.348_{\pm0.007}(20)$ | $0.193_{\pm0.0}(35)$ | $0.549_{\pm0.004}(1)$ | $0.257_{\pm0.028}(29)$ | $0.404_{\pm0.021}(11)$ | $0.209_{\pm0.003}(33)$ | $0.373_{\pm0.027}(15)$ | $0.425_{\pm0.018}(8)$ | $0.458_{\pm0.013}(4)$ | $0.445_{\pm0.044}(5)$ | $0.08_{\pm0.023}(43)$ | $0.403_{\pm0.002}(12)$ |
| shuttle | $0.965_{\pm0.005}(19)$ | $0.456_{\pm0.028}(37)$ | $0.95_{\pm0.004}(26)$ | $0.478_{\pm0.274}(36)$ | $0.967_{\pm0.000}(16)$ | $0.976_{\pm0.000}(12)$ | $0.985_{\pm0.007}(7)$ | $0.981_{\pm0.00}(8)$ | $0.952_{\pm0.014}(24)$ | $0.5_{\pm0.34}(34)$ | $0.905_{\pm0.006}(30)$ | $0.975_{\pm0.002}(13)$ |
| skin | $0.712_{\pm0.034}(9)$ | $0.448_{\pm0.002}(25)$ | $0.303_{\pm0.0}(37)$ | $0.578_{\pm0.018}(20)$ | $0.645_{\pm0.001}(15)$ | $0.534_{\pm0.002}(22)$ | $0.638_{\pm0.008}(16)$ | $0.65_{\pm0.002}(14)$ | $0.297_{\pm0.032}(39)$ | $0.515_{\pm0.075}(23)$ | $0.625_{\pm0.000}(18)$ | $0.664_{\pm0.001}(13)$ |
| smtp | $0.453_{\pm0.061}(18)$ | $0.16_{\pm0.055}(25)$ | $0.585_{\pm0.037}(5)$ | $0.003_{\pm0.0}(38)$ | $0.367_{\pm0.041}(21)$ | $0.009_{\pm0.001}(35)$ | $0.009_{\pm0.001}(35)$ | $0.633_{\pm0.051}(3)$ | $0.183_{\pm0.165}(24)$ | $0.086_{\pm0.053}(29)$ | $0.01_{\pm0.00}(34)$ | $0.636_{\pm0.047}(2)$ |
| Avg Ranking | 16.91 | 30.09 | 21.36 | 22.64 | 13.91 | 24.0 | 22.18 | 11.09 | 23.64 | 25.55 | 25.09 | 11.55 |

| Dataset | QMCD | Sampling | ABOD | COF | INNE | KNN | KPCA | LOF | PCA | | |
|---|---|---|---|---|---|---|---|---|---|---|---|
| ALOI | $0.062_{\pm0.0}(39)$ | $0.071_{\pm0.000}(25)$ | $0.158_{\pm0.000}(3)$ | $NAN$ | $0.072_{\pm0.001}(20)$ | $0.115_{\pm0.002}(6)$ | $NAN$ | $0.142_{\pm0.000}(4)$ | $0.07_{\pm0.000}(26)$ | | |
| celeba | $0.044_{\pm0.0}(38)$ | $0.181_{\pm0.031}(4)$ | $0.041_{\pm0.0}(40)$ | $NAN$ | $0.106_{\pm0.004}(21)$ | $0.064_{\pm0.0}(34)$ | $NAN$ | $0.038_{\pm0.001}(41)$ | $0.181_{\pm0.001}(4)$ | | |
| cover | $0.071_{\pm0.001}(31)$ | $0.144_{\pm0.03}(23)$ | $0.028_{\pm0.0}(37)$ | $NAN$ | $0.189_{\pm0.007}(19)$ | $0.033_{\pm0.0}(35)$ | $NAN$ | $0.022_{\pm0.0}(39)$ | $0.117_{\pm0.000}(26)$ | | |
| donors | $0.169_{\pm0.002}(31)$ | $0.319_{\pm0.063}(21)$ | $0.101_{\pm0.0}(39)$ | $NAN$ | $0.143_{\pm0.007}(34)$ | $0.157_{\pm0.0}(32)$ | $NAN$ | $0.104_{\pm0.0}(40)$ | $0.224_{\pm0.0}(29)$ | | |
| fraud | $0.416_{\pm0.02}(15)$ | $0.268_{\pm0.007}(27)$ | $0.02_{\pm0.0}(36)$ | $NAN$ | $0.219_{\pm0.11}(32)$ | $0.08_{\pm0.0}(34)$ | $NAN$ | $0.004_{\pm0.0}(40)$ | $0.228_{\pm0.003}(31)$ | | |
| http | $0.708_{\pm0.01}(20)$ | $0.893_{\pm0.002}(14)$ | $0.014_{\pm0.0}(36)$ | $NAN$ | $0.747_{\pm0.207}(19)$ | $0.014_{\pm0.0}(36)$ | $NAN$ | $0.012_{\pm0.0}(38)$ | $0.602_{\pm0.003}(21)$ | | |
| magic.gamma | $0.749_{\pm0.006}(30)$ | $0.8_{\pm0.009}(18)$ | $0.788_{\pm0.002}(22)$ | $0.635_{\pm0.005}(42)$ | $0.753_{\pm0.006}(29)$ | $0.808_{\pm0.001}(16)$ | $0.56_{\pm0.006}(44)$ | $0.657_{\pm0.005}(40)$ | $0.703_{\pm0.001}(35)$ | | |
| mammography | $0.14_{\pm0.005}(40)$ | $0.378_{\pm0.056}(13)$ | $0.043_{\pm0.001}(45)$ | $0.11_{\pm0.046}(42)$ | $0.254_{\pm0.02}(30)$ | $0.23_{\pm0.013}(32)$ | $0.038_{\pm0.002}(46)$ | $0.143_{\pm0.007}(39)$ | $0.271_{\pm0.006}(27)$ | | |
| shuttle | $0.684_{\pm0.014}(32)$ | $0.965_{\pm0.007}(19)$ | $0.171_{\pm0.002}(41)$ | $NAN$ | $0.633_{\pm0.12}(33)$ | $0.203_{\pm0.000}(40)$ | $NAN$ | $0.211_{\pm0.002}(39)$ | $0.942_{\pm0.004}(28)$ | | |
| skin | $0.366_{\pm0.002}(30)$ | $0.702_{\pm0.017}(10)$ | $0.342_{\pm0.0}(34)$ | $NAN$ | $0.298_{\pm0.03}(38)$ | $0.332_{\pm0.0}(35)$ | $NAN$ | $0.359_{\pm0.001}(31)$ | $0.253_{\pm0.001}(42)$ | | |
| smtp | $0.022_{\pm0.006}(32)$ | $0.451_{\pm0.052}(19)$ | $0.002_{\pm0.0}(39)$ | $NAN$ | $0.488_{\pm0.113}(9)$ | $0.105_{\pm0.014}(27)$ | $NAN$ | $0.017_{\pm0.0}(33)$ | $0.454_{\pm0.054}(15)$ | | |
| Avg Ranking | 30.73 | 17.55 | 33.82 | 42.0 | 25.82 | 29.73 | 45.0 | 34.27 | 25.82 | | |

Table 12: AUROC results on 9 high-dimensional datasets, where we compare TCCM to 44 baselines (with 5 independent runs). We report the mean ± std (rank).

| Dataset | TCCM (Ours) | AE | AE-1SVM | ALAD | AnoGAN | DAGMM | DeepSVDD | DIF | DROCC | DTE-Cat | DTE-DDPM | DTE-Gaussian |
|---|---|---|---|---|---|---|---|---|---|---|---|---|
| backdoor | 0.948±0.014(5) | 0.935±0.002(7) | 0.919±0.002(11) | 0.488±0.139(38) | 0.58±0.15(31) | 0.451±0.15(39) | 0.551±0.043(33) | 0.926±0.001(9) | 0.5±0.0(34) | 0.918±0.004(12) | 0.606±0.008(29) | 0.925±0.003(10) |
| campaign | 0.785±0.017(8) | 0.815±0.011(1) | 0.79±0.006(6) | 0.527±0.041(42) | 0.758±0.021(20) | 0.602±0.04(39) | 0.731±0.025(25) | 0.676±0.01(35) | 0.59±0.123(40) | 0.787±0.005(7) | 0.708±0.007(28) | 0.701±0.063(30) |
| census | 0.715±0.004(4) | 0.721±0.003(3) | 0.705±0.005(10) | 0.407±0.062(37) | 0.685±0.037(16) | 0.512±0.028(31) | 0.693±0.015(13) | 0.579±0.016(29) | 0.5±0.0(32) | 0.693±0.008(13) | 0.637±0.002(22) | 0.493±0.039(34) |
| InternetAds | 0.872±0.017(6) | 0.883±0.006(1) | 0.882±0.006(2) | 0.373±0.026(40) | 0.595±0.054(24) | 0.871±0.027(20) | NAN | 0.683±0.019(21) | 0.55±0.018(28) | 0.492±0.07(35) | 0.85±0.03(10) | 0.868±0.018(7) |
| mnist | 0.933±0.018(7) | 0.936±0.003(4) | 1.0±0.0(1) | 0.542±0.034(44) | 0.871±0.01(20) | 0.762±0.061(32) | 0.811±0.027(27) | 0.882±0.01(19) | 0.852±0.022(24) | 0.902±0.017(17) | 0.815±0.009(26) | 0.859±0.005(23) |
| musk | 1.0±0.0(1) | 1.0±0.0(1) | 1.0±0.0(1) | 0.512±0.1(41) | 0.962±0.05(30) | 0.899±0.06(34) | 0.987±0.042(19) | 0.999±0.001(17) | 0.281±0.177(44) | 1.0±0.0(1) | 1.0±0.0(1) | 1.0±0.0(1) |
| optdigits | 0.9±0.036(7) | 0.87±0.004(11) | 0.682±0.03(23) | 0.479±0.043(38) | 0.547±0.112(30) | 0.613±0.202(28) | 0.542±0.176(32) | 0.626±0.066(25) | 0.859±0.049(13) | 0.849±0.021(15) | 0.618±0.027(27) | 0.883±0.01(10) |
| SpamBase | 0.866±0.006(1) | 0.816±0.011(14) | 0.813±0.012(15) | 0.54±0.01(37) | 0.825±0.01(10) | 0.633±0.034(35) | 0.796±0.031(22) | 0.5±0.029(38) | 0.777±0.006(25) | 0.848±0.01(5) | 0.784±0.011(23) | 0.784±0.016(23) |
| speech | 0.549±0.009(5) | 0.476±0.007(27) | 0.471±0.007(35) | 0.472±0.035(33) | 0.486±0.028(23) | 0.498±0.034(17) | 0.514±0.035(14) | 0.476±0.028(27) | 0.527±0.042(9) | 0.523±0.009(12) | 0.521±0.061(13) | 0.531±0.029(7) |
| Avg Ranking | 4.89 | 7.67 | 12.33 | 38.89 | 22.67 | 31.88 | 24.0 | 26.11 | 28.44 | 10.22 | 21.0 | 16.11 |

| Dataset | DTE-IG | DTE-NP | GANomaly | GOAD | ICL | LUNAR | MCM | MO_GAAL | PlanarFlow | SLAD | SO_GAAL | VAE |
|---|---|---|---|---|---|---|---|---|---|---|---|---|
| backdoor | 0.938±0.01(6) | 0.951±0.001(3) | 0.795±0.057(18) | 0.495±0.077(37) | NAN | 0.954±0.001(2) | 0.785±0.016(8) | 0.859±0.037(15) | NAN | 0.5±0.0(34) | 0.77±0.123(19) | 0.911±0.001(4) |
| campaign | 0.717±0.061(27) | 0.785±0.002(8) | 0.682±0.038(34) | 0.393±0.02(44) | 0.811±0.005(2) | 0.731±0.003(25) | 0.785±0.016(8) | 0.65±0.053(37) | 0.687±0.078(33) | 0.763±0.005(19) | 0.696±0.023(32) | 0.781±0.001(11) |
| census | 0.66±0.014(20) | 0.722±0.001(2) | 0.695±0.022(12) | NAN | NAN | 0.674±0.003(19) | NAN | 0.59±0.051(26) | NAN | 0.616±0.106(25) | 0.573±0.037(30) | 0.706±0.001(8) |
| InternetAds | 0.809±0.03(12) | 0.795±0.032(14) | 0.772±0.035(15) | 0.651±0.103(23) | NAN | 0.856±0.005(8) | NAN | 0.453±0.044(36) | 0.796±0.023(13) | 0.853±0.03(9) | 0.423±0.056(37) | 0.881±0.004(3) |
| mnist | 0.793±0.05(29) | 0.944±0.002(2) | 0.791±0.073(30) | 0.44±0.13(46) | 0.913±0.01(12) | 0.926±0.001(11) | 0.928±0.01(10) | 0.686±0.106(36) | 0.809±0.023(28) | 0.912±0.006(14) | 0.681±0.078(37) | 0.935±0.002(5) |
| musk | 1.0±0.0(1) | 1.0±0.0(1) | 0.954±0.02(35) | 1.0±0.0(1) | 1.0±0.0(1) | 1.0±0.0(1) | 1.0±0.0(1) | 0.893±0.132(35) | 0.875±0.103(37) | 1.0±0.0(1) | 0.889±0.162(36) | 1.0±0.0(1) |
| optdigits | 0.868±0.08(12) | 0.961±0.004(5) | 0.724±0.167(21) | 0.902±0.144(20) | 0.981±0.004(2) | 0.997±0.001(1) | 0.852±0.084(14) | 0.245±0.071(45) | 0.429±0.14(41) | 0.942±0.01(6) | 0.26±0.105(44) | 0.686±0.057(22) |
| SpamBase | 0.744±0.049(28) | 0.851±0.005(3) | 0.825±0.015(10) | 0.405±0.188(43) | 0.843±0.007(6) | 0.83±0.005(9) | 0.82±0.014(12) | 0.431±0.105(42) | 0.837±0.025(8) | 0.851±0.007(3) | 0.398±0.098(44) | 0.812±0.013(16) |
| speech | 0.445±0.029(44) | 0.567±0.011(2) | 0.492±0.031(20) | 0.504±0.079(16) | 0.565±0.036(3) | 0.57±0.063(1) | 0.472±0.007(33) | 0.458±0.034(42) | 0.493±0.047(19) | 0.511±0.05(15) | 0.457±0.031(43) | 0.471±0.006(35) |
| Avg Ranking | 19.89 | 4.44 | 17.89 | 32.62 | 4.33 | 8.0 | 13.0 | 34.89 | 25.57 | 14.0 | 35.78 | 12.67 |

| Dataset | CBLOF | CD | ECOD | FB | GMM | HBOS | IForest | KDE | LMDD | LODA | MCD | OCSVM |
|---|---|---|---|---|---|---|---|---|---|---|---|---|
| backdoor | 0.714±0.04(22) | 0.744±0.127(21) | 0.846±0.001(17) | 0.95±0.006(4) | 0.928±0.0(8) | 0.713±0.005(23) | 0.746±0.024(20) | 0.905±0.001(14) | NAN | 0.411±0.226(40) | 0.851±0.102(16) | 0.626±0.004(28) |
| campaign | 0.772±0.002(16) | 0.768±0.02(18) | 0.77±0.001(17) | 0.675±0.043(36) | 0.799±0.001(4) | 0.775±0.043(14) | 0.738±0.018(22) | 0.775±0.001(14) | 0.699±0.03(31) | 0.584±0.04(44) | 0.791±0.005(5) | 0.776±0.001(13) |
| census | 0.708±0.003(6) | 0.5±0.0(32) | 0.658±0.174(21) | 0.582±0.006(28) | 0.706±0.001(8) | 0.624±0.001(23) | 0.624±0.012(23) | 0.723±0.002(1) | NAN | 0.561±0.036(26) | 0.575±0.122(25) | 0.705±0.001(11) |
| InternetAds | 0.697±0.044(19) | 0.5±0.0(32) | 0.678±0.007(20) | 0.746±0.041(16) | 0.881±0.016(3) | 0.403±0.05(39) | 0.419±0.079(38) | 0.841±0.019(11) | 0.561±0.026(26) | 0.575±0.122(25) | NAN | 0.705±0.044(18) |
| mnist | 0.913±0.003(12) | 0.605±0.117(41) | 0.747±0.004(33) | 0.929±0.004(9) | 0.926±0.001(11) | 0.611±0.006(39) | 0.864±0.012(21) | 0.945±0.002(15) | 0.722±0.009(35) | 0.728±0.074(34) | 0.891±0.047(18) | 0.91±0.003(15) |
| musk | 1.0±0.0(1) | 0.671±0.008(38) | 0.954±0.02(35) | 1.0±0.0(1) | 1.0±0.0(1) | 1.0±0.0(1) | 0.931±0.018(33) | 1.0±0.0(1) | 1.0±0.0(1) | 1.0±0.0(1) | 0.999±0.001(17) | 1.0±0.0(1) |
| optdigits | 0.835±0.02(16) | 0.383±0.024(42) | 0.609±0.004(29) | 0.975±0.003(3) | 0.813±0.005(18) | 0.899±0.004(8) | 0.809±0.064(19) | 0.973±0.004(4) | 0.475±0.019(39) | 0.498±0.184(36) | 0.652±0.034(24) | 0.626±0.01(25) |
| SpamBase | 0.811±0.01(17) | 0.491±0.023(39) | 0.851±0.005(3) | 0.757±0.024(27) | 0.797±0.012(21) | 0.772±0.007(26) | 0.819±0.017(13) | 0.857±0.004(2) | 0.661±0.05(32) | 0.681±0.01(31) | 0.799±0.04(20) | 0.813±0.017(17) |
| speech | 0.473±0.006(32) | 0.482±0.006(25) | 0.471±0.006(35) | 0.496±0.006(18) | 0.527±0.005(9) | 0.476±0.006(27) | 0.468±0.023(40) | 0.531±0.059(7) | 0.489±0.016(21) | 0.474±0.006(30) | 0.526±0.005(11) | 0.468±0.006(40) |
| Avg Ranking | 15.67 | 30.78 | 25.78 | 15.78 | 9.22 | 25.44 | 26.11 | 6.11 | 26.43 | 30.33 | 17.0 | 18.67 |

| Dataset | QMCD | Sampling | ABOD | COF | INNE | KNN | KPCA | LOF | PCA |
|---|---|---|---|---|---|---|---|---|---|
| backdoor | 0.567±0.002(32) | 0.678±0.021(24) | 0.668±0.001(25) | NAN | 0.606±0.025(29) | 0.666±0.001(26) | NAN | 0.664±0.001(27) | 0.5±0.0(43) |
| campaign | 0.805±0.006(3) | 0.74±0.028(21) | 0.706±0.002(29) | 0.434±0.001(36) | 0.738±0.003(22) | 0.736±0.001(24) | NAN | 0.631±0.043(38) | 0.5±0.0(43) |
| census | 0.681±0.001(18) | 0.707±0.006(7) | NAN | NAN | 0.689±0.005(15) | 0.682±0.001(17) | NAN | 0.585±0.001(27) | 0.5±0.0(32) |
| InternetAds | 0.5±0.001(32) | 0.707±0.042(17) | 0.528±0.022(30) | 0.266±0.034(41) | 0.548±0.04(29) | 0.561±0.041(26) | 0.505±0.039(31) | 0.223±0.025(42) | 0.493±0.007(34) |
| mnist | 0.639±0.007(38) | 0.904±0.011(16) | 0.766±0.006(31) | 0.594±0.009(42) | 0.863±0.006(22) | 0.546±0.018(43) | 0.61±0.04(40) | 0.5±0.0(45) | |
| musk | 0.036±0.007(46) | 1.0±0.0(1) | 1.0±0.0(1) | 0.056±0.002(45) | 0.449±0.046(43) | 0.997±0.001(28) | 0.499±0.042(42) | 0.623±0.007(39) | 0.999±0.0(25) |
| optdigits | 0.133±0.003(46) | 0.814±0.011(17) | 0.45±0.005(40) | 0.512±0.008(33) | 0.506±0.026(34) | 0.292±0.003(43) | 0.498±0.095(36) | 0.544±0.008(31) | 0.5±0.0(35) |
| SpamBase | 0.696±0.037(29) | 0.806±0.01(19) | 0.47±0.007(40) | 0.365±0.015(45) | 0.688±0.017(30) | 0.64±0.01(34) | 0.59±0.02(36) | 0.352±0.012(46) | 0.469±0.006(41) |
| speech | 0.413±0.007(46) | 0.483±0.014(24) | 0.557±0.015(4) | 0.416±0.021(45) | 0.474±0.006(30) | 0.469±0.008(39) | 0.489±0.039(21) | 0.482±0.009(25) | 0.47±0.006(38) |
| Avg Ranking | 32.22 | 16.22 | 31.11 | 41.5 | 26.56 | 30.67 | 34.5 | 35.0 | 36.33 |

Table 13: AUPRC results on 9 high-dimensional datasets, where we compare TCCM to 44 baselines (with 5 independent runs). We report the mean ± std (rank).

| Dataset | TCCM (Ours) | AE | AE-1SVM | ALAD | AnoGAN | DAGMM | DeepSVDD | DIF | DROCC | DTE-Cat | DTE-DDPM | DTE-Gaussian |
|---|---|---|---|---|---|---|---|---|---|---|---|---|
| backdoor | 0.835±0.063(8) | 0.868±0.002(4) | 0.86±0.006(5) | 0.059±0.03(35) | 0.071±0.003(32) | 0.051±0.014(36) | 0.069±0.012(33) | 0.696±0.03(10) | 0.048±0.0(38) | 0.631±0.037(12) | 0.086±0.002(26) | 0.804±0.017(9) |
| campaign | 0.49±0.019(15) | 0.504±0.013(5) | 0.206±0.006(8) | 0.096±0.046(36) | 0.229±0.039(42) | 0.492±0.022(13) | 0.449±0.035(23) | 0.381±0.04(18) | 0.297±0.13(41) | 0.495±0.014(11) | 0.443±0.009(25) | 0.41±0.06(29) |
| census | 0.213±0.005(3) | 0.217±0.004(2) | 0.206±0.008(8) | 0.118±0.001(36) | 0.178±0.016(19) | 0.13±0.014(14) | 0.194±0.014(11) | 0.126±0.01(33) | 0.117±0.03(34) | 0.118±0.002(20) | 0.18±0.002(18) | 0.143±0.004(26) |
| InternetAds | 0.829±0.042(4) | 0.869±0.011(1) | 0.868±0.011(2) | 0.255±0.006(40) | 0.433±0.114(24) | NAN | 0.486±0.05(23) | 0.325±0.051(31) | 0.405±0.07(25) | 0.725±0.091(11) | 0.531±0.108(20) | 0.865±0.026(6) |
| mnist | 0.747±0.048(5) | 0.737±0.009(6) | 0.716±0.016(11) | 0.212±0.003(45) | 0.622±0.019(21) | 0.494±0.086(31) | 0.55±0.038(25) | 0.624±0.021(20) | 0.636±0.03(18) | 0.644±0.029(18) | 0.557±0.012(23) | 0.666±0.021(17) |
| musk | 1.0±0.0(1) | 1.0±0.0(1) | 1.0±0.0(1) | 0.092±0.082(44) | 0.745±0.181(30) | 0.641±0.194(32) | 0.881±0.178(29) | 0.989±0.017(25) | 0.135±0.13(42) | 1.0±0.0(1) | 0.997±0.006(22) | 1.0±0.0(1) |
| optdigits | 0.267±0.092(9) | 0.175±0.004(15) | 0.08±0.009(25) | 0.053±0.006(38) | 0.058±0.012(34) | 0.092±0.068(23) | 0.066±0.03(31) | 0.074±0.012(27) | 0.201±0.055(12) | 0.174±0.024(16) | 0.075±0.009(26) | 0.241±0.018(11) |
| SpamBase | 0.899±0.004(1) | 0.849±0.01(14) | 0.849±0.011(14) | 0.619±0.009(37) | 0.863±0.007(9) | 0.699±0.022(33) | 0.836±0.028(21) | 0.569±0.026(41) | 0.829±0.009(22) | 0.87±0.011(7) | 0.829±0.01(22) | 0.822±0.01(24) |
| speech | 0.058±0.005(2) | 0.054±0.007(22) | 0.039±0.002(22) | 0.033±0.002(38) | 0.041±0.006(16) | 0.04±0.006(17) | 0.034±0.002(36) | 0.034±0.003(36) | 0.04±0.0(17) | 0.047±0.005(8) | 0.04±0.005(17) | 0.049±0.007(5) |
| Avg Ranking | 5.56 | 7.56 | 10.33 | 39.44 | 22.0 | 30.0 | 26.11 | 28.44 | 27.78 | 11.67 | 22.11 | 14.44 |

| Dataset | DTE-IG | DTE-NP | GANomaly | GOAD | ICL | LUNAR | MCM | MO_GAAL | PlanarFlow | SLAD | SO_GAAL | VAE |
|---|---|---|---|---|---|---|---|---|---|---|---|---|
| backdoor | 0.851±0.006(7) | 0.642±0.003(11) | 0.153±0.099(19) | 0.062±0.01(34) | NAN | 0.899±0.004(1) | NAN | 0.259±0.075(15) | NAN | 0.048±0.0(38) | 0.13±0.072(20) | 0.856±0.001(6) |
| campaign | 0.447±0.025(24) | 0.492±0.004(13) | 0.385±0.058(33) | 0.176±0.01(44) | 0.498±0.009(7) | 0.435±0.009(26) | 0.493±0.01(12) | 0.369±0.052(35) | 0.397±0.085(31) | 0.485±0.007(17) | 0.398±0.027(30) | 0.499±0.003(6) |
| census | 0.188±0.003(15) | 0.217±0.003(2) | 0.188±0.013(15) | NAN | NAN | 0.165±0.001(23) | 0.181±0.017(21) | 0.154±0.017(24) | NAN | 0.168±0.046(22) | 0.146±0.125(25) | 0.2±0.001(7) |
| InternetAds | 0.722±0.06(12) | 0.661±0.113(13) | 0.639±0.101(14) | 0.505±0.078(22) | NAN | 0.771±0.025(9) | NAN | 0.365±0.06(29) | 0.61±0.076(16) | 0.735±0.046(10) | 0.356±0.07(30) | 0.867±0.012(3) |
| mnist | 0.549±0.091(26) | 0.769±0.02(3) | 0.495±0.072(30) | 0.264±0.06(43) | 0.715±0.109(13) | 0.775±0.023(1) | 0.717±0.03(11) | 0.433±0.095(34) | 0.529±0.03(29) | 0.728±0.021(7) | 0.414±0.059(36) | 0.726±0.006(8) |
| musk | 1.0±0.0(1) | 1.0±0.0(1) | 1.0±0.0(1) | 0.714±0.283(31) | 1.0±0.0(1) | 1.0±0.0(1) | 1.0±0.0(1) | 0.6±0.304(34) | 0.546±0.263(36) | 1.0±0.0(1) | 0.599±0.323(35) | 1.0±0.0(1) |
| optdigits | 0.282±0.117(8) | 0.406±0.026(5) | 0.113±0.046(21) | 0.193±0.132(13) | 0.606±0.04(2) | 0.926±0.03(1) | 0.181±0.079(14) | 0.035±0.003(45) | 0.048±0.012(41) | 0.34±0.04(7) | 0.036±0.006(44) | 0.081±0.017(24) |
| SpamBase | 0.796±0.041(26) | 0.875±0.005(5) | 0.859±0.011(10) | 0.555±0.145(42) | 0.884±0.004(3) | 0.856±0.007(11) | 0.854±0.012(12) | 0.601±0.06(39) | 0.875±0.022(5) | 0.888±0.008(2) | 0.577±0.066(40) | 0.845±0.012(18) |
| speech | 0.03±0.002(44) | 0.054±0.005(5) | 0.042±0.01(15) | 0.037±0.007(31) | 0.033±0.002(36) | 0.043±0.008(13) | 0.056±0.015(4) | 0.038±0.001(22) | 0.036±0.003(32) | 0.036±0.007(32) | 0.031±0.004(44) | 0.038±0.002(22) |
| Avg Ranking | 18.11 | 6.56 | 17.56 | 32.5 | 6.5 | 8.56 | 11.83 | 33.0 | 27.14 | 15.11 | 33.44 | 11.0 |

| Dataset | CBLOF | CD | ECOD | FB | GMM | HBOS | IForest | KDE | LMDD | LODA | MCD | OCSVM |
|---|---|---|---|---|---|---|---|---|---|---|---|---|
| backdoor | 0.095±0.013(23) | 0.242±0.19(16) | 0.169±0.001(18) | 0.87±0.099(3) | 0.87±0.001(2) | 0.086±0.001(26) | 0.089±0.01(25) | 0.455±0.008(14) | NAN | 0.072±0.07(30) | 0.202±0.089(17) | 0.077±0.003(29) |
| campaign | 0.485±0.006(16) | 0.469±0.026(20) | 0.502±0.004(3) | 0.307±0.067(39) | 0.517±0.001(2) | 0.508±0.002(22) | 0.462±0.02(22) | 0.478±0.003(19) | 0.429±0.023(27) | 0.302±0.074(40) | 0.482±0.014(18) | 0.497±0.003(8) |
| census | 0.202±0.003(10) | 0.315±0.035(35) | NAN | 0.126±0.002(32) | 0.2±0.001(11) | 0.139±0.028(23) | 0.14±0.047(27) | 0.216±0.006(4) | NAN | 0.135±0.04(30) | 0.397±0.137(27) | 0.205±0.005(9) |
| InternetAds | 0.529±0.109(21) | 0.315±0.0(35) | 0.619±0.009(15) | 0.565±0.007(17) | 0.775±0.068(7) | 0.26±0.027(39) | 0.265±0.037(38) | 0.773±0.05(8) | 0.399±0.091(26) | 0.397±0.137(27) | NAN | 0.546±0.115(18) |
| mnist | 0.684±0.011(14) | 0.281±0.12(41) | 0.303±0.007(40) | 0.716±0.025(11) | 0.725±0.012(9) | 0.215±0.006(44) | 0.539±0.043(28) | 0.774±0.018(3) | 0.46±0.112(32) | 0.419±0.115(35) | 0.583±0.093(22) | 0.683±0.006(15) |
| musk | 1.0±0.0(1) | 0.132±0.005(43) | 0.604±0.005(33) | 1.0±0.0(1) | 1.0±0.0(1) | 1.0±0.0(1) | 0.355±0.052(37) | 1.0±0.0(1) | 0.992±0.007(24) | 0.997±0.006(22) | 0.987±0.01(27) | 1.0±0.0(1) |
| optdigits | 0.14±0.015(18) | 0.042±0.002(42) | 0.098±0.006(22) | 0.511±0.024(3) | 1.0±0.0(1) | 1.0±0.0(1) | 0.132±0.004(19) | 1.0±0.0(1) | 0.052±0.002(39) | 0.057±0.015(35) | 0.072±0.007(26) | 0.068±0.009(28) |
| SpamBase | 0.846±0.017(17) | 0.537±0.014(43) | 0.682±0.008(34) | 0.732±0.028(30) | 0.837±0.011(20) | 0.781±0.047(27) | 0.854±0.021(12) | 0.883±0.004(4) | 0.704±0.042(32) | 0.769±0.067(28) | 0.814±0.026(25) | 0.847±0.024(16) |
| speech | 0.038±0.001(22) | 0.038±0.001(22) | 0.04±0.009(17) | 0.044±0.001(11) | 0.056±0.007(4) | 0.044±0.006(11) | 0.032±0.002(41) | 0.065±0.006(1) | 0.045±0.008(10) | 0.043±0.006(12) | 0.043±0.002(13) | 0.038±0.001(22) |
| Avg Ranking | 15.78 | 29.22 | 24.0 | 17.44 | 8.22 | 20.44 | 27.44 | 6.33 | 31.67 | 21.29 | 16.33 |

| Dataset | QMCD | Sampling | ABOD | COF | INNE | KNN | KPCA | LOF | PCA |
|---|---|---|---|---|---|---|---|---|---|
| backdoor | 0.051±0.0(36) | 0.081±0.012(28) | 0.095±0.04(25) | NAN | 0.072±0.006(30) | 0.109±0.001(21) | NAN | 0.107±0.001(22) | 0.048±0.0(38) |
| campaign | 0.496±0.012(10) | 0.466±0.036(21) | 0.347±0.003(36) | NAN | 0.392±0.005(32) | 0.413±0.002(28) | NAN | 0.313±0.001(38) | 0.202±0.0(43) |
| census | 0.207±0.001(7) | 0.199±0.003(13) | 0.094±0.0(37) | NAN | 0.172±0.003(21) | 0.181±0.0(17) | NAN | 0.137±0.001(29) | 0.117±0.0(34) |
| InternetAds | 0.315±0.0(35) | 0.536±0.105(19) | 0.339±0.011(32) | 0.227±0.01(41) | 0.325±0.025(34) | 0.366±0.032(28) | 0.328±0.019(33) | 0.207±0.006(42) | 0.306±0.004(37) |
| mnist | 0.358±0.01(37) | 0.679±0.021(16) | 0.445±0.009(33) | 0.317±0.016(39) | 0.543±0.02(27) | 0.555±0.014(24) | 0.269±0.018(42) | 0.347±0.013(38) | 0.169±0.0(46) |
| musk | 0.032±0.0(46) | 1.0±0.0(1) | 0.042±0.006(45) | 0.141±0.015(41) | 0.931±0.031(28) | 0.191±0.003(40) | 0.279±0.143(39) | 0.326±0.009(38) | 0.98±0.0(27) |
| optdigits | 0.031±0.0(46) | 0.127±0.006(20) | 0.098±0.02(22) | 0.061±0.001(33) | 0.051±0.0(40) | 0.036±0.0(43) | 0.054±0.011(37) | 0.068±0.002(29) | 0.056±0.0(36) |
| SpamBase | 0.751±0.0(29) | 0.843±0.011(19) | 0.43±0.002(44) | 0.499±0.008(45) | 0.674±0.016(35) | 0.725±0.01(31) | 0.648±0.011(36) | 0.485±0.046(46) | 0.522±0.004(44) |
| speech | 0.026±0.0(46) | 0.035±0.001(34) | 0.049±0.01(7) | 0.029±0.003(45) | 0.035±0.002(34) | 0.038±0.003(30) | 0.033±0.004(38) | 0.04±0.002(17) | 0.038±0.001(22) |
| Avg Ranking | 32.44 | 19.0 | 30.33 | 40.67 | 31.22 | 28.22 | 37.5 | 33.22 | 36.33 |

