# OpenReview forum: "Scalable, Explainable and Provably Robust Anomaly Detection with One-Step Flow Matching"
_NeurIPS.cc/2025/Conference — NeurIPS 2025 poster_

### Official Review · Reviewer_Gjui · 2025-06-30

**Clarity:** 3
**Significance:** 2
**Originality:** 2
**Rating:** 4
**Confidence:** 4

**Summary:**

This paper proposes a novel semi-supervised AD method for tabular data using the concept of flowing matching. Specifically, the author uses a neural network to predict the time-dependent velocity field in the transformation between normal data distribution and a degenerate Dirac distribution at the origin. The method is evaluated on the popular benchmark ADbench, and the  method outperforms all baselines in the average rank, also the authors provides several theoretical guarantees for the method and detailed experimental results for ablation studies.

**Questions:**

**Method**:
See weakness. If the authors address my concerns properly, i will raise my score.

**Experimental Results**:

1. In the experimental results of the paper, the performance of $k$NN is much worse than most of methods. However, in [1],  following the same experimental setting, $k$NN is almost the most powerful semi-supervised AD method. Why the perfromance of $k$NN is so different in the paper? Also, there are several recent AD methods based on generative models and AE to be mentioned and compared, e.g. [2] [3] [4].

[1] Livernoche, Victor, et al. "On diffusion modeling for anomaly detection." arXiv preprint arXiv:2305.18593 (2023).

[2] Liu et al. Unsupervised Anomaly Detection by Robust Density Estimation, AAAI 2022

[3] Rozner et al. Anomaly Detection with Variance Stabilized Density Estimation, UAI 2024

[4] Lindenbaum, et al. Transductive and inductive outlier detection with robust autoencoders. UAI 2024


2. In the time comparison part, the authors compare TCCM with two non-parametric methods: KDE and DTENP, and LUNAR which requires graph converting. Then the authors claim that "TCCM achieves considerably faster inference than most deep learning baselines". I think this is quite unfair since the time complexity of the $3$ methods is at least $\mathcal{O}(n^2d)$ and there are many faster deep learning methods like auto-encoder, DeepSVDD, MCM and so on.

**Ethical Concerns:**

["NO or VERY MINOR ethics concerns only"]

**Final Justification:**

The authors solved most of my concerns, I decide to raise my score.

**Limitations:**

Yes.

**Paper Formatting Concerns:**

1. The authors should include a brief introduction on generative mode based anomaly detection methods in main context.

2. In Figure (a) and (b), it is better to also include the average performance of each methods.

**Quality:**

2

**Strengths And Weaknesses:**

**Strengths:**

1. The authors provide complete experimental results. 2. The method demonstrates good experimental performance on a wide range of datasets. 3. The paper is easy to follow and well written.

**Weaknesses:**
1. In the limitations of existing studies, the authors mention that "rely on restrictive architectural constraints (e.g., no biases, bounded
activations) to avoid representation collapse". For TCCM, i think representation or model collapse could also happen, e.g., consider the neural network is a one-layer MLP, $f_θ([\mathbf{z}; \mathrm{Embed}(t)])=[-I,\mathbf{0}][\mathbf{z}; \mathrm{Embed}(t)]^T=-\mathbf{z}$. In this case, the loss attains minimal but the model can not identify normal data and anomalous data, and it seems that this model collapse is not mentioned and solved in the paper.

2. Usually, in flow matching or score matching, the points in the original distribution and points in the target distribution correspond one-to-one. But in the paper authors use "a degenerate Dirac distribution at the origin" as the target distribution, which contains only one point, indicating a relationship of one-to-many.  This does not make sense to me.

3. In the training process of TCCM, for different timestep $t$, $\mathbf{z}_t=\mathbf{z}_0$, and this is different from original flow matching. The only change in $\tilde{\mathbf{z}}_t$ is the time embedding. If  $\mathbf{z}_t$ does not change for different $t$, then it is not a "flow".  In my understanding, the trajectory is a straight line and you should sample from the line $\mathbf{z}_t=\frac{\mathbf{z}-\mathbf{0}}{t}$ to train TCCM. For this reason, TCCM is more like an AE on augmented data combining self-supervised learning and auto-encoder, which is not novel.

---

> ### Author Rebuttal · Authors · 2025-07-30
>
> We thank the reviewer for the summary and positive comments. Below, we address the concerns point by point.
> > Weaknesses 1:  In the limitations of existing studies, the authors mention that “rely on restrictive architectural constraints (e.g., no biases, bounded activations) to avoid representation collapse". For TCCM, I think representation or model collapse could also happen, e.g., consider the neural network is a one-layer MLP,  $f_{\theta}([\mathbf{z}, \text{Embed}(t)]) = [\mathbf{I}, \mathbf{0}] [\mathbf{z}; \text{Embed}(t)]^T = \mathbf{z}$.  In this case, the loss attains minimal but the model cannot identify normal data and anomalous data, and it seems that this model collapse is not mentioned and solved in the paper.
>
>
> We thank the reviewer for raising the concern regarding potential representation collapse. The illustrative case  $f_{\theta}([\mathbf{z},\text{Embed}(t)] )$ $ = [\mathbf{I},\mathbf{0}][\mathbf{z};\text{Embed}(t)]^{T} = \mathbf{z}$
> assumes a highly restricted architecture—a single-layer linear mapping without bias or activation—where the time embedding has no effect. While this is a valid theoretical edge case, it does not reflect the architecture used in TCCM.
>
> In practice, **TCCM** uses a multi-layer MLP with ReLU activations and high-dimensional sinusoidal time embeddings, explicitly concatenated to the input. This enables the model to learn complex, time-varying contraction dynamics, making trivial identity (or partial identity) mappings unlikely.
>
> Unlike prior methods such as Deep SVDD [Ruff2018Deep]—which require architectural constraints (e.g., no bias, bounded activations) to avoid collapse—TCCM imposes no such restrictions. Instead, collapse is discouraged through the use of time-conditioned supervision, multi-time-step training, and implicit regularization from architectural nonlinearity.
>
> Empirically, we observe no evidence of collapse: training loss does not flatten trivially, anomaly scores vary meaningfully across test points, and anomaly detection accuracy remains consistently strong across datasets.
>
> We will clarify this point in the final version and explicitly discuss how our design mitigates collapse without relying on handcrafted constraints.
>
> **Reference**
>
> - [Ruff2018Deep] Ruff, Lukas, et al. "Deep one-class classification." ICML, 2018.
>
> >Weaknesses 2: Usually, in flow matching or score matching, the points in the original distribution and points in the target distribution correspond one-to-one. But in the paper authors use ``a degenerate Dirac distribution at the origin" as the target distribution, which contains only one point, indicating a relationship of one-to-many. This does not make sense to me.
>
> We thank the reviewer for this insightful comment regarding the use of a degenerate (Dirac) target distribution in our formulation. Below we clarify the motivation and appropriateness of this design in the context of anomaly detection.
>
> **(1) Different objective from conventional flow matching.**
> Classical flow matching and score matching methods are typically used for *generative modeling*, where samples from the source and target distributions correspond one-to-one to ensure reconstruction quality. In contrast, our goal is not generation but *anomaly detection*—we care about measuring how well a sample follows contraction dynamics, not whether it can be mapped to a specific output point. As such, we do not require a bijective mapping between distributions.
>
> **(2) One-to-many mapping is intentional and beneficial.**
> The use of a Dirac distribution at the origin introduces a *consistent supervision signal*: every normal point learns to contract toward the same target. This simplifies optimization and injects a strong geometric prior that improves both efficiency and robustness. The resulting one-to-many relationship does not harm performance—in fact, it supports generalization, as anomalies that do not conform to this contraction pattern are naturally identified as outliers.
>
> **(3) Similarity to Deep SVDD, but extended to continuous flows.**
> This idea mirrors the intuition behind **Deep SVDD**, which learns to map normal data into a compact region (typically a small hypersphere) in latent space. TCCM generalizes this notion to a continuous-time setting via a learned vector field, which guides contraction toward the origin across time. We note that Deep SVDD does *not* collapse all samples to the center but instead penalizes distance to a hypersphere center—our design is even more aggressive in promoting compactness.
>
> **(4) Clarification is already provided in Appendix A.2 and Appendix C.1.**
> We have discussed (1) the difference of TCCM from generative models in *Appendix A.2* (see paragraph and *“Difference from Generative Models”*), and (2) the connection and distinction between TCCM, flow matching, and diffusion models in detail in *Appendix C.1* (*“Relation to Flow Matching and Diffusion Modeling”*).
>
> **In summary,** the “one-to-many” design is not only valid in our setting, but also *aligned with the goals of semi-supervised anomaly detection*. Our theoretical and empirical results (e.g., Figure 1 and Section 5) demonstrate its effectiveness.
>
> >Weakness 3: In the training process of TCCM, for different timestep $t$,
> $\mathbf{z} _ {t}=\mathbf{z} _ {0}$, and this is different from original flow matching. The only change in $\tilde{\mathbf{z}}$ is the time embedding. If
> $\mathbf{z} _ {t}$ does not change for different $t$, then it is not a ``flow". In my understanding, the trajectory is a straight line and you should sample from the line $\mathbf{z} _ {t}=\frac{\mathbf{z}-\mathbf{0}}{t-0}$ to train TCCM. For this reason, TCCM is more like an AE on augmented data combining self-supervised learning and auto-encoder, which is not novel.
>
> We thank the reviewer for the thoughtful comments regarding the design of our training procedure and its relation to classical flow matching. Below we clarify the rationale and provide empirical and conceptual justifications.
>
> **(1) Time-conditioned vector fields without interpolated inputs.**
> It is correct that during training, TCCM always takes the original input $\mathbf{z}_0 = \mathbf{z}_t$, and the variation across time is introduced solely via the time embedding. Unlike standard flow matching, which supervises the velocity field on interpolated samples along a trajectory (e.g., $\mathbf{z}_t = t\mathbf{z} + (1 - t)\mathbf{0}$), TCCM directly supervises the field at the initial position $\mathbf{z}$ with a fixed target vector $-\mathbf{z}$.
>
> This is *intentional and appropriate* for our task: our model learns a time-conditioned vector field $\mathbf{f}_\theta(\mathbf{z}, t)$ that outputs the *instantaneous velocity* at location $\mathbf{z}$ and time $t$, rather than simulating or reconstructing trajectories. This formulation enables consistent, efficient learning without requiring time-dependent input sampling.
>
> **(2) Empirical justification via ablation.**
> We conduct an ablation study comparing our design against a variant that trains on interpolated inputs $t \cdot \mathbf{z}$ (i.e., $\mathbf{z}_t = t\mathbf{z}$). Results across multiple datasets show that this modification *hurts performance*, validating our choice of fixed inputs with time-conditioned supervision. We will include these results in the appendix.
>
> **(3) Distinction from autoencoders.**
> We respectfully disagree with the claim that TCCM reduces to an autoencoder with augmented inputs. Autoencoders are trained to reconstruct $\mathbf{z}$ using a decoding loss (e.g., $\|\hat{\mathbf{z}} - \mathbf{z}\|$), whereas TCCM learns a contraction vector field supervised by directional alignment $\mathbf{f}_\theta(\mathbf{z}, t) \approx -\mathbf{z}$. This leads to fundamentally different inductive biases and optimization dynamics. TCCM does not reconstruct inputs—it measures their deviation from contraction, enabling effective anomaly detection with strong empirical performance and interpretable scoring (see Eq. (5)).
>
> >Experimental Results: 1. ...Why the performance of
> kNN is so different in the paper? ...recent AD methods to be mentioned and compared [2] [3] [4].
>
> **(1) On the discrepancy in `kNN` performance:**
> The stronger results in [1] may stem from two key differences:
> (1) *Subsampling*: [1] subsamples datasets to 50,000 samples, while we use full datasets (up to 600K), making `kNN` more prone to scalability issues.
> (2) *Protocol*: [1] treats `kNN` as inductive (trained on normal samples), whereas we follow PyOD’s transductive setting (no separation), aligning with its official leaderboard. These factors explain the gap.
>
> **(2) On more baselines:**
> We thank the reviewer for pointing out [2,3,4]. We will refer to them in the revised version. However, it is challenging to implement (code is not given for some of them) and/or integrate them into our pipeline, and run all experiments during the rebuttal phase.
>
> >Experimental Results: 2. ...the authors compare TCCM with two non-parametric methods: ... Then the authors claim that ``TCCM achieves considerably faster inference than most deep learning baselines". I think this is quite unfair since the time complexity of the 3 methods is at least $O(n^{2}d)$ and there are many faster deep learning methods like auto-encoder, DeepSVDD, MCM and so on.
>
> We will include full runtime results (training, inference, and total) for all 45 algorithms in *Appendix D.2*. We also add a scatter plot of *inference time vs. AUROC* across all deep learning methods. Among 24 deep models, TCCM ranks 8th in inference time, while achieving the best AUROC. Although a few methods (e.g., DeepSVDD, GANomaly) are marginally faster, they suffer from significantly lower detection performance. In contrast, TCCM offers a strong trade-off between speed and effectiveness—making it one of the most efficient high-performing methods overall.
>
> >Formatting Concerns:
>
> Formatting suggestions will be taken in the revised version.

---

> > ### Comment · Reviewer_Gjui · 2025-08-01
> >
> > I appreciate the authors' response and their efforts in addressing several of my concerns. However, I maintain my original score for the following reasons:
> >
> > - **Unfair and Confusing Experimental Setting:**
> >
> >  The comparison between methods is compromised by the inconsistent use of inductive and transductive settings. While some methods are evaluated under an inductive setting, others (e.g., kNN) are tested in a transductive setting, which includes anomalous data in the training phase. This is particularly detrimental to methods like kNN, skewing their reported performance. Although the authors claim that this is for "aligning with its official leaderboard", to the bset of my knowledge, most paper [1] [2] [3] concluded that kNN is almost the most effective baseline in the same setting. For a fair comparison, all methods should be evaluated under the same setting (either inductive or transductive). Additionally, it is unclear whether hyperparameters (e.g., training epochs for TCCM) were tuned equally for all methods. Without such consistency, the performance results lack reliability.
> >
> >  [1] Livernoche, Victor, et al. "On diffusion modeling for anomaly detection."
> >
> >  [2] Thimonier et al. "Beyond Individual Input for Deep Anomaly Detection on Tabular Data." .
> >
> >  [3] Shenkar, Tom, and Lior Wolf. "Anomaly detection for tabular data with internal contrastive learning."
> >
> > - **Ambiguous Relationship Between TCCM and AE on Augmented Data:**
> >
> > The distinction between TCCM and AE remains unclear. Predicting $-z$ in TCCM versus predicting $z$ in AE does not appear fundamentally different. Furthermore, in Algorithm 1 (line 1159), for different time setp, $\hat{z}^{(i)}$ varies only in the time-embedding component, not in the data component, which contradicts the principles of flow matching—data should evolve in change of time step. This oversight raises concerns about the methodological soundness of the proposed approach.

---

> ### Author Response · Authors · 2025-08-05
> **Unified Inductive Evaluation and Architectural Ablation: TCCM vs. AE+TE**
>
> ### **(1) On Evaluation Setting: Inductive vs. Transductive Comparison)**
>
> We appreciate the reviewer’s concern regarding protocol consistency. In response, we have conducted a comprehensive re-evaluation of **all baselines under the inductive (semi-supervised) setting**, where models are trained solely on normal data.
> - For classical methods such as kNN, LOF, and PCA, we adopt their inductive variants as implemented in PyOD, following the  practice in DTE.
> - For space reasons, we report the top 20 methods below (full results will be included in the final version):
>
> | Detector           | Rank_PR ↓ | Rank_ROC ↓ |
> |--------------------|---------|----------|
> | TCCM(Ours) | 7.02    | 7.28     |
> | LUNAR              | 9.21    | 8.96     |
> | DTE-NP   | 9.36    | 7.34     |
> | KNN        | 10.74   | 8.89     |
> | KDE                | 10.74   | 10.74    |
> | AutoEncoder        | 15.72   | 15.19    |
> | ABOD       | 15.74   | 14.45    |
> | INNE      | 16.36   | 14.91    |
> | CBLOF              | 16.70   | 15.57    |
> | LOF        | 16.81   | 15.32    |
> | DTECategorical     | 16.87   | 15.49    |
> | OCSVM              | 17.04   | 17.68    |
> | ICL                | 17.17   | 19.57    |
> | DTEGaussian        | 17.21   | 18.53    |
> | KPCA       | 17.36   | 16.32    |
> | GMM                | 17.36   | 16.43    |
> | DTEInverseGamma    | 18.43   | 19.98    |
> | Sampling           | 19.04   | 17.09    |
> | SLAD               | 19.32   | 19.13    |
> | MCM                | 19.34   | 19.38    |
>
> As seen, TCCM ranks 1st in both PR and ROC metrics across all baselines, including kNN, LOF, and PCA in their inductive forms. We will include these new results and clear labeling of each method’s setting in the final submission.
>
>  ### **(2) On the Distinction Between TCCM and AE with Time Embedding (AE+TE)**
>
> To clarify, we conducted an ablation study comparing our method (TCCM) against an autoencoder baseline with the same time embedding input.
>
> **(2.1) Architectural Comparison**
> While both models ingest $[\mathbf{x}, \text{Embed}(t)]$, their parameterizations and learning objectives are different:
>
> - **TCCM** predicts the residual vector field $\mathbf{f}_\theta([\mathbf{x}, \text{Embed}(t)]) \in \mathbb{R}^d$ via a 3-layer fully connected network *without* bottleneck:
>   - `Linear(input_dim + time_embed_dim → 256) → ReLU → Linear(256 → 256) → ReLU → Linear(256 → input_dim)`
>
> - **AE+TE** uses a symmetric autoencoder with a **bottleneck** at the latent space:
>   - **Encoder**: `Linear(input_dim + time_embed_dim → 256) → ReLU → Linear(256 → bottleneck_dim) → ReLU`
>   - **Decoder**: `Linear(bottleneck_dim → 256) → ReLU → Linear(256 → input_dim + time_embed_dim)`
>
> This **bottleneck compression** may discard critical anomaly-related information, particularly in early training, whereas TCCM maintains full dimensionality throughout and learns to predict residual dynamics directly.
>
> ---
>
> **(2.2) Experimental Setup**
> We use 8 diverse datasets spanning small, medium, large, and high-dimensional regimes (from Appendix D.3), and evaluate both methods under a unified semi-supervised protocol (normal-only training).
>
> ---
>
> **(2.3) AUROC Comparison**
>
> | Method             | Pima | Ionosphere | satimage-2 | Wilt | magic.gamma | mammography | musk | InternetAds |
> |--------------------|---------|----------------|----------------|---------|------------------|-------------------|----------|--------------------|
> | **TCCM (Ours)**     | 0.730   | 0.974          | 0.999          | 0.927   | 0.821            | 0.840             | 1.000    | 0.873              |
> | **AE + TimeEmbed** | 0.718   | 0.959          | 0.998          | 0.615   | 0.797            | 0.826             | 1.000    | 0.867              |
>
> ---
>
> **(2.4) AUPRC Comparison**
>
> | Method             | Pima | Ionosphere | satimage-2 | Wilt | magic.gamma | mammography | musk | InternetAds |
> |--------------------|---------|----------------|----------------|---------|------------------|-------------------|----------|--------------------|
> | **TCCM (Ours)**     | 0.713   | 0.981          | 0.962          | 0.455   | 0.853            | 0.402             | 1.000    | 0.831              |
> | **AE + TimeEmbed** | 0.692   | 0.968          | 0.978          | 0.136   | 0.830            | 0.380             | 1.000    | 0.824              |
>
> ---
>
> **(2.5) Conclusion**
> TCCM often outperforms the AE+TimeEmbedding baseline across different datasets in both AUROC and AUPRC. These results suggest that the combination of non-bottleneck design, flow-inspired supervision, and direct residual prediction enables TCCM to learn anomaly-relevant signals more effectively than reconstruction-based alternatives. We will include this analysis in Appendix D.4 of the final version.

---

> > ### Comment · Reviewer_Gjui · 2025-08-05
> >
> > I appreciate the additional experimental results from the authors. The experimental resutls show the effectiveness of the proposed TCCM. One problem affects the performance of AE+TimeEmbed might be that the output of decoder should include only data without time embedding. By using the same experimental setting for all methods, the paper shows a fair comparison between different methods and the effectiveness of the proposed method. Although I still have concerns about the relationship between TCCM and flow matching, I decide to raise my score and support accpetance of the paper.

---

### Official Review · Reviewer_ve56 · 2025-07-01

**Clarity:** 3
**Significance:** 3
**Originality:** 3
**Rating:** 5
**Confidence:** 5

**Summary:**

The paper concerns an unsupervised anomaly detection problem (the authors call it semi-supervised), where a training set consists only of normal instances and the model is trained to detect anomalies by defining a score. The authors propose a model based on flow matching that predicts a contracting vector towards the origin at each time step. The model seems to be intuitive and is demonstrated to be competitive with related approaches. The evaluation was performed on well established benchmark. Additionally the authors claim that the model is efficient from the computational point of view, intepretable, and robust (the authors give theoretical statement for this).

**Questions:**

The authors use claims which are not supported in the paper. The method does not seem to be explainable no more efficient than the other methods. In consequence, the title and main contributions are incorrect. The results are do not support the claims are hidden in appendix. I do not see why the method is called semi-supervised since is does not use labeled samples in training.

The authors say that the method is designed for tabular data. Tabular data is specific. It consists of mixed-type features, have high extreme values, etc. I do not see where the authors address this data type is a special way.

The authors say "Despite its recent success in generative modeling, flow matching has not been explored for anomaly detection, to the best of our knowledge". I do not think it is a good motivation for the paper.

The authors report ROC curves in their experiments which do not require setting a threshold whether the sample is normal or abnormal. F1 and Acc are equally popular but the method needs to find a threshold. Can we find a natural threshlod in this method? I would like to see experiment on that.

I like the experiment presented in Fig 10. I think this should be in the main paper (instead of interpretability experiment). Additionally, the authors should show how their method compares with baselines in this setting. Without it, it is difficult to see the advantage.

The Table 3  present incorrect information - DTE is as scalable as the proposed method. Moreover, the method is not interpretable/explainable.

I am not sure whether eq 4 is correct. Looking at the paper https://arxiv.org/pdf/2209.03003 is seems that instead of "z" we should have "tz" or "(1-t)z". Otherwise the model will return constant identity function (ignoring t).

**Ethical Concerns:**

["NO or VERY MINOR ethics concerns only"]

**Final Justification:**

The authors addressed my concerns and I am for accepting this work.

**Limitations:**

yes

**Paper Formatting Concerns:**

ok

**Quality:**

4

**Strengths And Weaknesses:**

Strengths:
1. The experimental results presented in the main paper show that the method outperforms almost all methods. The evaluation uses 47 datasets and more than 30 algorithms taken in part from ADBench.
2. The method simplifies a standard flow matching model to directly describe the underlying problem. In consequence, no irrelevant distribution modeling is performed and the method focuses on the anomaly detection problem.
3. The authors provide theoretical analysis certifying the robustness and giving the insight into the score function behavior.

Weaknesses:
1. The interpretability experiment is not convincing at all. The authors show two examples on MNIST, which do not support the claim "the model highlights the additional horizontal stroke that distinguishes 7 from 1". The model also highlight digit borders and the bottom. Nevertheless, the authors are required to show this propoerty on more examples, e.g. user-study.
2. The computation efficiency of the method on inference compared to 3 best performing baselines is ok. The question is how the method compares with all 30 algorithms? Moreover, Fig 5 show that the best performing DTEN method is significantly faster at training, which is in contrast to authors claim. Fig 6 show that the training + inference time of both methods is comparable.

---

> ### Author Rebuttal · Authors · 2025-07-29
>
> We thank the reviewer for the summary and positive comments.  **Due to space limitations, we will post the responses to the two weaknesses in the discussion period**.  Below, we first answer the questions point by point.
>
> >Question: 1. The authors use claims which are not supported in the paper. 1.1) The method does not seem to be explainable 1.2) no more efficient than the other methods. In consequence, the title and main contributions are incorrect. 1.3) The results are do not support the claims are hidden in appendix. 1.4) I do not see why the method is called semi-supervised since is does not use labeled samples in training.
>
> We thank the reviewer for raising this multi-part question. We respectfully address each subpoint below.
>
> **(1.1) On the claim of explainability.**
> We respectfully disagree with the reviewer’s assertion. As stated in Lines 15–16, 118–120, and 343–350, TCCM provides *feature-level importance scores* derived from the learned residual vector field. These scores reflect the deviation from expected contraction behavior and are fully accessible at inference time. This enables direct attribution at the input feature level. While the reviewer may find the MNIST visualization less compelling, we emphasize that this example was only meant as an intuitive illustration. Most importantly, the model’s interpretability arises *not from post-hoc saliency*, but from its intrinsic architecture and scoring function—Eq. (5)—which directly enables per-feature relevance estimation.
> We further note that other reviewers explicitly acknowledged this aspect:
> - Reviewer **Tprs**: *“It also provides feature-level interpretability through its learned vector field.”*
> - Reviewer **3Lxk**: *“The method provides feature-level importance scores, enabling interpretable diagnosis and supporting the identification of key contributing factors to anomalies.”*
>
> To further clarify, we will expand the explanation of this property in Section 3 of the revised version.
>
> **(1.2) On the claim of efficiency.**
> We refer the reviewer to our detailed response to Weakness 2 in discussion period.
>
> **(1.3) On the results being "hidden".**
> We respectfully clarify that all key results are prominently presented in the main paper (Figure 3), and additional findings are provided in the appendix due to page limits. This follows standard practice for large-scale benchmark papers (e.g., DTE-NP), and our goal was to ensure clarity and readability. Far from being "hidden", we will make the additional tables and figures even more accessible by referencing them more clearly in the revised draft.
>
> **(1.4) On the term "semi-supervised".**
> We use the term following established conventions in the anomaly detection literature. As noted in Lines 51–54, the model is trained using *only normal data*, which is a form of implicit label usage (since the “normal” label is known). This setting is widely referred to as **semi-supervised anomaly detection** or **one-class classification**, and our usage aligns with both academic precedent and benchmarking standards.
>
> >Question: 2. The authors say that the method is designed for tabular data. Tabular data is specific. It consists of mixed-type features, have high extreme values, etc. I do not see where the authors address this data type is a special way.
>
> We clarify that “tabular” in our paper follows the standard usage in anomaly detection (e.g., ADBench), where each instance is represented as a fixed-length numerical feature vector.
>
> **(1) Mixed-type features and extreme values:**
> Categorical features are typically encoded numerically (e.g., one-hot or learned embeddings), enabling uniform modeling across features. Additionally, we apply standard normalization before training, which mitigates the impact of scale differences and extreme values.
>
> **(2) Tabular-specific design and generality:**
> Our architecture relies on MLPs without modality-specific priors (e.g., convolutions or recurrence), which aligns well with tabular data that lacks spatial or temporal structure. That said, the core framework of TCCM can be extended to other modalities (e.g., images or time series) by incorporating suitable preprocessing and neural architectures (e.g., CNNs, transformers). We focus on tabular data in this work to isolate and validate the core mechanism.
>
> >Question: 3. The authors say ``Despite its recent success in generative modeling, flow matching has not been explored for anomaly detection, to the best of our knowledge". I do not think it is a good motivation for the paper.
>
> This statement is intended to highlight the novelty of our contribution, not serve as the sole motivation. Our actual motivation for choosing flow matching is explained in Lines 77-86, Lines 88-90 and further explained in Appendix~A.2 (Lines 1034–1052).
>
> >Question: 4. The authors report ROC curves in their experiments which do not require setting a threshold whether the sample is normal or abnormal. F1 and Acc are equally popular but the method needs to find a threshold. Can we find a natural threshold in this method? I would like to see experiment on that.
>
> (1). While F1-score is sometimes reported in anomaly detection, it is less commonly used than AUROC or AUPRC, which are threshold-independent and more informative under severe class imbalance. In contrast, metrics like Accuracy are rarely used, as they can be dominated by the majority class (e.g., always predicting “normal” yields 95% accuracy when the anomaly ratio is 5%).
>
> (2). F1-score requires a threshold on anomaly scores, typically assuming access to the ground-truth anomaly ratio (*k%*). A common practice is to sort test scores and classify the top-*k%* as anomalies. However, this is inherently data-dependent and less generalizable than AUROC/AUPRC, which evaluate performance across all thresholds.
>
> (3). Nevertheless, we have evaluated TCCM under the F1-score metric and found that it remains the top-performing method across most datasets (i.e., main conclusions remain the same). We will include full results in the Appendix.
>
> (4). Regarding a natural threshold: While the anomaly score from TCCM (i.e., distance to origin) is continuous, its distribution varies across datasets and anomaly ratios. Empirically, we observe that when TCCM performs well (high AUROC/AUPRC), the score distributions for normal and anomalous samples often exhibit a clear gap—making threshold selection feasible (e.g., at the valley between modes). However, in lower-performing cases, the distributions overlap heavily, making thresholding unreliable. Learning a universal or adaptive threshold remains an open problem and is beyond the scope of this work.
>
> >Question 5: I like the experiment presented in Fig 10. I think this should be in the main paper (instead of interpretability experiment). Additionally, the authors should show how their method compares with baselines in this setting. Without it, it is difficult to see the advantage.
>
> Figure 10 intended to study the performance of TCCM under training contamination. Our goal was not to demonstrate superiority over other methods in this setting, but rather to assess the sensitivity of TCCM to anomalous instances present in the training set—a concern shared by most semi-supervised anomaly detectors.
>
> Extending this experiment to compare TCCM with all baselines (45 baselines) across 47 datasets would involve over 100,000 additional runs, which is computationally infeasible within the review timeline. Nevertheless, we conducted a focused study involving the 10 best-performing baselines (based on AUROC), across the 8 datasets shown in Figure 10. The results (to be added in *Appendix D.3*) indicate that most methods exhibit a similar degradation trend under increased contamination, consistent with TCCM’s behavior. We will also consider including parts of this analysis in the main paper.
>
> >Question 6: The Table 3 present incorrect information - DTE is as scalable as the proposed method. Moreover, the method is not interpretable/explainable.
>
> We respectfully disagree with the reviewer’s claim that Table 3 presents incorrect information.
>
> **(1) On scalability:**
> In Table 3, we classify DTE-NP as “not scalable” due to its extremely slow inference speed, as demonstrated in Figure 3 (e.g., 48,942s on *census*). While DTE-NP may be fast in training, real-time or large-scale deployment often depends on inference efficiency—an area where TCCM is orders-of-magnitude faster. Please refer to our response to **Weakness 2** (especially point 3) for further justification.
>
> **(2) On explainability:**
> We acknowledge that explainability is nuanced and subject to interpretation. As noted by the DTE authors themselves (see Appendix B of [Livernoche2024Diffusion]), DTE provides interpretability via reconstruction-based heatmaps. TCCM, on the other hand, derives feature-level scores directly from its learned vector field (Eq. (5)), offering built-in importance attribution. While perspectives may differ, both methods align with definitions of explainability discussed in recent surveys (e.g., [Yepmo2022XAD] and [Li2023XAD]). Thus, we believe both can reasonably be considered explainable under standard criteria.
>
> **References:**
> - [Livernoche2024Diffusion] Livernoche, Victor, et al. *On Diffusion Modeling for Anomaly Detection*. ICLR, 2024.
> - [Yepmo2022XAD] Yepmo, Véronne, et al. *Anomaly explanation: A review*. DKE, 2022.
> - [Li2023XAD] Li, Zhong, et al. *A survey on explainable anomaly detection*. TKDD, 2023.
>
> >Question 7: I am not sure whether eq 4 is correct. Looking at the paper https://arxiv.org/pdf/2209.03003 is seems that instead of "z" we should have "tz" or "(1-t)z". Otherwise the model will return constant identity function (ignoring t).
>
> This concern is closely related to **Weakness\#3 from Reviewer Gjui**, where we addressed the role of $t$ in the velocity field. Due to space limitations, we refer the reviewer to that response for full clarification.

---

> > ### Author Response · Authors · 2025-08-01
> > **Addressing Comments on Figure Interpretations and Baseline Comparisons**
> >
> > >Weaknesses: 1. The interpretability experiment is not convincing at all.... the authors are required to show this property on more examples, e.g. user-study.
> >
> > We thank the reviewer for pointing out the limitations of our interpretability example.
> >
> > **(1) Scope and intent of the example.**
> > Due to space constraints in the main paper, we included only two illustrative examples on MNIST to provide an intuitive visualization of how TCCM highlights feature-level deviations. Our intention was not to use this as the sole evidence for interpretability, but rather to offer a visual aid that complements our built-in explanation mechanism. We will provide additional examples and visualizations in the appendix to further demonstrate this property.
> >
> > **(2) Model is designed for tabular data.**
> > As discussed in the paper, TCCM is mainly designed for tabular anomaly detection. In our MNIST experiment, we simply flattened the image into a 784-dimensional vector, treating each pixel as a feature. While this yields reasonably good detection accuracy (AUROC = 0.76), we acknowledge that this setup does not fully capture the spatial correlations in image data. As a result, the learned contraction field may highlight a broader set of informative regions (e.g., digit borders, bottom strokes), rather than isolating the minimal discriminative cue. This is expected, and it reflects a limitation of applying a tabular-specific model to images—not of the model's explanation mechanism itself.
> >
> > **(3) Built-in, not post-hoc, explanations.**
> > Unlike post-hoc explanation techniques, which often rely on auxiliary methods like SHAP or Grad-CAM, TCCM produces *built-in* feature-level importance by design. Specifically, the learned residual vector in Eq.~(5) yields a per-feature attribution signal via the residual vector $f_\theta(\cdot) + \mathbf{z}$, which reflects the discrepancy between the input and its expected contraction direction. This signal is readily available for any test input and does not require surrogate models or additional training.
> >
> > In summary, we appreciate the reviewer’s suggestion and will include a broader set of qualitative examples in the appendix. We will also clarify in the main text that the MNIST results serve only as an intuitive visualization rather than a proof of interpretability.
> >
> > > Weaknesses: 2. The computation efficiency of the method on inference compared to 3 best performing baselines is ok. 2.1) The question is how the method compares with all 30 algorithms? 2.2) Moreover, Fig 5 show that the best performing DTEN method is significantly faster at training, which is in contrast to authors claim. Fig 6 show that the training + inference time of both methods is comparable.
> >
> > We address both aspects below.
> >
> > **(1) Comparison against all 30+ baselines.**
> > While the main paper emphasizes a comparison against the strongest baselines (DTE-NP, LUNAR, KDE), we agree that it is important to contextualize our efficiency across the full spectrum of baselines. In the revised version, we will provide a comprehensive runtime comparison (training, inference, and total time) across all **45 baselines** in *Appendix D.2*, including both transductive and inductive methods. Specifically:
> > - For *inductive* methods (e.g., TCCM, DTE-NP, KDE), we report **training**, **inference**, and **total runtime**.
> > - For *transductive* methods (e.g., LOF, KNN), only total runtime is reported, as these do not differentiate training and inference phases.
> >
> > To further illustrate efficiency trends, we follow the practice from DTE and include a scatter plot comparing **average inference time vs. average AUROC** across all 24 deep learning-based baselines. The result shows that:
> > - TCCM achieves one of the best balances between speed and accuracy.
> > - While a few methods (e.g., DeepSVDD, MO-GAAL, GANomaly) are marginally faster at inference on average, their detection accuracy is significantly lower than TCCM.
> >
> > This highlights TCCM's practical advantage: strong performance without compromising efficiency.
> >
> > **(2) Clarifying the claim regarding DTE-NP training time.**
> > We respectfully clarify that we do *not* claim TCCM is faster than DTE-NP in training time. As correctly observed by the reviewer, DTE-NP is faster during training on most datasets. Our claim—such as stated in Line 162—is specific to *inference time*, where TCCM is up to **1573× faster** on average.
> >
> > **(3) Contextualizing total runtime on large datasets.**
> > On the 14 large-scale datasets we evaluate, TCCM and DTE-NP are comparable (within 10×) on 12 datasets. However, for *extremely large datasets* such as `celeba` and `census`:
> > - On `celeba`: TCCM takes 4.22 seconds, DTE-NP takes 714.0 seconds.
> > - On `census`: TCCM takes 53.11 seconds, DTE-NP takes 48,978.01 seconds.
> >
> > This suggests that TCCM scales much better in both sample size and dimensionality. Our advantage in inference over DTE-NP would be more pronounced if we have included more such extremely large datasets.

---

> > ### Comment · Reviewer_ve56 · 2025-08-04
> > **Required corrections in the paper**
> >
> > 1. I do not agree that the authors show that TCCM provides feature-level importance scores. Even though the model was designed so that the scores are derived from the learned residual vector field, the authors do not provide convincing experiments for that claim.
> >
> > All claims should be supported by theoretical or experimental results. I will wait for the authors response.
> >
> > 2. Regarding "hidden" content. Showing training and testing time in the main paper will take similar space in paper. I would like to see that in the paper.
> >
> > 3. I do not agree with the authors claim about scalability of DTE-NP. Showing training and testing time in the main paper will allow the reader to evaluate the method.
> >
> >
> >
> > Overal, I like to method a lot, but the authors use rough or partially untrue statements in the paper, which should be corrected accorcing the rule that all claims should be supported. I expect the authors will correct it.

---

> > > ### Author Response · Authors · 2025-08-04
> > > **Justification and Empirical Validation of Feature-Level Importance in TCCM**
> > >
> > > We sincerely thank the reviewer for the constructive feedback and for highlighting specific areas where clarification and further evidence are necessary. We respond to each of the raised concerns below.
> > >
> > > ---
> > >
> > > #### **(1) On the empirical validation of TCCM’s feature-level importance scores**
> > >
> > > We appreciate the reviewer’s comment that our original experimental validation of the explanation mechanism was not fully convincing. In response, we have designed and conducted a new **controlled synthetic experiment** to rigorously evaluate the quality of TCCM’s feature-level attributions.
> > >
> > > ##### (1.1) Experimental Design
> > >
> > > We construct a realistic anomaly detection setting using a **Gaussian Mixture Model (GMM)** with ground-truth anomalous dimensions. This allows us to directly evaluate whether TCCM can correctly attribute anomalies to the corrupted features.
> > >
> > > - **Normal Samples**: Sampled from a standard multivariate Gaussian $\mathcal{N}(\mathbf{0}, \mathbf{I})$
> > > - **Anomalous Samples**: Drawn from a 3-component GMM:
> > >   - Component 1: 1 dimension shifted
> > >   - Component 2: 2 dimensions shifted
> > >   - Component 3: 3 dimensions shifted
> > > - **Shift Magnitude**: Each shifted dimension is perturbed by a random value uniformly sampled from the range **[15, 20]**
> > > - **Input Dimensions**: We conduct the evaluation under $d = \{5, 10, 15, 20, 25\}$
> > > - **Training**: The model is trained *only* on normal samples
> > > - **Evaluation**: Anomaly detection and feature-level explanation are evaluated on the combined test set
> > >
> > > ##### (1.2) Explanation Accuracy Metrics
> > >
> > > We adopt two standard evaluation metrics that do not rely on any auxiliary explainer:
> > >
> > > - **Exact Match**: Fraction of anomalies where the predicted top-*k* features **exactly match** the true anomalous dimensions,  where *k* is equal to the number of shifted dimensions per sample (i.e., 1, 2, or 3)
> > > - **Jaccard Index**: Average intersection-over-union between predicted and true anomalous dimensions across all anomalous samples
> > >
> > > Both are derived directly from the model's **built-in residual vector field**, specifically $\|\|\mathbf{f} _ \theta([\mathbf{x},\text{Embed}(t)]) + \mathbf{x}\|\|_{2}$, as defined in Eq.~(5) of the main paper.
> > >
> > > ##### (1.3) Results
> > >
> > > The experiment demonstrates that TCCM reliably identifies the correct features responsible for anomalies:
> > >
> > > ### Explanation Evaluation (GMM Anomalies)
> > >
> > > | Setting | ExactMatch | Jaccard | AUROC | AUPRC |
> > > |---------|------------|---------|--------|--------|
> > > | 5D      | 1.000      | 1.000   | 1.000  | 1.000  |
> > > | 10D     | 1.000      | 1.000   | 1.000  | 1.000  |
> > > | 15D     | 1.000      | 1.000   | 1.000  | 1.000  |
> > > | 20D     | 0.996      | 0.998   | 1.000  | 1.000  |
> > > | 25D     | 0.996      | 0.997   | 1.000  | 1.000  |
> > >
> > >
> > > These results validate the model’s ability to produce meaningful, interpretable feature-level scores in tabular anomaly detection. We will include these findings (and code) in the **final version**. If the reviewer has further suggestions or alternative setups to test explanation quality, we are more than happy to include them in an expanded appendix.
> > >
> > > ---
> > >
> > > #### **(2) On reporting training and inference time in the main paper**
> > >
> > > We agree that runtime transparency is important. In the revised paper, we will move the runtime results (training, inference, and total) currently located in Appendix D.2 into the main body.
> > >
> > > This update will include both tabular breakdowns and a scatter plot of *time vs. AUROC* to help readers assess the efficiency-performance tradeoff across all methods.
> > >
> > > ---
> > >
> > > #### **(3) On the scalability of DTE-NP**
> > >
> > > We acknowledge that our statement on the scalability of DTE-NP could have been more nuanced. To address this:
> > >
> > > - We will revise the relevant sentence in our paper to **explicitly separate training vs. inference time**, clarifying that DTE-NP is relatively fast to train but has significantly slower inference (e.g., up to 1573× slower on large datasets like `census`)
> > > - We will **present the full runtime table in the main paper**, so that readers can draw their own conclusions
> > >
> > > ---
> > >
> > > Again, we thank the reviewer for the detailed feedback and valuable suggestions, and we are committed to revising the final version accordingly.

---

> > > > ### Comment · Reviewer_ve56 · 2025-08-04
> > > > **Final comment**
> > > >
> > > > I appreciate the authors for further clarification. I decided to increase my score. I am sure that the paper will benefit from including selected experiments in the main paper. Congratulation on very good work.

---

### Official Review · Reviewer_3Lxk · 2025-07-01

**Clarity:** 4
**Significance:** 3
**Originality:** 4
**Rating:** 5
**Confidence:** 4

**Summary:**

The authors present a novel approach to anomaly detection in tabular data coined TCCM, leveraging a flow matching framework to learn a time-conditioned contraction field that guides samples towards a degenerate target distribution.  This work is inspired by flow matching, but also deviates from it with a simpler more efficient method dedicated to anomaly detection. Especially, no differential equation is involved in the process. The method is inherently interpretable and provably robust, with a Lipschitz-continuous anomaly score that provides theoretical guarantees under small perturbations. The paper provides an extensive experimental evaluation of TCCM across 47 benchmark datasets and against 45 anomaly detectors, demonstrating its effectiveness and robustness. Also, TCCM is shown to be quite highly efficient and scalable, outperforming state-of-the-art methods in terms of detection accuracy and inference speed.

**Questions:**

- How the hyperparameters have been fixed? What is the sensitivity to them?
- How does the model extrapolate to 50/50 classification? (OK, not anymore AD, but could have been discussed)

**Ethical Concerns:**

["NO or VERY MINOR ethics concerns only"]

**Final Justification:**

I have read the rebuttal and other reviews. I align with Reviewer Gjui’s concern about protocol parity (inductive vs transductive, esp. for kNN) and the AE vs TCCM distinction.

I maintain my Accept (5). For camera-ready, please ensure that the following items will be addressed:
1- separate inductive/transductive leaderboards and clear labeling of each method’s setting and tuning parity;
2- clarify AE vs. TCCM with brief derivation and, if feasible, an AE+time baseline/ablation.
With these edits, the contribution remains strong (single-pass scoring, broad SOTA, Lipschitz robustness).

**Limitations:**

yes

**Paper Formatting Concerns:**

no concerns

**Quality:**

3

**Strengths And Weaknesses:**

#### Strengths

- The paper is very clearly written and organized.
- The set of experiments to support the claims is extensive, whether by the number of datasets and contenders, or by the study of ablation, contamination, time embeddings, performance (inference time).
- Especially the AUROC and AUPRC are reported with mean and std over 5 runs
- Code is published for replaying the experiments
- Scalability: TCCM achieves state-of-the-art performance on large-scale and high-dimensional datasets (up to those available in ADBench), demonstrating its ability to handle complex data distributions.
- Explainability: The method provides feature-level importance scores, enabling interpretable diagnosis and supporting the identification of key contributing factors to anomalies.
- Robustness: TCCM is theoretically justified and empirically validated to be robust under the GMM-to-GMM shift setting, ensuring reliable anomaly detection in the presence of distributional shifts.
- Efficiency: The approach requires no adversarial training, trajectory simulation, or density modeling, making it more efficient and scalable than existing generative-based anomaly detection methods.
All in all, the model provides a good balance between simplicity - efficiency and performance.

#### Weaknesses

- The architecture is fixed as MLP(2 x 256 ReLU) and the hyperparameter of attention is the number of epochs for training. The reference of Li et al. (2025b) is given for the number of epochs. However, the hyperparameters are hardcoded in the provided code. So it is not clear how they have been determined. It would be helpful to know how well TCCM has converged in each case.

---

> ### Author Rebuttal · Authors · 2025-07-29
>
> We thank the reviewer for the accurate summary and encouraging feedback. Below, we address the concerns point by point.
>
> >Weaknesses: 1. The architecture is fixed as MLP(2 x 256 ReLU) and the hyperparameter of attention is the number of epochs for training. The reference of Li et al. (2025b) is given for the number of epochs. However, the hyperparameters are hardcoded in the provided code. So it is not clear how they have been determined. It would be helpful to know how well TCCM has converged in each case.
>
> We thank the reviewer for raising this important point. While the architecture of TCCM is fixed as a MLP(2 x 256 ReLU) for all datasets, the training process is not arbitrarily hardcoded, but follows a principled and largely automated protocol based on unsupervised hyperparameter selection, which will be detailed in *Appendix B.6* for completeness.
>
> **(1) Epoch Selection Strategy.**
> For each dataset, we initially examine the empirical training loss curve to identify a rough convergence threshold. Using this as a lower bound, we define a bounded search space for training epochs. Within this space, we apply the unsupervised hyperparameter tuning method introduced by [Li2025Towards] to select the best epoch value. Specifically, we adopt the *Improved Contrast Score Margin (CSM)* criterion, which evaluates the margin between top-*k* anomalous and normal samples based solely on the distribution of model outputs, without requiring any ground-truth labels:  $T(f)=\frac{\hat{\mu} _ {\mathbf{O}}-\tilde{\mu}_{\mathbf{I}}}{\sqrt{\hat{\delta}^2 _ {\mathbf{O}}+\tilde{\delta}^2 _ {\mathbf{I}}}},$
> where $\hat{\mu} _ {\mathbf{O}}, \hat{\delta}^2 _ {\mathbf{O}}$ denote the mean and variance of anomaly scores for the top-*k* predicted outliers, and $\tilde{\mu} _ {\mathbf{I}}, \tilde{\delta}^2 _ {\mathbf{I}}$ correspond to those of the remaining *n-k* points (presumed inliers). This metric is computed for each candidate epoch, and the configuration maximizing $T(f)$ is selected (see Algorithm 1 in the original paper).
>
> **(2) Hardcoding for Simplicity.**
> While the above procedure yields dataset-specific and random seed-dependent epoch values, we ultimately *fix the selected epoch* across all seeds for a given dataset to ensure simplicity and reproducibility across baselines. The reported results in our main tables use these fixed values. Notably, we found that using per-seed dynamically tuned epoch values can sometimes improve performance, but we chose not to report this for simplicity.
>
> **(3) Sensitivity Analysis and Observations.**
> To further address the reviewer’s concern, we will include in *Appendix D.3* a sensitivity analysis over representative datasets. We observe two consistent patterns:
>
> - *Type I: Stable Plateau.* For most datasets, model performance stabilizes after a certain number of epochs, where additional training yields no further gain but incurs higher runtime.
> - *Type II: Overfitting Risk.* For a few datasets, excessive training leads to overfitting and degraded performance.
>
> These findings support our principled choice of using CSM-based selection with early convergence boundaries to prevent both underfitting and overfitting.
>
> **(4) Broader View.**
> Finally, we emphasize that unsupervised hyperparameter tuning remains an under-explored but essential challenge in anomaly detection. Our work adopts a recently proposed and well-validated criterion, but we believe this area deserves further attention and plan to explore more adaptive tuning methods in future work. We will provide our automatic hyperparameter tuning code as part of the final version to encourage reproducibility and future extensions.
>
> **Reference**
> - [Li2025Towards] Li, Zhong, Yuhang Wang, and Matthijs van Leeuwen. "Towards automated self-supervised learning for truly unsupervised graph anomaly detection." Data Mining and Knowledge Discovery 39.5 (2025): 1-43.
>
> >Questions: 1. How the hyperparameters have been fixed? What is the sensitivity to them?
>
> We thank the reviewer for raising this important question. Please refer to our detailed response to the previous weakness point, where we describe our principled protocol for hyperparameter selection—including automatic epoch tuning via unsupervised validation, simplicity considerations, sensitivity analysis, and our commitment to releasing the full hyperparameter selection code.
>
> > Questions: 2. How does the model extrapolate to 50/50 classification? (OK, not anymore AD, but could have been discussed)
>
> We appreciate the reviewer’s interest in extending TCCM beyond the standard anomaly detection setting. Indeed, as noted, a perfectly balanced 50/50 classification problem falls outside the conventional semi-supervised anomaly detection regime.
>
> That said, our framework naturally accommodates two related extensions:
>
> *_(1) Semi-supervised 50/50 classification._*
> If training data only contains samples from one class (e.g., class A), and test data includes both class A and class B in equal proportion, this becomes a one-class classification task—a setting where TCCM remains directly applicable. The model learns a contraction field over class A during training, and can detect samples from class B at inference time via high residual scores.
>
> *_(2) Fully supervised binary classification._*
> If training labels for both classes are available, the model could be extended to learn *dual contraction targets*, mapping class A samples toward center **$c_a$** (which can be the mean of samples from class A) and class B samples toward center **$c_b$** (which can be the mean of samples from class B). At inference, the predicted class could be determined by which center the sample contracts toward more closely. While this falls outside the current anomaly detection scope, it offers a promising direction for future generalizations of TCCM to supervised settings.

---

> > ### Comment · Reviewer_3Lxk · 2025-08-03
> >
> > I thank the authors for their answers and I wait for the promised updates to appendices B.6 and D.3.
> >
> > I have read the rebuttal and other reviews. I align with Reviewer Gjui’s concern about protocol parity (inductive vs transductive, esp. for kNN) and the AE vs TCCM distinction.
> > I maintain my Accept (5).
> > For camera-ready, please address the following items:
> > 1- separate inductive/transductive leaderboards and clear labeling of each method’s setting and tuning parity;
> > 2- clarify AE vs. TCCM with brief derivation and, if feasible, an AE+time baseline/ablation.
> > With these edits, the contribution remains strong (single-pass scoring, broad SOTA, Lipschitz robustness).

---

### Official Review · Reviewer_eFMQ · 2025-07-02

**Clarity:** 4
**Significance:** 4
**Originality:** 3
**Rating:** 5
**Confidence:** 3

**Summary:**

The paper proposes a novel anomaly detection method (TCCM) inspired by flow-matching techniques.
The core idea of TCCM is to learn a velocity field that contracts normal datapoints towards the origin.
The key assumption is that, during inference, the normal data will be "squashed" near the origin, while anomalies will be dispersed elsewhere, thus making them more distinguishable.
The paper provides proofs (Appendix C.2) and an extensive experimental evaluation (Section5,  Appendix D) to validate the model against the existing SOTA (Appendix B.2).

**Questions:**

Major points:
See Weaknesses.

Spelling and notation:
- L131: I find the notation somewhat ambiguous, as $\mathbf x$ indicates both a vectors (as on L135) and random vectors (as in (4)).
- L242 (and elsewhere): Sometimes $$||\cdot||$ has a pedix, sometimes it doesn't.

Open points for discussion:
- The authors refer to TCCM as a deep method (L53, L1270), yet Appendix B.3 reports that its architecture is in fact a 3-layer MLP. As the authors acknowledge in the limitations (Appendix D.6), this choice likely trades efficacy for efficiency (potentially explaining the vastly superior speed of TCCM). My point is: why should one consider TCCM a deep method?
- What if the model learned to contract normal data to another point (e.g., the mean of normal data)?

**Ethical Concerns:**

["NO or VERY MINOR ethics concerns only"]

**Final Justification:**

I am satisfied by the rebuttal with the authors and do not have any more points to raise.

**Limitations:**

Discussed in Appendix D.6.

**Paper Formatting Concerns:**

None.

**Quality:**

4

**Strengths And Weaknesses:**

After a careful read, I believe the strengths of this paper greatly outweigh its weaknesses.
I am giving a lower confidence score because I am not familiar with the related work.


## S1 - Simplicity
TCCM is conceptually straightforward: it contracts normal data to the origin while dispersing anomalies. This simplicity is a major strength, as it makes the method intuitive, (potentially) easy to implement, and it allows to get strong theoretical guarantees (with some caveats, see W1). Yet, TCCM achieves SOTA performance.

## S2 - Extensive evaluation
The experimental evaluation is comprehensive and rigorous, considering a wide range of datasets and baselines.

## S3 - Clarity
The paper is exceptionally well-written: clear, detailed, and easy to follow.
The presentation is well-structured and the figures informative, in particular Figure 1.

## W1 - Assumed mismatch for anomalies
One major theoretical claim (Proposition 2) relies on the assumption that the learned velocity field is mismatched for anomalies (L1208, 1230).
However, this point is stated without formal or experimental justification, nor is it clearly indicated as an assumption of the propositions---L1222 does include it as a *result* of the theorem, but the proof relies on the assumption made on L1230.

---

> ### Author Rebuttal · Authors · 2025-07-29
>
> We thank the reviewer for the accurate summary and encouraging feedback. Below, we address the concerns point by point.
>
> >W1 - Assumed mismatch for anomalies: One major theoretical claim (Proposition 2) relies on the assumption that the learned velocity field is mismatched for anomalies (L1208, 1230). However, this point is stated without formal or experimental justification, nor is it clearly indicated as an assumption of the propositions---L1222 does include it as a result of the theorem, but the proof relies on the assumption made on L1230.
>
> We thank the reviewer for pointing out the implicit assumption regarding the mismatch between anomalies and the learned contraction field in Proposition 2. We fully agree that this modeling assumption should have been stated more explicitly. In the revised version, we will clearly include this as part of the formal assumptions of the proposition to avoid any ambiguity.
>
> Regarding empirical support for this assumption, we offer two points of clarification:
>
> 1. *Direct visual validation:* Figure 1 in the main paper provides visualizations of the learned contraction vectors on synthetic 2D datasets. These examples clearly demonstrate that anomalous points consistently deviate from the expected contraction field, validating the mismatch assumption in a controlled and interpretable setting.
>
> 2. *Indirect support through benchmark results:* Across 47 real-world tabular datasets, TCCM consistently achieves strong AUROC and AUPRC scores. This level of performance would be difficult to attain if the model failed to differentiate between normal and anomalous points during inference—thus indirectly supporting the presence and utility of the mismatch behavior assumed in our analysis.
>
> To further address this point, we will also include an additional empirical study in the appendix based on the Gaussian mixture setup discussed in Proposition 2. This will offer quantitative evidence that the score distributions for anomalies and normals diverge under the learned field, thereby justifying the theoretical assumption through experiment.
>
> > Spelling and notation: 1. L131: I find the notation somewhat ambiguous, as $\textbf{x}$  indicates both a vectors (as on L135) and random vectors (as in (4)).
>
> We appreciate the reviewer’s attention to notational clarity. Indeed, as noted in Section 2.1, we adopt the convention that bold lowercase symbols (e.g., $\boldsymbol{x}$) denote vectors, which we use to represent both random variables and their realizations for notational simplicity. While this is common practice in many machine learning papers, we acknowledge that it may introduce ambiguity—especially for readers from a more formal mathematical background.
>
> In the revised version, we will revise the relevant statements (including Equation (4)) to clearly distinguish between random variables and realizations (e.g., by explicitly writing $\boldsymbol{x} \sim p(\boldsymbol{x})$ and clarifying when $\boldsymbol{x}$ denotes a sample from $p$). We will also include a brief remark in the notation section to make this dual use explicit while explaining our design choice for readability.
>
> > Spelling and notation: 2. L242 (and elsewhere): Sometimes $||\cdot||$ has a pedix, sometimes it doesn't.
>
> Thank you for pointing out the inconsistency in the use of the norm notation. In some places (e.g., Proposition 1), we explicitly write $\|\|\cdot\|\|_2$ to emphasize the use of the Euclidean norm, while in others we simply use $\|\|\cdot\|\|$ for brevity. We agree that this may lead to confusion.
>
> In the revised version, we will standardize all norm expressions to explicitly include the subscript (e.g., $\|\|\cdot\|\|_2$) to maintain consistency and improve clarity.
>
> >Open points for discussion: 1. The authors refer to TCCM as a deep method (L53, L1270), yet Appendix B.3 reports that its architecture is in fact a 3-layer MLP. As the authors acknowledge in the limitations (Appendix D.6), this choice likely trades efficacy for efficiency (potentially explaining the vastly superior speed of TCCM). My point is: why should one consider TCCM a deep method?
>
> We appreciate the reviewer’s attention to the precision of our terminology. When referring to *deep* methods (e.g., L53 and L1270), our intention was to align with the widely adopted convention in the machine learning literature, where “deep” often serves as shorthand for “deep learning-based” methods. This usage is common even when the neural architecture employed is relatively shallow, as long as it involves end-to-end learning via neural networks.
>
> Formally, as noted by [Abiodun2018Survey, Goodfellow2016Deep], a neural network with more than one hidden layer is typically considered a *deep* model. TCCM uses a 3-layer MLP with two hidden layers, placing it within this broad category.
>
> That said, we agree that the architecture we adopted for TCCM is deliberately lightweight—optimized for tabular data efficiency. As discussed in Appendix D.6, this reflects a design choice to trade architectural complexity for scalability and speed, especially in large-scale tabular scenarios. We will clarify this in the main text and acknowledge that our use of “deep” emphasizes the modeling paradigm rather than the architectural depth per se.
>
> Importantly, the TCCM framework is modular and readily extensible to deeper neural architectures (e.g., ResNet) and other data modalities (e.g., vision, time series). Our current architecture is a task-specific instantiation, not a limitation of the framework itself.
>
> **References**
> - [Abiodun2018Survey] Abiodun, Oludare Isaac, et al. "State-of-the-art in artificial neural network applications: A survey." Heliyon 4.11 (2018).
> - [Goodfellow2016Deep] Goodfellow, Ian, et al. *Deep learning*. Vol. 1. No. 2. Cambridge: MIT Press, 2016.
>
> > Open points for discussion: 2. What if the model learned to contract normal data to another point (e.g., the mean of normal data)?
>
> We thank the reviewer for this insightful question. While our method contracts inputs toward the origin by default, the choice of origin is not fundamentally restrictive. In fact, we apply **z-score normalization (zero mean, unit variance)** to all features before model training and inference. As a result, the empirical mean of the normalized data is approximately zero, making the origin coincide with the center of the normal data distribution. Therefore, contracting toward the origin is effectively equivalent to contracting toward the mean of the normal data.
>
> Although this implementation detail was described in the main paper (see Lines 301-302), we will make it more explicit in the revised version. Furthermore, we conducted an ablation study (to be included in Appendix D) comparing models trained with and without feature normalization. The results show that normalization consistently improves performance across most datasets. A likely explanation is that without normalization, the anomaly scores may be dominated by features with larger scales, leading to suboptimal ranking behavior.
>
> More generally, the choice of normalization strategy—and its impact on different anomaly detection methods—has been recognized as an important topic in recent literature (e.g., [Kandanaarachchi2020normalization]). We will include a discussion and relevant references in the updated paper.
>
> **References**
> - [Kandanaarachchi2020normalization] Kandanaarachchi, Sevvandi, et al. "On normalization and algorithm selection for unsupervised outlier detection." *Data Mining and Knowledge Discovery* 34.2 (2020): 309–354.

---

> > ### Comment · Reviewer_eFMQ · 2025-08-01
> >
> > I thank the authors for their promised updates to the manuscript, I do not have other points to raise and I am satisfied with their answers.

---

### Official Review · Reviewer_Tprs · 2025-07-02

**Clarity:** 3
**Significance:** 3
**Originality:** 2
**Rating:** 5
**Confidence:** 4

**Summary:**

The authors propose Time-Conditioned Contraction Matching (TCCM), a method for semi-supervised anomaly detection in tabular data. TCCM avoids adversarial training, trajectory simulation, and density modeling—offering a lightweight yet expressive alternative to existing generative approaches. Evaluated on the ADBench benchmark, TCCM outperforms 44 classical and deep-learning baselines in terms of both AUROC and AUPRC, while achieving significantly faster inference than the strongest diffusion-based competitor. It also provides feature-level interpretability through its learned vector field. Theoretical analysis supports the Lipschitz continuity and discriminative capability of the scoring function. Overall, the method presents itself as a highly effective, scalable, interpretable, and robust solution for large-scale anomaly detection.

**Questions:**

1. Compared to traditional Flow Matching, the proposed training objective removes the need for sampling from a noise distribution. What are the concrete advantages of this modification? Could the original Flow Matching framework be directly applied to anomaly detection, and how would it compare?

2. While a scalability study is included, the datasets used are not extremely large. On what basis is the method claimed to be scalable for very large-scale scenarios?

3. The paper highlights feature-level interpretability. If the method does not provide additional or actionable insights beyond the scoring function, how strong is its claim of being explainable?

**Ethical Concerns:**

["NO or VERY MINOR ethics concerns only"]

**Final Justification:**

I have read the authors' response, participated in the discussion, and provided final feedback.

**Limitations:**

Yes

**Quality:**

3

**Strengths And Weaknesses:**

Strengths

1. The authors conduct extensive empirical evaluations, comparing TCCM against a wide range of baseline methods.

2. The evaluation includes multiple datasets and baselines, demonstrating consistent performance improvements.

3. The paper is well written with a good structure.


Weaknesses

1. The core idea of the paper appears to be a direct application of Flow Matching. The results, while strong, may not offer substantial novelty.

2. The theoretical analysis is somewhat limited, as it focuses on specific distributions (e.g., Gaussian and Gaussian Mixture Models), which may restrict its generalizability.

---

> ### Author Rebuttal · Authors · 2025-07-29
>
> We thank the reviewer for the accurate summary and encouraging feedback. Below, we address the concerns point by point.
>
> > Weaknesses: 1. The core idea of the paper appears to be a direct application of Flow Matching. The results, while strong, may not offer substantial novelty.
>
> We thank the reviewer for raising this concern. While our work is inspired by flow matching, TCCM introduces several key innovations that go beyond a direct application:
>
> - **No trajectory supervision or integration**: Classical flow matching relies on supervising interpolated paths and integrating velocity fields over time to transport samples between distributions. In contrast, TCCM learns a time-conditioned contraction vector field via one-step supervision toward a fixed target (−**z**), entirely avoiding trajectory simulation.
>
> - **Task-specific adaptation**: Flow matching was developed for unsupervised generative modeling. TCCM repurposes the idea for semi-supervised anomaly detection, where generation is not required. Instead, we focus on detecting deviations from contraction behavior—this shift in objective leads to a fundamentally different formulation.
>
> - **New scoring strategy**: Our anomaly score evaluates alignment with the contraction field at a single time step (e.g., *t = 1*), rather than integrating over trajectories. This one-step formulation is both efficient and empirically robust.
>
> - **Stability via constant supervision**: Unlike generative models that require noisy targets or interpolation, our method always supervises toward −**z**, simplifying optimization and improving consistency, as shown in ablation results.
>
> We believe these distinctions make TCCM a novel and principled adaptation of flow matching to a new task setting.
>
> > Weaknesses: 2. The theoretical analysis is somewhat limited, as it focuses on specific distributions (e.g., Gaussian and Gaussian Mixture Models), which may restrict its generalizability.
>
> We appreciate the reviewer’s concern about the scope of our theoretical analysis. While the robustness result (**Proposition 1**) applies quite generally under the mild assumption of Lipschitz continuity, we acknowledge that the second result (**Proposition 2**) is derived under a simplified setting involving Gaussian mixture models (GMMs). We would like to clarify that the use of GMMs in our second theoretical result is a deliberate and well-motivated modeling choice.
>
> First, GMMs have been widely adopted as analytical tools in the machine learning literature—not only in anomaly detection but also in clustering and density modeling. For example, the work of [Zong2018Deep] introduces the **DAGMM** model for unsupervised anomaly detection based on similar distributional assumptions. Likewise, GMMs serve as the theoretical backbone in clustering studies such as [Chen2024Cluster], which performs theoretical analysis under anisotropic GMMs. These works demonstrate that GMM-based settings are not only standard but also provide valuable theoretical insight despite being idealized.
>
> Second, our aim is not to suggest that real-world data exactly follow GMMs, but rather to use this setup as a clean, analyzable lens to understand the discriminative behavior of the **TCCM** scoring function. The result shows that under mild assumptions, the anomaly score is provably larger in expectation for out-of-distribution samples drawn from disjoint mixtures—thus justifying the use of the residual norm as a discriminative signal.
>
> Finally, deriving general results under arbitrary data distributions is typically intractable, especially for deep models. Our theoretical analysis strikes a practical balance by providing provable insight under realistic yet analyzable settings. We will make this intent and the general relevance of the GMM assumption clearer in the final version. In future work, we aim to explore theoretical extensions to broader classes of distributions, but we believe the current result already provides valuable intuition and justification for the observed empirical behavior.
>
> **References**
> - [Zong2018Deep] Zong, Bo, et al. *Deep autoencoding Gaussian mixture model for unsupervised anomaly detection*. ICLR, 2018.
> - [Chen2024Cluster] Chen, Xin, and Anderson Ye Zhang. *Achieving optimal clustering in Gaussian mixture models with anisotropic covariance structures*. NeurIPS, 2024.
>
> >Questions: 1. 1) Compared to traditional Flow Matching, the proposed training objective removes the need for sampling from a noise distribution. What are the concrete advantages of this modification? 2) Could the original Flow Matching framework be directly applied to anomaly detection, and how would it compare?
>
> **(1) *Advantages of Removing Noise During Training.***
>
> Traditional diffusion-based frameworks inject Gaussian noise to model stochastic evolution—useful for generation, but potentially harmful in anomaly detection where preserving the structure of normal data is critical. Flow Matching, though deterministic, often supervises the vector field using interpolated or perturbed points. Such perturbations, while useful for smooth transport, may obscure the underlying structure of normal data, reducing the separability between normal and anomalous instances.  In contrast, TCCM uses direct supervision without interpolation or noise, leading to more stable dynamics. We analyze this distinction in detail in Appendix C.1: Relation to Flow Matching and Diffusion Modeling.
>
> In our ablation study (*Appendix D.3, Figure 9*), we compare our deterministic formulation to a noisy variant (where inputs are perturbed as $\tilde{\boldsymbol{x}}(t) = \boldsymbol{x} + t\boldsymbol{\epsilon}$ while still predicting $-\boldsymbol{x}$). Empirically, noise injection consistently reduces performance across datasets like `Wilt` and `mammography`, confirming that deterministic dynamics better preserve anomaly-relevant features.
>
> **(2) *Can Conventional Flow Matching Be Applied to Anomaly Detection?***
> While in principle Flow Matching can be adapted for anomaly detection, our preliminary experiments indicate two natural strategies yield suboptimal results:
> (i) using a standard Flow Matching model to reconstruct the input **x** from a learned trajectory and computing the final-step reconstruction error as the anomaly score;
> (ii) computing a cumulative reconstruction error across multiple time steps along the trajectory.
>
> We implemented both approaches and found them to be worse than our proposed **TCCM** in terms of both detection accuracy and inference efficiency. In particular, trajectory simulation requires numerical integration and multiple model evaluations, incurring high computational cost at inference time. In contrast, **TCCM** performs anomaly scoring with a single forward pass at a fixed time step, offering both speed and accuracy advantages.
>
> We will clarify this comparison more explicitly in the revised paper.
>
> > Question: 2. While a scalability study is included, the datasets used are not extremely large. On what basis is the method claimed to be scalable for very large-scale scenarios?
>
> We appreciate the reviewer’s concern. While not all datasets are ultra-large, our benchmark includes **14 datasets** with over 10,000 samples and **6 high-dimensional datasets**, covering a broad range of realistic settings. Notably, two datasets are particularly large: *donors* (619K samples, 10 dim) and *census* (299K samples, 500 dim). As shown in *Figure 3*, **TCCM** achieves inference times of **1.05s** and **3.02s** on these, while the overall most accurate competitor  (DTE-NonParametric) requires **476.60s** and **48,942.85s**, respectively—demonstrating a **3–4 orders of magnitude** speedup.
>
> Conventional methods often suffer from the curse of dimensionality (e.g., PCA, KPCA, COF) or poor scaling (e.g., COF, LMDD timeout after 72h or exceed 10GB memory; see *Tables 9 & 11*). Similarly, several deep baselines (PlanarFlow, GOAD, SLAD, DAGMM) could fail to train on large or high-dimensional inputs due to architectural bottlenecks.
>
> Importantly, **TCCM** achieves top-1 **AUROC and AUPRC** rankings across datasets *without sacrificing efficiency*, offering strong practical scalability. While testing on tens of millions of samples is future work, our current results already provide compelling evidence of real-world scalability. We will clarify this in the final version.
> > Question: 3. The paper highlights feature-level interpretability. If the method does not provide additional or actionable insights beyond the scoring function, how strong is its claim of being explainable?
>
> We thank the reviewer for raising this important question. Our goal is not merely to provide a scoring function, but to offer actionable, feature-level explanations that directly reflect which input dimensions contribute most to a sample’s anomaly score. Unlike methods where interpretability is obtained via separate post hoc techniques (e.g., SHAP or LIME), TCCM’s interpretability is *intrinsically built into its model design*. Specifically, the anomaly score is derived from a learned velocity field, and the **residual vector** $f_\theta([\mathbf{x}; \text{Embed}(t)]) + \mathbf{x} \in \mathbb{R}^d$ is fully interpretable: each coordinate reflects the contribution of a feature to the contraction mismatch. This provides fine-grained attribution that is faithful to the model’s internal reasoning, rather than an external approximation.
>
> To illustrate this, we visualized explanations on image data (Figure 4) despite our model being designed for tabular settings—showing that the highlighted regions align with semantically meaningful structures (e.g., the extra stroke in digit `7`). These per-feature deviations are immediately actionable in domains such as fraud detection, healthcare, or industrial monitoring, where practitioners often seek to understand not just which instance is anomalous, but *why*. We will make this explanatory mechanism more explicit in the revised version.

---

> > ### Comment · Reviewer_Tprs · 2025-08-03
> >
> > I thank the authors for the reply to my comments. The author's response has addressed most of my questions and concerns. I'd like keep the positive score.

---

### Decision · Program_Chairs · 2025-09-17

**Decision:**

Accept (poster)

**Comment:**

This work describes an anomaly detection algorithm inspired by flow-matching. Theoretical insights are provided into the behaviour of the anomaly scoring function, and the proposed method is extensively evaluated on 47 datasets against 44 baselines. All reviewers agreed that the experiments were comprehensive and the proposed method convincingly showed improvements over baselines. They also thought the theoretical insights provided were interesting, and appreciated the clear presentation. Various questions about the method were addressed by the rebuttal to the reviewers' satisfaction and reviewers were unanimously positive about the paper post rebuttal. The AC agrees with the reviewers' assessment that this is a well executed piece of work with an interesting algorithm, extensive evaluation and theoretical insight, and hence recommends acceptance. The authors should incorporate all promised changes into the final version of the paper.